# Exact closed-form Gaussian moments of residual layers

## Abstract

We study the problem of propagating the mean and covariance of a general multi-variate Gaussian distribution through a deep (residual) neural network using layer-by-layer moment matching. We close a longstanding gap by deriving exact moment matching for the probit, GeLU, ReLU (as a limit of GeLU), Heaviside (as a limit of probit), and sine activation functions; for both feedforward and generalized residual layers. On random networks, we find orders-of-magnitude improvements in the KL divergence error metric, up to a millionfold, over popular alternatives. On real data, we find competitive statistical calibration for inference under epistemic uncertainty in the input. On a variational Bayes network, we show that our method attains hundredfold improvements in KL divergence from Monte Carlo ground truth over a state-of-the-art deterministic inference method. We also give an *a priori* error bound and a preliminary analysis of stochastic feedforward neurons, which have recently attracted general interest.

## 1 Introduction

We are interested in inference of the output distribution of a neural network when the input is Gaussian-distributed. There are at least four reasons to study this problem.

1. The pushforward distribution of a neural network can shed light on local robustness of the network to "typical" (as opposed to worst-case) input perturbations (Wright et al., 2024).

2. Uncertainty propagation allows a neural network trained on a population with certain inputs to make predictions on a shifted distribution of uncertain inputs (Bibi et al., 2018).

3. Deterministic distribution propagation can replace Monte Carlo in both training and inference of variational Bayes neural networks with random weights (Frey & Hinton, 1999; Wu et al., 2019; Petersen et al., 2024; Wright et al., 2024; Rui Li & Trapp, 2025).

4. Analytical moments can be used to understand how activation functions behave in deep networks, by applying the Central Limit Theorem in the wide limit (He et al., 2015).

This problem is of basic importance, easily stated in elementary terms, and widely studied. Yet even in the simplest case of a single hidden layer with two neurons, no existing method can compute the mean and covariance exactly, in closed form.

From the perspective of numerical analysis, a numerical approximation to a functional (such as moments with respect to a measure) should be judged by the largest class of functions on which it is exact (Press, 2007, Chapter 4). Jacobian linearization and the unscented transforms (Julier et al., 1995; Julier, 2002) are exact for linear functions. In one dimension, Gauss-Hermite quadrature using $n$ points is exact for the first moment of polynomials of degree $2n - 1$, and for the second moment of polynomials of degree $n - 1$. But in dimension $d$, Gauss-Hermite quadrature needs $n^d$ points to achieve the same accuracy. In search of efficiency, the activation function-based calculations in Abdelaziz et al. (2015); Huber (2020); Wagner et al. (2022); Akgül et al. (2025); Bibi et al. (2018); Wu et al. (2019) are exact only for one-layer networks *of a single neuron* (they approximate the covariance between neurons). A recent theoretical work proves optimal approximation for a single ReLU neuron (Petersen et al., 2024). The series expansion of Wright et al. (2024) is formally correct for a single hidden layer if taken to infinity, but must be computed by hand and truncated at finitely many terms.

In this work, we have managed to achieve, for the first time, exact moment propagation (to machine precision) for activation functions ReLU, used in the earliest deep networks; GeLU (Hendrycks & Gimpel, 2017), used in frontier LLMs; sine, conceived to overcome frequency bias (Parascandolo et al., 2017; Ziyin et al., 2020) and later appreciated for physics-informed learning (Sitzmann et al., 2020); probit, a close approximation to the logistic sigmoid (Huber, 2020); Heaviside, used in Bayes networks (Wright et al., 2024); as well as general residual connections. We derive closed-form means and full covariance matrices.

On multi-layer networks, our method applies a variational Gaussian approximation, for which we provide *a priori* error bounds. We argue that this approximation not only prudently trades off between accuracy and scalability, but also essentially saturates a hard theoretical limit: exact integration of a general deep neural network under high-dimensional uncertainty is known to be #P-hard in the number of input neurons (Feischl & Zehetgruber, 2025).[1]

**Contribution.** We derive exact first and second moment matching for propagating the mean and covariance matrix of a Gaussian distribution through a single layer of a (residual) neural network (Lemma 1). For deeper neural networks, the single-layer formula is chained over layers, with orders-of-magnitude accuracy improvement over other popular techniques (§5.1). We demonstrate applications of uncertainty propagation to inference on real data: regression on a noisy input (§2) and classification with missing features (§5.3). We discuss hard instances on which methods fail (§4), and then prove a soft guarantee on the method's accuracy in favorable conditions (§3). Finally, we briefly explore a possibility related to stochastic activations (§5.5).

**Related work: local approximation.** Existing results on distribution propagation through neural networks can be taxonomized by the assumptions they impose on the input distribution (Appendix A, Table 4). Some works assume a small covariance matrix. Then the mean and covariance can be propagated by Jacobian linearization (Titensky et al., 2018; Nagel & Huber, 2022; Petersen et al., 2024; Jungmann et al., 2025; Bergna et al., 2025; Rui Li & Trapp, 2025). This formula is justified by the delta method (van der Vaart, 1998, Chap. 3), dates back to Gauss's study of heavenly bodies (Gauss, 1857, Chapter 187), and is taught in textbooks (Taylor, 1997). The Unscented Transformation, used in Astudillo & Neto (2011); Abdelaziz et al. (2015), is also justified by a Taylor expansion (Julier, 2002).

**Related work: analytical approximation.** For certain activation functions including ReLU and GeLU, the mean and the diagonal of the covariance matrix can be computed explicitly, but (prior to this paper) there is no closed-form known for off-diagonal covariances of a hidden layer. Some works set them to zero—the mean-field assumption (Huber, 2020; Goulet et al., 2021; Wagner et al., 2022; Rui Li & Trapp, 2025; Bergna et al., 2025; Akgül et al., 2025; Rui Li & Trapp, 2025). Bibi et al. (2018) uses an analytical approximation around zero mean, and Wu et al. (2019) uses an analytical approximation around infinite mean. For the logistic activation function $\sigma(x) = (1 + e^{-x})^{-1}$; Astudillo & Neto (2011); Abdelaziz et al. (2015), and Huber (2020) approximate $\sigma$ with another function having closed-form Gaussian moments, such as a piecewise exponential function or a rescaled Normal CDF $\Phi$. Appendix A, Table 5 catalogs the literature on moment approximations for activation functions. We exemplify the failure modes of the above methods by giving single-hidden-layer counterexamples in §4.

For a general activation function, Wright et al. (2024) uses a Fourier transform to derive the exact mean and covariance matrix of a general activation function as a formal power series in $\rho$, the inter-neuron correlation. The series must be truncated, as each coefficient needs to be derived by hand.

Our work supersedes the ReLU, GeLU, and Heaviside moment derivations of Frey & Hinton (1999); Bibi et al. (2018); Wu et al. (2019); Huber (2020); Akgül et al. (2025); Wright et al. (2024), as well as the sine moment derivations of Sitzmann et al. (2020) by virtue of being exact on single layers. While we do not have exact integrals for the logistic sigmoid function, we do for $\Phi$, which is a similarly shaped function.[2]

## 2 METHODOLOGY

---

[1] We discuss the runtime of our method in App. O.

[2] In fact, Huber (2020) proposes to train a network using logistic activation and compute its moments using a $\Phi$ surrogate network. Our numerical experiments simply train a $\Phi$ network directly.

The activation function of a neural network is denoted $\sigma : \mathbb{R} \to \mathbb{R}$ and applies elementwise. Except for parameters $A, b, C, d$, capital letters refer to random variables. The layers of a neural network are indicated by superscripts, e.g. $A^k$ is a matrix of parameters for the $k$th layer. If $X$ is a square-integrable random vector, the notation $\mathrm{N}\, X$ refers to a random variable distributed as $\mathcal{N}(\mathbb{E}\, X, \mathrm{Cov}\, X)$.

A neural network is a composition of layer functions $\mathbb{R}^n \to \mathbb{R}^m$

$$g(x; A, b, C, d) = \sigma(Ax + b) + Cx + d, \tag{1}$$

where $A \in \mathbb{R}^{m \times n}, b \in \mathbb{R}^m, C \in \mathbb{R}^{m \times n}, d \in \mathbb{R}^m$ are parameters.

**Definition 1.** *A neural network with $\ell$ layers is the function $f : \mathbb{R}^{n_x} \to \mathbb{R}^{n_y}$ defined by*

$$
\begin{aligned}
f(x) &= f^\ell(x) \\
f^k(x) &= g(f^{k-1}(x); A^k, b^k, C^k, d^k) && \forall k \in \{1 \ldots \ell\} \\
f^0(x) &= x
\end{aligned}
$$

Stated formally, the problem of uncertainty propagation studied in this paper is:

**Problem 1.** *Let $f$ be a neural network with $\ell$ layers. Given $X \sim \mathcal{N}(\mu, \Sigma)$, characterize the distribution of $Y_0 = f(X)$.*

After layer-wise Gaussian approximation, this problem reduces to:

**Problem 2.** *Given $X \sim \mathcal{N}(\mu, \Sigma)$ and $A, b, C, d$; find exact expressions for $\mathbb{E}\, g(X; A, b, C, d)$ and $\mathrm{Cov}\, g(X; A, b, C, d)$.*

### 2.1 OUR ANALYTIC METHOD $Y_{\mathrm{ana}}$

Our method, like Wright et al. (2024), re-approximates each layer by a Gaussian sharing its first two moments:

**Definition 2.** *Let $f$ be a neural network with $\ell$ layers. Given $X \sim \mathcal{N}(\mu, \Sigma)$, the moment-matching Gaussian approximation of $f(X)$, is the random variable $Y_{\mathrm{ana}}$ defined by*

$$
\begin{aligned}
Y_{\mathrm{ana}} &= Y^\ell \\
Y^k &= \mathrm{N}\, g(Y^{k-1}; A^k, b^k, C^k, d^k) && \forall k \in \{1 \ldots \ell\} \\
Y^0 &= X
\end{aligned}
$$

According to basic index manipulation, three transcendental functions are needed to compute the first two Gaussian moments of a layer defined by (1).

**Definition 3.** *Given a nonlinear function $\sigma : \mathbb{R} \to \mathbb{R}$, the functions $M_\sigma : \mathbb{R} \times \mathbb{R}_+ \to \mathbb{R}$ and $K_\sigma, L_\sigma : \mathbb{R}^2 \times \mathbb{R}_{\geq 0}^{2 \times 2} \to \mathbb{R}$ are*

$$M_\sigma(\mu; \nu) = \mathbb{E}\, \sigma(X), \qquad\qquad X \sim \mathcal{N}(\mu, \nu)$$

$$K_\sigma(\mu_1, \mu_2; \nu_{11}, \nu_{22}, \nu_{12}) = \mathrm{Cov}(\sigma(X_1), \sigma(X_2)), \quad \begin{pmatrix} X_1 \\ X_2 \end{pmatrix} \sim \mathcal{N}\left( \begin{pmatrix} \mu_1 \\ \mu_2 \end{pmatrix}, \begin{pmatrix} \nu_{11} & \nu_{12} \\ \nu_{12} & \nu_{22} \end{pmatrix} \right)$$

$$L_\sigma(\mu_1; \nu_{11}, \nu_{22}, \nu_{12}) = \mathrm{Cov}(\sigma(X_1), X_2), \qquad \begin{pmatrix} X_1 \\ X_2 \end{pmatrix} \sim \mathcal{N}\left( \begin{pmatrix} \mu_1 \\ \star \end{pmatrix}, \begin{pmatrix} \nu_{11} & \nu_{12} \\ \nu_{12} & \nu_{22} \end{pmatrix} \right)$$

**Lemma 1.** *For some activation function $\sigma$, let $g$ be the function defined by $g_\sigma(x; A, b, C, d) = \sigma(Ax + b) + Cx + d$. Let $X \sim \mathcal{N}(\mu, \Sigma)$. Then*

$$\left( \mathbb{E}\, g_\sigma(X; A, b, C, d) \right)_i = M_\sigma(\mu_i; \nu_{ii}) + (C\mu)_i + d_i$$

*and*

$$
\begin{aligned}
\left( \mathrm{Cov}\, g_\sigma(X; A, b, C, d) \right)_{i,j} = {}& K_\sigma\left( \mu_i, \mu_j; \nu_{ii}, \nu_{jj}, \nu_{ij} \right) \\
& + L_\sigma\left( \mu_i; \nu_{ii}, \tau_{jj}, \kappa_{ij} \right) + L_\sigma\left( \mu_j; \nu_{jj}, \tau_{ii}, \kappa_{ji} \right) \\
& + \tau_{ij}.
\end{aligned}
$$

*where for all valid indices $(i, j)$,*

$$\mu_i = (A\mu + b)_i \qquad\qquad \tau_{ij} = (C\Sigma C^\mathsf{T})_{i,j}$$
$$\nu_{ij} = (A\Sigma A^\mathsf{T})_{i,j} \qquad\qquad \kappa_{ij} = (A\Sigma C^\mathsf{T})_{i,j}$$

With a five-dimensional domain, these functions are too complex to be represented by a look-up table. We compute them analytically for:

**probit** in App. D by expressing $\Phi(x) = \mathbb{E}\left[Z \leq x\right]$ in terms of an auxiliary standard Normal $Z$, generalizing (due to the inclusion of an affine term) and superseding the results of Huber (2020, § III.B) and Wright et al. (2024, App. B.4).

**GeLU** in App. E by repeated applications of the multivariate Stein's lemma and the Gaussian ODE $\phi'(x) + x\phi(x) = 0$, generalizing and superseding the result of Wright et al. (2024, App. B.2).

**ReLU** in App. F by using the Dominated Convergence Theorem to take the pointwise limit

$$\mathrm{ReLU}(x) = \lim_{\lambda \to \infty} \lambda^{-1} \mathrm{GeLU}(\lambda x),$$

generalizing and superseding the results of Frey & Hinton (1999, App. C.3), Bibi et al. (2018, §3), Wu et al. (2019, App A.2.2), Huber (2020, § III.C), and Wright et al. (2024, App. B.2)

**Heaviside** in App. G by using the Dominated Convergence Theorem to take the pointwise limit

$$\mathrm{Heaviside}(x) = \lim_{\lambda \to \infty} \Phi(\lambda x),$$

generalizing and superseding the results of Frey & Hinton (1999, App. C.2), Wu et al. (2019, App. A.2.1), and Wright et al. (2024, App. B.2).

**sine** in App. H by combining the characteristic function of the Normal distribution with the trigonometric identity $\sin(x) = \left(e^{ix} - e^{-ix}\right)/(2i)$, generalizing and superseding the results in Sitzmann et al. (2020, App. 1).

The calculations are interesting in themselves because they are probabilistic in nature: we never resort to Riemann integrals against the Gaussian density, instead working with higher-level properties of Gaussian variables derived in App. C.

The baseline methods $Y_{\mathrm{mfa}}$, mean-field; $Y_{\mathrm{lin}}$, linear; $Y_{\mathrm{u'95}}$, unscented'95; and $Y_{\mathrm{u'02}}$, unscented'02 are presented in Appendix B.

## 2.2 Ground truth(s) $Y_0$ and $Y_1$

The true distribution is

$$Y_0 = f(X),$$

and the pseudo-true distribution is

$$Y_1 = \mathcal{N}\left(\mathbb{E}\,Y_0, \mathrm{Cov}\,Y_0\right).$$

Whereas $Y_0$ is the ideal answer to Problem 1, $Y_1$ is the closest Gaussian approximation (by KL divergence) to $Y_0$. We obtain $Y_0$ and $Y_1$ in baselines by quasi-Monte Carlo simulation.

In §5.1, we evaluate each method by computing the KL divergence of its Normal approximation from $Y_1$. This measures how close the method's Normal approximation is to the best Normal approximation. We also compare each Normal approximation to the true (non-Normal) distribution $Y_0$ using Wasserstein distance.

## 3 Theoretical Guarantees

In Appendix I, we give a theoretical bound on the dissimilarity between the laws of $Y_0$ and $Y$, as measured in Wasserstein distance:

**Definition 4.** *Let $X$ and $Y$ be random variables taking values $\mathbb{R}^n$. The Wasserstein distance $d_W(X,Y)$ is*

$$d_{\mathrm{W}}(X,Y) = \sup_{\|\nabla h\|_\infty \leq 1} \mathbb{E}(h(X) - h(Y)).$$

We prove that $d_{\mathrm{W}}(Y_0, Y)$ can be decomposed as the final step of a recursion

error at layer $k \leq$ (Lipschitz constant of layer $k$) (error at layer $k-1$)

$$+ \text{(non-normality of layer } k)$$

where all of the terms can be bounded in terms of the input distribution and the network weights. Even though this bound is too loose to be depended upon in practice, it lends attribution to the sources of error in our approach; in particular, that non-normality arises as a multilayer interaction between variance and nonlinearity.

## 4 ADVERSARIAL EXAMPLES

The Introduction claims that linear and unscented propagation are only exact for low-order (1 and 2, respectively) polynomials. We support this claim by applying them to general smooth functions.

**Example 1** (for $Y_{\mathrm{lin}}$). *Consider the network $Y = \sin(X)$, where $X \sim \mathcal{N}(0, \sigma^2)$. Then $Y_{\mathrm{lin}} = \mathcal{N}(0, \sigma^2)$. But in fact $\mathrm{Var}\, Y = (1 - e^{-\sigma^2})/2$ which tends to $1/2$ for large $\sigma^2$. So by increasing $\sigma^2$, we can make $Y_{\mathrm{lin}}$ arbitrarily wrong.*

**Example 2** (for $Y_{\mathrm{u95}}$ and $Y_{\mathrm{u02}}$). *Suppose that $X \sim \mathcal{N}(0,1)$, and the sigma points are $X \in \{-\alpha, 0, \alpha\}$ for some $\alpha > 0$. Then on the neural network $Y = \sin(\alpha^{-1}\pi X)$, $Y_{\mathrm{u95}}$ and $Y_{\mathrm{u02}}$ will be identically zero and arbitrarily wrong.*

The Introduction claims that mean-field propagation is only exact for networks consisting of a single neuron. We support this claim by applying it to a network with multiple neurons.

**Example 3** (for $Y_{\mathrm{mfa}}$). *Consider the following (linear) network, with scalar input $X$, hidden $Y^1 \in \mathbb{R}^m$, and scalar output $Y$.*

$$Y = \frac{1}{m} \sum_{i=1}^{m} Y_i^1$$

$$Y_i^1 = X$$

$$X \sim \mathcal{N}(0,1)$$

*The mean-field approximation treats each $Y_i^1$ as independent $\mathcal{N}(0,1)$, so it concludes $Y \sim \mathcal{N}(0, m^{-1})$. But $Y$ is identical to $X \sim \mathcal{N}(0,1)$. So by increasing $m$, we can make $Y_{\mathrm{mfa}}$ arbitrarily wrong.*

However the mean and variance of our method $Y_{\mathrm{ana}}$ are exact on all single-hidden-layer networks, which includes all of the examples above. Unlike the works cited in Appendix A, we push our method past the breaking point, by an explicit multi-layer network combining strong non-normality with strong nonlinearity.

**Example 4** (for $Y_{\mathrm{ana}}$). *Consider the following network, where input $X$, output $Y$, and hidden $Y^1$ are scalars; $u(x) = \mathbf{1}_{x \geq 0}$ is the Heaviside function; and $\alpha$ is a weight.*

$$Y = \alpha u(Y^1 - 3)$$

$$Y^1 = 2u(X)$$

$$X \sim \mathcal{N}(0,1)$$

*At the hidden layer, $Y^1$ is approximated by $\mathcal{N}(1,1)$. Therefore $Y_{\mathrm{ana}}$ is approximated by some nondegenerate Normal distribution, scaled by $\alpha$. By increasing $\alpha$, we can make $Y_{\mathrm{ana}}$ arbitrarily wrong: $Y_0$ is identically zero because $Y^1 - 3$ is always negative.*

| distribution | $d_{\mathrm{W}}(\cdot, Y_0)$ | $D_{\mathrm{KL}}(\cdot \| Y_1)$ |
|---|---|---|
| pseudo-true ($Y_1$) | $9.531 \times 10^{-2} \pm 2.9 \times 10^{-5}$ | $0$ |
| analytic | $1.111 \times 10^{-1} \pm 3.5 \times 10^{-5}$ | $2.086 \times 10^{-3} \pm 1.2 \times 10^{-6}$ |
| mean-field | $3.101 \times 10^{-1} \pm 3.7 \times 10^{-5}$ | $5.797 \times 10^{-2} \pm 7.0 \times 10^{-6}$ |
| linear | $1.929 \times 10^{+1} \pm 2.5 \times 10^{-4}$ | $1.627 \times 10^{+2} \pm 7.7 \times 10^{-3}$ |
| unscented'95 | $4.135 \times 10^{-1} \pm 3.2 \times 10^{-5}$ | $1.353 \times 10^{-1} \pm 1.3 \times 10^{-5}$ |
| unscented'02 | $8.884 \times 10^{+2} \pm 1.0 \times 10^{-2}$ | $3.067 \times 10^{+5} \pm 1.4 \times 10^{+1}$ |

Table 1: Summary statistics for `Network(architecture=small, weights=trained, activation=probit residual), variance=large`. $d_{\mathrm{W}}(\cdot, Y_0)$ is the Wasserstein distance to $Y_0$, the ground truth (lower is better), and $d_{\mathrm{KL}}(\cdot \| Y_1)$ is the KL divergence from $Y_1$, the pseudo-true Normal distribution having ground truth moments (lower is better).

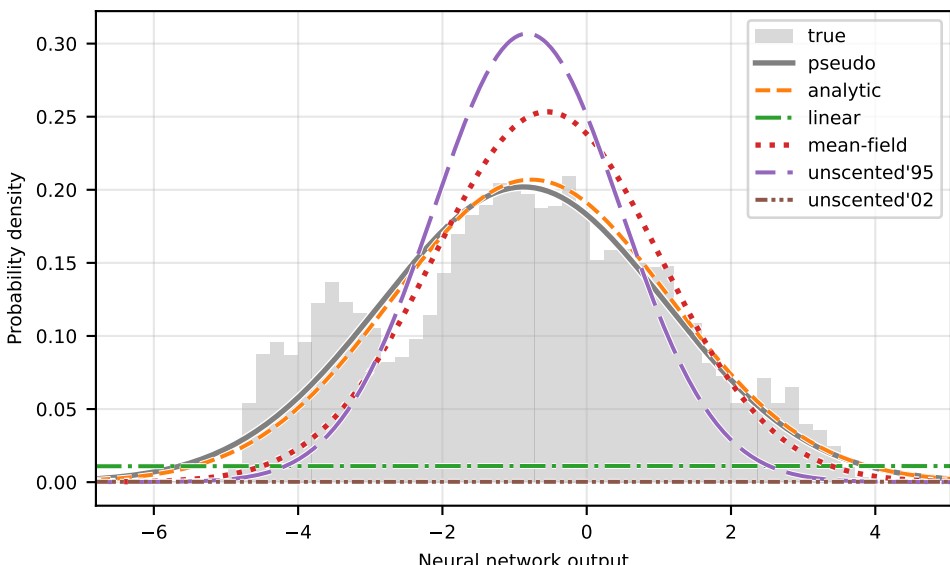

Figure 1: Probability distributions for `Network(architecture=small, weights=trained, activation=probit residual), variance=large`.

## 5 EXAMPLES, APPLICATIONS, AND EXTENSIONS

### 5.1 RANDOM NETWORKS

We apply our method and other benchmarks to 38 different ensembles of random neural networks with more than one hidden layer. They are designed to cover a range of architectures, weights, and activations, and to stress-test the assumptions of layer-by-layer moment matching. We sample one neural network from each ensemble and evaluate the goodness of approximation of the output distribution for three input distributions. In each case we evaluate $Y_0, Y_1, Y_{\mathrm{ana}}, Y_{\mathrm{mfa}}, Y_{\mathrm{lin}}, Y_{\mathrm{u95}}$, and $Y_{\mathrm{u02}}$, and compare distance to the true distribution $d_{\mathrm{W}}(\cdot, Y_0)$ and KL divergence from the pseudo-true distribution $d_{\mathrm{KL}}(\cdot, Y_1)$.

See Appendix N for full specifications of the 114 test cases, the Monte Carlo methodology, and the full results. For the case of large input variance, across all networks, our method is typically **one hundred** times better than Unscented'95, **ten thousand** times better than linear, and **one million** times better than Unscented'02 (measured by KL divergence from the pseudo-true distribution), as seen in Fig. 2. Visualizations for the medium- and small- input variance cases are available in Appendix N.1.

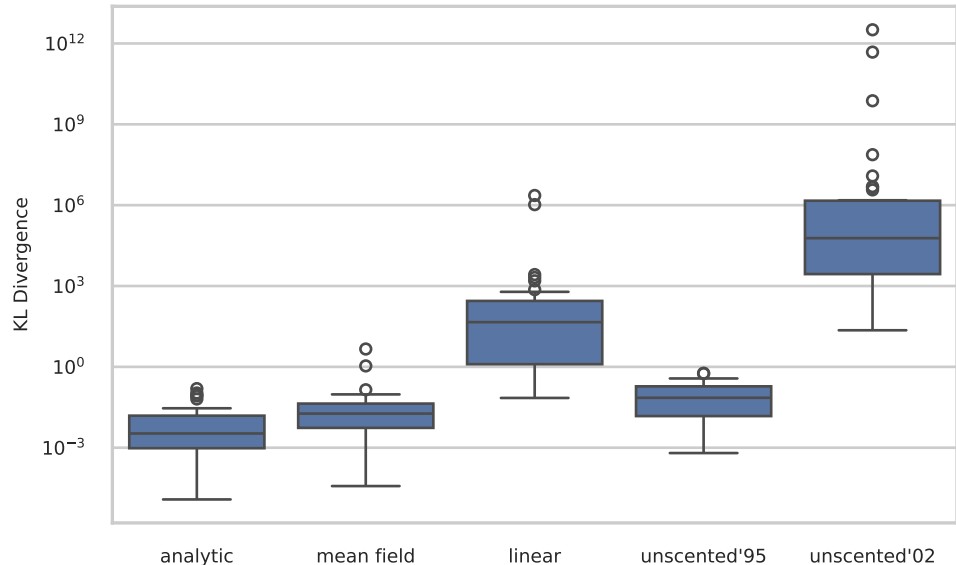

Figure 2: Comparison of goodness of approximation (lower KL divergence is better) for all random neural networks, grouped by approximation method, in the large input variance scenario.

One example, which we reproduce here in Table 1 and Fig. 1, is the large-variance case of a small network with trained weights and a probit-residual activation function, in which the output distribution is evidently non-normal, but our method is still closest to the true distribution (in Wasserstein distance) and to the pseudo-true distribution (in KL divergence).

Many examples; such as the small-variance test case with wide architecture, trained weights, and sine activation; exhibit the underdispersion of the mean-field approximation predicted in Example 3. It is remarkable that even in the small-variance regime when our method, linearization, and the unscented transformations are all justified by the delta method and asymptotically equivalent, our method is often still closest to the pseudo-true distribution.

## 5.2 INPUT UNCERTAINTY: REGRESSION

A generative neural network trained for regression on noiseless inputs is used to make predictions on noisy inputs. At inference, the network is provided with a perturbed input and the covariance matrix of the perturbation. The prediction is a distribution over the output.

We apply this procedure to the California Housing dataset (Kelley Pace & Barry, 1997). We report the average log probability density of the test $y$ under the predictive distribution $\hat{Y}$ in Table. 2, as well as coverage and interval width for nominal 95% confidence intervals. The analytic moment propagation method has the highest log probability density and the closest-to-nominal coverage. All uncertainty propagation methods outperform "certain," which ignores the uncertainty in the input.

The model specification, training, inference, and Monte Carlo method are detailed in Appendix. J.

## 5.3 INPUT UNCERTAINTY: BINARY CLASSIFICATION

A neural network trained for binary classification (Bernoulli regression) is used to make predictions on a shifted distribution of missing features. The missing features are filled by linear regression, leading to input uncertainty that needs to be propagated to the output probability $\hat{p}$. Uncertainty propagation is needed because the calibration of $\hat{p}$ matters in applications such as betting on the outcome of a binary event (Cover & Thomas, 2006, Example 6.1.1).

We apply this to the Taiwanese bankruptcy prediction dataset (Liang et al., 2016).

| Method | log pdf | coverage (%) | interval width |
|---|---|---|---|
| certain | $-4.452 \pm 4.5 \times 10^{-2}$ | $49.7 \pm 2.0 \times 10^{-1}$ | 1.67 |
| analytic | $-1.420 \pm 7.6 \times 10^{-3}$ | $96.0 \pm 1.5 \times 10^{-1}$ | $4.06 \pm 3.5 \times 10^{-3}$ |
| mean field | $-1.647 \pm 2.7 \times 10^{-3}$ | $99.8 \pm 4.0 \times 10^{-2}$ | $6.87 \pm 4.4 \times 10^{-4}$ |
| linear | $-1.851 \pm 3.2 \times 10^{-3}$ | $97.3 \pm 5.4 \times 10^{-2}$ | $7.73 \pm 5.5 \times 10^{-3}$ |
| unscented'95 | $-1.457 \pm 8.8 \times 10^{-3}$ | $93.6 \pm 1.8 \times 10^{-1}$ | $3.89 \pm 4.3 \times 10^{-3}$ |
| unscented'02 | $-2.529 \pm 1.2 \times 10^{-3}$ | $99.7 \pm 1.2 \times 10^{-2}$ | $18.85 \pm 1.5 \times 10^{-2}$ |

Table 2: Performance of different uncertainty propagation methods on the California housing regression problem. Both coverage (closer to 95% is better) and interval width (smaller is better) are reported for nominal 95% confidence intervals.

| Method for predicting $\hat{p}$ | log probability of correct label |
|---|---|
| certain | $-0.202 \pm 2.9 \times 10^{-2}$ |
| analytic | $-0.145 \pm 1.8 \times 10^{-2}$ |
| mean field | $-0.163 \pm 2.2 \times 10^{-2}$ |
| linear | $-0.202 \pm 2.9 \times 10^{-2}$ |
| unscented'95 | $-0.159 \pm 2.1 \times 10^{-2}$ |
| unscented'02 | $-0.144 \pm 1.8 \times 10^{-2}$ |

Table 3: Average (with standard error) log predicted probability (higher is better) of the correct label on test data in the Taiwanese bankruptcy dataset.

The missing data dramatically degrades the discrimination of $\hat{p}$ across all uncertainty propagation methods, decreasing the area under the receiver operating characteristic by roughly 0.2 (App. K, Fig. 5). However, the analytic uncertainty-aware prediction has the best-calibrated probabilities up to sampling and numerical uncertainty in our results (Table. 3). All uncertainty propagation methods outperform "certain" prediction, which ignores the uncertainty in the imputation step.

We give the details of the model specification, training, and inference in Appendix. K.

### 5.4 WEIGHT UNCERTAINTY: VARIATIONAL BAYES NETWORKS

We apply our distributional approximation to variational inference for Bayesian networks. In this case, the input $x$ is fixed and the weights $w$ are random with a prior distribution $p(w)$. The network predicts $y \sim p(y \mid w, x)$. The posterior distribution is approximated variationally as $q(w; \theta)$ by optimizing the evidence lower bound (Blundell et al., 2015):

$$\theta^* = \arg \min_{\theta} \left\{ \mathbb{E}_{w \sim q(w;\theta)} \log p(y \mid w, x) + D_{\mathrm{KL},w}(q(w;\theta) \mid p(w)) \right\}. \tag{2}$$

Our example uses the GeLU activation function and, like Wright et al. (2024, §4.3), a single hidden layer; for other details, see App. L.

Monte Carlo Variational Inference (MCVI) is a stochastic gradient method that approximates $\mathbb{E}_{w \sim q(w;\theta)}$ with Monte Carlo samples. In order to reduce the gradient variance and computational cost, Wu et al. (2019); Petersen et al. (2024); Wright et al. (2024); Rui Li & Trapp (2025) use moment-matching deterministic approximations to the evidence lower bound, and then apply the same deterministic approximation to evaluate the predictive distribution. While this benchmarking strategy assesses the end-to-end learning process, it does not necessarily reflect accurate distributional approximations: intuitively, we can expect a sufficiently expressive variational model to be *robust* to systematic errors in the distribution propagation. On one hand, this robustness favors end-to-end inference (training and testing with the same distributional approximation), but on the other hand, it means that end-to-end inference fails to interrogate whether the approximation itself is accurate.[3]

---

[3]"Learning can still improve a bound on the log likelihood of the data even when the posterior distribution over hidden states is computed incorrectly"; see also the surrounding discussion in Frey & Hinton (1999, §1.2).

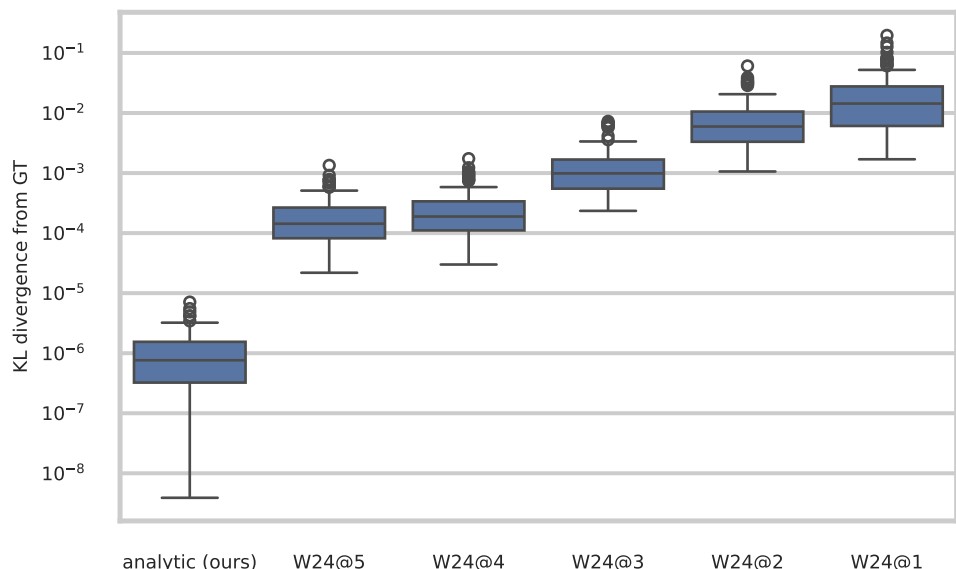

Figure 3: KL divergence (lower is better) between pseudo-true (ground truth moments) predictive distribution (by Monte Carlo) and approximations for the concrete compressive strength dataset. W24@$k$ means the $k$th partial sum of the GeLU covariance series of Wright et al. (2024, Appendix B.3).

Because our interest is in the correctness of distributional approximation, our experiments differ from the works cited in the following ways:

1. We train the Bayesian network using MCVI with a large Monte Carlo batch size. After training has converged, the Monte Carlo predictions from this network are taken as ground truth mean and variance.

2. We test the Bayesian network using Monte Carlo on the variational posterior, as well as six different techniques to propagate distributions through activation functions: the power series of Wright et al. (2024), expanded to 1–5 terms, and our method.

3. The figure of merit is the KL divergence (lower is better) between the pseudo-true Gaussian distribution (via Monte Carlo) and each deterministic approximation.

We applied this method to four regression datasets from the UCI Machine Learning Repository. The data references and full results are in App. L; we highlight only the concrete compressive strength dataset here. As Fig. 3 shows, the power series expansion of Wright et al. (2024) shows a "dose response" i.e. becomes more accurate as more terms are added. But the most accurate approximation to the Monte Carlo predictive distribution is attained using exact moment matching, our method. It is not exact because, for a Bayes network with even a single hidden layer, we use a moment-matching Gaussian multiplication approximation (Goulet et al., 2021, eqq. 3–6).

## 5.5 Stochastic activations

As publicly acclaimed last year, the earliest conceptions of an artificial neural networks were, like biological neural networks, stochastic (Davour). Later, a neuron's stochastic activation was replaced by a deterministic sigmoid that represented its average behavior. Today, the biological similitude has ceased to be a driving motivation, yet as a curiosity, we analyze the output distribution of a neural

---

For a toy example, take $Z \sim \mathcal{N}(0, 1)$; if the uncertainty propagation formula were $aZ \sim \mathcal{N}(0, 2a)$, which is patently incorrect, the variational model could learn to approximate the target distribution $\mathcal{N}(0, \sigma^2)$ using the parameter $a = \sigma^2/2$. If the approximations were adequate, one would expect the different inference methods to be indistinguishable from each other and from MCVI, as they are in Wright et al. (2024, Table 3).

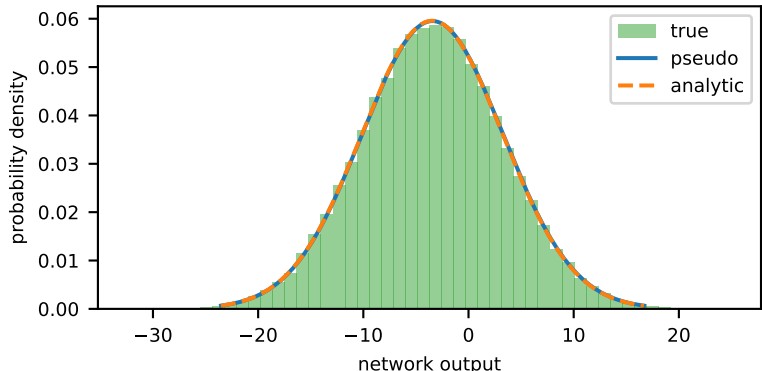

Figure 4: Output distribution of a stochastic neural network, pseudo-true Normal distribution ("pseudo"), and layer-by-layer moment-matched Normal distribution ("analytic").

network whose activations are random processes modeled after a stochastic neuron:

$$\tilde{\sigma}(x, U) = 2\mathbf{1}_{U<\Phi(x)} - 1, \tag{3}$$

where at each artificial neuron, $U \sim \text{Uniform}(0, 1)$ is an independent random variable.

In Appendix. M we derive a moment-matching approximation to the distribution of the output of a stochastic neural network. Applying this formula to 1 million samples from a stochastic version of the "deep" neural network of §5.1 with a constant zero input results in a normal distribution with a good subjective agreement (Fig. 4). Whether this line of inquiry deserves further methodological development we reserve for future work.

## 6 NOVELTY AND SIGNIFICANCE

Until now, the first and second moments of a neural network layer have been approximated in various ways (independence, approximate activation function, truncated power series, etc.). We compute them exactly for many popular activation functions. Our networks also generalize from previous work in uncertainty propagation by allowing for residual connections, which are common in modern neural networks.

This discovery enables layer-wise moment matching in deep networks, which we demonstrate to be orders of magnitude more accurate than other uncertainty propagation methods.

We finally demonstrate that this uncertainty propagation method is effective for uncertainty-aware inference on real data, as well as for deterministic inference on variational Bayes networks.

# 7 REPRODUCIBILITY STATEMENT

In the Supplementary Material, executable Python scripts are in `demo/`. They reference libraries in `src/` and generate output in `docs/`. `test_case.py` and `analyze_results.py` generate the results in §N. To reproduce §J, run `california_housing.py`. To reproduce §K, run `classification.py`. To reproduce §M, run `stochastic.py`. Full reproduction code for the discussion version will be posted later in the ICLR review cycle.

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

## SUPPLEMENTARY MATERIAL

Owing to the sheer length of the supplementary material, which includes a programmatically generated section of exhaustive random neural network test cases (§N), we begin with a table of contents to re-orient the reader.

## CONTENTS

## A SUPPLEMENT TO LITERATURE REVIEW

| Assumption | References |
|---|---|
| $\Sigma$ is small (linearized) | Titensky et al. (2018); Nagel & Huber (2022) |
| | Petersen et al. (2024); Jungmann et al. (2025) |
| | Bergna et al. (2025); Rui Li & Trapp (2025) |
| $\Sigma$ is small (unscented) | Astudillo & Neto (2011); Abdelaziz et al. (2015) |
| $\Sigma$ is diagonal | Abdelaziz et al. (2015); Huber (2020) |
| | Goulet et al. (2021); Rui Li & Trapp (2025) |
| | Wagner et al. (2022); Akgül et al. (2025) |
| $\mu = 0$ | Bibi et al. (2018) |
| $\mu \to \infty$ | Wu et al. (2019) |
| no assumptions | Wright et al. (2024) |
| | this paper |

Table 4: Comparison of assumptions imposed on Gaussian approximations of neural network layers with input $\mathcal{N}(\mu, \Sigma)$.

| Activation function | References |
|---|---|
| piecewise linear | Bibi et al. (2018); Huber (2020) |
| | Wright et al. (2024); Akgül et al. (2025) |
| | Wu et al. (2019) |
| logistic ($\approx$ piecewise exponential) | Astudillo & Neto (2011); Abdelaziz et al. (2015) |
| logistic ($\approx \Phi$) | Huber (2020) |
| Heaviside | Wu et al. (2019); Wright et al. (2024) |
| GeLU | Wright et al. (2024) |
| (sin, $\Phi$, GeLU, ReLU, Heaviside) + affine | this paper (**exact**) |

Table 5: Activation functions for which moment propagation has been approximated. Note two distinct approaches to approximating the logistic function.

## B BASELINES

### B.1 THE MEAN-FIELD APPROXIMATION $Y_{\text{mfa}}$

Following Huber (2020); Wagner et al. (2022); Akgül et al. (2025), we define the mean-field analytic approximation by assuming that neurons in the same hidden layer are independent:

$$
\begin{aligned}
Y_{\text{mfa}} &= Y^\ell \\
Y^k &= \mathcal{N}(\mu^k, \Sigma^k) && \forall k \in \{1 \ldots \ell\} \\
\mu^k &= \mathbb{E}\, g(Y^{k-1}; A^k, b^k, C^k, d^k) \\
\Sigma^k_{ij} &= \begin{cases} \left[\text{Cov}\, g(Y^{k-1}; A^k, b^k, C^k, d^k)\right]_{ij}, & i = j \\ 0, & \text{else} \end{cases} && \text{(mean-field assumption)} \\
Y^0 &= X
\end{aligned}
$$

### B.2 THE LINEAR APPROXIMATION $Y_{\text{lin}}$

Following Titensky et al. (2018); Nagel & Huber (2022); Petersen et al. (2024); Jungmann et al. (2025), we define the linear approximation as the asymptotically Normal output distribution (pursuant to the delta method) in the limit of small input variance:

$$
Y_{\text{lin}} = \mathcal{N}(f(\mathbb{E}\, X), \nabla f(\mathbb{E}\, X)\, \text{Cov}\, X \nabla f(\mathbb{E}\, X)^\intercal)
$$

## B.3 The unscented approximation(s) $Y_{\mathrm{u95}}$ and $Y_{\mathrm{u02}}$

An unscented transformation is a quadrature rule for approximating a probability measure on $\mathbb{R}^n$ by $2n + 1$ point masses whose locations (called sigma points) and weights satisfy certain first- and second-order moment matching conditions. This technique was developed to improve upon linearization in nonlinear Kalman filtering (Julier et al., 1995; Julier & Uhlmann, 1997; Julier et al., 2000; Julier, 2002; Wan & Van Der Merwe, 2000; Julier & Uhlmann, 2004).

There are (at least) two unscented transformations: a one-parameter family, which we refer to as Unscented'95; and a three-parameter family, which we refer to as Unscented'02.

- Unscented'95 (Julier et al., 1995; Julier & Uhlmann, 1997; Julier et al., 2000), used in Astudillo & Neto (2011); Abdelaziz et al. (2015), is a one-parameter family of unscented transformations, parameterized by a single shape hyperparameter $\kappa$.
- Unscented'02 (Julier, 2002; Wan & Van Der Merwe, 2000) is a three-parameter family of unscented transformations, parameterized by three shape hyperparameters $(\alpha, \beta, \kappa)$. This version, with default values tuned to be drastically more localized in $X$-space than Unscented'95, is usually called "the" unscented transformation in current filtering research (Jiang et al., 2025) and tooling (Ljung, 2025).

## C Preliminaries: Gaussian integrals

**Definition 5.** *The bivariate normal CDF $\Phi_2$ is defined by*

$$\Phi_2(h, k; \rho) = \mathbb{P}\left[Z_1 \le h, Z_2 \le k\right],$$

*where*

$$\begin{pmatrix} Z_1 \\ Z_2 \end{pmatrix} \sim \mathcal{N}\left( \begin{pmatrix} 0 \\ 0 \end{pmatrix}, \begin{pmatrix} 1 & \rho \\ \rho & 1 \end{pmatrix} \right).$$

**Remark 1.** *While we consider it a valid "closed form" atom, the bivariate normal CDF can be a difficult transcendental function to evaluate. Many software packages such as SciPy (Wagner et al., 2022) implement the multivariate normal CDF by a (quasi-) Monte Carlo integration over $\mathbb{R}^n$, which is too expensive for our purposes. Furthermore, we frequently use the expression $\Phi(h, k; \rho) - \Phi(h, k; 0)$, which is vulnerable to cancellation error for extreme values of $h$, $k$, and $\rho$. To avoid this, we use 10-point Gaussian quadrature of the one-dimensional proper integral*

$$\Phi_2(h, k; \rho) - \Phi_2(h, k; 0) = \int_0^\rho \partial'_\rho \Phi_2(h, k; \rho') \, \mathrm{d}\rho',$$

*using*

$$\partial_\rho \Phi_2(h, k; \rho) = \frac{1}{2\pi\sqrt{1 - \rho^2}} e^{-\frac{1}{2(1-\rho^2)}\left(h^2 + k^2 - 2\rho h k\right)},$$

*a helpful identity found in Drezner & Wesolowsky (1990).*

**Definition 6.** *The bivariate Normal density is*

$$\phi_2(h, k; \rho) = \frac{\partial^2}{\partial h \partial k} \, \Phi_2(h, k; \rho).$$

**Definition 7.** *The partial derivative of the joint CDF is*

$$\Phi_{2;1}(h, k; \rho) = \frac{\partial}{\partial h} \, \Phi_2(h, k; \rho) \tag{4}$$

**Lemma 2.** *The function $\Phi_{2;1}$ satisfies*

$$\Phi_{2;1}(h, k; \rho) = \phi(h)\Phi\left( \frac{k - \rho h}{\sqrt{1 - \rho^2}} \right).$$

*Proof.* Letting $Z_1, Z_2$ be standard normal with correlation $\rho$, there is a probabilistic interpretation of $\Phi_{2;1}$:

$$\Phi_{2;1}(h, k; \rho) = \underbrace{f_{Z_1}(h)}_{\text{marginal density}} \underbrace{\mathbb{P}\left(Z_2 \leq k \mid Z_1 = h\right)}_{\text{conditional cdf}}$$

$$= \phi(h)\Phi\left(\frac{k - \rho h}{\sqrt{1 - \rho^2}}\right).$$

$\square$

**Lemma 3** (Multivariate Stein's lemma). *Let $(X_1, \ldots, X_n)$ be a multivariate normal random vector. Then*

$$\mathbb{E}(X_1 - \mathbb{E} X_1)f(X_1, \ldots, X_n) = \sum_{i=1}^{n} \text{Cov}(X_1, X_i)\, \mathbb{E}\left[\partial_{X_i} f(X_1, \ldots, X_n)\right]$$

**Lemma 4** (Stein simplification of $L_\sigma$). *Let $L_\sigma$ be defined as in Definition 3, and suppose that $\sigma$ is differentiable. Then*

$$L_\sigma(\mu_1; \nu_{11}, \nu_{22}, \nu_{12}) = \nu_{12}\, \mathbb{E}\, \sigma'(Z_1), \qquad Z_1 \sim \mathcal{N}(\mu_1, \nu_{11}).$$

*Proof.* Straightforward application of Lemma 3. $\square$

**Lemma 5** (Univariate integrals). *If $Z \sim \mathcal{N}(\mu, \nu)$, then*

$$\mathbb{E}\, \Phi(Z) = \Phi\left(\frac{\mu}{\sqrt{1 + \nu}}\right) \tag{5a}$$

$$\mathbb{E}\, \phi(Z) = \frac{1}{\sqrt{1 + \nu}}\phi\left(\frac{\mu}{\sqrt{1 + \nu}}\right) \tag{5b}$$

$$\mathbb{E}\, Z\Phi(Z) = \frac{1}{(1 + \nu)^{3/2}}\phi\left(\frac{\mu}{\sqrt{1 + \nu}}\right) \tag{5c}$$

*Proof of* (5a). Introducing an independent $Z \sim \mathcal{N}(0, 1)$, we have

$$\mathbb{E}\, \Phi(Z) = \mathbb{E}\, \mathbb{P}\left[Z \leq X \mid X\right] \tag{6}$$
$$= \mathbb{P}\left[Z \leq X\right] \qquad \text{(by the law of total probability)}$$
$$= \mathbb{P}\left[Z - X \leq 0\right] \tag{7}$$

We conclude by noting that the random variable $Z - X$ has a Normal distribution with mean $-\mu$ and variance $1 + \nu$. $\square$

*Proof of* (5b). We use $\phi = \Phi'$.

$$\mathbb{E}\, \phi(Z) = \mathbb{E}\, \frac{\mathrm{d}}{\mathrm{d}t}\bigg|_{t=0} \Phi(Z + t) \tag{8}$$

$$= \frac{\mathrm{d}}{\mathrm{d}t}\bigg|_{t=0} \mathbb{E}\, \Phi(Z + t) \qquad \text{(dominated convergence theorem)}$$

$$= \frac{\mathrm{d}}{\mathrm{d}t}\bigg|_{t=0} \Phi\left(\frac{\mu + t}{\sqrt{1 + \nu}}\right) \qquad \text{(by (5a))}$$

$\square$

*Proof of* (5c). We use the Gaussian ODE

$$\phi'(x) + x\phi(x) = 0. \tag{9}$$

Centering and applying Lemma 3,

$$\mathbb{E}\, Z\phi(Z) = \mathbb{E}(Z - \mu)\phi(Z) + \mu\, \mathbb{E}\, \phi(Z) \qquad\qquad \text{(centering)}$$
$$= \nu\, \mathbb{E}\, \phi'(Z) + \mu\, \mathbb{E}\, \phi(Z) \qquad\qquad \text{(Stein's)}$$
$$= -\nu\, \mathbb{E}\, Z\Phi(Z) + \mu\, \mathbb{E}\, \phi(Z) \qquad\qquad \text{(Gaussian ODE)}$$

Collecting like terms and solving for (I),

$$(1 + \nu)\, \mathbb{E}\, Z\Phi(Z) = \mu\, \mathbb{E}\, \phi(Z) \tag{10}$$

$$\mathbb{E}\, Z\Phi(Z) = \frac{\mu}{1 + \nu}\, \mathbb{E}\, \phi(Z) \tag{11}$$

$$= \frac{\mu_1}{(1 + \nu_{11})^{3/2}} \phi\left(\frac{\mu_1}{\sqrt{1 + \nu_{11}}}\right) \qquad \text{(by (5b))}$$

$\square$

**Lemma 6** (Bivariate integrals). *Let*

$$\begin{pmatrix} X_1 \\ X_2 \end{pmatrix} \sim \mathcal{N}\left(\begin{pmatrix} \mu_1 \\ \mu_2 \end{pmatrix}, \begin{pmatrix} \nu_{11} & \nu_{12} \\ \nu_{12} & \nu_{22} \end{pmatrix}\right).$$

*Then*

$$\mathbb{E}\, \Phi(X_1)\Phi(X_2) = \Phi_2\left(\frac{\mu_1}{\sqrt{1 + \nu_{11}}}, \frac{\mu_2}{\sqrt{1 + \nu_{22}}}; \frac{\nu_{12}}{\sqrt{(1 + \nu_{11})(1 + \nu_{22})}}\right) \tag{12a}$$

$$\mathbb{E}\, \phi(X_1)\Phi(X_2) = \frac{1}{\sqrt{1 + \nu_{11}}} \Phi_{2;1}\left(\frac{\mu_1}{\sqrt{1 + \nu_{11}}}, \frac{\mu_2}{\sqrt{1 + \nu_{22}}}; \frac{\nu_{12}}{\sqrt{(1 + \nu_{11})(1 + \nu_{22})}}\right) \tag{12b}$$

$$\mathbb{E}\, \phi(X_1)\phi(X_2) = \frac{1}{\sqrt{(1 + \nu_{11})(1 + \nu_{22})}} \phi_2\left(\frac{\mu_1}{\sqrt{1 + \nu_{11}}}, \frac{\mu_2}{\sqrt{1 + \nu_{22}}}; \frac{\nu_{12}}{\sqrt{(1 + \nu_{11})(1 + \nu_{22})}}\right) \tag{12c}$$

$$\mathbb{E}\, \phi'(X_1)\Phi(X_2) = \frac{-\mu_1}{1 + \nu_{11}}\, \mathbb{E}\, \phi(X_1)\Phi(X_2) + \frac{-\nu_{12}}{1 + \nu_{11}}\, \mathbb{E}\, \phi(X_1)\phi(X_2) \tag{12d}$$

*Proof of* (12a). Introduce independent $Z_1, Z_2 \sim \mathcal{N}(0, 1)$. Then

$$\mathbb{E}\, \Phi(X_1)\Phi(X_2) = \mathbb{E}\, \mathbb{P}\left[Z_1 \leq X_1 \mid X_1\right] \mathbb{P}\left[Z_2 \leq X_2 \mid X_2\right] \tag{13}$$
$$= \mathbb{E}\, \mathbb{P}\left[Z_1 \leq X_1, Z_2 \leq X_2 \mid X_1, X_2\right] \qquad \text{(independence)}$$
$$= \mathbb{P}\left[Z_1 \leq X_1, Z_2 \leq X_2\right] \qquad \text{(by the law of total probability)}$$
$$= \mathbb{P}\left[Z_1 - X_1 \leq 0, Z_2 - X_2 \leq 0\right]. \tag{14}$$

We conclude by using the fact that $(Z_1 - X_1, Z_2 - X_2)$ is jointly Normal with distribution

$$\begin{pmatrix} Z_1 - X_1 \\ Z_2 - X_2 \end{pmatrix} \sim \mathcal{N}\left(\begin{pmatrix} -\mu_1 \\ -\mu_2 \end{pmatrix}, \begin{pmatrix} 1 + \nu_{11} & \nu_{12} \\ \nu_{12} & 1 + \nu_{22} \end{pmatrix}\right). \tag{15}$$

$\square$

*Proof of* (12b). Using $\phi = \Phi'$, we have

$$\mathbb{E}\,\phi(X_1)\Phi(X_2)$$

$$= \mathbb{E}\,\frac{\mathrm{d}}{\mathrm{d}t}\bigg|_{t=0}\Phi(X_1+t)\Phi(X_2) \tag{16}$$

$$= \frac{\mathrm{d}}{\mathrm{d}t}\bigg|_{t=0}\mathbb{E}\,\Phi(X_1+t)\Phi(X_2) \qquad\text{(dominated convergence theorem)}$$

$$= \frac{\mathrm{d}}{\mathrm{d}t}\bigg|_{t=0}\mathbb{P}\left(Z_1-X_1-t\le 0, Z_2-X_2\le 0\right) \qquad\text{(introducing } Z_1, Z_2\sim\mathcal{N}(0,1))$$

$$= \frac{\mathrm{d}}{\mathrm{d}t}\bigg|_{t=0}\Phi_2\left(\frac{\mu_1+t}{\sqrt{1+\nu_{11}}}, \frac{\mu_2}{\sqrt{1+\nu_{22}}}; \frac{\nu_{12}}{\sqrt{(1+\nu_{11})(1+\nu_{22})}}\right) \tag{17}$$

$$= \frac{1}{\sqrt{1+\nu_{11}}}\Phi_{2;1}\left(\frac{\mu_1}{\sqrt{1+\nu_{11}}}, \frac{\mu_2}{\sqrt{1+\nu_{22}}}; \frac{\nu_{12}}{\sqrt{(1+\nu_{11})(1+\nu_{22})}}\right) \tag{18}$$

$$\square$$

*Proof of* (12c). Using $\phi = \Phi'$ in both terms,

$$\mathbb{E}\,\phi(X_1)\phi(X_2) = \mathbb{E}\,\frac{\partial^2}{\partial t\partial s}\bigg|_{t=0,s=0}\Phi(X_1+t)\Phi(X_2+s) \tag{19}$$

$$= \frac{\partial^2}{\partial t\partial s}\bigg|_{t=0,s=0}\mathbb{E}\,\Phi(X_1+t)\Phi(X_2+s) \qquad\text{(dominated convergence theorem)}$$

$$= \frac{\partial^2}{\partial t\partial s}\bigg|_{t=0,s=0}\mathbb{P}\left(Z_1-X_1-t\le 0, Z_2-X_2-s\le 0\right)$$

$$\text{(introducing } Z_1, Z_2\sim\mathcal{N}(0,1))$$

$$= \frac{\partial^2}{\partial t\partial s}\bigg|_{t=0,s=0}\Phi_2\left(\frac{\mu_1+t}{\sqrt{1+\nu_{11}}}, \frac{\mu_2+s}{\sqrt{1+\nu_{22}}}; \frac{\nu_{12}}{\sqrt{(1+\nu_{11})(1+\nu_{22})}}\right) \tag{20}$$

$$= \frac{1}{\sqrt{1+\nu_{11}}\sqrt{1+\nu_{22}}}\phi_2\left(\frac{\mu_1}{\sqrt{1+\nu_{11}}}, \frac{\mu_2}{\sqrt{1+\nu_{22}}}; \frac{\nu_{12}}{\sqrt{(1+\nu_{11})(1+\nu_{22})}}\right)$$

$$\text{(pdf is derivative of cdf)}$$

$$\square$$

*Proof of* (12d). We use the Gaussian ODE

$$\phi'(x) + x\phi(x) = 0. \tag{21}$$

Using this fact,

$$\mathbb{E}\,\phi'(X_1)\Phi(X_2) = -\mathbb{E}\,X_1\phi(X_1)\Phi(X_2) \tag{22}$$

$$= -\mu_1\,\mathbb{E}\,\phi(X_1)\Phi(X_2) - \mathbb{E}(X_1-\mu_1)\phi(X_1)\Phi(X_2) \qquad\text{(centering)}$$

$$= -\mu_1\,\mathbb{E}\,\phi(X_1)\Phi(X_2) - \nu_{11}\underbrace{\mathbb{E}\,\phi'(X_1)\Phi(X_2)}_{\text{same as LHS}} - \nu_{12}\,\mathbb{E}\,\phi(X_1)\phi(X_2) \quad\text{(Lemma 3)}$$

Collecting like terms and solving for $\mathbb{E}\,\phi'(X_1)\Phi(X_2)$,,

$$\mathbb{E}\,\phi'(X_1)\Phi(X_2) = \frac{-\mu_1}{1+\nu_{11}}\,\mathbb{E}\,\phi(X_1)\Phi(X_2) + \frac{-\nu_{12}}{1+\nu_{11}}\,\mathbb{E}\,\phi(X_1)\phi(X_2). \tag{23}$$

$$\square$$

# D   DERIVATION OF UNCERTAINTY PROPAGATION FORMULAS FOR PROBIT ACTIVATION

In this appendix, we derive the $M_\sigma$, $K_\sigma$, and $L_\sigma$ functions (Def. 3) for the normal CDF activation function

$$\sigma(x) = \sqrt{\frac{2}{\pi}} \int_0^x e^{-\frac{1}{2}u^2} du = 2\Phi(x) - 1, \quad \Phi(x) = \mathbb{P}_{Z \sim \mathcal{N}(0,1)}(Z \leq x) \tag{24}$$

**Lemma 7.** *Let $M_\sigma$ be defined as in Def. 3, with $\sigma$ as in (24).*

$$M_\sigma(\mu; \nu) = \sigma\left(\frac{\mu}{\sqrt{1+\nu}}\right)$$

*Proof.* Let $X \sim \mathcal{N}(\mu, \nu)$.

$$\mathbb{E}\,\sigma(X) = -1 + 2\,\mathbb{E}\,\Phi(X) \tag{25}$$

This is a direct application of Lemma 5, (5a). $\square$

**Lemma 8.** *Let $K_\sigma$ be defined as in Def. 3, with $\sigma$ as in (24).*

$$K(\mu_1, \mu_2; \nu_{11}, \nu_{22}, \nu_{12}) = 4\,\Phi_2\left(\frac{\mu_1}{\sqrt{1+\nu_{11}}}, \frac{\mu_2}{\sqrt{1+\nu_{22}}}; \rho'\right)\Bigg|_{\rho'=0}^{\rho'=\frac{\nu_{12}}{\sqrt{(1+\nu_{11})(1+\nu_{22})}}},$$

*where $\Phi_2$ is the bivariate normal CDF.*

*Proof.* The covariance may be expressed as

$$\text{Cov}(\sigma(X_1), \sigma(X_2)) = 4\,\text{Cov}(\Phi(X_1), \Phi(X_2)) \tag{26}$$
$$= 4\,\mathbb{E}\,\Phi(X_1)\Phi(X_2) - 4\,\mathbb{E}\,\Phi(X_1)\,\mathbb{E}\,\Phi(X_2) \tag{27}$$

Now apply Lemma 6 to the first term. $\square$

**Lemma 9.** *Let $L_\sigma$ be defined as in Def. 3, with $\sigma$ as in (24). Then*

$$L_\sigma(\mu_1; \nu_{11}, \nu_{22}, \nu_{12}) = 2\frac{\nu_{12}}{\sqrt{1+\nu_{11}}}\phi\left(\frac{\mu_1}{\sqrt{1+\nu_{11}}}\right).$$

*Proof.* Using Lemma 4, we have

$$L_\sigma(\mu_1; \nu_{11}, \nu_{22}, \nu_{12}) = \nu_{12}\,\mathbb{E}\,\sigma'(X_1) \tag{28}$$
$$= 2\nu_{12}\,\mathbb{E}\,\phi(X_1) \tag{29}$$

where $\phi = \Phi'$. By Lemma 5, applied to $X_1 \sim \mathcal{N}(\mu_1, \nu_{11})$,

$$\mathbb{E}\,\phi(X_1) = \frac{1}{\sqrt{1+\nu_{11}}}\phi\left(\frac{\mu_1}{\sqrt{1+\nu_{11}}}\right). \tag{30}$$

$\square$

# E   DERIVATION OF UNCERTAINTY PROPAGATION FORMULAS FOR GELU ACTIVATION

In this appendix, we derive the $M_\sigma$, $K_\sigma$ and $L_\sigma$ functions (Def. 3) for the GeLU activation function

$$\sigma(x) = x\Phi(x). \tag{31}$$

**Lemma 10.** *Let $M_\sigma$ be defined as in Def. 3, with $\sigma$ as in (31). Then*

$$M_\sigma(\mu; \nu) = \frac{\nu}{\sqrt{1+\nu}}\phi\left(\frac{\mu}{\sqrt{1+\nu}}\right) + \mu\Phi\left(\frac{\mu}{\sqrt{1+\nu}}\right).$$

*Proof.* Let $X \sim \mathcal{N}(\mu, \nu)$ and $Z \sim \mathcal{N}(0, 1)$. Centering $X$,

$$
\begin{aligned}
M_\sigma(\mu; \nu) = \mathbb{E}\, X\Phi(X) &= \mathbb{E}(X - \mu)\Phi(X) + \mu\, \mathbb{E}\, \Phi(X) \\
&= \mathbb{E}(X - \mu)\Phi(X) + \mu\Phi\left(\frac{\mu}{\sqrt{1 + \nu}}\right) && \text{(using Lemma 5, (5a))} \\
&= \nu\, \mathbb{E}\, \phi(X) + \mu\Phi\left(\frac{\mu}{\sqrt{1 + \nu}}\right) && \text{(using Lemma 3)} \\
&= \frac{\nu}{\sqrt{1 + \nu}}\phi\left(\frac{\mu}{\sqrt{1 + \nu}}\right) + \mu\Phi\left(\frac{\mu}{\sqrt{1 + \nu}}\right) && \text{(using Lemma 5, (5b))}
\end{aligned}
$$

$\square$

**Lemma 11.** *Let $K_\sigma$ be defined as in Def. 3, with $\sigma$ as in* (31). *Then*

$$
\begin{aligned}
&K_\sigma(\mu_1, \mu_2; \nu_{11}, \nu_{22}, \nu_{12}) \\
&= \left(\mu_1\nu_{12} + \mu_2\nu_{11} - \frac{\mu_1\nu_{12}\nu_{11}}{1 + \nu_{11}}\right) \frac{1}{\sqrt{1 + \nu_{11}}}\Phi_{2;1}\left(\frac{\mu_1}{\sqrt{1 + \nu_{11}}}, \frac{\mu_2}{\sqrt{1 + \nu_{22}}}; \frac{\nu_{12}}{\sqrt{(1 + \nu_{11})(1 + \nu_{22})}}\right) \\
&\quad + \left(\mu_2\nu_{12} + \mu_1\nu_{22} - \frac{\mu_2\nu_{12}\nu_{22}}{1 + \nu_{22}}\right) \frac{1}{\sqrt{1 + \nu_{22}}}\Phi_{2;1}\left(\frac{\mu_2}{\sqrt{1 + \nu_{22}}}, \frac{\mu_1}{\sqrt{1 + \nu_{11}}}; \frac{\nu_{12}}{\sqrt{(1 + \nu_{11})(1 + \nu_{22})}}\right) \\
&\quad + \frac{\nu_{11}\nu_{22} + \nu_{12}^2\left(1 - \frac{\nu_{11}}{1 + \nu_{11}} - \frac{\nu_{22}}{1 + \nu_{22}}\right)}{\sqrt{(1 + \nu_{11})(1 + \nu_{22})}}\phi_2\left(\frac{\mu_1}{\sqrt{1 + \nu_{11}}}, \frac{\mu_2}{\sqrt{1 + \nu_{22}}}; \frac{\nu_{12}}{\sqrt{(1 + \nu_{11})(1 + \nu_{22})}}\right) \\
&\quad + (\mu_1\mu_2 + \nu_{12})\, \Phi_2\left(\frac{\mu_1}{\sqrt{1 + \nu_{11}}}, \frac{\mu_2}{\sqrt{1 + \nu_{22}}}; \frac{\nu_{12}}{\sqrt{(1 + \nu_{11})(1 + \nu_{22})}}\right) \\
&\quad - M_\sigma(\mu_1; \nu_{11})M_\sigma(\mu_2; \nu_{22})
\end{aligned}
$$

*Proof.* Let

$$
\begin{pmatrix} X_1 \\ X_2 \end{pmatrix} \sim \mathcal{N}\left(\begin{pmatrix} \mu_1 \\ \mu_2 \end{pmatrix}, \begin{pmatrix} \nu_{11} & \nu_{12} \\ \nu_{12} & \nu_{22} \end{pmatrix}\right). \tag{32}
$$

Our strategy is to use the formula

$$
\mathrm{Cov}(\sigma(X_1), \sigma(X_2)) = \mathbb{E}\, \sigma(X_1)\sigma(X_2) - \mathbb{E}\, \sigma(X_1)\, \mathbb{E}\, \sigma(X_2)
$$

and use Lemma 10 for the second term. For the cross term, we first center the random variables:

$$
\begin{aligned}
\mathbb{E}\, X_1\Phi(X_1)X_2\Phi(X_2) &= \underbrace{\mathbb{E}(X_1 - \mu_1)\Phi(X_1)(X_2 - \mu_2)\Phi(X_2)}_{\text{(I)}} \\
&\quad + \underbrace{\mathbb{E}\, \mu_1\Phi(X_1)(X_2 - \mu_2)\Phi(X_2)}_{\text{(II)}} \\
&\quad + \underbrace{\mathbb{E}(X_1 - \mu_1)\Phi(X_1)\mu_2\Phi(X_2)}_{\text{(III)}} \\
&\quad + \underbrace{\mathbb{E}\, \mu_1\Phi(X_1)\mu_2\Phi(X_2)}_{\text{(IV)}} \tag{33}
\end{aligned}
$$

For term (I) of (33),

$$
\begin{aligned}
\text{(I)} &= \mathbb{E}(X_1 - \mu_1)\Phi(X_1)(X_2 - \mu_2)\Phi(X_2) \tag{34} \\
&= \sum_{i=1}^{2} \mathrm{Cov}(X_1, X_i)\, \mathbb{E}\, \partial_{X_i}\left(\Phi(X_1)(X_2 - \mu_2)\Phi(X_2)\right) && \text{(Lemma 3)} \\
&= \underbrace{\nu_{11}\, \mathbb{E}\, \phi(X_1)(X_2 - \mu_2)\Phi(X_2)}_{\text{(I.a)}} + \nu_{12}\, \mathbb{E}\, \Phi(X_1)\Phi(X_2) + \underbrace{\nu_{12}\, \mathbb{E}\, \Phi(X_1)(X_2 - \mu_2)\phi(X_2)}_{\text{(I.b)}} \tag{35}
\end{aligned}
$$

We apply Lemma 3 to (I.a) and (I.b) to get

$$\text{(I.a)} + \text{(I.b)} = \nu_{11}\, \mathbb{E}\, \phi(X_1)(X_2 - \mu_2)\Phi(X_2) + \nu_{12}\, \mathbb{E}\, \Phi(X_1)(X_2 - \mu_2)\phi(X_2) \tag{36}$$

$$= \nu_{11}\, \mathbb{E}\, \big[\nu_{12}\phi'(X_1)\Phi(X_2) + \nu_{22}\phi(X_1)\phi(X_2)\big]$$
$$+ \nu_{12}\, \mathbb{E}\, \big[\nu_{12}\phi(X_1)\phi(X_2) + \nu_{22}\Phi(X_1)\phi'(X_2)\big] \tag{37}$$

$$= \nu_{12}\nu_{11}\, \underbrace{\mathbb{E}\, \phi'(X_1)\Phi(X_2)}_{\text{(I.c)}} + \nu_{12}\nu_{22}\, \underbrace{\mathbb{E}\, \Phi(X_1)\phi'(X_2)}_{\text{(I.d)}}$$
$$+ \left(\nu_{11}\nu_{22} + \nu_{12}^2\right) \mathbb{E}\, \phi(X_1)\phi(X_2) \tag{38}$$

By Lemma 6, (12d), we have

$$\text{(I.c)} = \frac{-\mu_1}{1 + \nu_{11}}\, \mathbb{E}\, \phi(X_1)\Phi(X_2) + \frac{-\nu_{12}}{1 + \nu_{11}}\, \mathbb{E}\, \phi(X_1)\phi(X_2)$$

$$\text{(I.d)} = \frac{-\mu_2}{1 + \nu_{22}}\, \mathbb{E}\, \Phi(X_1)\phi(X_2) + \frac{-\nu_{12}}{1 + \nu_{22}}\, \mathbb{E}\, \phi(X_1)\phi(X_2)$$

Combining these last two equations with (35) and (38), (I) becomes

$$\text{(I)} = \nu_{12}\nu_{11} \left( \frac{-\mu_1}{1 + \nu_{11}}\, \mathbb{E}\, \phi(X_1)\Phi(X_2) + \frac{-\nu_{12}}{1 + \nu_{11}}\, \mathbb{E}\, \phi(X_1)\phi(X_2) \right)$$
$$+ \nu_{12}\nu_{22} \left( \frac{-\mu_2}{1 + \nu_{22}}\, \mathbb{E}\, \Phi(X_1)\phi(X_2) + \frac{-\nu_{12}}{1 + \nu_{22}}\, \mathbb{E}\, \phi(X_1)\phi(X_2) \right)$$
$$+ \left(\nu_{11}\nu_{22} + \nu_{12}^2\right) \mathbb{E}\, \phi(X_1)\phi(X_2)$$
$$+ \nu_{12}\, \mathbb{E}\, \Phi(X_1)\Phi(X_2) \tag{39}$$

which simplifies to

$$\text{(I)} = \frac{-\mu_1 \nu_{12}\nu_{11}}{1 + \nu_{11}}\, \mathbb{E}\, \phi(X_1)\Phi(X_2) + \frac{-\mu_2 \nu_{12}\nu_{22}}{1 + \nu_{22}}\, \mathbb{E}\, \Phi(X_1)\phi(X_2)$$
$$+ \left[ \nu_{11}\nu_{22} + \nu_{12}^2 \left( 1 - \frac{\nu_{11}}{1 + \nu_{11}} - \frac{\nu_{22}}{1 + \nu_{22}} \right) \right] \mathbb{E}\, \phi(X_1)\phi(X_2)$$
$$+ \nu_{12}\, \mathbb{E}\, \Phi(X_1)\Phi(X_2) \tag{40}$$

For terms (II-III) of (33), we apply Lemma 3 to get

$$\text{(II)} = \mathbb{E}\, \mu_1 \Phi(X_1)(X_2 - \mu_2)\Phi(X_2) = \mu_1 \left( \nu_{12}\, \mathbb{E}\, \phi(X_1)\Phi(X_2) + \nu_{22}\, \mathbb{E}\, \Phi(X_1)\phi(X_2) \right)$$

$$\text{(III)} = \mathbb{E}(X_1 - \mu_1)\Phi(X_1)\mu_2 \Phi(X_2) = \mu_2 \left( \nu_{11}\, \mathbb{E}\, \phi(X_1)\Phi(X_2) + \nu_{12}\, \mathbb{E}\, \Phi(X_1)\phi(X_2) \right)$$

Now (33) becomes:

$$\mathbb{E}\, X_1 \Phi(X_1) X_2 \Phi(X_2)$$
$$= \left( \mu_1 \nu_{12} + \mu_2 \nu_{11} - \frac{\mu_1 \nu_{12}\nu_{11}}{1 + \nu_{11}} \right) \underbrace{\mathbb{E}\, \phi(X_1)\Phi(X_2)}_{\text{(V.a)}}$$
$$+ \left( \mu_2 \nu_{12} + \mu_1 \nu_{22} - \frac{\mu_2 \nu_{12}\nu_{22}}{1 + \nu_{22}} \right) \underbrace{\mathbb{E}\, \Phi(X_1)\phi(X_2)}_{\text{(V.b)}}$$
$$+ \left[ \nu_{11}\nu_{22} + \nu_{12}^2 \left( 1 - \frac{\nu_{11}}{1 + \nu_{11}} - \frac{\nu_{22}}{1 + \nu_{22}} \right) \right] \underbrace{\mathbb{E}\, \phi(X_1)\phi(X_2)}_{\text{(V.c)}}$$
$$+ (\mu_1 \mu_2 + \nu_{12}) \underbrace{\mathbb{E}\, \Phi(X_1)\Phi(X_2)}_{\text{(V.d)}} \tag{41}$$

By Lemma 6, (12b),

$$(\text{V.a}) = \frac{1}{\sqrt{1+\nu_{11}}} \Phi_{2;1} \left( \frac{\mu_1}{\sqrt{1+\nu_{11}}}, \frac{\mu_2}{\sqrt{1+\nu_{22}}}; \frac{\nu_{12}}{\sqrt{(1+\nu_{11})(1+\nu_{22})}} \right) \tag{42}$$

and

$$(\text{V.b}) = \frac{1}{\sqrt{1+\nu_{22}}} \Phi_{2;1} \left( \frac{\mu_2}{\sqrt{1+\nu_{22}}}, \frac{\mu_1}{\sqrt{1+\nu_{11}}}; \frac{\nu_{12}}{\sqrt{(1+\nu_{11})(1+\nu_{22})}} \right). \tag{43}$$

By Lemma 6, (12c),

$$(\text{V.c}) = \frac{1}{\sqrt{1+\nu_{11}}\sqrt{1+\nu_{22}}} \phi_2 \left( \frac{\mu_1}{\sqrt{1+\nu_{11}}}, \frac{\mu_2}{\sqrt{1+\nu_{22}}}; \frac{\nu_{12}}{\sqrt{(1+\nu_{11})(1+\nu_{22})}} \right). \tag{44}$$

By Lemma 6, (12a),

$$(\text{V.d}) = \Phi_2 \left( \frac{\mu_1}{\sqrt{1+\nu_{11}}}, \frac{\mu_2}{\sqrt{1+\nu_{22}}}; \frac{\nu_{12}}{\sqrt{(1+\nu_{11})(1+\nu_{22})}} \right) \tag{45}$$

The final form of (41) is

$$\mathbb{E}\, X_1 \Phi(X_1) X_2 \Phi(X_2)$$

$$= \left( \mu_1 \nu_{12} + \mu_2 \nu_{11} - \frac{\mu_1 \nu_{12} \nu_{11}}{1+\nu_{11}} \right) \frac{1}{\sqrt{1+\nu_{11}}} \Phi_{2;1} \left( \frac{\mu_1}{\sqrt{1+\nu_{11}}}, \frac{\mu_2}{\sqrt{1+\nu_{22}}}; \frac{\nu_{12}}{\sqrt{(1+\nu_{11})(1+\nu_{22})}} \right)$$

$$+ \left( \mu_2 \nu_{12} + \mu_1 \nu_{22} - \frac{\mu_2 \nu_{12} \nu_{22}}{1+\nu_{22}} \right) \frac{1}{\sqrt{1+\nu_{22}}} \Phi_{2;1} \left( \frac{\mu_2}{\sqrt{1+\nu_{22}}}, \frac{\mu_1}{\sqrt{1+\nu_{11}}}; \frac{\nu_{12}}{\sqrt{(1+\nu_{11})(1+\nu_{22})}} \right)$$

$$+ \frac{\nu_{11}\nu_{22} + \nu_{12}^2 \left( 1 - \frac{\nu_{11}}{1+\nu_{11}} - \frac{\nu_{22}}{1+\nu_{22}} \right)}{\sqrt{(1+\nu_{11})(1+\nu_{22})}} \phi_2 \left( \frac{\mu_1}{\sqrt{1+\nu_{11}}}, \frac{\mu_2}{\sqrt{1+\nu_{22}}}; \frac{\nu_{12}}{\sqrt{(1+\nu_{11})(1+\nu_{22})}} \right)$$

$$+ (\mu_1 \mu_2 + \nu_{12})\, \Phi_2 \left( \frac{\mu_1}{\sqrt{1+\nu_{11}}}, \frac{\mu_2}{\sqrt{1+\nu_{22}}}; \frac{\nu_{12}}{\sqrt{(1+\nu_{11})(1+\nu_{22})}} \right) \tag{46}$$

The conclusion follows from subtracting $M(\mu_1; \nu_{11}) M(\mu_2; \nu_{22})$. $\qquad\square$

**Lemma 12.** *Let $L_\sigma$ be defined as in Def. 3, with $\sigma$ as in* (31)*. Then*

$$L_\sigma(\mu_1; \nu_{11}, \nu_{22}, \nu_{12}) = \nu_{12} \frac{\mu_1}{(1+\nu_{11})^{3/2}} \phi \left( \frac{\mu_1}{\sqrt{1+\nu_{11}}} \right) + \nu_{12} \Phi \left( \frac{\mu_1}{\sqrt{1+\nu_{11}}} \right). \tag{47}$$

*Proof.* By Lemma 4,

$$L_\sigma(\mu_1; \nu_{11}, \nu_{22}, \nu_{12}) = \nu_{12}\, \mathbb{E}\, \sigma'(X_1), \qquad\qquad X_1 \sim \mathcal{N}(\mu_1, \nu_{11})$$

$$= \nu_{12} \underbrace{\mathbb{E}\, X_1 \phi(X_1)}_{\text{(I)}} + \nu_{12} \underbrace{\mathbb{E}\, \Phi(X_1)}_{\text{(II)}} \qquad\qquad \text{(product rule)}$$

By Lemma 5, (5c)

$$(\text{I}) = \frac{\mu_1}{(1+\nu_{11})^{3/2}} \phi \left( \frac{\mu_1}{\sqrt{1+\nu_{11}}} \right). \tag{48}$$

Term (II) is given by

$$(\text{II}) = \mathbb{E}\, \Phi(X_1) = \Phi \left( \frac{\mu_1}{\sqrt{1+\nu_{11}}} \right) \tag{49}$$

Applying terms (I)–(II),

$$L_\sigma(\mu_1; \nu_{11}, \nu_{22}, \nu_{12}) = \nu_{12} \frac{\mu_1}{(1+\nu_{11})^{3/2}} \phi \left( \frac{\mu_1}{\sqrt{1+\nu_{11}}} \right) + \nu_{12} \Phi \left( \frac{\mu_1}{\sqrt{1+\nu_{11}}} \right) \tag{50}$$

$\square$

# F  DERIVATION OF UNCERTAINTY PROPAGATION FORMULAS FOR RELU ACTIVATION

In this appendix, we derive the uncertainty propagation formulas for ReLU activation

$$\sigma(x) = \max(0, x) \tag{51}$$

using the identity

$$\mathrm{ReLU}(x) = \lim_{\alpha \to \infty} \alpha^{-1} \mathrm{GeLU}(\alpha x). \tag{52}$$

See Muthén (1990) for another way to derive $M$ and $K$.

**Lemma 13.** *Let $M_\sigma$ be defined as in Def. 3, with $\sigma$ as in* (51). *Then*

$$M_\sigma(\mu; \nu) = \sqrt{\nu}\phi\left(\frac{\mu}{\sqrt{\nu}}\right) + \mu\Phi\left(\frac{\mu}{\sqrt{\nu}}\right).$$

*Proof.* By the dominated convergence theorem,

$$M_\sigma(\mu; \nu) = \lim_{\alpha \to \infty} \alpha^{-1}\, \mathbb{E}\, M_{\mathrm{GeLU}}(\alpha\mu; \alpha^2\nu) \tag{53}$$

$$= \lim_{\alpha \to \infty} \alpha^{-1}\left[\frac{\alpha^2\nu}{\sqrt{1 + \alpha^2\nu}}\phi\left(\frac{\alpha\mu}{\sqrt{1 + \alpha^2\nu}}\right) + \mu\Phi\left(\frac{\alpha\mu}{\sqrt{1 + \alpha^2\nu}}\right)\right] \quad \text{(by Lemma 10)}$$

$$= \sqrt{\nu}\phi\left(\frac{\mu}{\sqrt{\nu}}\right) + \mu\Phi\left(\frac{\mu}{\sqrt{\nu}}\right) \tag{54}$$

$\square$

**Lemma 14.** *Let $K_\sigma$ be defined as in Def. 3, with $\sigma$ as in* (51). *Then*

$$K_\sigma(\mu_1, \mu_2; \nu_{11}, \nu_{22}, \nu_{12}) = \mu_2\sqrt{\nu_{11}}\Phi_{2;1}\left(\frac{\mu_1}{\sqrt{\nu_{11}}}, \frac{\mu_2}{\sqrt{\nu_{22}}}; \frac{\nu_{12}}{\sqrt{\nu_{11}\nu_{22}}}\right)$$

$$+ \mu_1\sqrt{\nu_{22}}\Phi_{2;1}\left(\frac{\mu_2}{\sqrt{\nu_{22}}}, \frac{\mu_1}{\sqrt{\nu_{11}}}; \frac{\nu_{12}}{\sqrt{\nu_{11}\nu_{22}}}\right)$$

$$+ \left(\sqrt{\nu_{11}\nu_{22}} - \frac{\nu_{12}^2}{\sqrt{\nu_{11}\nu_{22}}}\right)\phi_2\left(\frac{\mu_1}{\sqrt{\nu_{11}}}, \frac{\mu_2}{\sqrt{\nu_{22}}}; \frac{\nu_{12}}{\sqrt{\nu_{11}\nu_{22}}}\right)$$

$$+ (\mu_1\mu_2 + \nu_{12})\Phi_2\left(\frac{\mu_1}{\sqrt{\nu_{11}}}, \frac{\mu_2}{\sqrt{\nu_{22}}}; \frac{\nu_{12}}{\sqrt{\nu_{11}\nu_{22}}}\right)$$

$$- M_\sigma(\mu_1; \nu_{11})M_\sigma(\mu_2; \nu_{22})$$

*Proof.* By the dominated convergence theorem,

$$\mathbb{E}\,\sigma(X_1)\sigma(X_2) = \mathbb{E} \lim_{\alpha \to \infty} \alpha^{-2}\, \mathrm{GeLU}(\alpha X_1)\, \mathrm{GeLU}(\alpha X_2) \tag{55}$$

$$= \lim_{\alpha \to \infty} \alpha^{-2}\, \mathbb{E}\, \mathrm{GeLU}(\alpha X_1)\, \mathrm{GeLU}(\alpha X_2) \tag{56}$$

To compute $\alpha^{-2} \mathbb{E} \operatorname{GeLU}(\alpha X_1) \operatorname{GeLU}(\alpha X_2)$, we use (46) (Proof of Lemma 11) while scaling $\mu$s by $\alpha$ and $\nu$s by $\alpha^2$.

$$\alpha^{-2} \mathbb{E} \operatorname{GeLU}(\alpha X_1) \operatorname{GeLU}(\alpha X_2)$$

$$= \frac{\alpha\mu_1\nu_{12} + \alpha\mu_2\nu_{11} - \alpha^3 \frac{\mu_1\nu_{12}\nu_{11}}{1+\alpha^2\nu_{11}}}{\sqrt{1+\alpha^2\nu_{11}}}$$

$$\Phi_{2;1}\left(\frac{\alpha\mu_1}{\sqrt{1+\alpha^2\nu_{11}}}, \frac{\alpha\mu_2}{\sqrt{1+\alpha^2\nu_{22}}}; \frac{\alpha^2\nu_{12}}{\sqrt{(1+\alpha^2\nu_{11})(1+\alpha^2\nu_{22})}}\right)$$

$$+ \frac{\alpha\mu_2\nu_{12} + \alpha\mu_1\nu_{22} - \alpha^3 \frac{\mu_2\nu_{12}\nu_{22}}{1+\alpha^2\nu_{22}}}{\sqrt{1+\alpha^2\nu_{22}}}$$

$$\Phi_{2;1}\left(\frac{\alpha\mu_2}{\sqrt{1+\alpha^2\nu_{22}}}, \frac{\alpha\mu_1}{\sqrt{1+\alpha^2\nu_{11}}}; \frac{\alpha^2\nu_{12}}{\sqrt{(1+\alpha^2\nu_{11})(1+\alpha^2\nu_{22})}}\right) \tag{57}$$

$$+ \frac{\alpha^2\nu_{11}\nu_{22} + \alpha^2\nu_{12}^2 \left(1 - \frac{\alpha^2\nu_{11}}{1+\alpha^2\nu_{11}} - \frac{\alpha^2\nu_{22}}{1+\alpha^2\nu_{22}}\right)}{\sqrt{(1+\alpha^2\nu_{11})(1+\alpha^2\nu_{22})}}$$

$$\phi_2\left(\frac{\alpha\mu_1}{\sqrt{1+\alpha^2\nu_{11}}}, \frac{\alpha\mu_2}{\sqrt{1+\alpha^2\nu_{22}}}; \frac{\alpha^2\nu_{12}}{\sqrt{(1+\alpha^2\nu_{11})(1+\alpha^2\nu_{22})}}\right)$$

$$+ (\mu_1\mu_2 + \nu_{12})\, \Phi_2\left(\frac{\alpha\mu_1}{\sqrt{1+\alpha^2\nu_{11}}}, \frac{\alpha\mu_2}{\sqrt{1+\alpha^2\nu_{22}}}; \frac{\alpha^2\nu_{12}}{\sqrt{(1+\alpha^2\nu_{11})(1+\alpha^2\nu_{22})}}\right)$$

Taking $\alpha \to \infty$,

$$\mathbb{E}\,\sigma(X_1)\sigma(X_2) = \lim_{\alpha\to\infty} \alpha^{-2} \mathbb{E} \operatorname{GeLU}(\alpha X_1) \operatorname{GeLU}(\alpha X_2)$$

$$= \mu_2\sqrt{\nu_{11}}\Phi_{2;1}\left(\frac{\mu_1}{\sqrt{\nu_{11}}}, \frac{\mu_2}{\sqrt{\nu_{22}}}; \frac{\nu_{12}}{\sqrt{\nu_{11}\nu_{22}}}\right)$$

$$+ \mu_1\sqrt{\nu_{22}}\Phi_{2;1}\left(\frac{\mu_2}{\sqrt{\nu_{22}}}, \frac{\mu_1}{\sqrt{\nu_{11}}}; \frac{\nu_{12}}{\sqrt{\nu_{11}\nu_{22}}}\right) \tag{58}$$

$$+ \left(\sqrt{\nu_{11}\nu_{22}} - \frac{\nu_{12}^2}{\sqrt{\nu_{11}\nu_{22}}}\right) \phi_2\left(\frac{\mu_1}{\sqrt{\nu_{11}}}, \frac{\mu_2}{\sqrt{\nu_{22}}}; \frac{\nu_{12}}{\sqrt{\nu_{11}\nu_{22}}}\right)$$

$$+ (\mu_1\mu_2 + \nu_{12})\, \Phi_2\left(\frac{\mu_1}{\sqrt{\nu_{11}}}, \frac{\mu_2}{\sqrt{\nu_{22}}}; \frac{\nu_{12}}{\sqrt{\nu_{11}\nu_{22}}}\right)$$

The conclusion follows after subtracting $\mathbb{E}\,\sigma(X_1)\,\mathbb{E}\,\sigma(X_2)$. $\qquad\square$

**Lemma 15.** *Let $L_\sigma$ be defined as in Def. 3, with $\sigma$ as in* (51). *Then*

$$L_\sigma(\mu_1; \nu_{11}, \nu_{22}, \nu_{12}) = \nu_{12}\Phi\left(\frac{\mu_1}{\sqrt{\nu_{11}}}\right).$$

*Proof.* By the dominated convergence theorem,

$$L_\sigma(\mu_1; \nu_{11}, \nu_{22}, \nu_{12})$$

$$= \lim_{\alpha\to\infty} \alpha^{-2} L_{\operatorname{GeLU}}(\alpha\mu_1; \alpha^2\nu_{11}, \alpha^2\nu_{22}, \alpha^2\nu_{12}) \tag{59}$$

$$= \lim_{\alpha\to\infty} \alpha^{-2} \left[\alpha^2\nu_{12}\frac{\alpha\mu_1}{(1+\alpha^2\nu_{11})^{3/2}}\phi\left(\frac{\alpha\mu_1}{\sqrt{1+\alpha^2\nu_{11}}}\right) + \alpha^2\nu_{12}\Phi\left(\frac{\alpha\mu_1}{\sqrt{1+\alpha^2\nu_{11}}}\right)\right]$$

$$\text{(by Lemma 12)}$$

$$= \nu_{12}\Phi\left(\frac{\mu_1}{\sqrt{\nu_{11}}}\right) \tag{60}$$

□

# G  DERIVATION OF UNCERTAINTY PROPAGATION FORMULAS FOR HEAVISIDE ACTIVATION

In this appendix, we state the $M_\sigma$, $K_\sigma$ and $L_\sigma$ functions (Def. 3) for the Heaviside activation function

$$\sigma(x) = \mathbf{1}_{x>0} \tag{61}$$

$$= \lim_{\alpha \to \infty} \left( 1/2 + 1/2\hat{\sigma}(\alpha x) \right) \tag{62}$$

where $\hat{\sigma}$ is the odd probit sigmoid defined as in (51). The idea of the proofs is to take dominated limits of the results in Appendix. D, similar to Appendix F's treatment of Appendix E, so we omit them.

**Lemma 16.** *Let $\sigma$ be the Heaviside function as in* (61). *Then the functions defined in Def. 3 are*

$$M_\sigma(\mu; \nu) = \Phi\left( \frac{\mu}{\sqrt{\nu}} \right),$$

$$K_\sigma(\mu_1, \mu_2; \nu_{11}, \nu_{22}, \nu_{12}) = \Phi_2\left( \frac{\mu_1}{\sqrt{\nu_{11}}}, \frac{\mu_2}{\sqrt{\nu_{22}}}; \rho' \right)\Bigg|_{\rho'=0}^{\rho'=\frac{\nu_{12}}{\sqrt{\nu_{11}\nu_{22}}}}, \quad and$$

$$L_\sigma(\mu_1; \nu_{11}, \nu_{22}, \nu_{12}) = \frac{\nu_{12}}{\sqrt{\nu_{11}}} \phi\left( \frac{\mu_1}{\sqrt{\nu_{11}}} \right).$$

# H DERIVATION OF UNCERTAINTY PROPAGATION FORMULAS FOR SINE ACTIVATION

In this appendix, we derive the $M_\sigma$, $K_\sigma$ and $L_\sigma$ functions (Def. 3) for the sinusoidal activation function

$$\sigma(x) = \sin(x). \tag{63}$$

We begin by recalling the identities that if $Z \sim \mathcal{N}(\mu, \nu)$,

$$\mathbb{E}\sin(Z) = e^{-\nu/2}\sin(\mu) \tag{64}$$

$$\mathbb{E}\cos(Z) = e^{-\nu/2}\cos(\mu) \tag{65}$$

These formulas, which can be derived from the characteristic function of the normal distribution, allow moments of $\sin(Z)$ and $\cos(Z)$ to be computed exactly. There is no need, as in Sitzmann et al. (2020, Lemma 1.6) to use analytic approximations in this step.

From (64) immediately follows:

**Lemma 17.** *Let $M_\sigma$ be defined as in Def. 3, with $\sigma$ as in* (63)*. Then*

$$M_\sigma(\mu; \nu) = e^{-\nu/2}\sin(\mu).$$

**Lemma 18.** *Let $K_\sigma$ be defined as in Def. 3, with $\sigma$ as in* (63)*. Then*

$$K_\sigma(\mu_1, \mu_2; \nu_{11}, \nu_{22}, \nu_{12}) = \frac{1}{2}\left[e^{\nu^* + \nu_{12}} - e^{\nu^*}\right]\cos(\mu_1 - \mu_2)$$
$$- \frac{1}{2}\left[e^{\nu^* - \nu_{12}} - e^{\nu^*}\right]\cos(\mu_1 + \mu_2),$$

*where*

$$\nu^* = -\frac{\nu_{11} + \nu_{22}}{2}.$$

*Proof.* Let

$$\begin{pmatrix} X_1 \\ X_2 \end{pmatrix} \sim \mathcal{N}\left(\begin{pmatrix} \mu_1 \\ \mu_2 \end{pmatrix}, \begin{pmatrix} \nu_{11} & \nu_{12} \\ \nu_{12} & \nu_{22} \end{pmatrix}\right). \tag{66}$$

Then by inserting (63), we have

$$\mathrm{Cov}(\sigma(X_1), \sigma(X_2)) = \mathbb{E}\sin(X_1)\sin(X_2) - \mathbb{E}\sin(X_1)\,\mathbb{E}\sin(X_2). \tag{67}$$

Using some trigonometric identities and (65), the first term of (67) becomes

$$\mathbb{E}\sin(X_1)\sin(X_2) = \frac{1}{2}\mathbb{E}\cos(\underbrace{X_1 - X_2}_{\mathcal{N}(\mu_1 - \mu_2, \nu_{11} + \nu_{22} - 2\nu_{12})}) - \frac{1}{2}\mathbb{E}\cos(\underbrace{X_1 + X_2}_{\mathcal{N}(\mu_1 + \mu_2, \nu_{11} + \nu_{22} + 2\nu_{12})}) \tag{68}$$

$$= \frac{1}{2}\exp\left(-\frac{\nu_{11} + \nu_{22}}{2} + \nu_{12}\right)\cos(\mu_1 - \mu_2)$$
$$- \frac{1}{2}\exp\left(-\frac{\nu_{11} + \nu_{22}}{2} - \nu_{12}\right)\cos(\mu_1 + \mu_2) \tag{69}$$

The second term of (67) becomes

$$\mathbb{E}\sin(X_1)\,\mathbb{E}\sin(X_2) = \frac{1}{2}\exp\left(-\frac{\nu_{11} + \nu_{22}}{2}\right)\cos(\mu_1 - \mu_2)$$
$$- \frac{1}{2}\exp\left(-\frac{\nu_{11} + \nu_{22}}{2}\right)\cos(\mu_1 + \mu_2) \tag{70}$$

The result follows from collecting like terms. $\square$

**Lemma 19.** *Let $L_\sigma$ be defined as in Def. 3, with $\sigma$ as in* (63)*. Then*

$$L_\sigma(\mu_1, \mu_2; \nu_{11}, \nu_{22}, \nu_{12}) = \nu_{12}e^{-\nu_{11}/2}\cos(\mu_1).$$

*Proof.* Let

$$\begin{pmatrix} X_1 \\ X_2 \end{pmatrix} \sim \mathcal{N}\left(\begin{pmatrix} \mu_1 \\ \mu_2 \end{pmatrix}, \begin{pmatrix} \nu_{11} & \nu_{12} \\ \nu_{12} & \nu_{22} \end{pmatrix}\right). \tag{71}$$

Using Lemma 4, we have

$$L_\sigma(\mu_1, \mu_2; \nu_{11}, \nu_{22}, \nu_{12}) = \nu_{12}\, \mathbb{E}\, \sigma'(X_1) \tag{72}$$

$$= \nu_{12}\, \mathbb{E} \cos(X_1) \tag{73}$$

$$= \nu_{12} e^{-\nu_{11}/2} \cos(\mu_1) \tag{74}$$

by (65). □

# I   THEORETICAL GUARANTEES

The objective of this section is to provide a theoretical analysis of the dissimilarity between $Y_0$ and $Y$. We recall the layer-by-layer Normal approximation resulting in $Y$ alongside the exact neural network formula resulting in $Y_0$.

$$Y = Y^\ell \qquad\qquad Y_0 = Y_0^\ell \qquad\qquad\qquad (75a)$$

$$Y^k = \mathrm{N}\, g^k(Y^{k-1}), \qquad Y_0^k = g^k(Y_0^{k-1}), \qquad k \in \{1 \ldots \ell\}, \qquad (75b)$$

$$Y^0 = X \qquad\qquad Y_0^0 = X, \qquad\qquad\qquad (75c)$$

where $g^k$ is the function

$$g^k(x) = g(x; A^k, b^k, C^k, d^k). \qquad\qquad (75d)$$

We will use the Wasserstein distance to measure the distance between $Y$ and $Y_0$.

The ultimate goal is to show that $d_{\mathrm{W}}(Y_0, Y)$ is small. We will build up this quantity by recursion through the layers of the neural network. The key idea is that the error induced by Normal approximation gets worse with every subsequent layer of the network.

## I.1   RECURSIVE TRIANGLE INEQUALITY

Our basic triangle inequality is

$$\Delta^k := d_{\mathrm{W}}(Y_0^k, Y^k) \qquad\qquad (76)$$

$$= d_{\mathrm{W}}(g^k(Y_0^{k-1}), Y^k) \qquad\qquad \text{(definition of hidden layer (75b))}$$

$$\leq d_{\mathrm{W}}(g^k(Y_0^{k-1}), g^k(Y^{k-1})) + d_{\mathrm{W}}(g^k(Y^{k-1}), Y^k) \qquad \text{(triangle inequality)}$$

$$\leq \left\| \nabla g^k \right\|_\infty d_{\mathrm{W}}(Y_0^{k-1}, Y^{k-1}) + d_{\mathrm{W}}(g^k(Y^{k-1}), Y^k) \qquad \text{(Lipschitz property of } d_{\mathrm{W}})$$

$$\leq \left\| \nabla g^k \right\|_\infty \Delta^{k-1} + d_{\mathrm{W}}(g^k(Y^{k-1}), Y^k) \qquad\qquad (77)$$

If $X$ is Gaussian, then $\Delta^0 = 0$. We need to fill two blanks to make this formula concrete.

## I.2   LIPSCHITZ CONSTANT

First, the Lipschitz constant $\left\| \nabla g^k \right\|_\infty$:

$$\nabla g(x; A, b, C, d) = \nabla \left[ \sigma(Ax + b) + Cx + d \right] \qquad\qquad (78)$$

$$= A^\mathsf{T} \operatorname{diag} \left\{ \sigma'(Ax + b) \right\} + C^\mathsf{T} \qquad\qquad (79)$$

To bound this quantity,

$$\left\| \nabla g(x; A, b, C, d) \right\| \leq \sup_x \left\| A^\mathsf{T} \operatorname{diag} \left\{ \sigma'(Ax + b) \right\} + C^\mathsf{T} \right\| \qquad\qquad (80)$$

$$\leq \left\| \sigma' \right\|_\infty \|A\| + \|C\| \qquad\qquad (81)$$

## I.3   NON-NORMALITY

Second, we need to bound the distance between $g^k(Y^{k-1})$ and $Y^k$. The importance of the recursion (77) is that at layer $k$, we assess the goodness of fit between $g^k(Y^{k-1})$, a nonlinear transformation of a Normal random vector, and $Y^k$, its approximant. As the input to $\sigma$ is Normal, the non-normality of $g^k(Y^{k-1})$ depends on the non-linearity of $g^k$, as measured by its second derivatives:

$$d_{\mathrm{W}}\left( g^k(Y^{k-1}), Y^k \right) \leq \frac{3}{\sqrt{2}} \left\| \Sigma_k^{-1} \right\| \left\| \Sigma_k \right\|^{1/2} \left\| A^k \Sigma_{k-1} \left( A^k \right)^\mathsf{T} \right\|^{3/2}$$

$$\cdot \left( \sum_{i=1}^{\mathsf{d}^k} \mathbb{E} \left\| \sigma''(Y_i^{k-1}) \right\|^4 \right)^{1/4} \left( \sum_{i=1}^{\mathsf{d}^k} \mathbb{E} \left\| \sigma'(Y_i^{k-1}) \right\|^4 \right)^{1/4} \qquad (82)$$

where $\mathsf{d}^k$ is the number of neurons in layer $k$ and $\Sigma^k = \operatorname{Cov} Y^k$. As a cruder approximation,

$$d_{\mathrm{W}}\left(g^k(Y^{k-1}), Y^k\right) \leq \frac{3}{\sqrt{2}}\left(\mathsf{d}^k\right)^{1/2} \|\sigma''\|_\infty \|\sigma'\|_\infty \left\|\Sigma_k^{-1}\right\| \|\Sigma_k\|^{1/2} \left\|A^k \Sigma_{k-1}\left(A^k\right)^\intercal\right\|^{3/2}$$

(83)

These inequalities are applications of Lem. 20, which follows from a result from the functional analysis of Gaussian spaces:

**Theorem 1** (Nourdin et al. (2009, Theorem 7.1))**.** *Fix $d \geq 2$, and let $C = \{C(i,j); i, j = 1, \ldots, d\}$ be a $d \times d$ positive definite matrix. Suppose that $F = (F_1, \ldots, F_d)$ is a $\mathbb{R}^d$-valued random vector such that $\mathbb{E}\, F_i = 0$ and $F_i \in \mathbb{D}^{2,4}$ for every $i = 1, \ldots, d$. Assume moreover that $F$ has covariance matrix $C$. Then*

$$d_W(F, \mathcal{N}_d(0, C)) \leq \frac{3}{\sqrt{2}} \|C^{-1}\| \|C\|^{1/2} \left(\sum_{i=1}^d \mathbb{E}\, \|D^2 F_i\|_{\mathrm{op}}^4\right)^{1/4} \left(\sum_{j=1}^d \mathbb{E}\, \|DF_j\|_{\mathfrak{H}}^4\right)^{1/4},$$

*where $\mathcal{N}_d(0, C)$ indicates a $d$-dimensional centered Gaussian vector, with covariance matrix equal to $C$, and $D$ is the Malliavin derivative with respect to the underlying isonormal process.*

This theorem is stated in terms of the deep generality of the Malliavin calculus for nonlinear functionals $F$ of infinite-dimensional Gaussian processes. We adapt it to an inequality for finite-dimensional Gaussian $X$:

**Lemma 20** (Second-order Poincaré inequality)**.** *Let $X \sim \mathcal{N}(\mu_X, \Sigma_X)$ be a $\mathbb{R}^n$-valued random vector and $Y = f(X)$ be a $\mathbb{R}^d$-valued random vector with mean $\mu$ and covariance matrix $\Sigma_Y$. Then*

$$d_{\mathrm{W}}\left(Y, \mathcal{N}(\mu, \Sigma_Y)\right) \leq \frac{3}{\sqrt{2}} \|\Sigma_Y^{-1}\| \|\Sigma_Y\|^{1/2} \|\Sigma_X\|^{3/2}$$
$$\cdot \left[\sum_{i=1}^d \left(\mathbb{E}\, \left\|\nabla^2 f_i(X)\right\|^4\right)^{1/4}\right] \left[\sum_{i=1}^d \left(\mathbb{E}\, \left\|\nabla f_i(X)\right\|^4\right)^{1/4}\right],$$

*where $\nabla^2 f_i(X)$ is the Hessian of $f_i$ evaluated at $X$, and $\nabla f_i(X)$ is the gradient of $f_i$ evaluated at $X$.*

While this lemma is technically a basic application of Thm. 1, some nontrivial prerequisites are required to get it right.[4] Before proving this lemma, we first provide an exposition of isonormal Gaussian processes and the spaces $\mathbb{D}^{m,p}$. This material is derived from Nualart (2006); Nourdin et al. (2010).

Let $\mathcal{I}$ be an abstract index set. A Gaussian process $\{X(i)\}_{i \in \mathcal{I}}$ is a family of $\mathbb{R}$-valued random variables such that for every finite subset $\mathcal{J} \subset \mathcal{I}$, the random variables $\{X(i)\}_{i \in \mathcal{J}}$ are jointly Normal. For example: let $Z$ be a multivariate Normal random vector taking values in $\mathbb{R}^n$. Let $\mathcal{I} = \{1, \ldots, n\}$. Define $X(i) = Z_i$ for $i \in \mathcal{I}$. Then $\{X(i)\}_{i \in \mathcal{I}}$ is a Gaussian process. Another example: in problems of learning a function $f : \mathbb{R}^n \to \mathbb{R}$ from data, the function of interest is modeled as a realization of a Gaussian process $\{f(x)\}_{x \in \mathbb{R}^n}$ with a covariance kernel $k(x, x')$ (Stein, 1999; Rasmussen & Williams, 2008).

The first example above had a finite index set. The second had a finite-dimensional index set. But the generality of Thm. 1 is amenable to an infinite-dimensional index set such as that obtained by a suitable restriction of $L^2$. For example, the modern theory of weak convergence is interested in identifying so-called Donsker classes of functions $h : \mathbb{R}^n \to \mathbb{R}$ such that for a sequence of i.i.d. $\mathbb{R}^n$-valued random variables $\{X_i\}_{i=1}^\infty$,

$$G(h) = \lim_{n \to \infty} \frac{1}{\sqrt{n}} \sum_{i=1}^n \left(h(X_i) - \mathbb{E}\, h(X)\right)$$

---

[4]cf. Karvonen & Särkkä (2025) which does not correctly scale with the input covariance.

converges in distribution to a Gaussian process indexed by $h$ (Van Der Vaart & Wellner, 1996).

Prepared with the vocabulary of abstract Gaussian processes, we introduce isonormal Gaussian processes:

**Definition 8** (Isonormal Gaussian process). *Let $\mathfrak{h}$ be a separable Hilbert space. A stochastic process $W = \{W(h), h \in \mathfrak{h}\}$ is an isonormal Gaussian process if it satisfies the following two properties:*

    *1. For every $h \in \mathfrak{h}$, $\mathbb{E}\, W(h) = 0$.*

    *2. For every $h_1, h_2 \in \mathfrak{h}$, $\mathbb{E}\, W(h_1)W(h_2) = \langle h_1, h_2 \rangle_{\mathfrak{h}}$.*

Let $\{W(h)\}_{h \in \mathfrak{h}}$ be an isonormal Gaussian process. Let $\mathcal{S}$ be the set of all random variables that depend "nicely" on a finite subset of $W$, i.e.

$$\mathcal{S} = \left\{ f(W(h_1), \ldots, W(h_n)) \mid f \in \mathcal{C}_{\mathrm{c}}^{\infty}(\mathbb{R}^n), h_1, \ldots, h_n \in \mathfrak{h} \right\}$$

where $\mathcal{C}_{\mathrm{c}}^{\infty}(\mathbb{R}^n)$ is the space of smooth functions with compact support. The Malliavin derivative is defined by

$$Df(W(h_1), \ldots, W(h_n)) = \sum_{i=1}^{n} \partial_i f(W(h_1), \ldots, W(h_n)) h_i.$$

The $m$th Malliavin derivative, which takes values in $\mathfrak{h}^{\otimes m}$, the $m$th tensor power of $\mathfrak{h}$, is defined iteratively. For $m \geq 1$ and $p \geq 1$, define the space $\mathbb{D}^{m,p}$ to be the closure of $\mathcal{S}$ under the norm[5]

$$\|F\|_{\mathbb{D}^{m,p}}^{p} = \mathbb{E}\,|F|^p + \sum_{i=1}^{m} \mathbb{E}\, \left\| D^i F \right\|_{\mathfrak{h}^{\otimes i}}^{p}.$$

*Proof of Lemma 20.* In order to apply Thm. 1, we need to represent $Y = f(X)$ as a member of $\mathbb{D}^{2,4}$. Without loss of generality, we can assume that $\mathbb{E}\, X = 0$. Next, we represent $X$ in terms of an isonormal Gaussian process. Let $\mathfrak{h}$ be the Hilbert space $\mathbb{R}^n$ equipped with the standard dot product. Let $\xi$ be a standard Normal random vector taking values in $\mathbb{R}^n$. Define $Z(h) = \langle h, \xi \rangle_{\mathfrak{h}}$ for $h \in \mathfrak{h}$. Then $\{Z(h)\}_{h \in \mathfrak{h}}$ is an isonormal Gaussian process and $\xi_i = Z(e_i)$ for $i = 1, \ldots, n$, where $e_i$ is the $i$th standard basis vector. Next, we note that

$$X \stackrel{\mathcal{D}}{=} \Sigma_X^{1/2} \xi$$

$$\stackrel{\mathcal{D}}{=} \Sigma_X^{1/2} \sum_{i=1}^{n} \xi_i e_i.$$

Therefore, by approximating $f$ using smooth bounded functions, we can write

$$Y = f(X)$$

$$= f\left( \Sigma_X^{1/2} \sum_{j=1}^{n} \xi_j e_j \right)$$

The Malliavin derivative of $Y$ is given by

$$DY_k = \sum_{i=1}^{n} \frac{\partial}{\partial \xi_i} f_k \left( \Sigma_X^{1/2} \sum_{j=1}^{n} \xi_j e_j \right) e_i$$

$$= \sum_{i=1}^{n} \left\{ Df_k \left( \Sigma_X^{1/2} \sum_{j=1}^{n} \xi_j e_j \right) \left[ \frac{\partial}{\partial \xi_i} \left( \Sigma_X^{1/2} \sum_{j=1}^{n} \xi_j e_j \right) \right] \right\} e_i$$

$$= \sum_{i=1}^{n} \left\{ Df_k \left( \Sigma_X^{1/2} \sum_{j=1}^{n} \xi_j e_j \right) \left[ \Sigma_X^{1/2} e_i \right] \right\} e_i$$

$$= \Sigma_X^{1/2} \nabla f_k(X),$$

---

[5]Compare to the Sobolev space $W^{m,p}$.

and likewise the second Malliavin derivative is (identifying $\mathfrak{h} \otimes \mathfrak{h}$ with $\mathbb{R}^{n \times n}$)

$$D^2 Y_k = \Sigma_X^{1/2} \nabla^2 f_k(X) \Sigma_X^{1/2}.$$

Thus the conclusion of Thm. 1 becomes:

$$d_W(Y, \mathcal{N}_d(\mu_Y, \Sigma_T)) \leq \frac{3}{\sqrt{2}} \|\Sigma_Y^{-1}\| \|\Sigma_Y\|^{1/2}$$

$$\cdot \underbrace{\left( \sum_{i=1}^{d} \mathbb{E} \|D^2 Y_i\|_{\text{op}}^4 \right)^{1/4}}_{\leq \|\Sigma_X\|^{2/2} \left( \sum_{i=1}^{d} \mathbb{E} \|\nabla^2 f_i(X)\|^4 \right)^{1/4}}$$

$$\cdot \underbrace{\left( \sum_{i=1}^{d} \mathbb{E} \|DY_i\|_{\mathfrak{h}}^4 \right)^{1/4}}_{\leq \|\Sigma_X\|^{1/2} \left( \sum_{i=1}^{d} \mathbb{E} \|\nabla f_i(X)\|^4 \right)^{1/4}}.$$

$$\square$$

## J  SUPPLEMENT TO §5.2

A neural mapping $f : \mathbb{R}^n \to \mathbb{R}$ and a variance $\sigma^2$ are chosen to maximize the log-likelihood of the generative model

$$y \mid x \sim \mathcal{N}(f(x), \sigma^2) \tag{84}$$

for a set of inputs $\{x_i\}_{i=1}^N$ and outputs $\{y_i\}_{i=1}^N$. At inference time, the input $x$ is corrupted to $\hat{X} = x + \epsilon$, where $\epsilon \sim \mathcal{N}(0, \Sigma_\epsilon)$ is a perturbation with a known distribution. We make the prediction

$$\hat{Y} \mid \hat{X} = f(\hat{X}) + \mathcal{N}(0, \sigma^2) \tag{85}$$

which we approximate by

$$\hat{Y} \mid \hat{X} \approx \mathcal{N}(\mathbb{E}(f(x) \mid \hat{X}), \sigma^2 + \text{Var}(f(x) \mid \hat{X})). \tag{86}$$

We apply this to the California housing dataset consisting of roughly 20,000 $(x, y)$ pairs, in which $y$ is the log median house price of a block from the 1990 United States Census, and $x$ is a vector of eight features of that block consisting of latitude, longitude, and other demographic information. At inference time, we corrupt the input $x$ with Gaussian noise.

### J.1  DATA

The California Housing dataset (Kelley Pace & Barry, 1997) consists of 20460 census blocks from the 1990 United States Census. We split this dataset into roughly 70% training, 10% validation, and 20% test. We pre-processed this dataset by log-transforming some of the variables and standardizing them using the means and standard deviations of the training set. The regressors are:

- latitude of the block
- longitude of the block
- median income of the block
- (logarithm) average number of rooms per household
- (logarithm) average number of bedrooms per household
- (logarithm) population of the block
- (logarithm) number of households in the block

The predictor is the log median house price of the block.

### J.2  NETWORK

We used a single-hidden-layer neural network to parameterize a model consisting of a superposition of a linear term and a sinusoidal term:

$$f(x) = C \sin(Ax + b) + C'x + d, \tag{87}$$

implemented using two layers:

$$f(x) = \begin{pmatrix} C \\ C' \end{pmatrix} f^1(x) + d, \tag{88}$$

$$f^1(x) = \sin\left( \begin{pmatrix} A \\ 0 \end{pmatrix} x + \begin{pmatrix} b \\ 0 \end{pmatrix} \right) + \begin{pmatrix} 0 \\ 1 \end{pmatrix} x + \begin{pmatrix} 0 \\ 0 \end{pmatrix}, \tag{89}$$

The number of hidden neurons (the number of rows of $A$) is 7. This number was the result of hand-tuning to reduce the subjective appearance of overfitting in the validation set.

### J.3  TRAINING

We initialized $A$ with i.i.d. Gaussian entries having mean zero and variance equal to the reciprocal of the number of columns. We initialized $b$ with i.i.d. $\text{Uniform}(-\pi, \pi)$. We initialized $C$, $C'$, and $d$ to zero.

We trained this network using the Adam optimizer on Optax (DeepMind et al., 2020) with learning rate $10^{-1}$ for the first 2000 steps and $10^{-2}$ for 6000 steps.

## J.4 INFERENCE

At inference time, we corrupt the input with Gaussian noise with zero variance in the latitude and longitude features, and 10X population covariance in the other six features. The two sources of uncertainty in the Monte Carlo simulation are the randomness of the train/val/test split (as judged relative to a hypothetical population) and the test data augmentation. In order to report a standard error that lumps both sources of uncertainty, we obtain 100 bootstrap samples of the test set. Within each bootstrap sample, we corrupt each input with 100 realizations of noise.

# K    SUPPLEMENT TO §5.3

A neural mapping $f : \mathbb{R}^n \to \mathbb{R}$ is trained to maximize the log-likelihood of the generative model

$$y \mid x \sim \text{Bernoulli}(p) \tag{90}$$
$$p \coloneqq \Phi(f(x)) \tag{91}$$

for a set of inputs $\{x_i\}_{i=1}^N$ and outputs $\{y_i\}_{i=1}^N$.

At inference time, the input $x$ is corrupted to $z = Px$, where $P$ is a wide measurement matrix. The situation in which $P$ is rank deficient corresponds to missing features. In order to use the model (90) for inference, we impute the missing features using linear regression:

$$\hat{X} \mid z = \mathcal{N}(\hat{\mu}, \hat{\Sigma}), \tag{92a}$$
$$\hat{\mu} = \mu + \Sigma P^\intercal \left( P \Sigma P^\intercal \right)^{-1} (z - \mu), \tag{92b}$$
$$\hat{\Sigma} = \Sigma - \left( \Sigma P^\intercal \right) \left( P \Sigma P^\intercal \right)^{-1} \left( P \Sigma \right). \tag{92c}$$

Here, $\mu$ and $\Sigma$ are the population mean and variance-covariance matrix of $x$, estimated from the training data.

Afterwards we make the uncertainty-aware prediction

$$\hat{Y} \mid \hat{X} \sim \text{Bernoulli}(\hat{p}) \tag{93}$$
$$\hat{p} \coloneqq \mathbb{E}\left[ \Phi(f(x)) \mid \hat{X} \right]. \tag{94}$$

At inference time, we drop all features except "Operating Gross Margin" and impute the rest of the balance sheet and cash flow features using linear regression following (92).

## K.1    DATA

The Taiwanese bankruptcy dataset (Liang et al., 2016) consists of 6819 instances of Taiwanese companies from 1999 to 2009. There are 95 features consisting of balance sheet (assets and liabilities) and cash flow information. The target variable is a binary indicator of bankruptcy. We preprocessed this dataset by standardizing the features using the means and standard deviations of the training set. We split this dataset into roughly 70% training, 10% validation, and 20% test.

## K.2    NETWORK

We used a single-hidden-layer neural network with a width of 200, depth of 3, and sine activation function:

$$f(x) = C \sin(Ax + b) + d \tag{95}$$

where $A$ is a $200 \times 95$ matrix, $b$ is a $200 \times 1$ vector, $C$ is a $1 \times 200$ vector, and $d$ is a $1 \times 1$ vector. We initialized $A$ with i.i.d. Gaussian entries having mean zero and variance equal to the reciprocal of the number of columns. We initialized $b$ with i.i.d. $\text{Uniform}(-\pi, \pi)$. We initialized $C$ to zero and $d = \Phi^{-1}(\mathbb{E}_{\text{train}} \, y)$.

## K.3    TRAINING

We trained this network using the AdamW optimizer (Loshchilov & Hutter, 2019; DeepMind et al., 2020) with learning rate $10^{-5}$ for 5000 steps.

## K.4    REPORTING

We report the average log predicted probability (higher is better) of the correct label on test data in the Taiwanese bankruptcy dataset. We also report the standard error of the mean.

## K.5    ADDITIONAL FIGURES

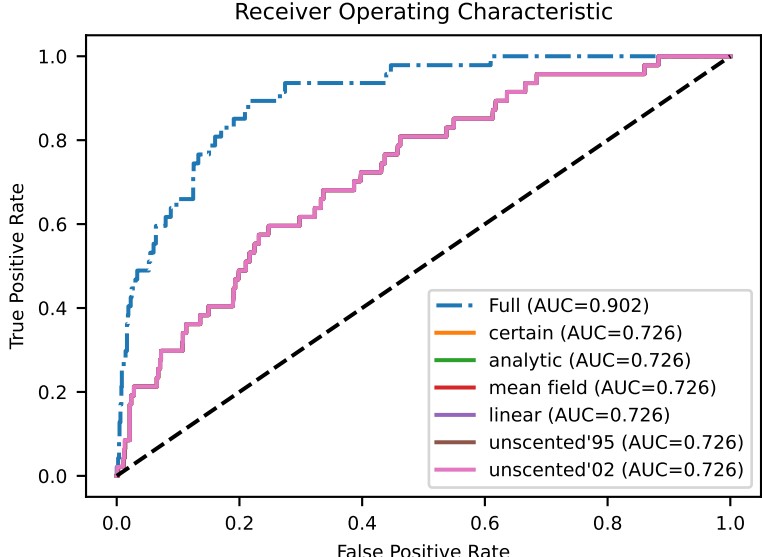

Figure 5: Receiver operator characteristic curve of test data in the Taiwanese bankruptcy dataset based on varying the threshold for $\hat{p}$.

## L  SUPPLEMENT TO §5.4

We expand on each of the terms in the evidence lower bound (2), repeated below.

$$\theta^* = \arg\min_\theta \left\{ \mathbb{E}_{w \sim q(w;\theta)} \log p(y \mid w, x) + D_{\text{KL},w}(q(w;\theta) \mid p(w)) \right\}.$$

Our neural network $f(x; w)$ consists of a single hidden layer. The weights $w$ encompass:

- $A$, $b$: pre-activation weights,
- $C$, $d$: post-activation weights, as well as
- $\lambda$: (homoscedastic) output log precision.

The log likelihood of $w$ is Normal:

$$\log p(y \mid w, x) = \frac{1}{2} \left( -\lambda + e^\lambda \left[ y - f(x; w) \right]^2 \right) + \frac{1}{2} \log \left[ 2\pi \right].$$

The prior distribution $p(w)$ is the Kaiming initialization of independent Normal distributions, zero mean in each matrix entry and variance equal to 2/fan-in. Like Wu et al. (2019); Petersen et al. (2024); Wright et al. (2024), the variational distribution $q(w; \theta)$ is a Gaussian matrix with independent entries, parameterized by means and log precisions.

We train using the Adam optimizer with a learning rate of 0.1 for 5 000 epochs. Each epoch draws 10 random samples from the variational distribution $q(w; \theta)$ per training instance and uses them to compute the gradient of the ELBO.

Each network takes about 2 minutes on a Nvidia T1200 GPU.

In reporting, we draw one million random variates from the variational posterior distribution for up to 100 instances from the test set.

The datasets are

- Combined cycle power plant (Pnar Tfekci, 2014)
- Concrete compressive strength (I-Cheng Yeh, 1998)
- Energy efficiency (we predict the heating load) (Athanasios Tsanas, 2012)
- Wine quality (Paulo Cortez, 2009)

## L.1 RESULTS

Table 6: KL divergence between ground truth predictive distribution (by Monte Carlo) and approximations. W24@$k$ means Wright et al. (2024) with $k$ terms in the series expansion. Mean and standard error of the mean (over the test set).

| Dataset | Method | KL divergence from GT |
|---|---|---|
| Combined Cycle Power Plant | analytic (ours) | $1.042 \times 10^{-6} \pm 1.0 \times 10^{-7}$ |
| | W24@5 | $2.460 \times 10^{-4} \pm 1.6 \times 10^{-5}$ |
| | W24@4 | $7.549 \times 10^{-4} \pm 6.0 \times 10^{-5}$ |
| | W24@3 | $1.336 \times 10^{-3} \pm 1.4 \times 10^{-4}$ |
| | W24@2 | $1.618 \times 10^{-3} \pm 2.0 \times 10^{-4}$ |
| | W24@1 | $1.794 \times 10^{-3} \pm 2.2 \times 10^{-4}$ |
| Concrete Compressive Strength | analytic (ours) | $1.100 \times 10^{-6} \pm 1.2 \times 10^{-7}$ |
| | W24@5 | $2.230 \times 10^{-4} \pm 2.3 \times 10^{-5}$ |
| | W24@4 | $2.973 \times 10^{-4} \pm 3.0 \times 10^{-5}$ |
| | W24@3 | $1.601 \times 10^{-3} \pm 1.7 \times 10^{-4}$ |
| | W24@2 | $9.629 \times 10^{-3} \pm 1.0 \times 10^{-3}$ |
| | W24@1 | $2.520 \times 10^{-2} \pm 3.2 \times 10^{-3}$ |
| Energy Efficiency | analytic (ours) | $1.069 \times 10^{-6} \pm 1.4 \times 10^{-7}$ |
| | W24@5 | $2.041 \times 10^{-4} \pm 1.2 \times 10^{-5}$ |
| | W24@4 | $2.459 \times 10^{-4} \pm 1.3 \times 10^{-5}$ |
| | W24@3 | $1.542 \times 10^{-3} \pm 7.8 \times 10^{-5}$ |
| | W24@2 | $7.640 \times 10^{-3} \pm 4.5 \times 10^{-4}$ |
| | W24@1 | $1.450 \times 10^{-2} \pm 9.5 \times 10^{-4}$ |
| Wine Quality | analytic (ours) | $1.616 \times 10^{-6} \pm 1.9 \times 10^{-7}$ |
| | W24@5 | $5.292 \times 10^{-4} \pm 4.2 \times 10^{-5}$ |
| | W24@4 | $1.764 \times 10^{-3} \pm 1.2 \times 10^{-4}$ |
| | W24@3 | $7.337 \times 10^{-3} \pm 5.8 \times 10^{-4}$ |
| | W24@2 | $1.672 \times 10^{-2} \pm 1.7 \times 10^{-3}$ |
| | W24@1 | $2.876 \times 10^{-2} \pm 4.1 \times 10^{-3}$ |

Table 7: Train and test log-likelihoods of the Bayesian networks for all datasets. We do not report experimental uncertainty, as these networks as taken as fixed for the purpose of approximating the output distribution.

| Dataset | Train log-likelihood | Test log-likelihood |
|---|---|---|
| Energy Efficiency | $-5.532 \times 10^{-1}$ | $-6.029 \times 10^{-1}$ |
| Concrete Compressive Strength | $-9.909 \times 10^{-1}$ | $-9.253 \times 10^{-1}$ |
| Wine Quality | $-1.292 \times 10^{0}$ | $-1.242 \times 10^{0}$ |
| Combined Cycle Power Plant | $-1.621 \times 10^{-1}$ | $-9.716 \times 10^{-2}$ |

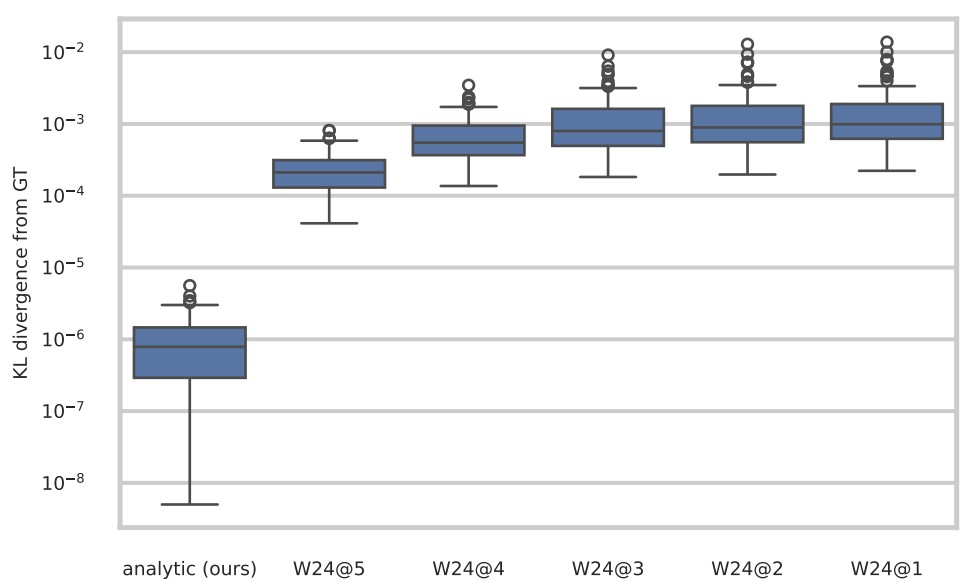

Figure 6: KL divergence between ground truth predictive distribution (by Monte Carlo) and approximations for the combined cycle power plant dataset. W24@$k$ means Wright et al. (2024) with $k$ terms in the series expansion.

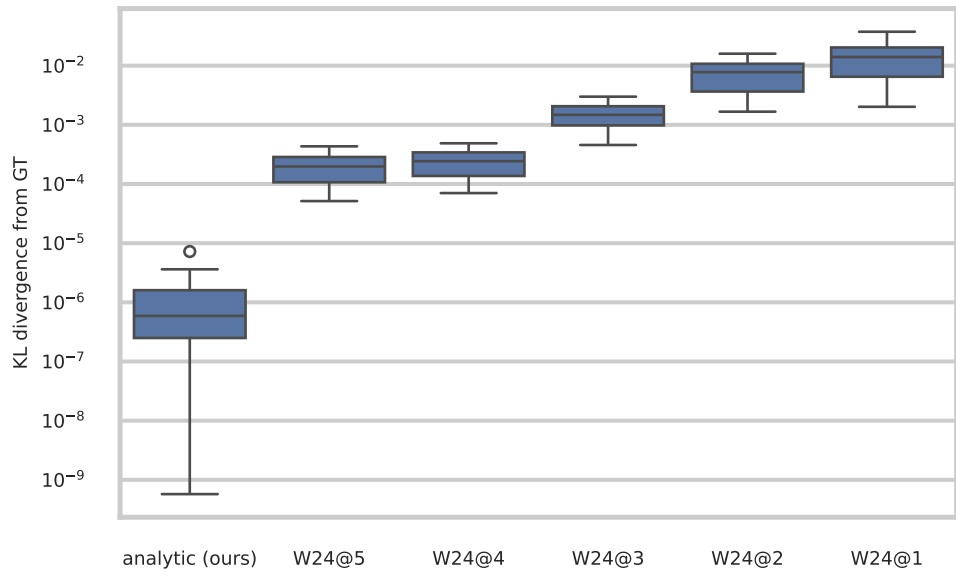

Figure 7: KL divergence between ground truth predictive distribution (by Monte Carlo) and approximations for the energy efficiency dataset. W24@$k$ means Wright et al. (2024) with $k$ terms in the series expansion.

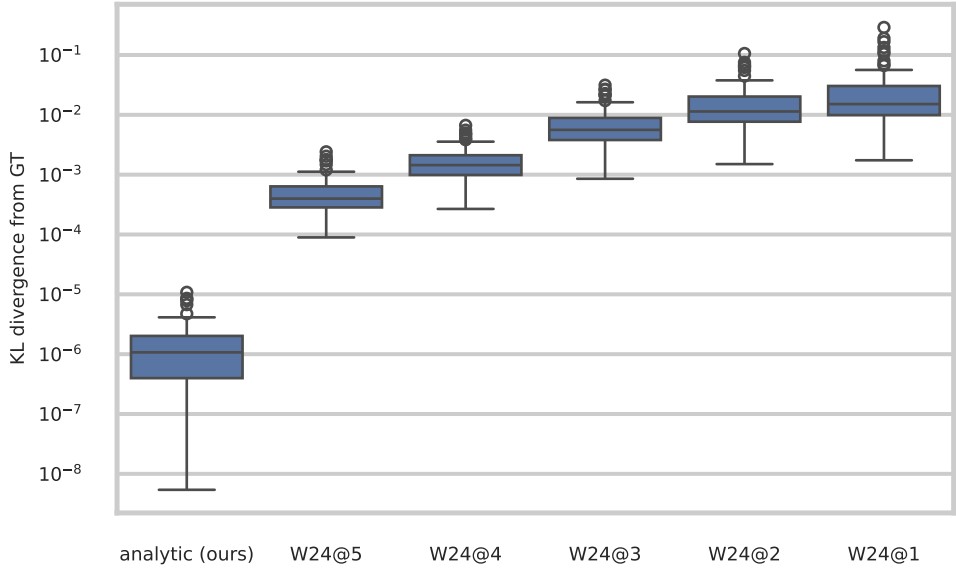

Figure 8: KL divergence between ground truth predictive distribution (by Monte Carlo) and approximations for the wine quality dataset. W24@$k$ means Wright et al. (2024) with $k$ terms in the series expansion.

## M  SUPPLEMENT TO §5.5

The random variable $\mathbf{1}_{U<\Phi(x)}$ has a Bernoulli distribution with parameter $\Phi(x)$, which contributes an additional diagonal variance term to Lemma 1 as we see from applying the Law of Total Covariance:

$$\left(\operatorname{Cov} g_{\tilde{\sigma}}(X; A, b, C, d)\right)_{i,j} = \left(\operatorname{Cov} \mathbb{E}\left[g_{\tilde{\sigma}}(X; A, b, C, d) \mid X\right]\right)_{i,j}$$

$$+ \left(\mathbb{E}\operatorname{Cov}\left[g_{\tilde{\sigma}}(X; A, b, C, d) \mid X\right]\right)_{i,j} \quad (96)$$

$$\left(\operatorname{Cov}\mathbb{E}\left[g_{\tilde{\sigma}}(X; A, b, C, d) \mid X\right]\right)_{i,j} = \left(\operatorname{Cov} g_{\sigma}(X; A, b, C, d)\right)_{i,j} \quad (97)$$

since $\mathbb{E}\left(\tilde{\sigma}(X, U) \mid X\right) = \sigma(X)$ and

$$\left(\mathbb{E}\operatorname{Cov}\left[g_{\tilde{\sigma}}(X; A, b, C, d) \mid X\right]\right)_{i,j} = 4\delta_{ij}\,\mathbb{E}\operatorname{Var}\left[\left(g_{\tilde{\sigma}}(X; A, b, C, d)\right)_i \mid X\right] \quad (98)$$

$$= 4\delta_{ij}\,\mathbb{E}\,\Phi(\xi_i)\left(1 - \Phi(\xi_i)\right) \quad (99)$$

where $\xi_i \sim \mathcal{N}(\mu_i, \nu_{ii})$. Using Owen's T function (Owen, 1980), we have

$$\mathbb{E}\,\Phi(\xi_i)\left(1 - \Phi(\xi_i)\right) = 2T\left(\frac{\mu_i}{\sqrt{1+\nu_{ii}}}, \frac{\nu_{ii}}{1+2\nu_{ii}}\right). \quad (100)$$

# N    SUPPLEMENT TO §5.1

We apply our method and other benchmarks to 38 different ensembles of random neural networks:

- network architecture $\in \{\text{wide}, \text{deep}\}$
- weights $\in \{\text{initialized}, \text{trained}\}$
- activation function $\in \{\text{probit}, \text{probit residual}, \text{sine}, \text{sine residual}\}$

(There are no trained networks containing Heaviside or Heaviside-residual layers.) From each ensemble, we sample one neural network and evaluate the goodness of approximation of the output distribution for input distributions:

- variance $\in \{\text{small}, \text{medium}, \text{large}\}$.

In each case we compare the distributions of:

- $Y_0$, the true distribution
- $Y_1$, the pseudo-true Gaussian distribution
- $Y_{\text{ana}}$, the analytic layer-wise Gaussian approximation (our method)
- $Y_{\text{mfa}}$, mean-field: applying our method for moment propagation and setting off-diagonal layer covariances to zero
- $Y_{\text{lin}}$, linearization-based moment propagation
- $Y_{\text{u}'95}$, unscented transform of the whole network using $\kappa = 2$
- $Y_{\text{u}'02}$, unscented transform of the whole network using $\alpha = 0.001, \beta = 2, \kappa = 2$

## ENSEMBLES OF NEURAL NETWORKS

This section specifies the 38 ensembles of random neural networks and the three ensembles of inputs used to produce the 114 test cases that follow. The neural networks in this example are parameterized as in Def. 1. Let $d_{\text{hidden}}$ and $w_{\text{hidden}}$ be the depth and width of the hidden layers, respectively. A random neural network $f : \mathbb{R}^3 \to \mathbb{R}$ is specified with $d_{\text{hidden}} + 1$ layers. The output layer is linear, with weights $C^{d_{\text{hidden}}+1}$ and biases $d^{d_{\text{hidden}}+1}$. The first hidden layer has $C^1 = 0_{w_{\text{hidden}} \times 1}$, $d^1 = 0_{w_{\text{hidden}} \times 1}$. The four degrees of freedom in the test cases are:

**architecture**
- if architecture = wide, then $d_{\text{hidden}} = 5$, $w_{\text{hidden}} = 400$;
- if architecture = deep, then $d_{\text{hidden}} = 20$, $w_{\text{hidden}} = 100$.

**weights**
- if weights = initialized:
  - $A$ matrices are initialized with i.i.d. Gaussian entries having mean zero and variance equal to the reciprocal of the number of columns times $\sqrt{2}$
  - if activation $\in \{\text{probit}, \text{probit residual}\}$, then $b$ vectors are initialized with independently sampled entries from $\mathcal{N}(0, 1)$
  - if activation $\in \{\text{sine}, \text{sine residual}\}$, then $b$ vectors are initialized with independently sampled entries from $\mathcal{U}(-\pi, \pi)$
  - if activation $\in \{\text{probit}, \text{sine}\}$, then $C$ matrices are initialized to the zero matrix.
  - if activation $\in \{\text{probit residual}, \text{sine residual}\}$, then square $C$ matrices are initialized to the identity matrix, and all other $C$ matrices are initialized to the zero matrix.
  - $d$ vectors are initialized to the zero vector.
- if weights = trained and activation $\notin \{\text{Heaviside}, \text{Heaviside residual}\}$, the following initialization as above, the neural network is trained to minimize the mean squared error loss on a pseudorandomly generated dataset of ten $(x, y)$ samples drawn from $\mathcal{N}(0, 1)$. Training consists of using the AdamW optimizer (Loshchilov & Hutter, 2019) with a learning rate of $10^{-6}$ for 30,000 iterations and until the loss is less than $10^{-8}$ (whichever is later). Implementation due to Optax, DeepMind et al. (2020)

**activation function**
- if activation $\in$ {probit, probit residual}, then $\sigma(x) = 2\Phi(x) - 1$ where $\Phi$ is the cumulative distribution function of the standard normal distribution
- if activation $\in$ {sine, sine residual}, then $\sigma(x) = \sin(x)$
- if activation $\in$ {GeLU, GeLU residual}, then $\sigma(x) = x\Phi(x)$.
- if activation $\in$ {ReLU, ReLU residual}, then $\sigma(x) = \max(0, x)$.
- if activation $\in$ {Heaviside, Heaviside residual}, then $\sigma(x) = \mathbf{1}_{\{x \geq 0\}}$

**variance**
- if variance = small, then $X \sim \mathcal{N}(0, 10^{-2}I)$
- if variance = medium, then $X \sim \mathcal{N}(0, I)$
- if variance = large, then $X \sim \mathcal{N}(0, 10^2 I)$

SIMULATION AND REPORTING

The only source of uncertainty in our numerical results is the "true distribution" of $Y_0 = f(X)$. For this we use twenty independent realizations of $N = 2^{16}$ quasi-Monte Carlo samples (Virtanen et al., 2020). We report statistical uncertainty in the form of mean $\pm$ standard error within the independent realizations. The distributional uncertainty is too small to visualize in figures, so we plot the pooled data without an uncertainty indication.

The Wasserstein distance between a Normal distribution and $Y_0$ is computed by

$$d_{\mathrm{W}}(\mathcal{N}(\mu, \sigma^2), Y_0) \approx \frac{1}{N} \sum_{i=1}^{N} \left| y_{(i)} - Q\left(\frac{i - 1/2}{N}\right) \right|$$

where $\{y_{(i)}\}_{i=1}^{N}$ are the quasi-Monte Carlo samples of $Y_0$ sorted in ascending order, and $Q$ is the quantile function of $\mathcal{N}(\mu, \sigma^2)$. When plotting distributions, we show $Y_0$ using a histogram with 50 bins.

The horizontal axes are scaled to include the 0.5th (99.5th) percentiles of the quasi-Monte Carlo samples of $Y_0$, or the mean minus (plus) three standard deviations of $Y_1$, whichever is smaller (greater).

The entire suite of 72 test cases takes (including initialization, training, quasi-Monte Carlo, and reporting) 18 minutes on an Ubuntu system with a 11th Gen Intel® Core™ i7-11850H CPU and 32GB of RAM.

## N.1 SUMMARIES

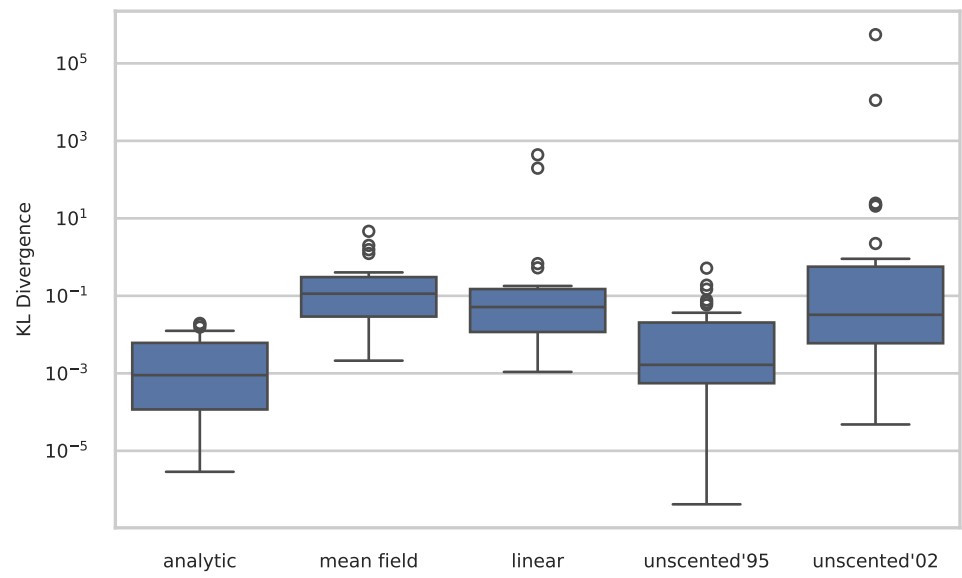

Figure 9: Comparison of goodness of approximation (lower KL divergence is better) for all random neural networks, grouped by approximation method, in the small input variance scenario.

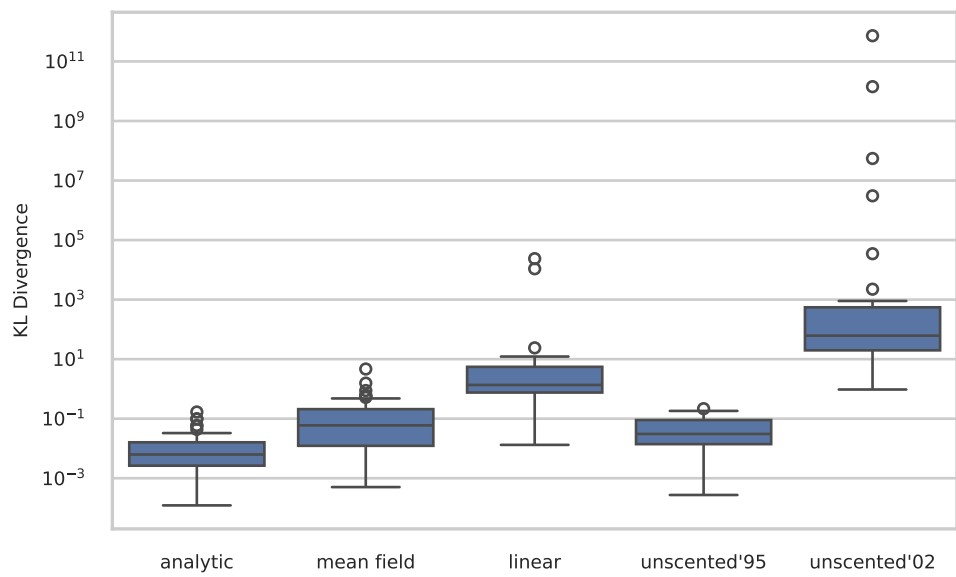

Figure 10: Comparison of goodness of approximation (lower KL divergence is better) for all random neural networks, grouped by approximation method, in the medium input variance scenario.

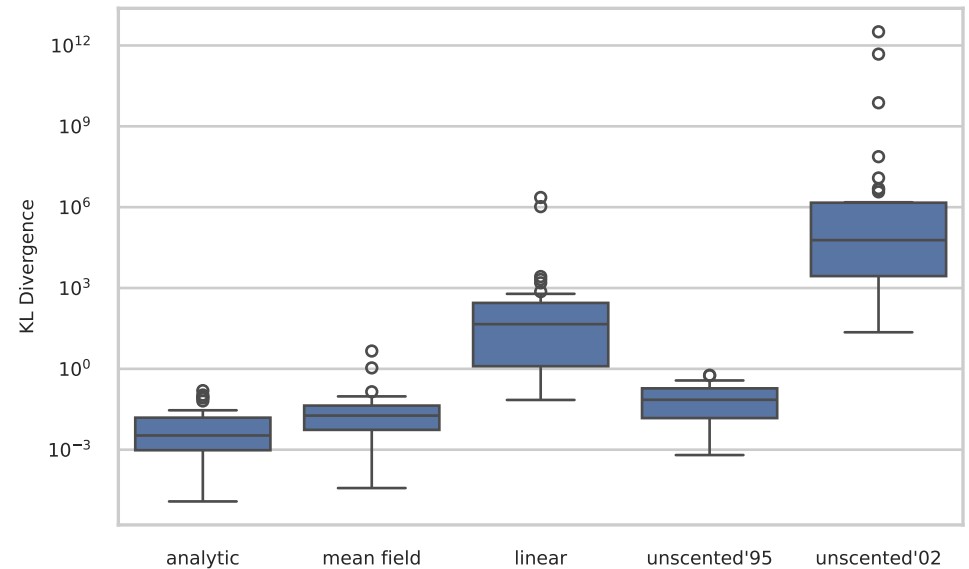

Figure 11: Comparison of goodness of approximation (lower KL divergence is better) for all random neural networks, grouped by approximation method, in the large input variance scenario.

| distribution | $\mu$ | $\sigma^2$ |
|---|---|---|
| pseudo-true $(Y_1)$ | $-3.856 \times 10^{-1} \pm 2.5 \times 10^{-8}$ | $4.863 \times 10^{-4} \pm 7.6 \times 10^{-9}$ |
| analytic | $-3.856 \times 10^{-1}$ | $4.632 \times 10^{-4}$ |
| mean-field | $-3.843 \times 10^{-1}$ | $7.883 \times 10^{-4}$ |
| linear | $-3.782 \times 10^{-1}$ | $4.533 \times 10^{-4}$ |
| unscented'95 | $-3.856 \times 10^{-1}$ | $4.485 \times 10^{-4}$ |
| unscented'02 | $-3.858 \times 10^{-1}$ | $5.698 \times 10^{-4}$ |

Table 8: Comparison of moments for Network(architecture=wide, weights=initialized, activation=probit), variance=small

| distribution | $d_{\mathrm{W}}(\cdot, Y_0)$ | $D_{\mathrm{KL}}(Y_1 \parallel \cdot)$ |
|---|---|---|
| pseudo-true $(Y_1)$ | $1.745 \times 10^{-2} \pm 9.5 \times 10^{-7}$ | $0$ |
| analytic | $1.738 \times 10^{-2} \pm 1.2 \times 10^{-6}$ | $5.828 \times 10^{-4} \pm 3.7 \times 10^{-7}$ |
| mean-field | $3.560 \times 10^{-2} \pm 1.3 \times 10^{-6}$ | $7.064 \times 10^{-2} \pm 4.9 \times 10^{-6}$ |
| linear | $4.981 \times 10^{-2} \pm 1.9 \times 10^{-7}$ | $5.747 \times 10^{-2} \pm 3.8 \times 10^{-7}$ |
| unscented'95 | $1.756 \times 10^{-2} \pm 9.0 \times 10^{-7}$ | $1.598 \times 10^{-3} \pm 6.1 \times 10^{-7}$ |
| unscented'02 | $2.102 \times 10^{-2} \pm 7.5 \times 10^{-7}$ | $6.681 \times 10^{-3} \pm 1.4 \times 10^{-6}$ |

Table 9: Comparison of statistical distances for Network(architecture=wide, weights=initialized, activation=probit), variance=small

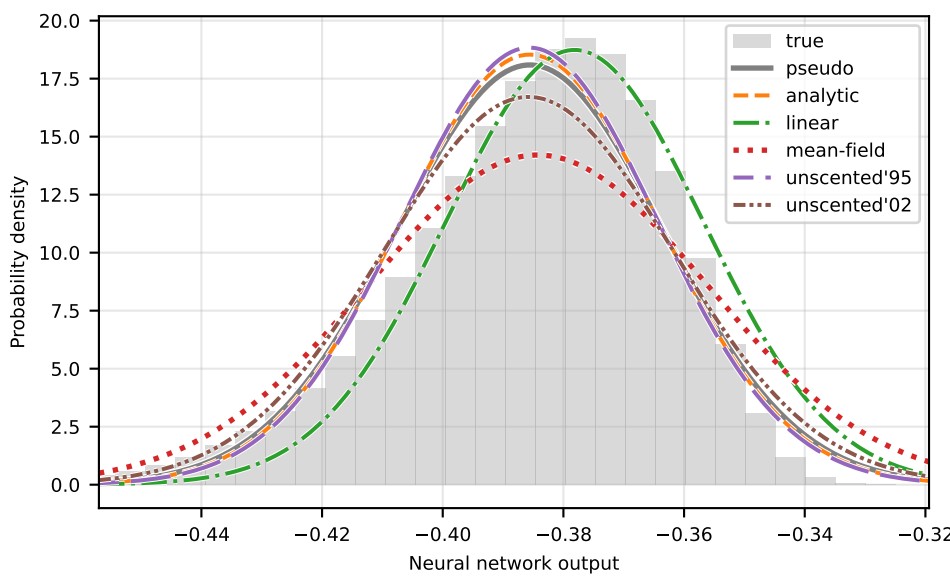

Figure 12: Probability distributions for Network(architecture=wide, weights=initialized, activation=probit), variance=small

| distribution | $\mu$ | $\sigma^2$ |
| --- | --- | --- |
| pseudo-true ($Y_1$) | $-5.609 \times 10^{-1} \pm 5.2 \times 10^{-7}$ | $1.946 \times 10^{-2} \pm 2.2 \times 10^{-7}$ |
| analytic | $-5.710 \times 10^{-1}$ | $1.773 \times 10^{-2}$ |
| mean-field | $-5.412 \times 10^{-1}$ | $2.545 \times 10^{-2}$ |
| linear | $-3.782 \times 10^{-1}$ | $4.533 \times 10^{-2}$ |
| unscented'95 | $-5.943 \times 10^{-1}$ | $8.656 \times 10^{-3}$ |
| unscented'02 | $-1.141 \times 10^{0}$ | $1.210 \times 10^{0}$ |

Table 10: Comparison of moments for Network(architecture=wide, weights=initialized, activation=probit), variance=medium

| distribution | $d_{\mathrm{W}}(\cdot, Y_0)$ | $D_{\mathrm{KL}}(Y_1 \parallel \cdot)$ |
| --- | --- | --- |
| pseudo-true ($Y_1$) | $3.924 \times 10^{-2} \pm 1.7 \times 10^{-6}$ | $0$ |
| analytic | $5.182 \times 10^{-2} \pm 3.1 \times 10^{-6}$ | $4.751 \times 10^{-3} \pm 6.0 \times 10^{-7}$ |
| mean-field | $5.895 \times 10^{-2} \pm 2.6 \times 10^{-6}$ | $2.964 \times 10^{-2} \pm 2.1 \times 10^{-6}$ |
| linear | $4.914 \times 10^{-1} \pm 2.3 \times 10^{-6}$ | $1.099 \times 10^{0} \pm 1.9 \times 10^{-5}$ |
| unscented'95 | $1.385 \times 10^{-1} \pm 3.3 \times 10^{-6}$ | $1.562 \times 10^{-1} \pm 3.2 \times 10^{-6}$ |
| unscented'02 | $2.423 \times 10^{0} \pm 8.2 \times 10^{-6}$ | $3.719 \times 10^{+1} \pm 4.5 \times 10^{-4}$ |

Table 11: Comparison of statistical distances for Network(architecture=wide, weights=initialized, activation=probit), variance=medium

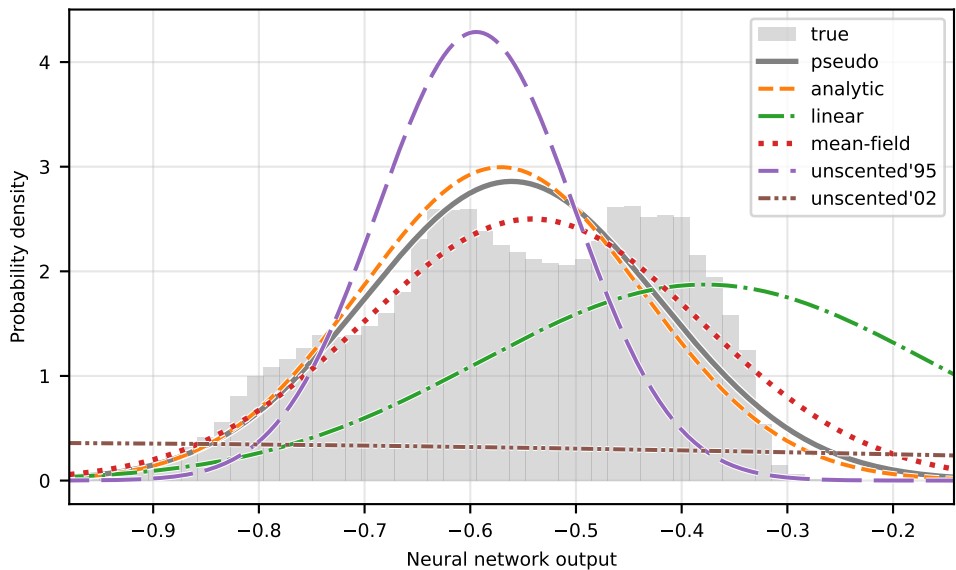

Figure 13: Probability distributions for Network(architecture=wide, weights=initialized, activation=probit), variance=medium

| distribution | $\mu$ | $\sigma^2$ |
|---|---|---|
| pseudo-true ($Y_1$) | $-6.141 \times 10^{-1} \pm 9.1 \times 10^{-6}$ | $3.372 \times 10^{-2} \pm 2.8 \times 10^{-6}$ |
| analytic | $-6.263 \times 10^{-1}$ | $3.250 \times 10^{-2}$ |
| mean-field | $-5.718 \times 10^{-1}$ | $4.610 \times 10^{-2}$ |
| linear | $-3.782 \times 10^{-1}$ | $4.533 \times 10^{0}$ |
| unscented'95 | $-6.768 \times 10^{-1}$ | $4.894 \times 10^{-2}$ |
| unscented'02 | $-7.669 \times 10^{+1}$ | $1.165 \times 10^{+4}$ |

Table 12: Comparison of moments for Network(architecture=wide, weights=initialized, activation=probit), variance=large

| distribution | $d_{\mathrm{W}}(\cdot, Y_0)$ | $D_{\mathrm{KL}}(Y_1 \parallel \cdot)$ |
|---|---|---|
| pseudo-true ($Y_1$) | $2.457 \times 10^{-2} \pm 1.6 \times 10^{-5}$ | $0$ |
| analytic | $3.666 \times 10^{-2} \pm 1.7 \times 10^{-5}$ | $2.566 \times 10^{-3} \pm 3.5 \times 10^{-6}$ |
| mean-field | $1.093 \times 10^{-1} \pm 2.0 \times 10^{-5}$ | $5.363 \times 10^{-2} \pm 2.0 \times 10^{-5}$ |
| linear | $3.637 \times 10^{0} \pm 8.9 \times 10^{-5}$ | $6.508 \times 10^{+1} \pm 5.6 \times 10^{-3}$ |
| unscented'95 | $1.561 \times 10^{-1} \pm 2.3 \times 10^{-5}$ | $9.772 \times 10^{-2} \pm 3.0 \times 10^{-5}$ |
| unscented'02 | $2.486 \times 10^{+2} \pm 5.2 \times 10^{-3}$ | $2.585 \times 10^{+5} \pm 2.1 \times 10^{+1}$ |

Table 13: Comparison of statistical distances for Network(architecture=wide, weights=initialized, activation=probit), variance=large

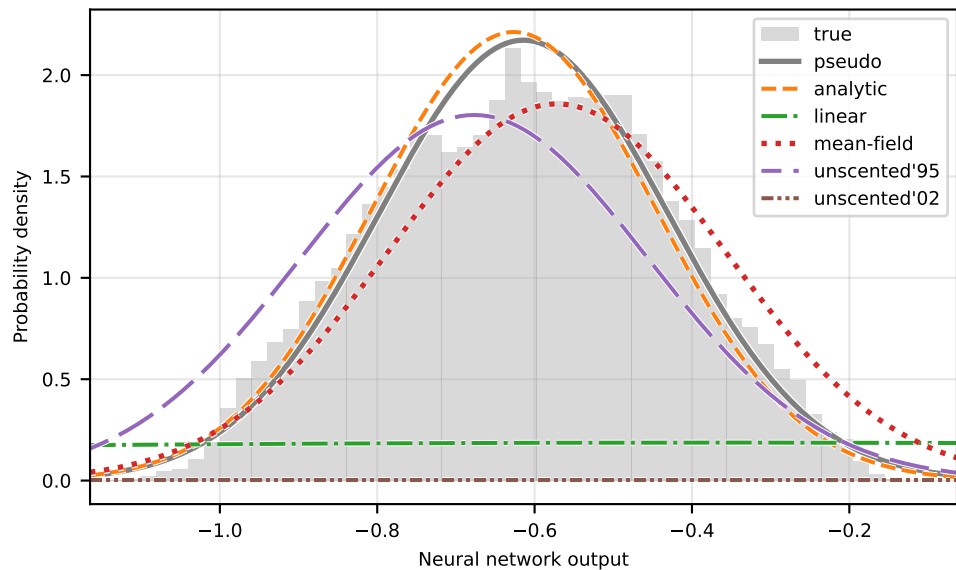

Figure 14: Probability distributions for Network(architecture=wide, weights=initialized, activation=probit), variance=large

| distribution | $\mu$ | $\sigma^2$ |
|---|---|---|
| pseudo-true ($Y_1$) | $+2.698 \times 10^{-1} \pm 3.2 \times 10^{-7}$ | $1.954 \times 10^{-1} \pm 1.2 \times 10^{-6}$ |
| analytic | $+2.691 \times 10^{-1}$ | $1.851 \times 10^{-1}$ |
| mean-field | $+3.184 \times 10^{-1}$ | $1.325 \times 10^{-3}$ |
| linear | $+3.554 \times 10^{-1}$ | $1.977 \times 10^{-1}$ |
| unscented'95 | $+2.673 \times 10^{-1}$ | $1.934 \times 10^{-1}$ |
| unscented'02 | $+2.623 \times 10^{-1}$ | $2.151 \times 10^{-1}$ |

Table 14: Comparison of moments for Network(architecture=wide, weights=trained, activation=probit), variance=small

| distribution | $d_{\mathrm{W}}(\cdot, Y_0)$ | $D_{\mathrm{KL}}(Y_1 \parallel \cdot)$ |
|---|---|---|
| pseudo-true ($Y_1$) | $8.960 \times 10^{-2} \pm 2.4 \times 10^{-6}$ | $0$ |
| analytic | $9.035 \times 10^{-2} \pm 2.6 \times 10^{-6}$ | $7.259 \times 10^{-4} \pm 1.6 \times 10^{-7}$ |
| mean-field | $4.805 \times 10^{-1} \pm 2.4 \times 10^{-6}$ | $2.006 \times 10^{0} \pm 3.0 \times 10^{-6}$ |
| linear | $1.288 \times 10^{-1} \pm 3.6 \times 10^{-7}$ | $1.879 \times 10^{-2} \pm 1.0 \times 10^{-7}$ |
| unscented'95 | $9.089 \times 10^{-2} \pm 2.5 \times 10^{-6}$ | $4.188 \times 10^{-5} \pm 2.7 \times 10^{-8}$ |
| unscented'02 | $9.728 \times 10^{-2} \pm 2.0 \times 10^{-6}$ | $2.525 \times 10^{-3} \pm 3.1 \times 10^{-7}$ |

Table 15: Comparison of statistical distances for Network(architecture=wide, weights=trained, activation=probit), variance=small

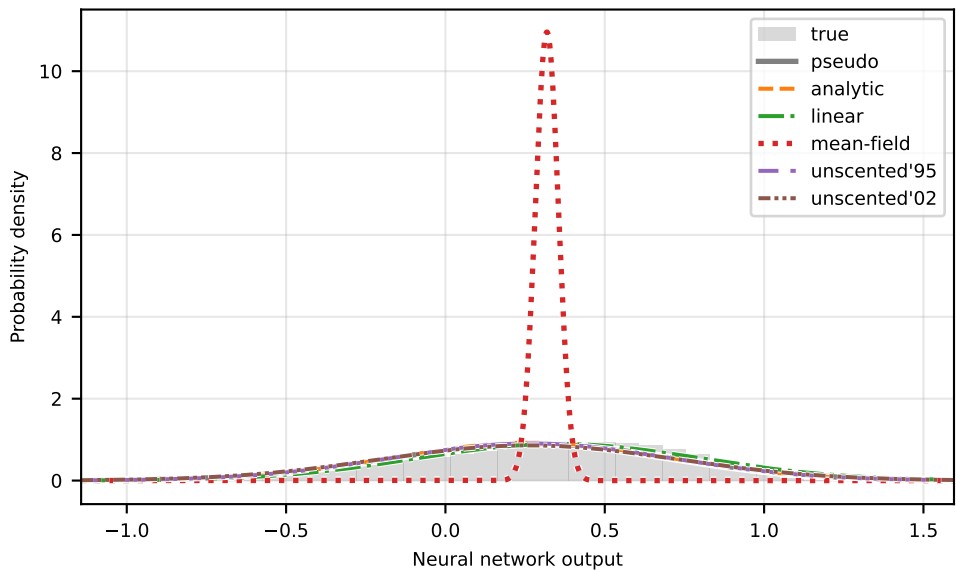

Figure 15: Probability distributions for Network(architecture=wide, weights=trained, activation=probit), variance=small

| distribution | $\mu$ | $\sigma^2$ |
|---|---|---|
| pseudo-true ($Y_1$) | $-8.887 \times 10^{-1} \pm 1.6 \times 10^{-6}$ | $2.394 \times 10^0 \pm 6.7 \times 10^{-6}$ |
| analytic | $-8.854 \times 10^{-1}$ | $1.654 \times 10^0$ |
| mean-field | $-6.635 \times 10^{-1}$ | $4.026 \times 10^{-2}$ |
| linear | $+3.554 \times 10^{-1}$ | $1.977 \times 10^{+1}$ |
| unscented'95 | $-9.842 \times 10^{-1}$ | $2.938 \times 10^0$ |
| unscented'02 | $-8.960 \times 10^0$ | $1.933 \times 10^{+2}$ |

Table 16: Comparison of moments for Network(architecture=wide, weights=trained, activation=probit), variance=medium

| distribution | $d_{\mathrm{W}}(\cdot, Y_0)$ | $D_{\mathrm{KL}}(Y_1 \parallel \cdot)$ |
|---|---|---|
| pseudo-true ($Y_1$) | $2.316 \times 10^{-1} \pm 1.0 \times 10^{-5}$ | $0$ |
| analytic | $2.970 \times 10^{-1} \pm 1.0 \times 10^{-5}$ | $3.033 \times 10^{-2} \pm 4.3 \times 10^{-7}$ |
| mean-field | $9.183 \times 10^{-1} \pm 7.8 \times 10^{-6}$ | $1.562 \times 10^0 \pm 1.3 \times 10^{-6}$ |
| linear | $1.915 \times 10^0 \pm 1.1 \times 10^{-5}$ | $2.898 \times 10^0 \pm 1.1 \times 10^{-5}$ |
| unscented'95 | $2.453 \times 10^{-1} \pm 9.7 \times 10^{-6}$ | $1.318 \times 10^{-2} \pm 3.2 \times 10^{-7}$ |
| unscented'02 | $9.627 \times 10^0 \pm 9.8 \times 10^{-6}$ | $5.129 \times 10^{+1} \pm 1.5 \times 10^{-4}$ |

Table 17: Comparison of statistical distances for Network(architecture=wide, weights=trained, activation=probit), variance=medium

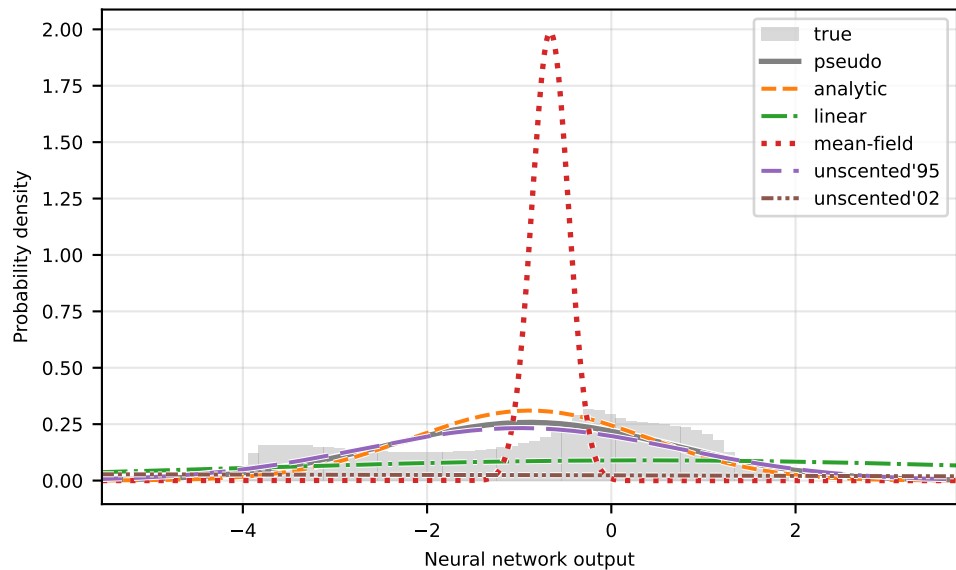

Figure 16: Probability distributions for Network(architecture=wide, weights=trained, activation=probit), variance=medium

| distribution | $\mu$ | $\sigma^2$ |
|---|---|---|
| pseudo-true ($Y_1$) | $-1.376 \times 10^0 \pm 1.6 \times 10^{-5}$ | $1.342 \times 10^0 \pm 3.8 \times 10^{-5}$ |
| analytic | $-1.316 \times 10^0$ | $9.442 \times 10^{-1}$ |
| mean-field | $-1.109 \times 10^0$ | $6.238 \times 10^{-2}$ |
| linear | $+3.554 \times 10^{-1}$ | $1.977 \times 10^{+3}$ |
| unscented'95 | $-1.393 \times 10^0$ | $1.282 \times 10^0$ |
| unscented'02 | $-9.306 \times 10^{+2}$ | $1.735 \times 10^{+6}$ |

Table 18: Comparison of moments for Network(architecture=wide, weights=trained, activation=probit), variance=large

| distribution | $d_{\mathrm{W}}(\cdot, Y_0)$ | $D_{\mathrm{KL}}(Y_1 \parallel \cdot)$ |
|---|---|---|
| pseudo-true ($Y_1$) | $1.432 \times 10^{-1} \pm 1.5 \times 10^{-5}$ | $0$ |
| analytic | $2.051 \times 10^{-1} \pm 1.4 \times 10^{-5}$ | $2.896 \times 10^{-2} \pm 4.2 \times 10^{-6}$ |
| mean-field | $7.035 \times 10^{-1} \pm 1.5 \times 10^{-5}$ | $1.084 \times 10^0 \pm 1.3 \times 10^{-5}$ |
| linear | $3.209 \times 10^{+1} \pm 2.4 \times 10^{-4}$ | $7.334 \times 10^{+2} \pm 2.1 \times 10^{-2}$ |
| unscented'95 | $1.550 \times 10^{-1} \pm 1.4 \times 10^{-5}$ | $6.325 \times 10^{-4} \pm 6.7 \times 10^{-7}$ |
| unscented'02 | $1.209 \times 10^{+3} \pm 8.5 \times 10^{-3}$ | $9.681 \times 10^{+5} \pm 2.7 \times 10^{+1}$ |

Table 19: Comparison of statistical distances for Network(architecture=wide, weights=trained, activation=probit), variance=large

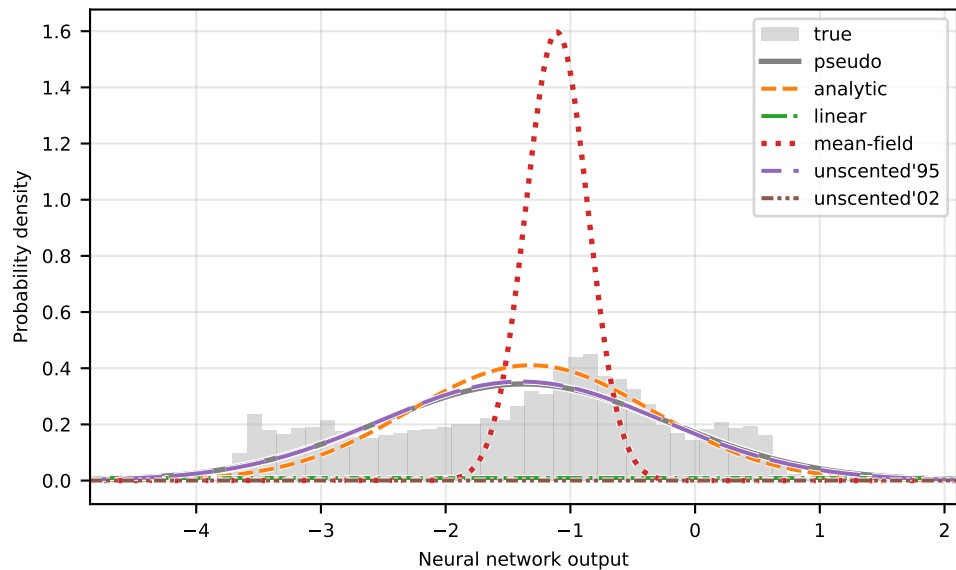

Figure 17: Probability distributions for Network(architecture=wide, weights=trained, activation=probit), variance=large

| distribution | $\mu$ | $\sigma^2$ |
|---|---|---|
| pseudo-true ($Y_1$) | $+1.754 \times 10^{-1} \pm 1.8 \times 10^{-7}$ | $4.923 \times 10^{-2} \pm 3.3 \times 10^{-7}$ |
| analytic | $+1.754 \times 10^{-1}$ | $4.870 \times 10^{-2}$ |
| mean-field | $+1.775 \times 10^{-1}$ | $3.412 \times 10^{-2}$ |
| linear | $+1.858 \times 10^{-1}$ | $5.340 \times 10^{-2}$ |
| unscented'95 | $+1.749 \times 10^{-1}$ | $5.137 \times 10^{-2}$ |
| unscented'02 | $+1.746 \times 10^{-1}$ | $5.365 \times 10^{-2}$ |

Table 20: Comparison of moments for Network(architecture=wide, weights=initialized, activation=probit residual), variance=small

| distribution | $d_{\mathrm{W}}(\cdot, Y_0)$ | $D_{\mathrm{KL}}(Y_1 \parallel \cdot)$ |
|---|---|---|
| pseudo-true ($Y_1$) | $2.925 \times 10^{-2} \pm 3.7 \times 10^{-6}$ | $0$ |
| analytic | $2.948 \times 10^{-2} \pm 3.1 \times 10^{-6}$ | $2.927 \times 10^{-5} \pm 3.6 \times 10^{-8}$ |
| mean-field | $6.750 \times 10^{-2} \pm 3.0 \times 10^{-6}$ | $2.988 \times 10^{-2} \pm 1.0 \times 10^{-6}$ |
| linear | $2.734 \times 10^{-2} \pm 3.9 \times 10^{-6}$ | $2.807 \times 10^{-3} \pm 2.8 \times 10^{-7}$ |
| unscented'95 | $2.957 \times 10^{-2} \pm 2.6 \times 10^{-6}$ | $4.640 \times 10^{-4} \pm 1.5 \times 10^{-7}$ |
| unscented'02 | $3.159 \times 10^{-2} \pm 4.1 \times 10^{-6}$ | $1.908 \times 10^{-3} \pm 3.0 \times 10^{-7}$ |

Table 21: Comparison of statistical distances for Network(architecture=wide, weights=initialized, activation=probit residual), variance=small

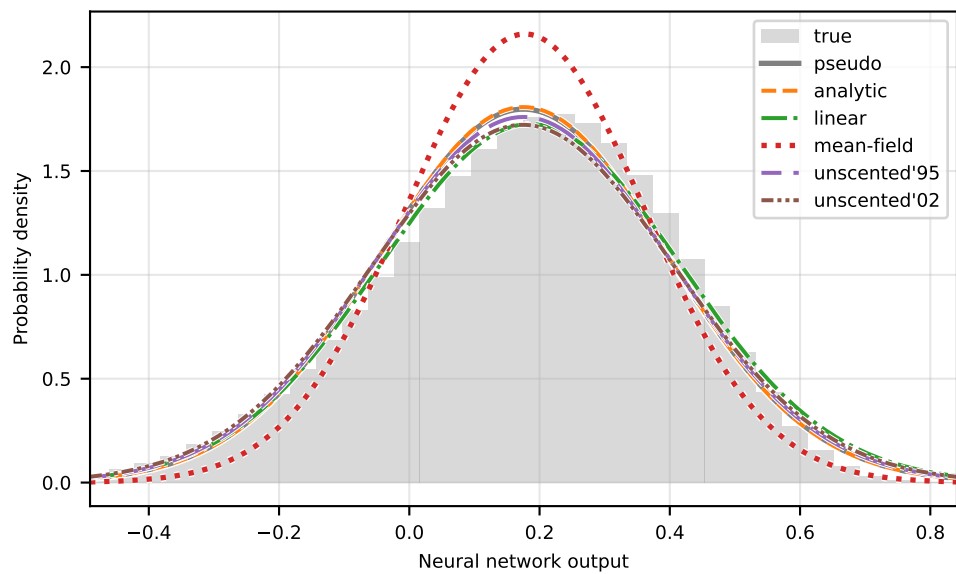

Figure 18: Probability distributions for Network(architecture=wide, weights=initialized, activation=probit residual), variance=small

| distribution | $\mu$ | $\sigma^2$ |
|---|---|---|
| pseudo-true ($Y_1$) | $-5.952 \times 10^{-2} \pm 3.6 \times 10^{-6}$ | $1.045 \times 10^0 \pm 9.2 \times 10^{-6}$ |
| analytic | $-2.266 \times 10^{-2}$ | $9.126 \times 10^{-1}$ |
| mean-field | $-6.728 \times 10^{-2}$ | $1.105 \times 10^0$ |
| linear | $+1.858 \times 10^{-1}$ | $5.340 \times 10^0$ |
| unscented'95 | $-1.497 \times 10^{-1}$ | $1.106 \times 10^0$ |
| unscented'02 | $-9.360 \times 10^{-1}$ | $7.856 \times 10^0$ |

Table 22: Comparison of moments for Network(architecture=wide, weights=initialized, activation=probit residual), variance=medium

| distribution | $d_{\mathrm{W}}(\cdot, Y_0)$ | $D_{\mathrm{KL}}(Y_1 \parallel \cdot)$ |
|---|---|---|
| pseudo-true ($Y_1$) | $1.107 \times 10^{-1} \pm 1.3 \times 10^{-5}$ | $0$ |
| analytic | $1.225 \times 10^{-1} \pm 1.0 \times 10^{-5}$ | $5.062 \times 10^{-3} \pm 5.5 \times 10^{-7}$ |
| mean-field | $1.094 \times 10^{-1} \pm 1.1 \times 10^{-5}$ | $8.233 \times 10^{-4} \pm 2.5 \times 10^{-7}$ |
| linear | $1.002 \times 10^0 \pm 9.5 \times 10^{-6}$ | $1.267 \times 10^0 \pm 1.9 \times 10^{-5}$ |
| unscented'95 | $1.445 \times 10^{-1} \pm 1.0 \times 10^{-5}$ | $4.702 \times 10^{-3} \pm 3.9 \times 10^{-7}$ |
| unscented'02 | $1.600 \times 10^0 \pm 9.8 \times 10^{-6}$ | $2.617 \times 10^0 \pm 3.2 \times 10^{-5}$ |

Table 23: Comparison of statistical distances for Network(architecture=wide, weights=initialized, activation=probit residual), variance=medium

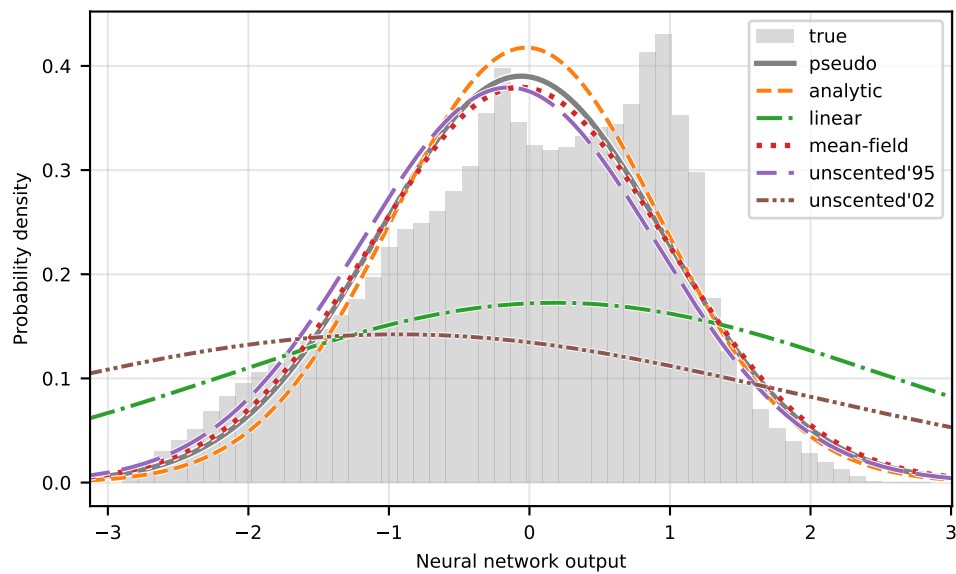

Figure 19: Probability distributions for Network(architecture=wide, weights=initialized, activation=probit residual), variance=medium

| distribution | $\mu$ | $\sigma^2$ |
|---|---|---|
| pseudo-true $(Y_1)$ | $+6.429 \times 10^{-3} \pm 3.2 \times 10^{-5}$ | $2.400 \times 10^{0} \pm 1.8 \times 10^{-4}$ |
| analytic | $+3.071 \times 10^{-2}$ | $2.001 \times 10^{0}$ |
| mean-field | $-1.149 \times 10^{-1}$ | $2.268 \times 10^{0}$ |
| linear | $+1.858 \times 10^{-1}$ | $5.340 \times 10^{+2}$ |
| unscented'95 | $-2.363 \times 10^{-1}$ | $1.624 \times 10^{0}$ |
| unscented'02 | $-1.120 \times 10^{+2}$ | $2.569 \times 10^{+4}$ |

Table 24: Comparison of moments for Network(architecture=wide, weights=initialized, activation=probit residual), variance=large

| distribution | $d_{\mathrm{W}}(\cdot, Y_0)$ | $D_{\mathrm{KL}}(Y_1 \parallel \cdot)$ |
|---|---|---|
| pseudo-true $(Y_1)$ | $1.315 \times 10^{-1} \pm 4.2 \times 10^{-5}$ | $0$ |
| analytic | $1.714 \times 10^{-1} \pm 4.4 \times 10^{-5}$ | $7.906 \times 10^{-3} \pm 6.3 \times 10^{-6}$ |
| mean-field | $1.460 \times 10^{-1} \pm 4.3 \times 10^{-5}$ | $3.846 \times 10^{-3} \pm 1.9 \times 10^{-6}$ |
| linear | $1.377 \times 10^{+1} \pm 3.0 \times 10^{-4}$ | $1.081 \times 10^{+2} \pm 8.3 \times 10^{-3}$ |
| unscented'95 | $2.530 \times 10^{-1} \pm 4.1 \times 10^{-5}$ | $4.584 \times 10^{-2} \pm 1.0 \times 10^{-5}$ |
| unscented'02 | $1.260 \times 10^{+2} \pm 2.4 \times 10^{-3}$ | $7.959 \times 10^{+3} \pm 6.0 \times 10^{-1}$ |

Table 25: Comparison of statistical distances for Network(architecture=wide, weights=initialized, activation=probit residual), variance=large

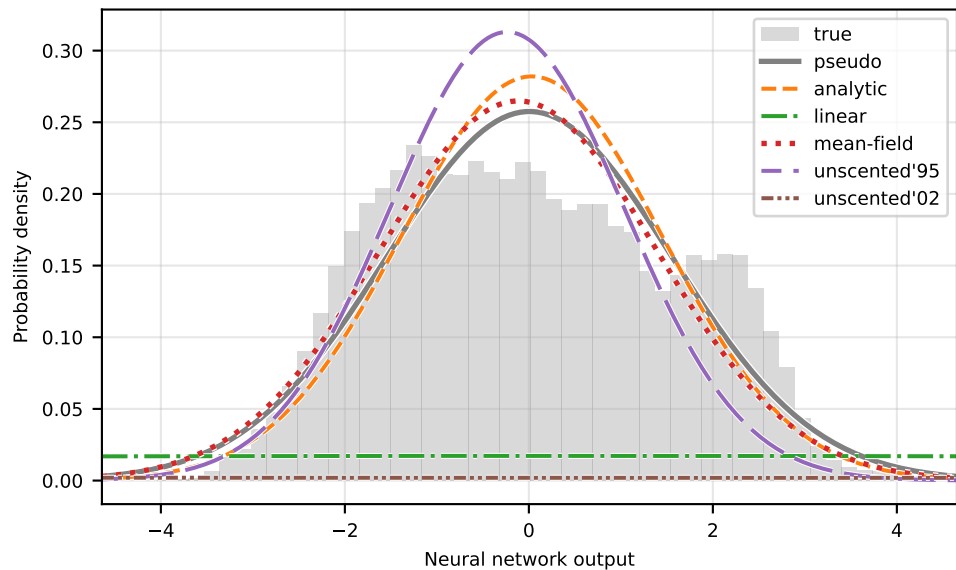

Figure 20: Probability distributions for Network(architecture=wide, weights=initialized, activation=probit residual), variance=large

| distribution | $\mu$ | $\sigma^2$ |
|---|---|---|
| pseudo-true ($Y_1$) | $-2.878 \times 10^{-2} \pm 2.5 \times 10^{-7}$ | $8.209 \times 10^{-2} \pm 4.9 \times 10^{-7}$ |
| analytic | $-2.882 \times 10^{-2}$ | $7.926 \times 10^{-2}$ |
| mean-field | $-6.754 \times 10^{-3}$ | $3.377 \times 10^{-2}$ |
| linear | $+2.228 \times 10^{-2}$ | $8.666 \times 10^{-2}$ |
| unscented'95 | $-3.091 \times 10^{-2}$ | $8.257 \times 10^{-2}$ |
| unscented'02 | $-3.498 \times 10^{-2}$ | $9.322 \times 10^{-2}$ |

Table 26: Comparison of moments for Network(architecture=wide, weights=trained, activation=probit residual), variance=small

| distribution | $d_{\mathrm{W}}(\cdot, Y_0)$ | $D_{\mathrm{KL}}(Y_1 \parallel \cdot)$ |
|---|---|---|
| pseudo-true ($Y_1$) | $3.621 \times 10^{-2} \pm 4.1 \times 10^{-6}$ | $0$ |
| analytic | $3.678 \times 10^{-2} \pm 4.5 \times 10^{-6}$ | $3.049 \times 10^{-4} \pm 1.0 \times 10^{-7}$ |
| mean-field | $1.522 \times 10^{-1} \pm 2.4 \times 10^{-6}$ | $1.528 \times 10^{-1} \pm 1.7 \times 10^{-6}$ |
| linear | $9.538 \times 10^{-2} \pm 4.8 \times 10^{-7}$ | $1.662 \times 10^{-2} \pm 2.9 \times 10^{-7}$ |
| unscented'95 | $3.773 \times 10^{-2} \pm 4.5 \times 10^{-6}$ | $3.607 \times 10^{-5} \pm 1.9 \times 10^{-8}$ |
| unscented'02 | $4.863 \times 10^{-2} \pm 2.3 \times 10^{-6}$ | $4.452 \times 10^{-3} \pm 4.1 \times 10^{-7}$ |

Table 27: Comparison of statistical distances for Network(architecture=wide, weights=trained, activation=probit residual), variance=small

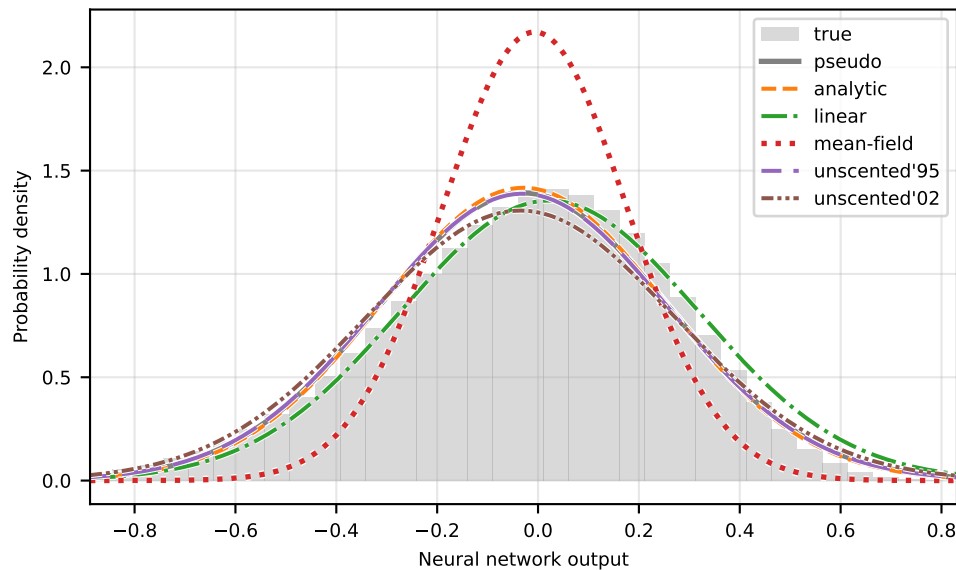

Figure 21: Probability distributions for Network(architecture=wide, weights=trained, activation=probit residual), variance=small

| distribution | $\mu$ | $\sigma^2$ |
|---|---|---|
| pseudo-true ($Y_1$) | $-4.463 \times 10^{-1} \pm 3.6 \times 10^{-6}$ | $9.455 \times 10^{-1} \pm 7.6 \times 10^{-6}$ |
| analytic | $-4.242 \times 10^{-1}$ | $7.310 \times 10^{-1}$ |
| mean-field | $-4.670 \times 10^{-1}$ | $1.116 \times 10^{0}$ |
| linear | $+2.228 \times 10^{-2}$ | $8.666 \times 10^{0}$ |
| unscented'95 | $-4.659 \times 10^{-1}$ | $9.297 \times 10^{-1}$ |
| unscented'02 | $-5.704 \times 10^{0}$ | $7.424 \times 10^{+1}$ |

Table 28: Comparison of moments for Network(architecture=wide, weights=trained, activation=probit residual), variance=medium

| distribution | $d_{\mathrm{W}}(\cdot, Y_0)$ | $D_{\mathrm{KL}}(Y_1 \parallel \cdot)$ |
|---|---|---|
| pseudo-true ($Y_1$) | $1.057 \times 10^{-1} \pm 8.3 \times 10^{-6}$ | $0$ |
| analytic | $1.598 \times 10^{-1} \pm 1.1 \times 10^{-5}$ | $1.548 \times 10^{-2} \pm 9.0 \times 10^{-7}$ |
| mean-field | $1.033 \times 10^{-1} \pm 1.0 \times 10^{-5}$ | $7.492 \times 10^{-3} \pm 7.2 \times 10^{-7}$ |
| linear | $1.591 \times 10^{0} \pm 1.1 \times 10^{-5}$ | $3.091 \times 10^{0} \pm 3.4 \times 10^{-5}$ |
| unscented'95 | $1.096 \times 10^{-1} \pm 8.9 \times 10^{-6}$ | $2.736 \times 10^{-4} \pm 1.1 \times 10^{-7}$ |
| unscented'02 | $7.592 \times 10^{0} \pm 1.9 \times 10^{-5}$ | $5.119 \times 10^{+1} \pm 4.3 \times 10^{-4}$ |

Table 29: Comparison of statistical distances for Network(architecture=wide, weights=trained, activation=probit residual), variance=medium

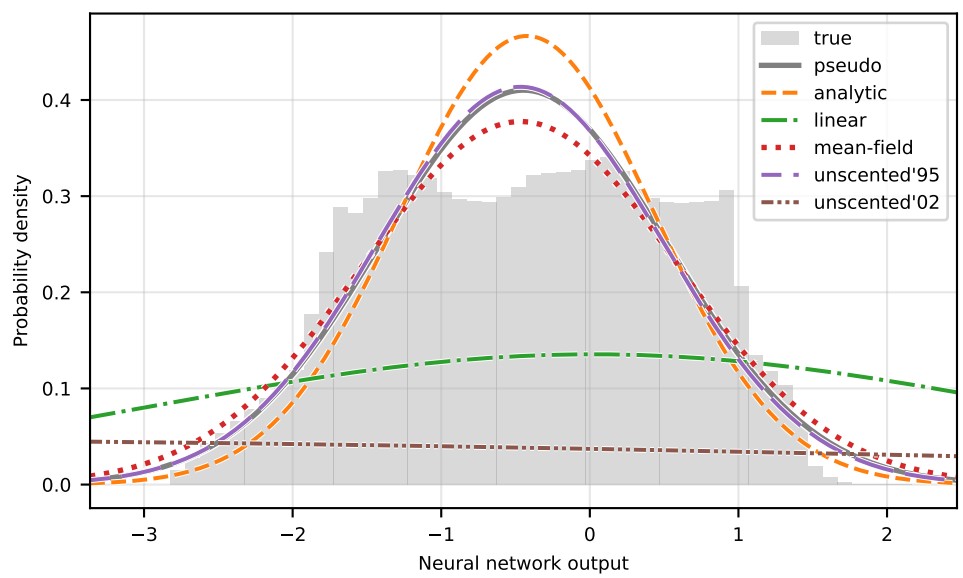

Figure 22: Probability distributions for Network(architecture=wide, weights=trained, activation=probit residual), variance=medium

| distribution | $\mu$ | $\sigma^2$ |
| --- | --- | --- |
| pseudo-true $(Y_1)$ | $-3.398 \times 10^{-1} \pm 3.1 \times 10^{-5}$ | $1.306 \times 10^0 \pm 1.4 \times 10^{-4}$ |
| analytic | $-3.099 \times 10^{-1}$ | $1.119 \times 10^0$ |
| mean-field | $-3.785 \times 10^{-1}$ | $2.281 \times 10^0$ |
| linear | $+2.228 \times 10^{-2}$ | $8.666 \times 10^{+2}$ |
| unscented'95 | $-4.517 \times 10^{-1}$ | $1.085 \times 10^0$ |
| unscented'02 | $-5.721 \times 10^{+2}$ | $6.556 \times 10^{+5}$ |

Table 30: Comparison of moments for Network(architecture=wide, weights=trained, activation=probit residual), variance=large

| distribution | $d_{\mathrm{W}}(\cdot, Y_0)$ | $D_{\mathrm{KL}}(Y_1 \parallel \cdot)$ |
| --- | --- | --- |
| pseudo-true $(Y_1)$ | $4.819 \times 10^{-2} \pm 5.0 \times 10^{-5}$ | $0$ |
| analytic | $9.476 \times 10^{-2} \pm 6.3 \times 10^{-5}$ | $6.010 \times 10^{-3} \pm 7.7 \times 10^{-6}$ |
| mean-field | $2.552 \times 10^{-1} \pm 6.7 \times 10^{-5}$ | $9.511 \times 10^{-2} \pm 4.0 \times 10^{-5}$ |
| linear | $2.110 \times 10^{+1} \pm 6.1 \times 10^{-4}$ | $3.282 \times 10^{+2} \pm 3.5 \times 10^{-2}$ |
| unscented'95 | $1.301 \times 10^{-1} \pm 4.1 \times 10^{-5}$ | $1.283 \times 10^{-2} \pm 8.1 \times 10^{-6}$ |
| unscented'02 | $7.484 \times 10^{+2} \pm 2.0 \times 10^{-2}$ | $3.763 \times 10^{+5} \pm 4.0 \times 10^{+1}$ |

Table 31: Comparison of statistical distances for Network(architecture=wide, weights=trained, activation=probit residual), variance=large

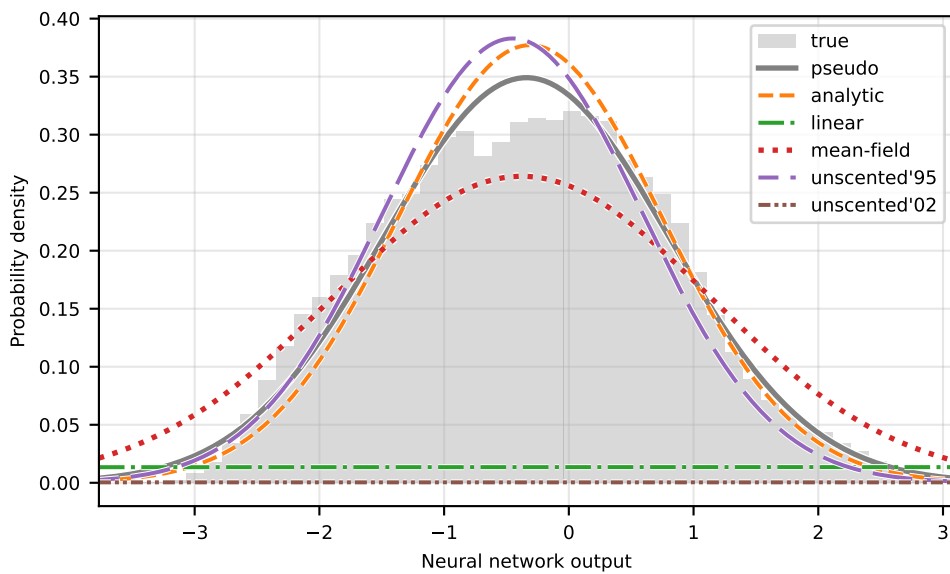

Figure 23: Probability distributions for Network(architecture=wide, weights=trained, activation=probit residual), variance=large

| distribution | $\mu$ | $\sigma^2$ |
|---|---|---|
| pseudo-true ($Y_1$) | $-7.232 \times 10^{-1} \pm 1.3 \times 10^{-7}$ | $3.095 \times 10^{-3} \pm 8.0 \times 10^{-8}$ |
| analytic | $-7.224 \times 10^{-1}$ | $2.803 \times 10^{-3}$ |
| mean-field | $-7.099 \times 10^{-1}$ | $8.446 \times 10^{-3}$ |
| linear | $-7.459 \times 10^{-1}$ | $2.995 \times 10^{-3}$ |
| unscented'95 | $-7.233 \times 10^{-1}$ | $2.532 \times 10^{-3}$ |
| unscented'02 | $-7.222 \times 10^{-1}$ | $4.117 \times 10^{-3}$ |

Table 32: Comparison of moments for Network(architecture=wide, weights=initialized, activation=sine), variance=small

| distribution | $d_{\mathrm{W}}(\cdot, Y_0)$ | $D_{\mathrm{KL}}(Y_1 \parallel \cdot)$ |
|---|---|---|
| pseudo-true ($Y_1$) | $2.493 \times 10^{-2} \pm 2.5 \times 10^{-6}$ | $0$ |
| analytic | $2.565 \times 10^{-2} \pm 2.1 \times 10^{-6}$ | $2.473 \times 10^{-3} \pm 1.2 \times 10^{-6}$ |
| mean-field | $1.421 \times 10^{-1} \pm 1.9 \times 10^{-6}$ | $3.909 \times 10^{-1} \pm 2.4 \times 10^{-5}$ |
| linear | $9.653 \times 10^{-2} \pm 3.5 \times 10^{-7}$ | $8.401 \times 10^{-2} \pm 1.1 \times 10^{-6}$ |
| unscented'95 | $2.625 \times 10^{-2} \pm 1.7 \times 10^{-6}$ | $9.448 \times 10^{-3} \pm 2.4 \times 10^{-6}$ |
| unscented'02 | $4.092 \times 10^{-2} \pm 2.4 \times 10^{-6}$ | $2.256 \times 10^{-2} \pm 4.3 \times 10^{-6}$ |

Table 33: Comparison of statistical distances for Network(architecture=wide, weights=initialized, activation=sine), variance=small

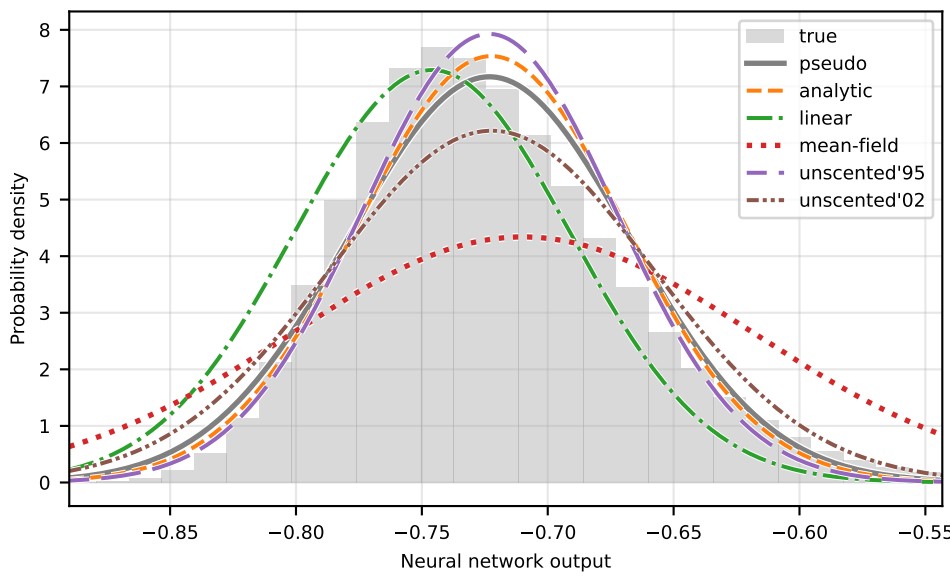

Figure 24: Probability distributions for Network(architecture=wide, weights=initialized, activation=sine), variance=small

| distribution | $\mu$ | $\sigma^2$ |
|---|---|---|
| pseudo-true ($Y_1$) | $-1.486 \times 10^{-1} \pm 7.9 \times 10^{-6}$ | $1.199 \times 10^{-1} \pm 1.1 \times 10^{-5}$ |
| analytic | $-9.940 \times 10^{-2}$ | $1.031 \times 10^{-1}$ |
| mean-field | $-2.147 \times 10^{-2}$ | $1.125 \times 10^{-1}$ |
| linear | $-7.459 \times 10^{-1}$ | $2.995 \times 10^{-1}$ |
| unscented'95 | $-1.529 \times 10^{-1}$ | $9.089 \times 10^{-2}$ |
| unscented'02 | $+1.623 \times 10^{0}$ | $1.152 \times 10^{+1}$ |

Table 34: Comparison of moments for Network(architecture=wide, weights=initialized, activation=sine), variance=medium

| distribution | $d_{\mathrm{W}}(\cdot, Y_0)$ | $D_{\mathrm{KL}}(Y_1 \parallel \cdot)$ |
|---|---|---|
| pseudo-true ($Y_1$) | $3.159 \times 10^{-2} \pm 2.3 \times 10^{-5}$ | $0$ |
| analytic | $9.243 \times 10^{-2} \pm 1.9 \times 10^{-5}$ | $1.553 \times 10^{-2} \pm 4.6 \times 10^{-6}$ |
| mean-field | $2.164 \times 10^{-1} \pm 1.5 \times 10^{-5}$ | $6.839 \times 10^{-2} \pm 1.1 \times 10^{-5}$ |
| linear | $1.015 \times 10^{0} \pm 2.1 \times 10^{-5}$ | $1.779 \times 10^{0} \pm 1.8 \times 10^{-4}$ |
| unscented'95 | $5.197 \times 10^{-2} \pm 1.6 \times 10^{-5}$ | $1.763 \times 10^{-2} \pm 1.1 \times 10^{-5}$ |
| unscented'02 | $4.814 \times 10^{0} \pm 1.2 \times 10^{-4}$ | $5.832 \times 10^{+1} \pm 5.6 \times 10^{-3}$ |

Table 35: Comparison of statistical distances for Network(architecture=wide, weights=initialized, activation=sine), variance=medium

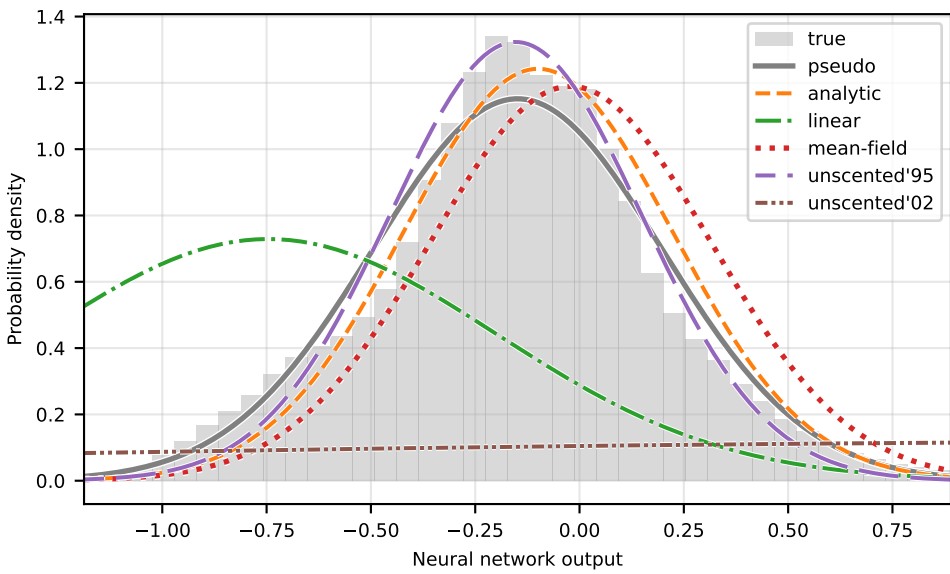

Figure 25: Probability distributions for Network(architecture=wide, weights=initialized, activation=sine), variance=medium

| distribution | $\mu$ | $\sigma^2$ |
|---|---|---|
| pseudo-true $(Y_1)$ | $+2.423 \times 10^{-1} \pm 1.5 \times 10^{-4}$ | $1.334 \times 10^{-1} \pm 1.1 \times 10^{-4}$ |
| analytic | $+2.380 \times 10^{-1}$ | $1.341 \times 10^{-1}$ |
| mean-field | $+2.362 \times 10^{-1}$ | $1.236 \times 10^{-1}$ |
| linear | $-7.459 \times 10^{-1}$ | $2.995 \times 10^{+1}$ |
| unscented'95 | $+2.682 \times 10^{-1}$ | $9.459 \times 10^{-2}$ |
| unscented'02 | $+2.360 \times 10^{+2}$ | $1.121 \times 10^{+5}$ |

Table 36: Comparison of moments for Network(architecture=wide, weights=initialized, activation=sine), variance=large

| distribution | $d_{\mathrm{W}}(\cdot, Y_0)$ | $D_{\mathrm{KL}}(Y_1 \parallel \cdot)$ |
|---|---|---|
| pseudo-true $(Y_1)$ | $3.009 \times 10^{-3} \pm 1.3 \times 10^{-4}$ | $0$ |
| analytic | $7.821 \times 10^{-3} \pm 2.0 \times 10^{-4}$ | $8.083 \times 10^{-5} \pm 4.8 \times 10^{-6}$ |
| mean-field | $1.992 \times 10^{-2} \pm 2.1 \times 10^{-4}$ | $1.565 \times 10^{-3} \pm 3.1 \times 10^{-5}$ |
| linear | $6.870 \times 10^{0} \pm 1.5 \times 10^{-3}$ | $1.127 \times 10^{+2} \pm 9.1 \times 10^{-2}$ |
| unscented'95 | $8.243 \times 10^{-2} \pm 1.7 \times 10^{-4}$ | $2.898 \times 10^{-2} \pm 1.1 \times 10^{-4}$ |
| unscented'02 | $5.469 \times 10^{+2} \pm 1.1 \times 10^{-1}$ | $6.284 \times 10^{+5} \pm 5.0 \times 10^{+2}$ |

Table 37: Comparison of statistical distances for Network(architecture=wide, weights=initialized, activation=sine), variance=large

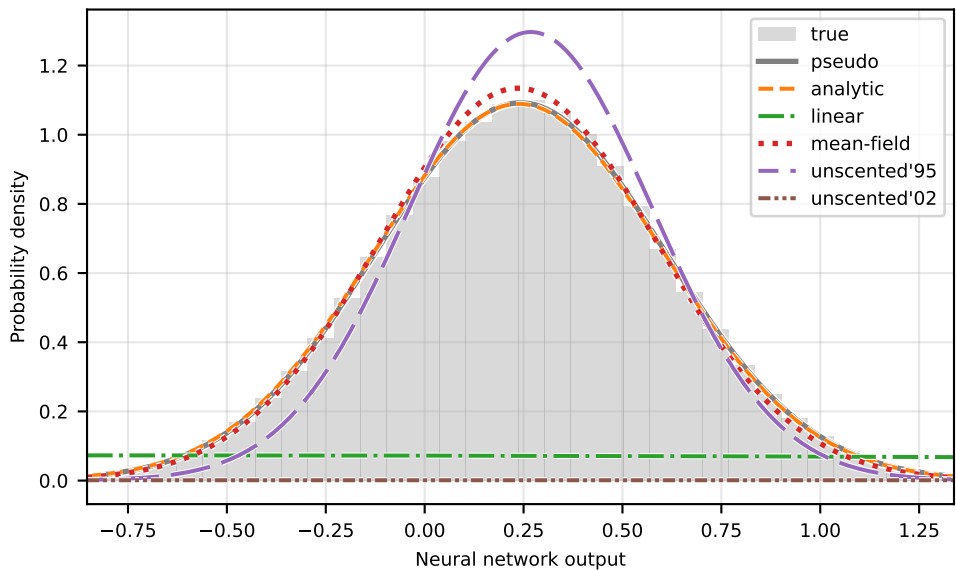

Figure 26: Probability distributions for Network(architecture=wide, weights=initialized, activation=sine), variance=large

| distribution | $\mu$ | $\sigma^2$ |
|---|---|---|
| pseudo-true ($Y_1$) | $-9.661 \times 10^{-2} \pm 2.2 \times 10^{-7}$ | $4.201 \times 10^{-2} \pm 2.9 \times 10^{-7}$ |
| analytic | $-9.662 \times 10^{-2}$ | $3.998 \times 10^{-2}$ |
| mean-field | $-5.941 \times 10^{-2}$ | $8.820 \times 10^{-3}$ |
| linear | $-3.517 \times 10^{-2}$ | $4.359 \times 10^{-2}$ |
| unscented'95 | $-9.971 \times 10^{-2}$ | $3.980 \times 10^{-2}$ |
| unscented'02 | $-1.027 \times 10^{-1}$ | $5.272 \times 10^{-2}$ |

Table 38: Comparison of moments for Network(architecture=wide, weights=trained, activation=sine), variance=small

| distribution | $d_{\mathrm{W}}(\cdot, Y_0)$ | $D_{\mathrm{KL}}(Y_1 \parallel \cdot)$ |
|---|---|---|
| pseudo-true ($Y_1$) | $5.363 \times 10^{-2} \pm 3.4 \times 10^{-6}$ | $0$ |
| analytic | $5.428 \times 10^{-2} \pm 3.3 \times 10^{-6}$ | $5.980 \times 10^{-4} \pm 1.6 \times 10^{-7}$ |
| mean-field | $1.947 \times 10^{-1} \pm 3.3 \times 10^{-6}$ | $4.018 \times 10^{-1} \pm 2.7 \times 10^{-6}$ |
| linear | $1.357 \times 10^{-1} \pm 4.7 \times 10^{-7}$ | $4.527 \times 10^{-2} \pm 4.6 \times 10^{-7}$ |
| unscented'95 | $5.685 \times 10^{-2} \pm 3.2 \times 10^{-6}$ | $8.258 \times 10^{-4} \pm 1.7 \times 10^{-7}$ |
| unscented'02 | $7.128 \times 10^{-2} \pm 3.3 \times 10^{-6}$ | $1.438 \times 10^{-2} \pm 8.9 \times 10^{-7}$ |

Table 39: Comparison of statistical distances for Network(architecture=wide, weights=trained, activation=sine), variance=small

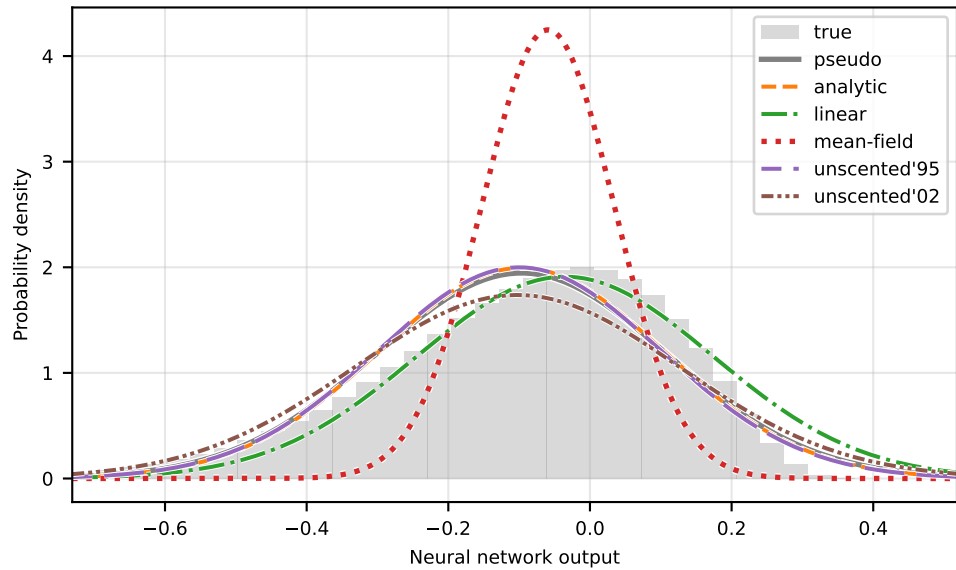

Figure 27: Probability distributions for Network(architecture=wide, weights=trained, activation=sine), variance=small

| distribution | $\mu$ | $\sigma^2$ |
|---|---|---|
| pseudo-true ($Y_1$) | $-1.067 \times 10^{-1} \pm 7.6 \times 10^{-6}$ | $2.428 \times 10^{-1} \pm 1.0 \times 10^{-5}$ |
| analytic | $-6.853 \times 10^{-2}$ | $1.456 \times 10^{-1}$ |
| mean-field | $+8.272 \times 10^{-3}$ | $1.152 \times 10^{-1}$ |
| linear | $-3.517 \times 10^{-2}$ | $4.359 \times 10^{0}$ |
| unscented'95 | $-7.132 \times 10^{-2}$ | $1.698 \times 10^{-1}$ |
| unscented'02 | $-6.790 \times 10^{0}$ | $9.563 \times 10^{+1}$ |

Table 40: Comparison of moments for Network(architecture=wide, weights=trained, activation=sine), variance=medium

| distribution | $d_{\mathrm{W}}(\cdot, Y_0)$ | $D_{\mathrm{KL}}(Y_1 \parallel \cdot)$ |
|---|---|---|
| pseudo-true ($Y_1$) | $1.000 \times 10^{-1} \pm 2.2 \times 10^{-5}$ | $0$ |
| analytic | $1.153 \times 10^{-1} \pm 1.2 \times 10^{-5}$ | $5.852 \times 10^{-2} \pm 8.0 \times 10^{-6}$ |
| mean-field | $1.785 \times 10^{-1} \pm 1.5 \times 10^{-5}$ | $1.374 \times 10^{-1} \pm 9.0 \times 10^{-6}$ |
| linear | $1.837 \times 10^{0} \pm 3.3 \times 10^{-5}$ | $7.042 \times 10^{0} \pm 3.7 \times 10^{-4}$ |
| unscented'95 | $9.049 \times 10^{-2} \pm 1.7 \times 10^{-5}$ | $3.107 \times 10^{-2} \pm 5.9 \times 10^{-6}$ |
| unscented'02 | $1.322 \times 10^{+1} \pm 1.6 \times 10^{-4}$ | $2.854 \times 10^{+2} \pm 1.2 \times 10^{-2}$ |

Table 41: Comparison of statistical distances for Network(architecture=wide, weights=trained, activation=sine), variance=medium

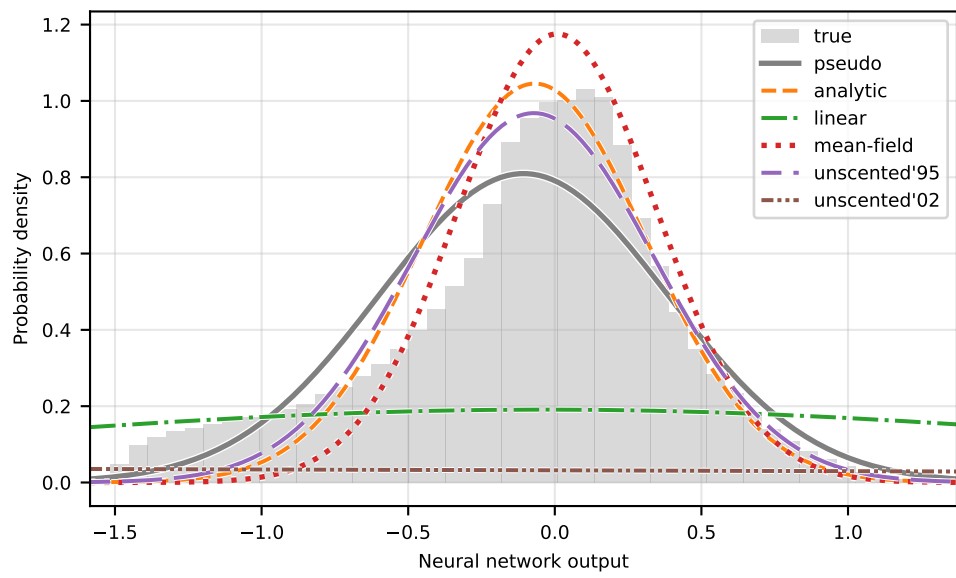

Figure 28: Probability distributions for Network(architecture=wide, weights=trained, activation=sine), variance=medium

| distribution | $\mu$ | $\sigma^2$ |
|---|---|---|
| pseudo-true ($Y_1$) | $+2.743 \times 10^{-1} \pm 1.4 \times 10^{-4}$ | $1.397 \times 10^{-1} \pm 1.1 \times 10^{-4}$ |
| analytic | $+2.702 \times 10^{-1}$ | $1.408 \times 10^{-1}$ |
| mean-field | $+2.699 \times 10^{-1}$ | $1.249 \times 10^{-1}$ |
| linear | $-3.517 \times 10^{-2}$ | $4.359 \times 10^{+2}$ |
| unscented'95 | $+3.025 \times 10^{-1}$ | $1.039 \times 10^{-1}$ |
| unscented'02 | $-6.753 \times 10^{+2}$ | $9.123 \times 10^{+5}$ |

Table 42: Comparison of moments for Network(architecture=wide, weights=trained, activation=sine), variance=large

| distribution | $d_{\mathrm{W}}(\cdot, Y_0)$ | $D_{\mathrm{KL}}(Y_1 \parallel \cdot)$ |
|---|---|---|
| pseudo-true ($Y_1$) | $2.929 \times 10^{-3} \pm 1.1 \times 10^{-4}$ | $0$ |
| analytic | $7.570 \times 10^{-3} \pm 1.8 \times 10^{-4}$ | $8.025 \times 10^{-5} \pm 4.3 \times 10^{-6}$ |
| mean-field | $2.646 \times 10^{-2} \pm 2.2 \times 10^{-4}$ | $3.105 \times 10^{-3} \pm 4.3 \times 10^{-5}$ |
| linear | $2.676 \times 10^{+1} \pm 5.4 \times 10^{-3}$ | $1.556 \times 10^{+3} \pm 1.2 \times 10^{0}$ |
| unscented'95 | $7.597 \times 10^{-2} \pm 1.6 \times 10^{-4}$ | $2.281 \times 10^{-2} \pm 9.4 \times 10^{-5}$ |
| unscented'02 | $1.546 \times 10^{+3} \pm 3.0 \times 10^{-1}$ | $4.898 \times 10^{+6} \pm 3.9 \times 10^{+3}$ |

Table 43: Comparison of statistical distances for Network(architecture=wide, weights=trained, activation=sine), variance=large

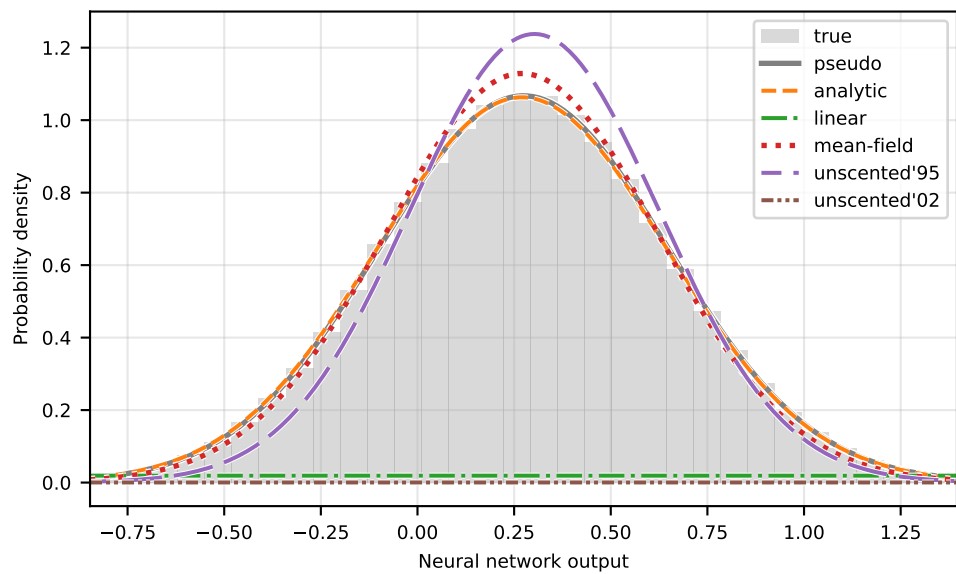

Figure 29: Probability distributions for Network(architecture=wide, weights=trained, activation=sine), variance=large

| distribution | $\mu$ | $\sigma^2$ |
|---|---|---|
| pseudo-true ($Y_1$) | $+1.778 \times 10^0 \pm 5.5 \times 10^{-7}$ | $5.871 \times 10^{-2} \pm 6.9 \times 10^{-7}$ |
| analytic | $+1.774 \times 10^0$ | $7.277 \times 10^{-2}$ |
| mean-field | $+1.646 \times 10^0$ | $1.393 \times 10^{-1}$ |
| linear | $+1.677 \times 10^0$ | $7.931 \times 10^{-2}$ |
| unscented'95 | $+1.789 \times 10^0$ | $5.573 \times 10^{-2}$ |
| unscented'02 | $+1.788 \times 10^0$ | $1.040 \times 10^{-1}$ |

Table 44: Comparison of moments for Network(architecture=wide, weights=initialized, activation=sine residual), variance=small

| distribution | $d_{\mathrm{W}}(\cdot, Y_0)$ | $D_{\mathrm{KL}}(Y_1 \parallel \cdot)$ |
|---|---|---|
| pseudo-true ($Y_1$) | $8.017 \times 10^{-2} \pm 3.5 \times 10^{-6}$ | $0$ |
| analytic | $7.802 \times 10^{-2} \pm 4.4 \times 10^{-6}$ | $1.251 \times 10^{-2} \pm 1.4 \times 10^{-6}$ |
| mean-field | $2.836 \times 10^{-1} \pm 4.0 \times 10^{-6}$ | $4.026 \times 10^{-1} \pm 9.1 \times 10^{-6}$ |
| linear | $2.043 \times 10^{-1} \pm 8.9 \times 10^{-7}$ | $1.112 \times 10^{-1} \pm 2.6 \times 10^{-6}$ |
| unscented'95 | $8.962 \times 10^{-2} \pm 3.5 \times 10^{-6}$ | $1.772 \times 10^{-3} \pm 2.4 \times 10^{-7}$ |
| unscented'02 | $1.538 \times 10^{-1} \pm 2.7 \times 10^{-6}$ | $1.008 \times 10^{-1} \pm 4.6 \times 10^{-6}$ |

Table 45: Comparison of statistical distances for Network(architecture=wide, weights=initialized, activation=sine residual), variance=small

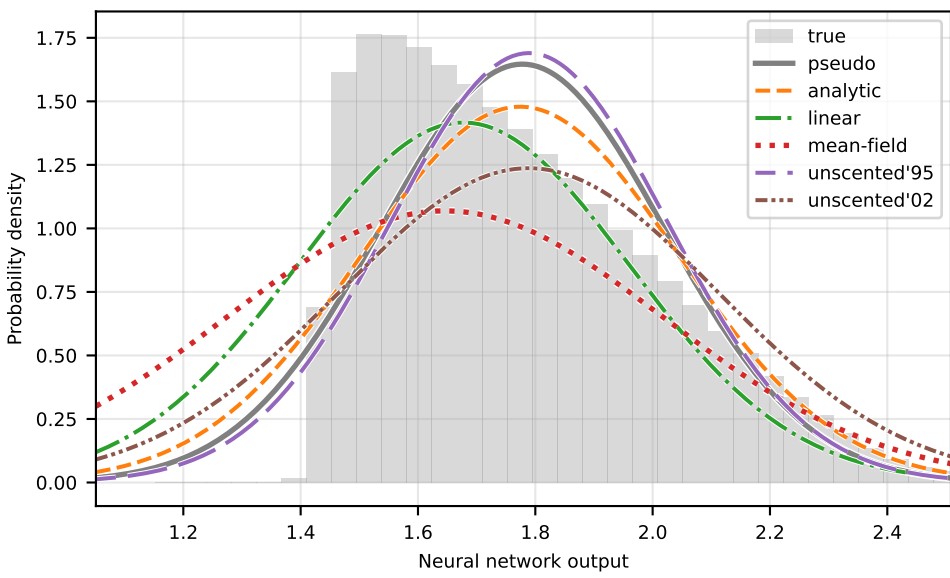

Figure 30: Probability distributions for Network(architecture=wide, weights=initialized, activation=sine residual), variance=small

| distribution | $\mu$ | $\sigma^2$ |
|---|---|---|
| pseudo-true $(Y_1)$ | $+7.436 \times 10^{-1} \pm 4.8 \times 10^{-5}$ | $2.285 \times 10^0 \pm 1.8 \times 10^{-4}$ |
| analytic | $+5.891 \times 10^{-1}$ | $1.899 \times 10^0$ |
| mean-field | $+3.448 \times 10^{-1}$ | $1.628 \times 10^0$ |
| linear | $+1.677 \times 10^0$ | $7.931 \times 10^0$ |
| unscented'95 | $+2.881 \times 10^{-1}$ | $3.174 \times 10^0$ |
| unscented'02 | $+1.279 \times 10^{+1}$ | $2.548 \times 10^{+2}$ |

Table 46: Comparison of moments for Network(architecture=wide, weights=initialized, activation=sine residual), variance=medium

| distribution | $d_{\mathrm{W}}(\cdot, Y_0)$ | $D_{\mathrm{KL}}(Y_1 \parallel \cdot)$ |
|---|---|---|
| pseudo-true $(Y_1)$ | $4.599 \times 10^{-2} \pm 4.1 \times 10^{-5}$ | $0$ |
| analytic | $1.512 \times 10^{-1} \pm 3.7 \times 10^{-5}$ | $1.326 \times 10^{-2} \pm 4.5 \times 10^{-6}$ |
| mean-field | $3.332 \times 10^{-1} \pm 4.8 \times 10^{-5}$ | $6.053 \times 10^{-2} \pm 6.2 \times 10^{-6}$ |
| linear | $1.045 \times 10^0 \pm 6.2 \times 10^{-5}$ | $8.041 \times 10^{-1} \pm 1.0 \times 10^{-4}$ |
| unscented'95 | $4.032 \times 10^{-1} \pm 4.7 \times 10^{-5}$ | $7.563 \times 10^{-2} \pm 2.7 \times 10^{-5}$ |
| unscented'02 | $1.245 \times 10^{+1} \pm 2.8 \times 10^{-4}$ | $8.465 \times 10^{+1} \pm 6.8 \times 10^{-3}$ |

Table 47: Comparison of statistical distances for Network(architecture=wide, weights=initialized, activation=sine residual), variance=medium

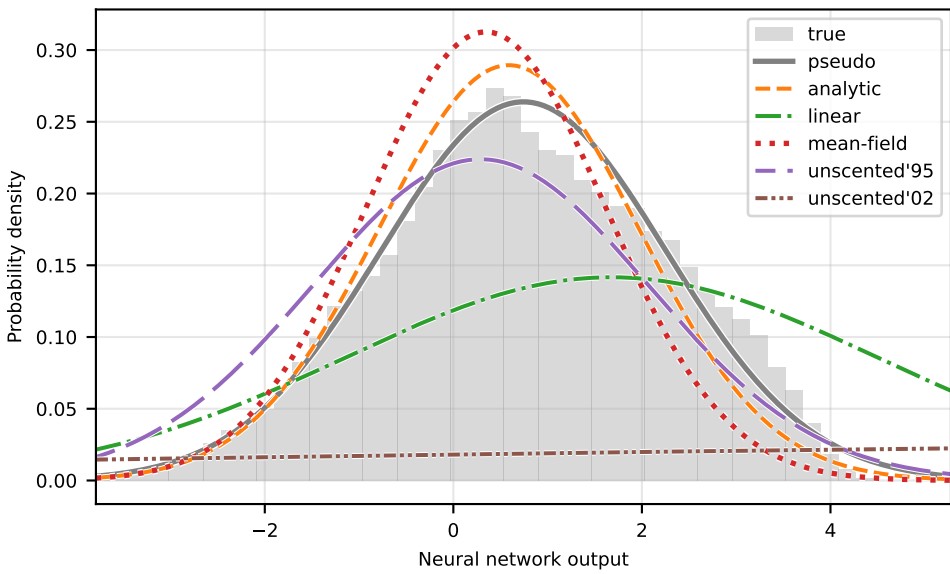

Figure 31: Probability distributions for Network(architecture=wide, weights=initialized, activation=sine residual), variance=medium

| distribution | $\mu$ | $\sigma^2$ |
|---|---|---|
| pseudo-true ($Y_1$) | $+5.654 \times 10^{-2} \pm 5.7 \times 10^{-4}$ | $1.868 \times 10^0 \pm 1.2 \times 10^{-3}$ |
| analytic | $+5.135 \times 10^{-2}$ | $1.862 \times 10^0$ |
| mean-field | $+4.378 \times 10^{-2}$ | $1.862 \times 10^0$ |
| linear | $+1.677 \times 10^0$ | $7.931 \times 10^{+2}$ |
| unscented'95 | $+1.063 \times 10^0$ | $2.373 \times 10^0$ |
| unscented'02 | $+1.113 \times 10^{+3}$ | $2.472 \times 10^{+6}$ |

Table 48: Comparison of moments for Network(architecture=wide, weights=initialized, activation=sine residual), variance=large

| distribution | $d_{\mathrm{W}}(\cdot, Y_0)$ | $D_{\mathrm{KL}}(Y_1 \parallel \cdot)$ |
|---|---|---|
| pseudo-true ($Y_1$) | $5.228 \times 10^{-3} \pm 2.2 \times 10^{-4}$ | $0$ |
| analytic | $6.261 \times 10^{-3} \pm 3.1 \times 10^{-4}$ | $1.346 \times 10^{-5} \pm 2.1 \times 10^{-6}$ |
| mean-field | $1.123 \times 10^{-2} \pm 4.7 \times 10^{-4}$ | $4.987 \times 10^{-5} \pm 4.2 \times 10^{-6}$ |
| linear | $1.832 \times 10^{+1} \pm 3.2 \times 10^{-3}$ | $2.095 \times 10^{+2} \pm 1.4 \times 10^{-1}$ |
| unscented'95 | $8.609 \times 10^{-1} \pm 5.4 \times 10^{-4}$ | $2.867 \times 10^{-1} \pm 4.5 \times 10^{-4}$ |
| unscented'02 | $1.330 \times 10^{+3} \pm 2.1 \times 10^{-1}$ | $9.932 \times 10^{+5} \pm 6.3 \times 10^{+2}$ |

Table 49: Comparison of statistical distances for Network(architecture=wide, weights=initialized, activation=sine residual), variance=large

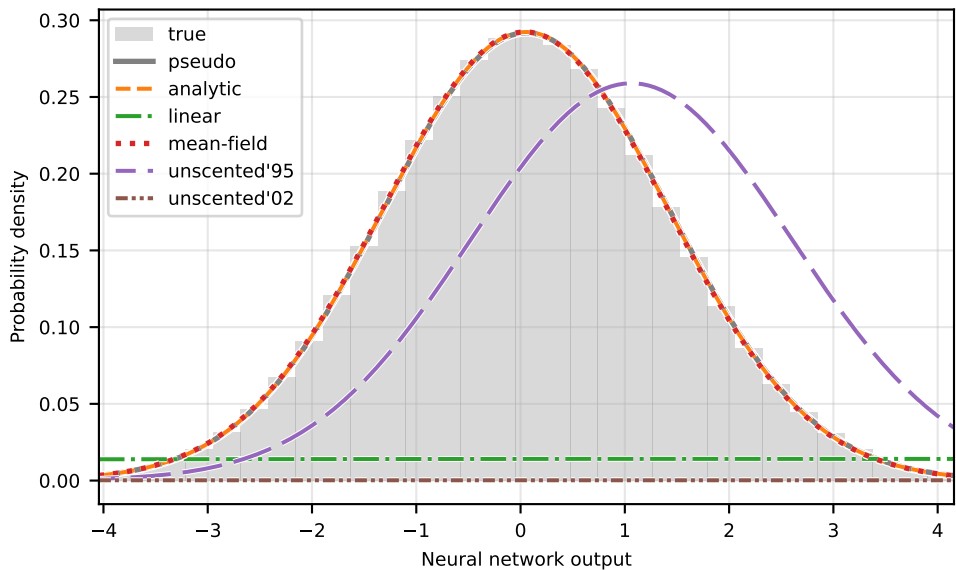

Figure 32: Probability distributions for Network(architecture=wide, weights=initialized, activation=sine residual), variance=large

| distribution | $\mu$ | $\sigma^2$ |
|---|---|---|
| pseudo-true ($Y_1$) | $+3.352 \times 10^{-1} \pm 5.0 \times 10^{-7}$ | $1.858 \times 10^{-1} \pm 1.4 \times 10^{-6}$ |
| analytic | $+3.330 \times 10^{-1}$ | $1.887 \times 10^{-1}$ |
| mean-field | $+3.550 \times 10^{-1}$ | $1.432 \times 10^{-1}$ |
| linear | $+3.526 \times 10^{-1}$ | $2.274 \times 10^{-1}$ |
| unscented'95 | $+3.373 \times 10^{-1}$ | $1.805 \times 10^{-1}$ |
| unscented'02 | $+3.176 \times 10^{-1}$ | $2.299 \times 10^{-1}$ |

Table 50: Comparison of moments for Network(architecture=wide, weights=trained, activation=sine residual), variance=small

| distribution | $d_{\mathrm{W}}(\cdot, Y_0)$ | $D_{\mathrm{KL}}(Y_1 \parallel \cdot)$ |
|---|---|---|
| pseudo-true ($Y_1$) | $2.299 \times 10^{-2} \pm 4.6 \times 10^{-6}$ | $0$ |
| analytic | $2.097 \times 10^{-2} \pm 4.2 \times 10^{-6}$ | $7.531 \times 10^{-5} \pm 6.1 \times 10^{-8}$ |
| mean-field | $7.953 \times 10^{-2} \pm 3.6 \times 10^{-6}$ | $1.664 \times 10^{-2} \pm 8.6 \times 10^{-7}$ |
| linear | $5.630 \times 10^{-2} \pm 3.5 \times 10^{-6}$ | $1.175 \times 10^{-2} \pm 8.3 \times 10^{-7}$ |
| unscented'95 | $2.748 \times 10^{-2} \pm 4.4 \times 10^{-6}$ | $2.177 \times 10^{-4} \pm 1.1 \times 10^{-7}$ |
| unscented'02 | $5.316 \times 10^{-2} \pm 2.7 \times 10^{-6}$ | $1.302 \times 10^{-2} \pm 9.1 \times 10^{-7}$ |

Table 51: Comparison of statistical distances for Network(architecture=wide, weights=trained, activation=sine residual), variance=small

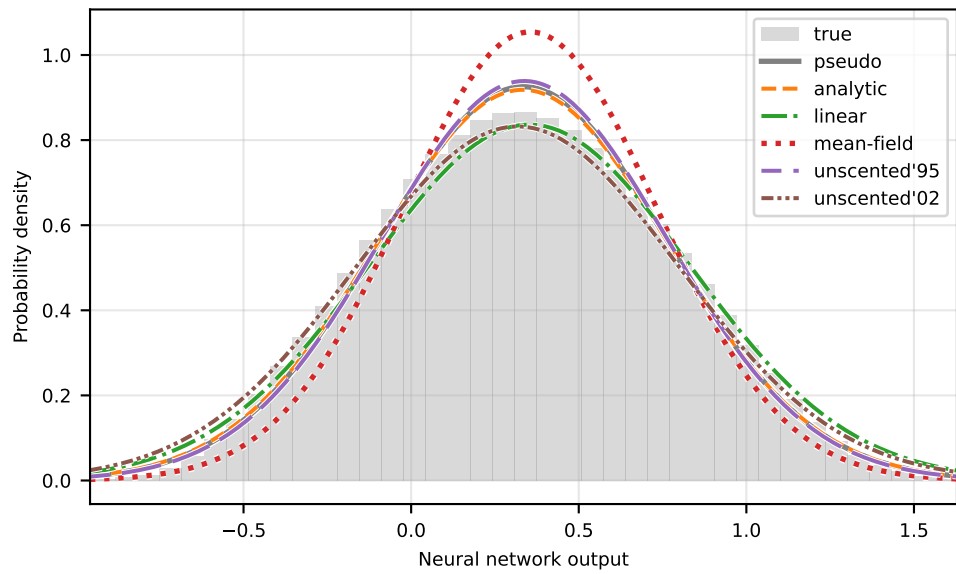

Figure 33: Probability distributions for Network(architecture=wide, weights=trained, activation=sine residual), variance=small

| distribution | $\mu$ | $\sigma^2$ |
|---|---|---|
| pseudo-true ($Y_1$) | $+1.823 \times 10^{-1} \pm 4.7 \times 10^{-5}$ | $1.672 \times 10^0 \pm 1.5 \times 10^{-4}$ |
| analytic | $+2.142 \times 10^{-1}$ | $1.758 \times 10^0$ |
| mean-field | $+7.102 \times 10^{-2}$ | $1.627 \times 10^0$ |
| linear | $+3.526 \times 10^{-1}$ | $2.274 \times 10^{+1}$ |
| unscented'95 | $+7.597 \times 10^{-2}$ | $3.013 \times 10^0$ |
| unscented'02 | $-3.146 \times 10^0$ | $4.722 \times 10^{+1}$ |

Table 52: Comparison of moments for Network(architecture=wide, weights=trained, activation=sine residual), variance=medium

| distribution | $d_{\mathrm{W}}(\cdot, Y_0)$ | $D_{\mathrm{KL}}(Y_1 \parallel \cdot)$ |
|---|---|---|
| pseudo-true ($Y_1$) | $5.071 \times 10^{-2} \pm 6.4 \times 10^{-5}$ | $0$ |
| analytic | $5.506 \times 10^{-2} \pm 4.5 \times 10^{-5}$ | $9.444 \times 10^{-4} \pm 1.9 \times 10^{-6}$ |
| mean-field | $1.046 \times 10^{-1} \pm 4.4 \times 10^{-5}$ | $3.885 \times 10^{-3} \pm 2.7 \times 10^{-6}$ |
| linear | $2.445 \times 10^0 \pm 9.5 \times 10^{-5}$ | $5.005 \times 10^0 \pm 5.5 \times 10^{-4}$ |
| unscented'95 | $3.110 \times 10^{-1} \pm 5.1 \times 10^{-5}$ | $1.099 \times 10^{-1} \pm 3.8 \times 10^{-5}$ |
| unscented'02 | $4.571 \times 10^0 \pm 1.5 \times 10^{-4}$ | $1.526 \times 10^{+1} \pm 1.6 \times 10^{-3}$ |

Table 53: Comparison of statistical distances for Network(architecture=wide, weights=trained, activation=sine residual), variance=medium

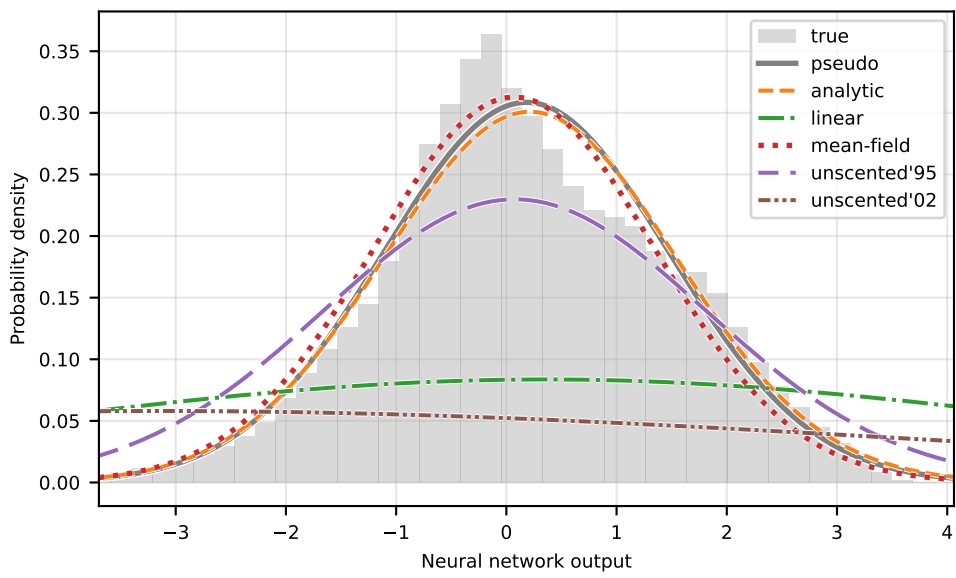

Figure 34: Probability distributions for Network(architecture=wide, weights=trained, activation=sine residual), variance=medium

| distribution | $\mu$ | $\sigma^2$ |
|---|---|---|
| pseudo-true $(Y_1)$ | $+1.018 \times 10^{-2} \pm 5.7 \times 10^{-4}$ | $1.859 \times 10^0 \pm 1.2 \times 10^{-3}$ |
| analytic | $+5.144 \times 10^{-3}$ | $1.855 \times 10^0$ |
| mean-field | $-9.526 \times 10^{-4}$ | $1.862 \times 10^0$ |
| linear | $+3.526 \times 10^{-1}$ | $2.274 \times 10^{+3}$ |
| unscented'95 | $+1.029 \times 10^0$ | $2.357 \times 10^0$ |
| unscented'02 | $-3.472 \times 10^{+2}$ | $2.438 \times 10^{+5}$ |

Table 54: Comparison of moments for Network(architecture=wide, weights=trained, activation=sine residual), variance=large

| distribution | $d_{\mathrm{W}}(\cdot, Y_0)$ | $D_{\mathrm{KL}}(Y_1 \parallel \cdot)$ |
|---|---|---|
| pseudo-true $(Y_1)$ | $4.921 \times 10^{-3} \pm 2.1 \times 10^{-4}$ | $0$ |
| analytic | $6.028 \times 10^{-3} \pm 3.1 \times 10^{-4}$ | $1.203 \times 10^{-5} \pm 1.9 \times 10^{-6}$ |
| mean-field | $9.905 \times 10^{-3} \pm 4.6 \times 10^{-4}$ | $3.769 \times 10^{-5} \pm 3.4 \times 10^{-6}$ |
| linear | $3.165 \times 10^{+1} \pm 5.4 \times 10^{-3}$ | $6.075 \times 10^{+2} \pm 3.9 \times 10^{-1}$ |
| unscented'95 | $8.729 \times 10^{-1} \pm 5.4 \times 10^{-4}$ | $2.946 \times 10^{-1} \pm 4.6 \times 10^{-4}$ |
| unscented'02 | $4.168 \times 10^{+2} \pm 6.7 \times 10^{-2}$ | $9.798 \times 10^{+4} \pm 6.3 \times 10^{+1}$ |

Table 55: Comparison of statistical distances for Network(architecture=wide, weights=trained, activation=sine residual), variance=large

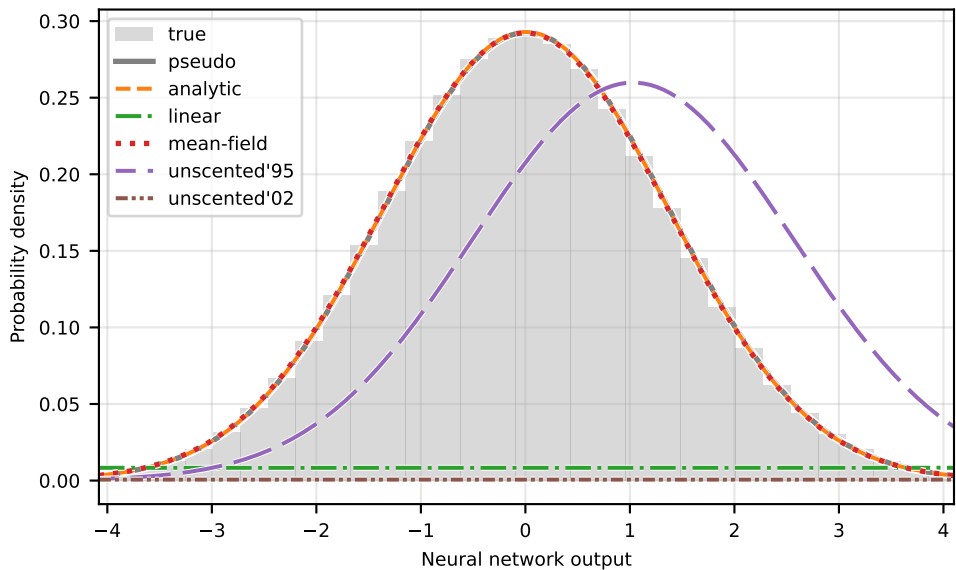

Figure 35: Probability distributions for Network(architecture=wide, weights=trained, activation=sine residual), variance=large

| distribution | $\mu$ | $\sigma^2$ |
|---|---|---|
| pseudo-true ($Y_1$) | $-2.777 \times 10^0 \pm 8.9 \times 10^{-8}$ | $2.682 \times 10^{-2} \pm 1.0 \times 10^{-7}$ |
| analytic | $-2.777 \times 10^0$ | $2.743 \times 10^{-2}$ |
| mean-field | $-2.783 \times 10^0$ | $8.697 \times 10^{-3}$ |
| linear | $-2.794 \times 10^0$ | $2.999 \times 10^{-2}$ |
| unscented'95 | $-2.777 \times 10^0$ | $2.814 \times 10^{-2}$ |
| unscented'02 | $-2.776 \times 10^0$ | $3.064 \times 10^{-2}$ |

Table 56: Comparison of moments for Network(architecture=wide, weights=initialized, activation=gelu), variance=small

| distribution | $d_\mathrm{W}(\cdot, Y_0)$ | $D_\mathrm{KL}(Y_1 \parallel \cdot)$ |
|---|---|---|
| pseudo-true ($Y_1$) | $1.772 \times 10^{-2} \pm 1.9 \times 10^{-6}$ | $0$ |
| analytic | $1.697 \times 10^{-2} \pm 2.5 \times 10^{-6}$ | $1.302 \times 10^{-4} \pm 4.5 \times 10^{-8}$ |
| mean-field | $1.423 \times 10^{-1} \pm 1.7 \times 10^{-6}$ | $2.257 \times 10^{-1} \pm 1.3 \times 10^{-6}$ |
| linear | $4.249 \times 10^{-2} \pm 1.1 \times 10^{-6}$ | $8.757 \times 10^{-3} \pm 2.3 \times 10^{-7}$ |
| unscented'95 | $1.685 \times 10^{-2} \pm 2.6 \times 10^{-6}$ | $5.865 \times 10^{-4} \pm 9.6 \times 10^{-8}$ |
| unscented'02 | $2.505 \times 10^{-2} \pm 1.2 \times 10^{-6}$ | $4.667 \times 10^{-3} \pm 2.8 \times 10^{-7}$ |

Table 57: Comparison of statistical distances for Network(architecture=wide, weights=initialized, activation=gelu), variance=small

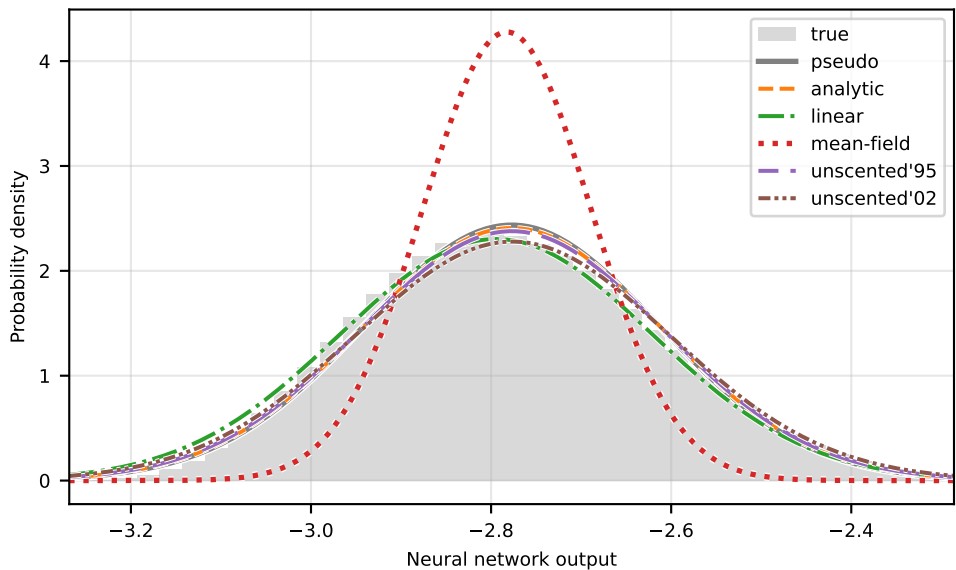

Figure 36: Probability distributions for Network(architecture=wide, weights=initialized, activation=gelu), variance=small

| distribution | $\mu$ | $\sigma^2$ |
|---|---|---|
| pseudo-true $(Y_1)$ | $-2.176 \times 10^0 \pm 4.3 \times 10^{-6}$ | $6.248 \times 10^{-1} \pm 1.4 \times 10^{-5}$ |
| analytic | $-2.127 \times 10^0$ | $5.620 \times 10^{-1}$ |
| mean-field | $-2.228 \times 10^0$ | $5.011 \times 10^{-1}$ |
| linear | $-2.794 \times 10^0$ | $2.999 \times 10^0$ |
| unscented'95 | $-2.106 \times 10^0$ | $6.547 \times 10^{-1}$ |
| unscented'02 | $-9.887 \times 10^{-1}$ | $9.520 \times 10^0$ |

Table 58: Comparison of moments for Network(architecture=wide, weights=initialized, activation=gelu), variance=medium

| distribution | $d_{\mathrm{W}}(\cdot, Y_0)$ | $D_{\mathrm{KL}}(Y_1 \parallel \cdot)$ |
|---|---|---|
| pseudo-true $(Y_1)$ | $2.103 \times 10^{-1} \pm 8.5 \times 10^{-6}$ | $0$ |
| analytic | $2.270 \times 10^{-1} \pm 9.0 \times 10^{-6}$ | $4.684 \times 10^{-3} \pm 1.0 \times 10^{-6}$ |
| mean-field | $1.903 \times 10^{-1} \pm 9.8 \times 10^{-6}$ | $1.346 \times 10^{-2} \pm 2.4 \times 10^{-6}$ |
| linear | $9.738 \times 10^{-1} \pm 1.0 \times 10^{-5}$ | $1.421 \times 10^0 \pm 4.8 \times 10^{-5}$ |
| unscented'95 | $2.438 \times 10^{-1} \pm 7.9 \times 10^{-6}$ | $4.571 \times 10^{-3} \pm 9.5 \times 10^{-7}$ |
| unscented'02 | $2.444 \times 10^0 \pm 1.9 \times 10^{-5}$ | $6.885 \times 10^0 \pm 1.9 \times 10^{-4}$ |

Table 59: Comparison of statistical distances for Network(architecture=wide, weights=initialized, activation=gelu), variance=medium

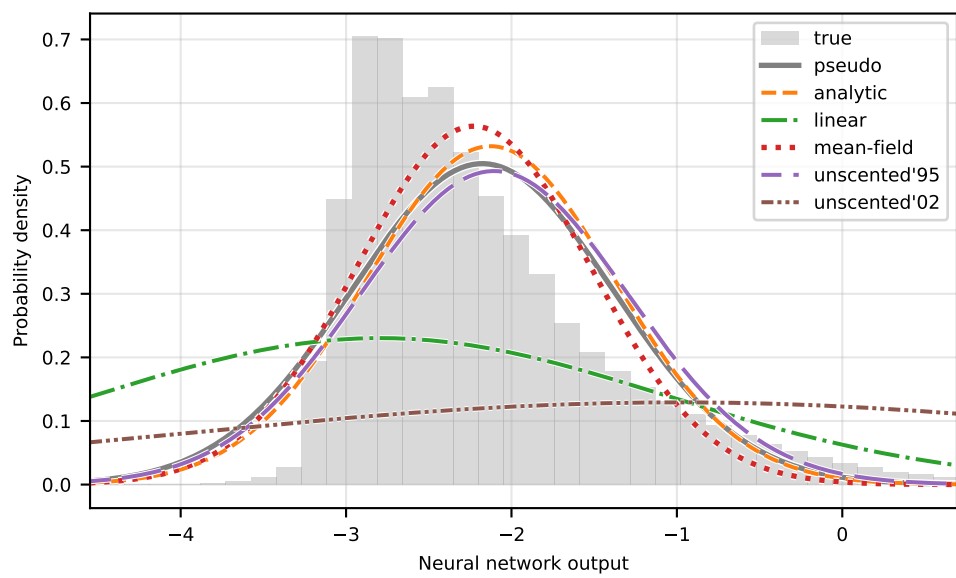

Figure 37: Probability distributions for Network(architecture=wide, weights=initialized, activation=gelu), variance=medium

| distribution | $\mu$ | $\sigma^2$ |
|---|---|---|
| pseudo-true ($Y_1$) | $+9.844 \times 10^{-1} \pm 5.5 \times 10^{-5}$ | $3.298 \times 10^{+1} \pm 1.3 \times 10^{-3}$ |
| analytic | $+8.293 \times 10^{-1}$ | $3.058 \times 10^{+1}$ |
| mean-field | $+5.103 \times 10^{-1}$ | $2.827 \times 10^{+1}$ |
| linear | $-2.794 \times 10^{0}$ | $2.999 \times 10^{+2}$ |
| unscented'95 | $+3.435 \times 10^{0}$ | $3.525 \times 10^{+1}$ |
| unscented'02 | $+1.777 \times 10^{+2}$ | $6.547 \times 10^{+4}$ |

Table 60: Comparison of moments for Network(architecture=wide, weights=initialized, activation=gelu), variance=large

| distribution | $d_{\mathrm{W}}(\cdot, Y_0)$ | $D_{\mathrm{KL}}(Y_1 \parallel \cdot)$ |
|---|---|---|
| pseudo-true ($Y_1$) | $4.269 \times 10^{-1} \pm 4.4 \times 10^{-5}$ | $0$ |
| analytic | $4.155 \times 10^{-1} \pm 4.9 \times 10^{-5}$ | $1.761 \times 10^{-3} \pm 1.4 \times 10^{-6}$ |
| mean-field | $4.163 \times 10^{-1} \pm 5.2 \times 10^{-5}$ | $9.056 \times 10^{-3} \pm 2.8 \times 10^{-6}$ |
| linear | $3.984 \times 10^{0} \pm 6.7 \times 10^{-5}$ | $3.159 \times 10^{0} \pm 1.7 \times 10^{-4}$ |
| unscented'95 | $1.057 \times 10^{0} \pm 5.0 \times 10^{-5}$ | $9.214 \times 10^{-2} \pm 6.4 \times 10^{-6}$ |
| unscented'02 | $1.035 \times 10^{+2} \pm 1.0 \times 10^{-3}$ | $1.462 \times 10^{+3} \pm 5.7 \times 10^{-2}$ |

Table 61: Comparison of statistical distances for Network(architecture=wide, weights=initialized, activation=gelu), variance=large

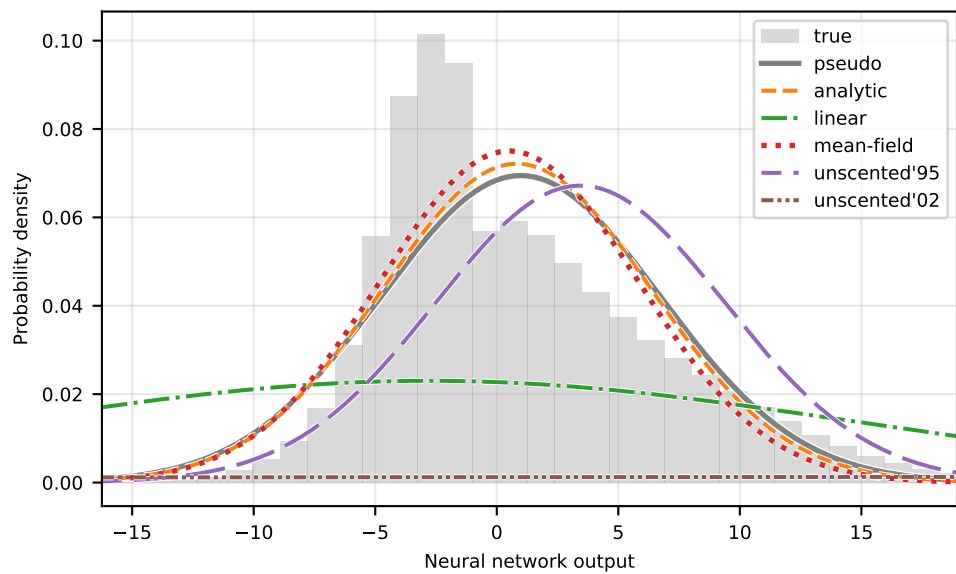

Figure 38: Probability distributions for Network(architecture=wide, weights=initialized, activation=gelu), variance=large

| distribution | $\mu$ | $\sigma^2$ |
|---|---|---|
| pseudo-true ($Y_1$) | $-6.976 \times 10^{-2} \pm 2.3 \times 10^{-7}$ | $4.419 \times 10^{-2} \pm 4.2 \times 10^{-7}$ |
| analytic | $-7.000 \times 10^{-2}$ | $4.266 \times 10^{-2}$ |
| mean-field | $-4.962 \times 10^{-2}$ | $9.842 \times 10^{-3}$ |
| linear | $-2.054 \times 10^{-2}$ | $4.617 \times 10^{-2}$ |
| unscented'95 | $-7.075 \times 10^{-2}$ | $4.375 \times 10^{-2}$ |
| unscented'02 | $-7.263 \times 10^{-2}$ | $5.159 \times 10^{-2}$ |

Table 62: Comparison of moments for Network(architecture=wide, weights=trained, activation=gelu), variance=small

| distribution | $d_{\mathrm{W}}(\cdot, Y_0)$ | $D_{\mathrm{KL}}(Y_1 \parallel \cdot)$ |
|---|---|---|
| pseudo-true ($Y_1$) | $4.228 \times 10^{-2} \pm 5.2 \times 10^{-6}$ | $0$ |
| analytic | $4.302 \times 10^{-2} \pm 4.3 \times 10^{-6}$ | $3.042 \times 10^{-4} \pm 1.6 \times 10^{-7}$ |
| mean-field | $1.926 \times 10^{-1} \pm 3.5 \times 10^{-6}$ | $3.668 \times 10^{-1} \pm 3.7 \times 10^{-6}$ |
| linear | $1.073 \times 10^{-1} \pm 4.8 \times 10^{-7}$ | $2.790 \times 10^{-2} \pm 4.5 \times 10^{-7}$ |
| unscented'95 | $4.315 \times 10^{-2} \pm 4.8 \times 10^{-6}$ | $3.611 \times 10^{-5} \pm 4.5 \times 10^{-8}$ |
| unscented'02 | $5.126 \times 10^{-2} \pm 4.7 \times 10^{-6}$ | $6.421 \times 10^{-3} \pm 8.0 \times 10^{-7}$ |

Table 63: Comparison of statistical distances for Network(architecture=wide, weights=trained, activation=gelu), variance=small

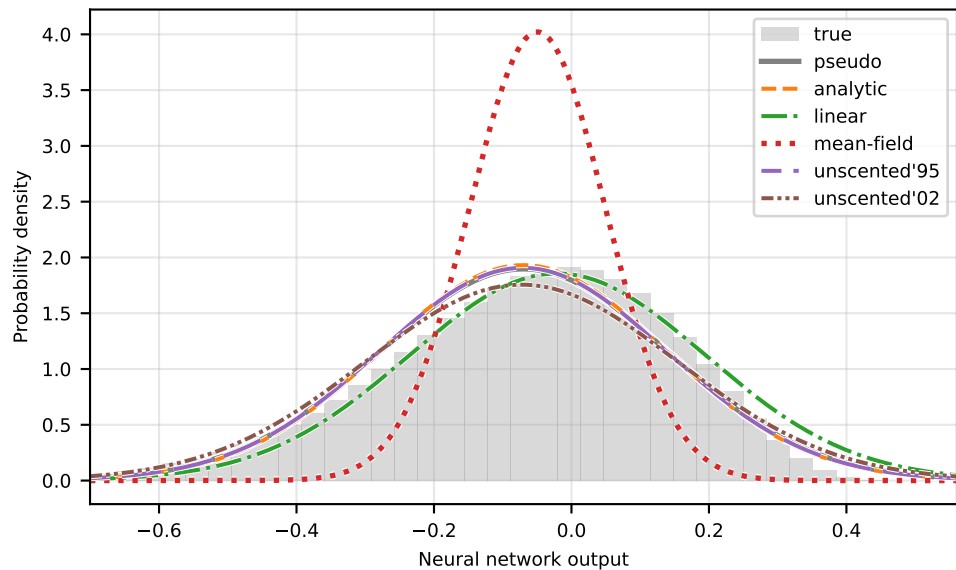

Figure 39: Probability distributions for Network(architecture=wide, weights=trained, activation=gelu), variance=small

| distribution | $\mu$ | $\sigma^2$ |
|---|---|---|
| pseudo-true $(Y_1)$ | $-5.119 \times 10^{-1} \pm 5.3 \times 10^{-6}$ | $1.097 \times 10^0 \pm 1.3 \times 10^{-5}$ |
| analytic | $-4.912 \times 10^{-1}$ | $1.173 \times 10^0$ |
| mean-field | $-6.076 \times 10^{-1}$ | $5.327 \times 10^{-1}$ |
| linear | $-2.054 \times 10^{-2}$ | $4.617 \times 10^0$ |
| unscented'95 | $-4.392 \times 10^{-1}$ | $9.772 \times 10^{-1}$ |
| unscented'02 | $-5.229 \times 10^0$ | $5.888 \times 10^{+1}$ |

Table 64: Comparison of moments for Network(architecture=wide, weights=trained, activation=gelu), variance=medium

| distribution | $d_{\mathrm{W}}(\cdot, Y_0)$ | $D_{\mathrm{KL}}(Y_1 \parallel \cdot)$ |
|---|---|---|
| pseudo-true $(Y_1)$ | $7.352 \times 10^{-2} \pm 1.1 \times 10^{-5}$ | $0$ |
| analytic | $7.357 \times 10^{-2} \pm 1.3 \times 10^{-5}$ | $1.333 \times 10^{-3} \pm 4.1 \times 10^{-7}$ |
| mean-field | $2.944 \times 10^{-1} \pm 9.8 \times 10^{-6}$ | $1.082 \times 10^{-1} \pm 3.0 \times 10^{-6}$ |
| linear | $9.054 \times 10^{-1} \pm 7.6 \times 10^{-6}$ | $9.951 \times 10^{-1} \pm 2.0 \times 10^{-5}$ |
| unscented'95 | $9.088 \times 10^{-2} \pm 1.1 \times 10^{-5}$ | $5.652 \times 10^{-3} \pm 8.1 \times 10^{-7}$ |
| unscented'02 | $6.447 \times 10^0 \pm 2.5 \times 10^{-5}$ | $3.447 \times 10^{+1} \pm 4.5 \times 10^{-4}$ |

Table 65: Comparison of statistical distances for Network(architecture=wide, weights=trained, activation=gelu), variance=medium

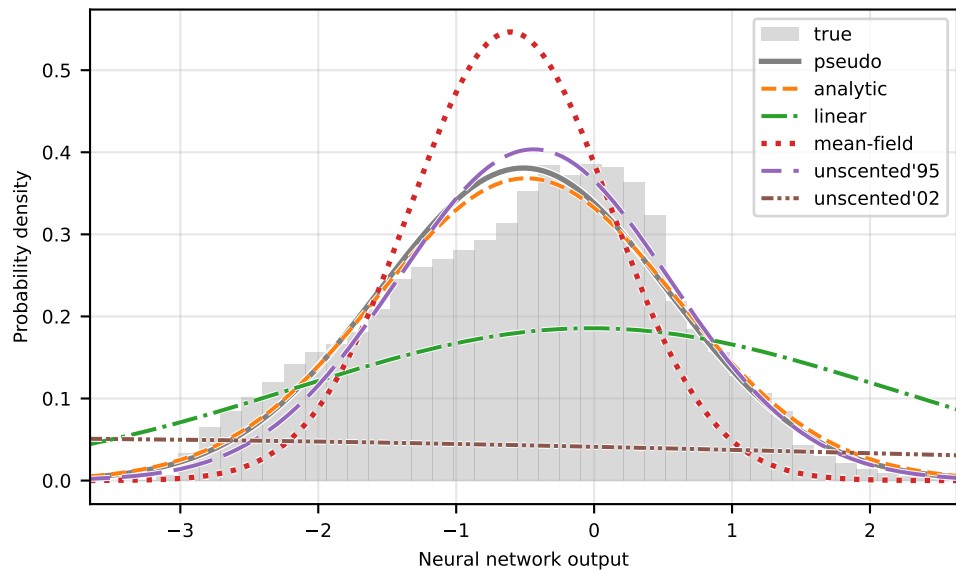

Figure 40: Probability distributions for Network(architecture=wide, weights=trained, activation=gelu), variance=medium

| distribution | $\mu$ | $\sigma^2$ |
|---|---|---|
| pseudo-true ($Y_1$) | $+2.787 \times 10^0 \pm 7.5 \times 10^{-5}$ | $3.069 \times 10^{+1} \pm 1.5 \times 10^{-3}$ |
| analytic | $+2.645 \times 10^0$ | $2.931 \times 10^{+1}$ |
| mean-field | $+2.512 \times 10^0$ | $2.951 \times 10^{+1}$ |
| linear | $-2.054 \times 10^{-2}$ | $4.617 \times 10^{+2}$ |
| unscented'95 | $+5.113 \times 10^0$ | $3.012 \times 10^{+1}$ |
| unscented'02 | $-5.207 \times 10^{+2}$ | $5.427 \times 10^{+5}$ |

Table 66: Comparison of moments for Network(architecture=wide, weights=trained, activation=gelu), variance=large

| distribution | $d_{\mathrm{W}}(\cdot, Y_0)$ | $D_{\mathrm{KL}}(Y_1 \parallel \cdot)$ |
|---|---|---|
| pseudo-true ($Y_1$) | $4.469 \times 10^{-1} \pm 3.9 \times 10^{-5}$ | $0$ |
| analytic | $4.410 \times 10^{-1} \pm 3.8 \times 10^{-5}$ | $8.506 \times 10^{-4} \pm 1.3 \times 10^{-6}$ |
| mean-field | $4.407 \times 10^{-1} \pm 4.1 \times 10^{-5}$ | $1.611 \times 10^{-3} \pm 1.3 \times 10^{-6}$ |
| linear | $5.473 \times 10^0 \pm 1.1 \times 10^{-4}$ | $5.794 \times 10^0 \pm 3.6 \times 10^{-4}$ |
| unscented'95 | $1.034 \times 10^0 \pm 4.3 \times 10^{-5}$ | $8.826 \times 10^{-2} \pm 8.0 \times 10^{-6}$ |
| unscented'02 | $3.086 \times 10^{+2} \pm 3.9 \times 10^{-3}$ | $1.330 \times 10^{+4} \pm 6.6 \times 10^{-1}$ |

Table 67: Comparison of statistical distances for Network(architecture=wide, weights=trained, activation=gelu), variance=large

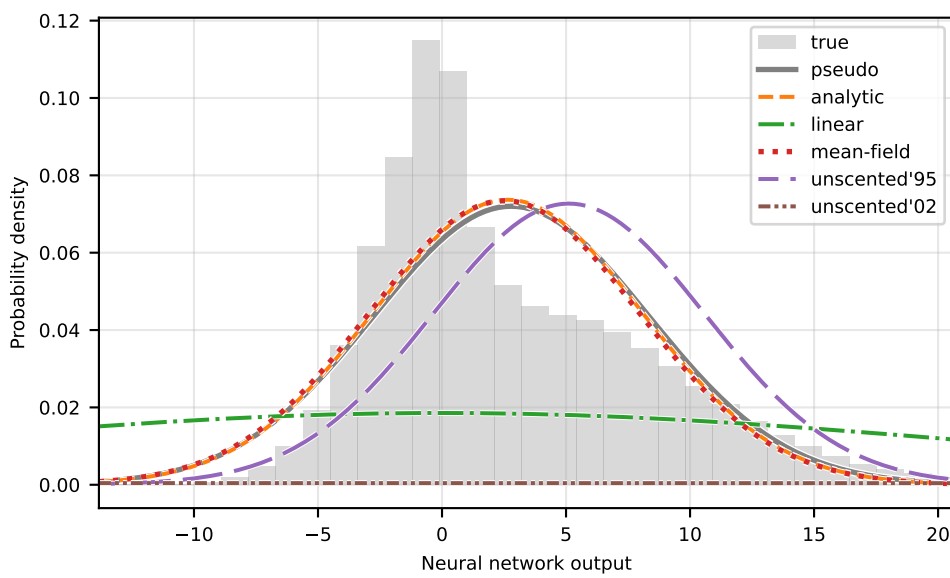

Figure 41: Probability distributions for Network(architecture=wide, weights=trained, activation=gelu), variance=large

| distribution | $\mu$ | $\sigma^2$ |
|---|---|---|
| pseudo-true ($Y_1$) | $-5.070 \times 10^0 \pm 3.8 \times 10^{-7}$ | $2.447 \times 10^{-1} \pm 1.4 \times 10^{-6}$ |
| analytic | $-5.069 \times 10^0$ | $2.424 \times 10^{-1}$ |
| mean-field | $-5.064 \times 10^0$ | $1.377 \times 10^{-1}$ |
| linear | $-5.098 \times 10^0$ | $2.464 \times 10^{-1}$ |
| unscented'95 | $-5.070 \times 10^0$ | $2.444 \times 10^{-1}$ |
| unscented'02 | $-5.069 \times 10^0$ | $2.480 \times 10^{-1}$ |

Table 68: Comparison of moments for Network(architecture=wide, weights=initialized, activation=gelu residual), variance=small

| distribution | $d_{\mathrm{W}}(\cdot, Y_0)$ | $D_{\mathrm{KL}}(Y_1 \parallel \cdot)$ |
|---|---|---|
| pseudo-true ($Y_1$) | $2.662 \times 10^{-2} \pm 3.2 \times 10^{-6}$ | $0$ |
| analytic | $2.694 \times 10^{-2} \pm 3.0 \times 10^{-6}$ | $2.256 \times 10^{-5} \pm 2.5 \times 10^{-8}$ |
| mean-field | $1.408 \times 10^{-1} \pm 2.7 \times 10^{-6}$ | $6.890 \times 10^{-2} \pm 1.2 \times 10^{-6}$ |
| linear | $3.954 \times 10^{-2} \pm 5.2 \times 10^{-7}$ | $1.593 \times 10^{-3} \pm 4.2 \times 10^{-8}$ |
| unscented'95 | $2.646 \times 10^{-2} \pm 3.4 \times 10^{-6}$ | $4.166 \times 10^{-7} \pm 3.2 \times 10^{-9}$ |
| unscented'02 | $2.755 \times 10^{-2} \pm 2.8 \times 10^{-6}$ | $4.789 \times 10^{-5} \pm 3.9 \times 10^{-8}$ |

Table 69: Comparison of statistical distances for Network(architecture=wide, weights=initialized, activation=gelu residual), variance=small

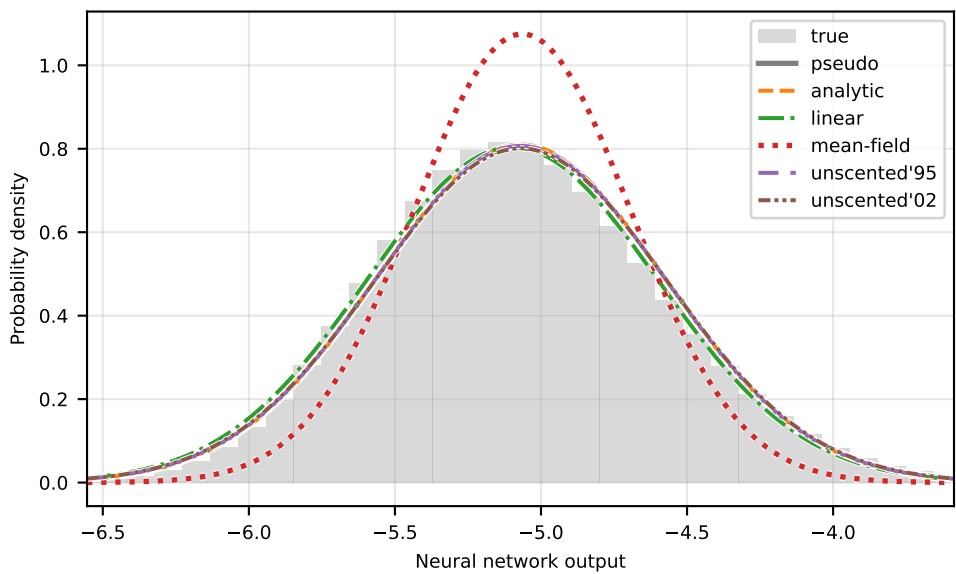

Figure 42: Probability distributions for Network(architecture=wide, weights=initialized, activation=gelu residual), variance=small

| distribution | $\mu$ | $\sigma^2$ |
|---|---|---|
| pseudo-true $(Y_1)$ | $-6.044 \times 10^0 \pm 1.3 \times 10^{-5}$ | $8.996 \times 10^0 \pm 1.2 \times 10^{-4}$ |
| analytic | $-6.034 \times 10^0$ | $7.057 \times 10^0$ |
| mean-field | $-6.139 \times 10^0$ | $8.948 \times 10^0$ |
| linear | $-5.098 \times 10^0$ | $2.464 \times 10^{+1}$ |
| unscented'95 | $-6.119 \times 10^0$ | $6.362 \times 10^0$ |
| unscented'02 | $-2.249 \times 10^0$ | $4.087 \times 10^{+1}$ |

Table 70: Comparison of moments for Network(architecture=wide, weights=initialized, activation=gelu residual), variance=medium

| distribution | $d_{\mathrm{W}}(\cdot, Y_0)$ | $D_{\mathrm{KL}}(Y_1 \parallel \cdot)$ |
|---|---|---|
| pseudo-true $(Y_1)$ | $1.041 \times 10^{-1} \pm 2.0 \times 10^{-5}$ | $0$ |
| analytic | $2.224 \times 10^{-1} \pm 1.0 \times 10^{-5}$ | $1.362 \times 10^{-2} \pm 1.4 \times 10^{-6}$ |
| mean-field | $1.218 \times 10^{-1} \pm 1.7 \times 10^{-5}$ | $5.037 \times 10^{-4} \pm 1.3 \times 10^{-7}$ |
| linear | $9.831 \times 10^{-1} \pm 1.2 \times 10^{-5}$ | $4.156 \times 10^{-1} \pm 1.2 \times 10^{-5}$ |
| unscented'95 | $2.747 \times 10^{-1} \pm 1.1 \times 10^{-5}$ | $2.714 \times 10^{-2} \pm 1.9 \times 10^{-6}$ |
| unscented'02 | $2.438 \times 10^0 \pm 1.5 \times 10^{-5}$ | $1.816 \times 10^0 \pm 3.2 \times 10^{-5}$ |

Table 71: Comparison of statistical distances for Network(architecture=wide, weights=initialized, activation=gelu residual), variance=medium

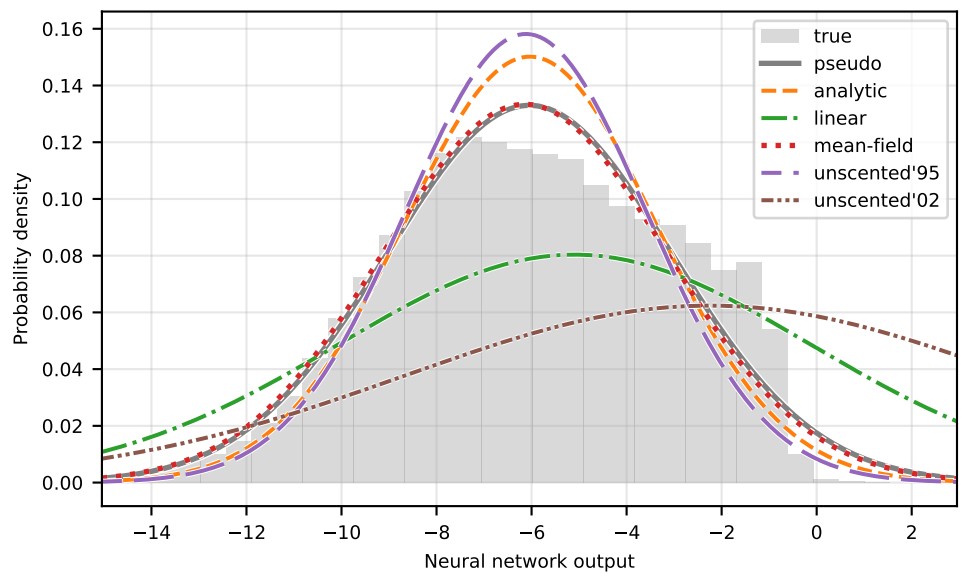

Figure 43: Probability distributions for Network(architecture=wide, weights=initialized, activation=gelu residual), variance=medium

| distribution | $\mu$ | $\sigma^2$ |
|---|---|---|
| pseudo-true $(Y_1)$ | $-3.746 \times 10^{+1} \pm 2.6 \times 10^{-4}$ | $7.905 \times 10^{+2} \pm 1.9 \times 10^{-2}$ |
| analytic | $-3.854 \times 10^{+1}$ | $5.569 \times 10^{+2}$ |
| mean-field | $-3.505 \times 10^{+1}$ | $7.171 \times 10^{+2}$ |
| linear | $-5.098 \times 10^{0}$ | $2.464 \times 10^{+3}$ |
| unscented'95 | $-3.841 \times 10^{+1}$ | $6.145 \times 10^{+2}$ |
| unscented'02 | $+2.797 \times 10^{+2}$ | $1.647 \times 10^{+5}$ |

Table 72: Comparison of moments for Network(architecture=wide, weights=initialized, activation=gelu residual), variance=large

| distribution | $d_{\mathrm{W}}(\cdot, Y_0)$ | $D_{\mathrm{KL}}(Y_1 \parallel \cdot)$ |
|---|---|---|
| pseudo-true $(Y_1)$ | $5.608 \times 10^{-1} \pm 6.8 \times 10^{-5}$ | $0$ |
| analytic | $9.376 \times 10^{-1} \pm 4.0 \times 10^{-5}$ | $2.814 \times 10^{-2} \pm 3.3 \times 10^{-6}$ |
| mean-field | $6.046 \times 10^{-1} \pm 5.8 \times 10^{-5}$ | $5.971 \times 10^{-3} \pm 1.5 \times 10^{-6}$ |
| linear | $6.194 \times 10^{0} \pm 6.0 \times 10^{-5}$ | $1.153 \times 10^{0} \pm 3.7 \times 10^{-5}$ |
| unscented'95 | $8.150 \times 10^{-1} \pm 2.7 \times 10^{-5}$ | $1.519 \times 10^{-2} \pm 2.5 \times 10^{-6}$ |
| unscented'02 | $7.554 \times 10^{+1} \pm 4.6 \times 10^{-4}$ | $1.646 \times 10^{+2} \pm 3.9 \times 10^{-3}$ |

Table 73: Comparison of statistical distances for Network(architecture=wide, weights=initialized, activation=gelu residual), variance=large

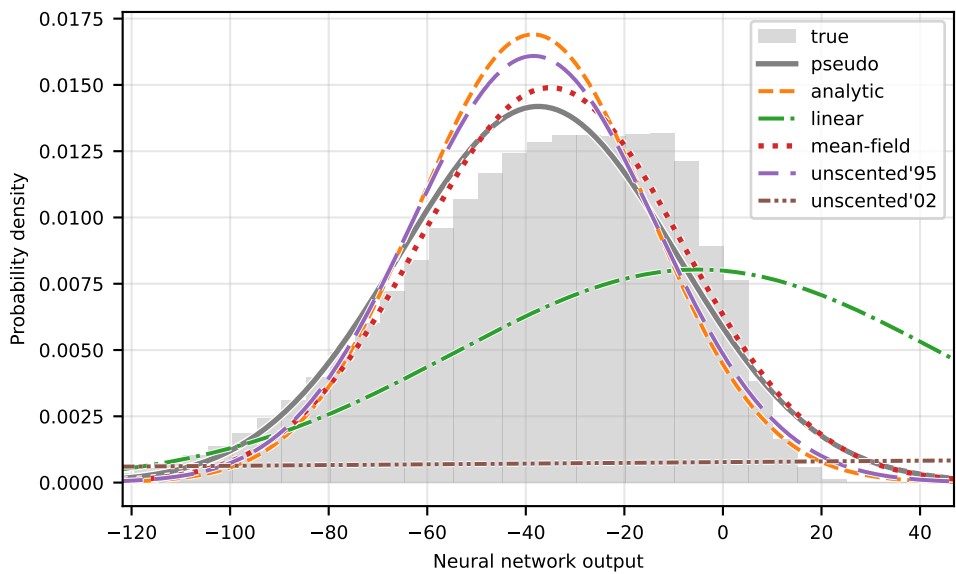

Figure 44: Probability distributions for Network(architecture=wide, weights=initialized, activation=gelu residual), variance=large

| distribution | $\mu$ | $\sigma^2$ |
|---|---|---|
| pseudo-true ($Y_1$) | $+1.997 \times 10^{-2} \pm 3.2 \times 10^{-7}$ | $7.824 \times 10^{-2} \pm 7.3 \times 10^{-7}$ |
| analytic | $+1.749 \times 10^{-2}$ | $7.865 \times 10^{-2}$ |
| mean-field | $+3.802 \times 10^{-2}$ | $1.411 \times 10^{-1}$ |
| linear | $+6.045 \times 10^{-2}$ | $7.470 \times 10^{-2}$ |
| unscented'95 | $+1.659 \times 10^{-2}$ | $7.546 \times 10^{-2}$ |
| unscented'02 | $+1.251 \times 10^{-2}$ | $7.929 \times 10^{-2}$ |

Table 74: Comparison of moments for Network(architecture=wide, weights=trained, activation=gelu residual), variance=small

| distribution | $d_{\mathrm{W}}(\cdot, Y_0)$ | $D_{\mathrm{KL}}(Y_1 \parallel \cdot)$ |
|---|---|---|
| pseudo-true ($Y_1$) | $1.176 \times 10^{-2} \pm 7.0 \times 10^{-6}$ | $0$ |
| analytic | $1.217 \times 10^{-2} \pm 6.1 \times 10^{-6}$ | $4.612 \times 10^{-5} \pm 2.7 \times 10^{-8}$ |
| mean-field | $1.536 \times 10^{-1} \pm 3.5 \times 10^{-6}$ | $1.089 \times 10^{-1} \pm 3.8 \times 10^{-6}$ |
| linear | $7.708 \times 10^{-2} \pm 3.3 \times 10^{-6}$ | $1.100 \times 10^{-2} \pm 2.1 \times 10^{-7}$ |
| unscented'95 | $1.055 \times 10^{-2} \pm 6.5 \times 10^{-6}$ | $3.966 \times 10^{-4} \pm 1.7 \times 10^{-7}$ |
| unscented'02 | $1.553 \times 10^{-2} \pm 7.9 \times 10^{-6}$ | $4.003 \times 10^{-4} \pm 7.4 \times 10^{-8}$ |

Table 75: Comparison of statistical distances for Network(architecture=wide, weights=trained, activation=gelu residual), variance=small

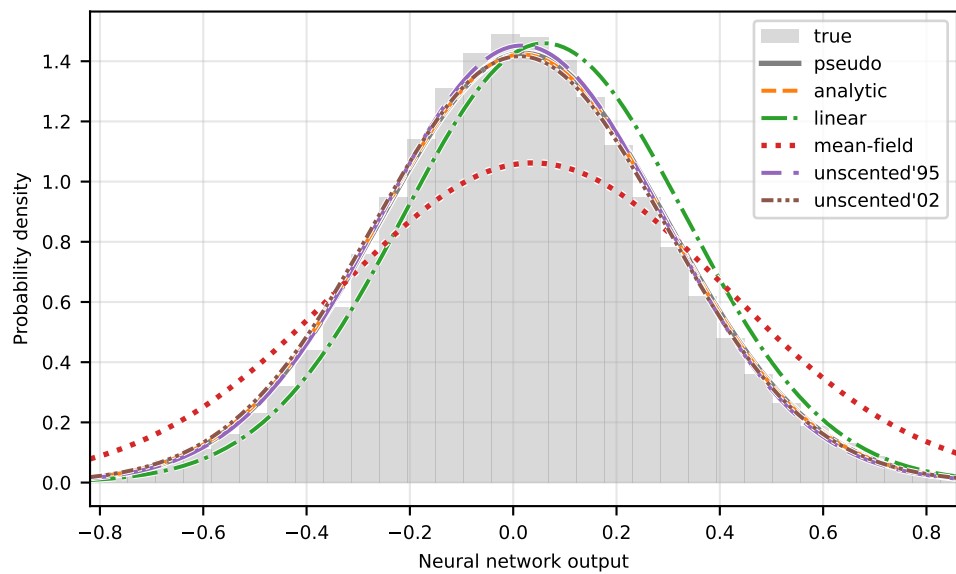

Figure 45: Probability distributions for Network(architecture=wide, weights=trained, activation=gelu residual), variance=small

| distribution | $\mu$ | $\sigma^2$ |
|---|---|---|
| pseudo-true ($Y_1$) | $-7.433 \times 10^{-1} \pm 1.2 \times 10^{-5}$ | $2.825 \times 10^{0} \pm 8.8 \times 10^{-5}$ |
| analytic | $-7.555 \times 10^{-1}$ | $3.514 \times 10^{0}$ |
| mean-field | $-8.641 \times 10^{-1}$ | $9.025 \times 10^{0}$ |
| linear | $+6.045 \times 10^{-2}$ | $7.470 \times 10^{0}$ |
| unscented'95 | $-7.988 \times 10^{-1}$ | $1.499 \times 10^{0}$ |
| unscented'02 | $-4.733 \times 10^{0}$ | $5.343 \times 10^{+1}$ |

Table 76: Comparison of moments for Network(architecture=wide, weights=trained, activation=gelu residual), variance=medium

| distribution | $d_{\mathrm{W}}(\cdot, Y_0)$ | $D_{\mathrm{KL}}(Y_1 \parallel \cdot)$ |
|---|---|---|
| pseudo-true ($Y_1$) | $1.348 \times 10^{-1} \pm 2.4 \times 10^{-5}$ | $0$ |
| analytic | $1.977 \times 10^{-1} \pm 1.5 \times 10^{-5}$ | $1.285 \times 10^{-2} \pm 3.8 \times 10^{-6}$ |
| mean-field | $8.580 \times 10^{-1} \pm 1.9 \times 10^{-5}$ | $5.193 \times 10^{-1} \pm 3.4 \times 10^{-5}$ |
| linear | $8.027 \times 10^{-1} \pm 1.2 \times 10^{-5}$ | $4.504 \times 10^{-1} \pm 2.9 \times 10^{-5}$ |
| unscented'95 | $2.733 \times 10^{-1} \pm 1.2 \times 10^{-5}$ | $8.263 \times 10^{-2} \pm 7.3 \times 10^{-6}$ |
| unscented'02 | $4.372 \times 10^{0} \pm 3.8 \times 10^{-5}$ | $1.031 \times 10^{+1} \pm 3.7 \times 10^{-4}$ |

Table 77: Comparison of statistical distances for Network(architecture=wide, weights=trained, activation=gelu residual), variance=medium

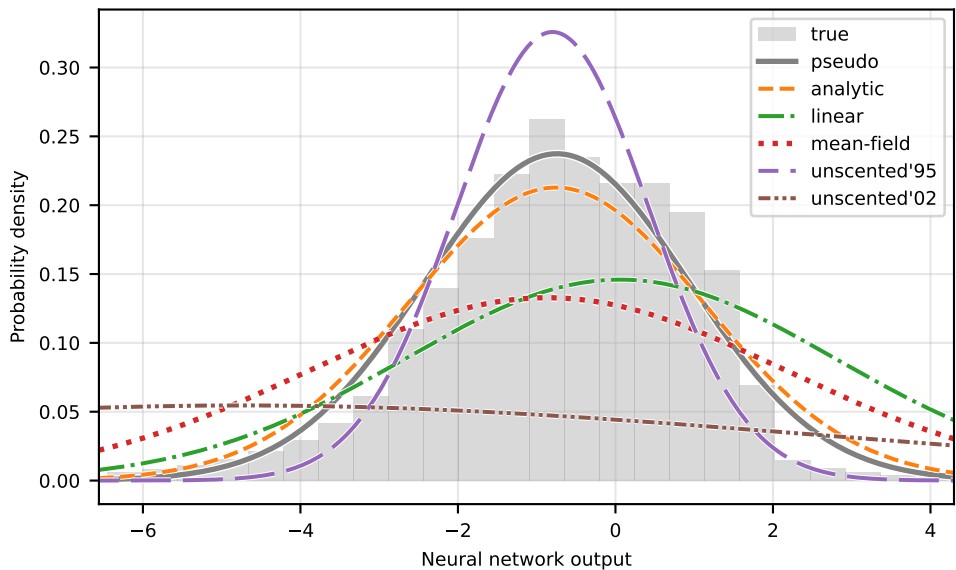

Figure 46: Probability distributions for Network(architecture=wide, weights=trained, activation=gelu residual), variance=medium

| distribution | $\mu$ | $\sigma^2$ |
|---|---|---|
| pseudo-true $(Y_1)$ | $-9.417 \times 10^0 \pm 2.5 \times 10^{-4}$ | $5.784 \times 10^{+2} \pm 1.3 \times 10^{-2}$ |
| analytic | $-9.746 \times 10^0$ | $4.943 \times 10^{+2}$ |
| mean-field | $-6.172 \times 10^0$ | $7.188 \times 10^{+2}$ |
| linear | $+6.045 \times 10^{-2}$ | $7.470 \times 10^{+2}$ |
| unscented'95 | $-8.083 \times 10^0$ | $6.251 \times 10^{+2}$ |
| unscented'02 | $-4.789 \times 10^{+2}$ | $4.595 \times 10^{+5}$ |

Table 78: Comparison of moments for Network(architecture=wide, weights=trained, activation=gelu residual), variance=large

| distribution | $d_{\mathrm{W}}(\cdot, Y_0)$ | $D_{\mathrm{KL}}(Y_1 \parallel \cdot)$ |
|---|---|---|
| pseudo-true $(Y_1)$ | $3.188 \times 10^{-1} \pm 6.6 \times 10^{-5}$ | $0$ |
| analytic | $3.979 \times 10^{-1} \pm 6.8 \times 10^{-5}$ | $5.957 \times 10^{-3} \pm 1.6 \times 10^{-6}$ |
| mean-field | $8.734 \times 10^{-1} \pm 7.2 \times 10^{-5}$ | $2.183 \times 10^{-2} \pm 3.0 \times 10^{-6}$ |
| linear | $1.954 \times 10^0 \pm 5.5 \times 10^{-5}$ | $9.551 \times 10^{-2} \pm 5.9 \times 10^{-6}$ |
| unscented'95 | $4.304 \times 10^{-1} \pm 6.8 \times 10^{-5}$ | $3.085 \times 10^{-3} \pm 1.0 \times 10^{-6}$ |
| unscented'02 | $1.325 \times 10^{+2} \pm 7.4 \times 10^{-4}$ | $5.839 \times 10^{+2} \pm 1.3 \times 10^{-2}$ |

Table 79: Comparison of statistical distances for Network(architecture=wide, weights=trained, activation=gelu residual), variance=large

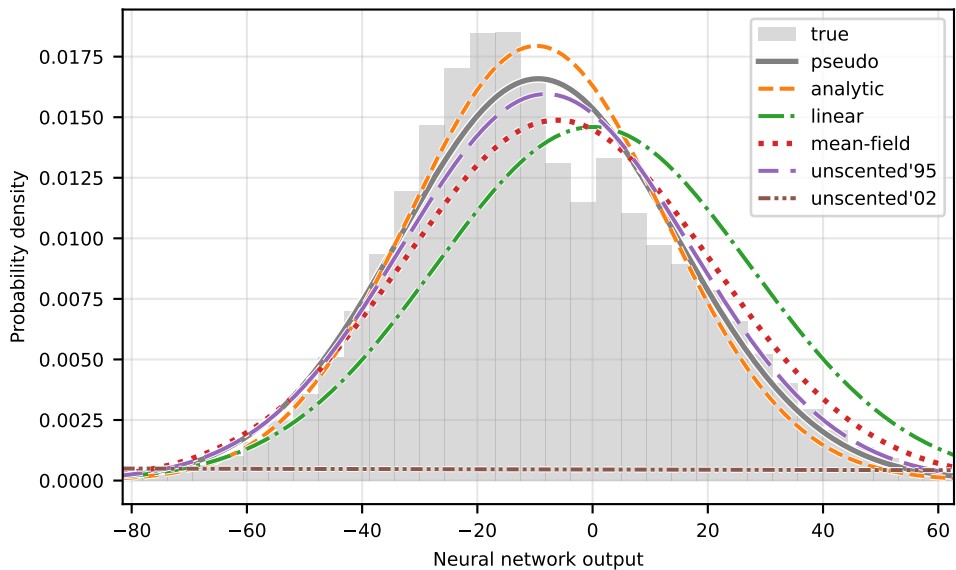

Figure 47: Probability distributions for Network(architecture=wide, weights=trained, activation=gelu residual), variance=large

| distribution | $\mu$ | $\sigma^2$ |
|---|---|---|
| pseudo-true $(Y_1)$ | $-2.548 \times 10^0 \pm 3.6 \times 10^{-7}$ | $1.672 \times 10^{-2} \pm 1.7 \times 10^{-7}$ |
| analytic | $-2.550 \times 10^0$ | $1.674 \times 10^{-2}$ |
| mean-field | $-2.551 \times 10^0$ | $7.878 \times 10^{-3}$ |
| linear | $-2.534 \times 10^0$ | $1.529 \times 10^{-2}$ |
| unscented'95 | $-2.552 \times 10^0$ | $1.753 \times 10^{-2}$ |
| unscented'02 | $-2.534 \times 10^0$ | $1.529 \times 10^{-2}$ |

Table 80: Comparison of moments for Network(architecture=wide, weights=initialized, activation=relu), variance=small

| distribution | $d_{\mathrm{W}}(\cdot, Y_0)$ | $D_{\mathrm{KL}}(Y_1 \parallel \cdot)$ |
|---|---|---|
| pseudo-true $(Y_1)$ | $1.237 \times 10^{-2} \pm 3.0 \times 10^{-6}$ | $0$ |
| analytic | $1.321 \times 10^{-2} \pm 2.2 \times 10^{-6}$ | $7.681 \times 10^{-5} \pm 3.2 \times 10^{-8}$ |
| mean-field | $9.102 \times 10^{-2} \pm 1.7 \times 10^{-6}$ | $1.120 \times 10^{-1} \pm 2.8 \times 10^{-6}$ |
| linear | $4.066 \times 10^{-2} \pm 1.5 \times 10^{-6}$ | $8.034 \times 10^{-3} \pm 3.8 \times 10^{-7}$ |
| unscented'95 | $1.699 \times 10^{-2} \pm 2.1 \times 10^{-6}$ | $8.407 \times 10^{-4} \pm 2.3 \times 10^{-7}$ |
| unscented'02 | $4.066 \times 10^{-2} \pm 1.5 \times 10^{-6}$ | $8.034 \times 10^{-3} \pm 3.8 \times 10^{-7}$ |

Table 81: Comparison of statistical distances for Network(architecture=wide, weights=initialized, activation=relu), variance=small

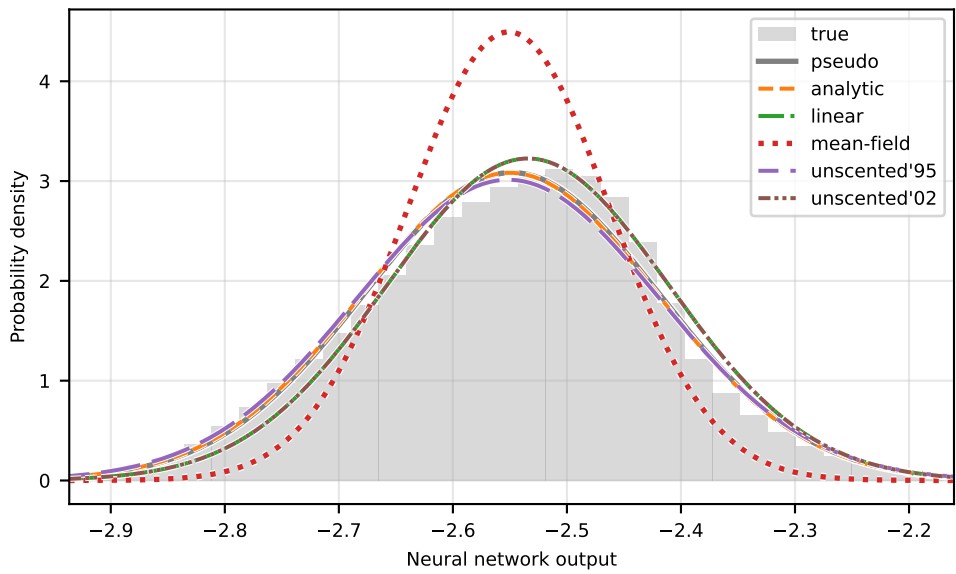

Figure 48: Probability distributions for Network(architecture=wide, weights=initialized, activation=relu), variance=small

| distribution | $\mu$ | $\sigma^2$ |
|---|---|---|
| pseudo-true ($Y_1$) | $-2.086 \times 10^0 \pm 8.0 \times 10^{-6}$ | $5.616 \times 10^{-1} \pm 1.5 \times 10^{-5}$ |
| analytic | $-2.047 \times 10^0$ | $5.068 \times 10^{-1}$ |
| mean-field | $-2.129 \times 10^0$ | $3.972 \times 10^{-1}$ |
| linear | $-2.534 \times 10^0$ | $1.529 \times 10^0$ |
| unscented'95 | $-1.992 \times 10^0$ | $6.390 \times 10^{-1}$ |
| unscented'02 | $+1.315 \times 10^{+1}$ | $4.933 \times 10^{+2}$ |

Table 82: Comparison of moments for Network(architecture=wide, weights=initialized, activation=relu), variance=medium

| distribution | $d_{\mathrm{W}}(\cdot, Y_0)$ | $D_{\mathrm{KL}}(Y_1 \parallel \cdot)$ |
|---|---|---|
| pseudo-true ($Y_1$) | $2.010 \times 10^{-1} \pm 1.2 \times 10^{-5}$ | $0$ |
| analytic | $2.124 \times 10^{-1} \pm 9.6 \times 10^{-6}$ | $3.897 \times 10^{-3} \pm 1.1 \times 10^{-6}$ |
| mean-field | $1.878 \times 10^{-1} \pm 1.0 \times 10^{-5}$ | $2.849 \times 10^{-2} \pm 4.2 \times 10^{-6}$ |
| linear | $5.572 \times 10^{-1} \pm 9.1 \times 10^{-6}$ | $5.394 \times 10^{-1} \pm 2.5 \times 10^{-5}$ |
| unscented'95 | $2.555 \times 10^{-1} \pm 1.4 \times 10^{-5}$ | $1.223 \times 10^{-2} \pm 3.0 \times 10^{-6}$ |
| unscented'02 | $2.468 \times 10^{+1} \pm 1.7 \times 10^{-4}$ | $6.418 \times 10^{+2} \pm 1.7 \times 10^{-2}$ |

Table 83: Comparison of statistical distances for Network(architecture=wide, weights=initialized, activation=relu), variance=medium

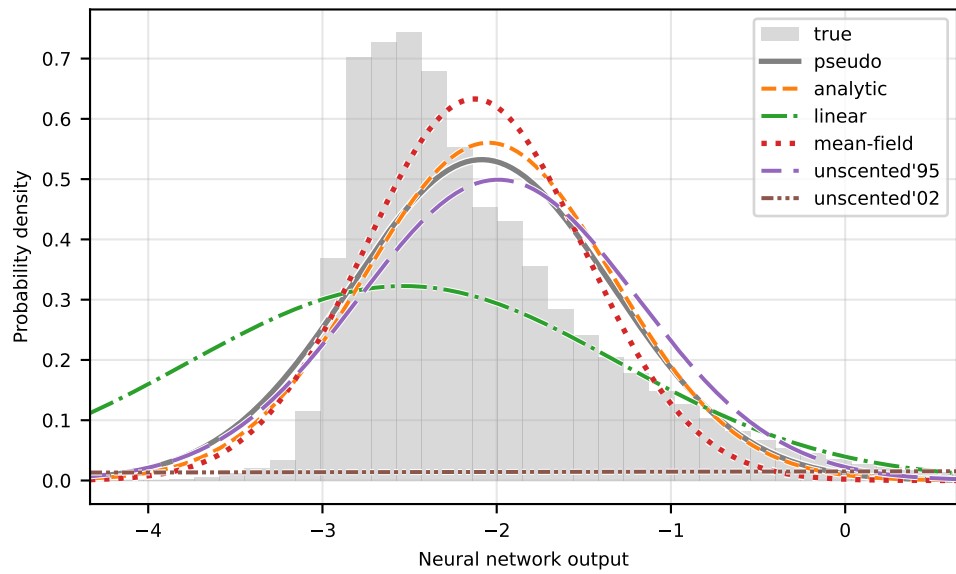

Figure 49: Probability distributions for Network(architecture=wide, weights=initialized, activation=relu), variance=medium

| distribution | $\mu$ | $\sigma^2$ |
|---|---|---|
| pseudo-true $(Y_1)$ | $+9.817 \times 10^{-1} \pm 5.8 \times 10^{-5}$ | $3.271 \times 10^{+1} \pm 1.2 \times 10^{-3}$ |
| analytic | $+8.530 \times 10^{-1}$ | $2.964 \times 10^{+1}$ |
| mean-field | $+5.436 \times 10^{-1}$ | $2.726 \times 10^{+1}$ |
| linear | $-2.534 \times 10^{0}$ | $1.529 \times 10^{+2}$ |
| unscented'95 | $+3.402 \times 10^{0}$ | $3.527 \times 10^{+1}$ |
| unscented'02 | $+7.403 \times 10^{+2}$ | $1.104 \times 10^{+6}$ |

Table 84: Comparison of moments for Network(architecture=wide, weights=initialized, activation=relu), variance=large

| distribution | $d_{\mathrm{W}}(\cdot, Y_0)$ | $D_{\mathrm{KL}}(Y_1 \parallel \cdot)$ |
|---|---|---|
| pseudo-true $(Y_1)$ | $4.287 \times 10^{-1} \pm 4.3 \times 10^{-5}$ | $0$ |
| analytic | $4.185 \times 10^{-1} \pm 4.8 \times 10^{-5}$ | $2.599 \times 10^{-3} \pm 1.6 \times 10^{-6}$ |
| mean-field | $4.187 \times 10^{-1} \pm 4.9 \times 10^{-5}$ | $1.075 \times 10^{-2} \pm 2.8 \times 10^{-6}$ |
| linear | $2.396 \times 10^{0} \pm 5.8 \times 10^{-5}$ | $1.255 \times 10^{0} \pm 7.3 \times 10^{-5}$ |
| unscented'95 | $1.049 \times 10^{0} \pm 4.7 \times 10^{-5}$ | $9.103 \times 10^{-2} \pm 5.8 \times 10^{-6}$ |
| unscented'02 | $4.327 \times 10^{+2} \pm 3.9 \times 10^{-3}$ | $2.522 \times 10^{+4} \pm 8.9 \times 10^{-1}$ |

Table 85: Comparison of statistical distances for Network(architecture=wide, weights=initialized, activation=relu), variance=large

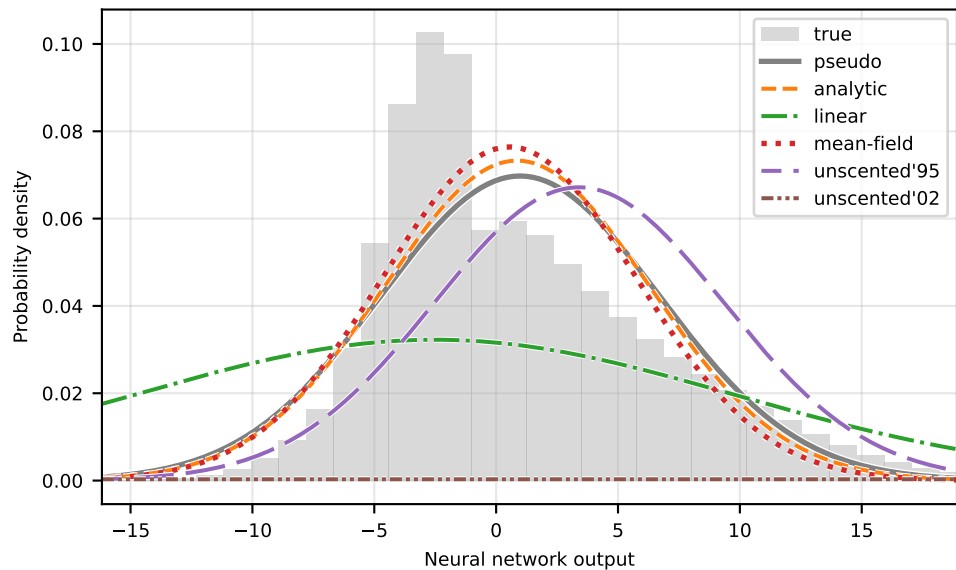

Figure 50: Probability distributions for Network(architecture=wide, weights=initialized, activation=relu), variance=large

| distribution | $\mu$ | $\sigma^2$ |
|---|---|---|
| pseudo-true ($Y_1$) | $-2.870 \times 10^{-1} \pm 4.3 \times 10^{-7}$ | $1.633 \times 10^{-2} \pm 2.5 \times 10^{-7}$ |
| analytic | $-2.877 \times 10^{-1}$ | $1.361 \times 10^{-2}$ |
| mean-field | $-2.822 \times 10^{-1}$ | $8.050 \times 10^{-3}$ |
| linear | $-2.263 \times 10^{-1}$ | $2.322 \times 10^{-2}$ |
| unscented'95 | $-2.794 \times 10^{-1}$ | $1.524 \times 10^{-2}$ |
| unscented'02 | $-2.263 \times 10^{-1}$ | $2.322 \times 10^{-2}$ |

Table 86: Comparison of moments for Network(architecture=wide, weights=trained, activation=relu), variance=small

| distribution | $d_{\mathrm{W}}(\cdot, Y_0)$ | $D_{\mathrm{KL}}(Y_1 \parallel \cdot)$ |
|---|---|---|
| pseudo-true ($Y_1$) | $6.259 \times 10^{-2} \pm 2.8 \times 10^{-6}$ | $0$ |
| analytic | $6.580 \times 10^{-2} \pm 3.4 \times 10^{-6}$ | $7.816 \times 10^{-3} \pm 1.3 \times 10^{-6}$ |
| mean-field | $9.224 \times 10^{-2} \pm 2.3 \times 10^{-6}$ | $1.008 \times 10^{-1} \pm 3.9 \times 10^{-6}$ |
| linear | $1.698 \times 10^{-1} \pm 9.1 \times 10^{-7}$ | $1.479 \times 10^{-1} \pm 4.1 \times 10^{-6}$ |
| unscented'95 | $5.729 \times 10^{-2} \pm 2.5 \times 10^{-6}$ | $2.914 \times 10^{-3} \pm 6.3 \times 10^{-7}$ |
| unscented'02 | $1.698 \times 10^{-1} \pm 9.1 \times 10^{-7}$ | $1.479 \times 10^{-1} \pm 4.1 \times 10^{-6}$ |

Table 87: Comparison of statistical distances for Network(architecture=wide, weights=trained, activation=relu), variance=small

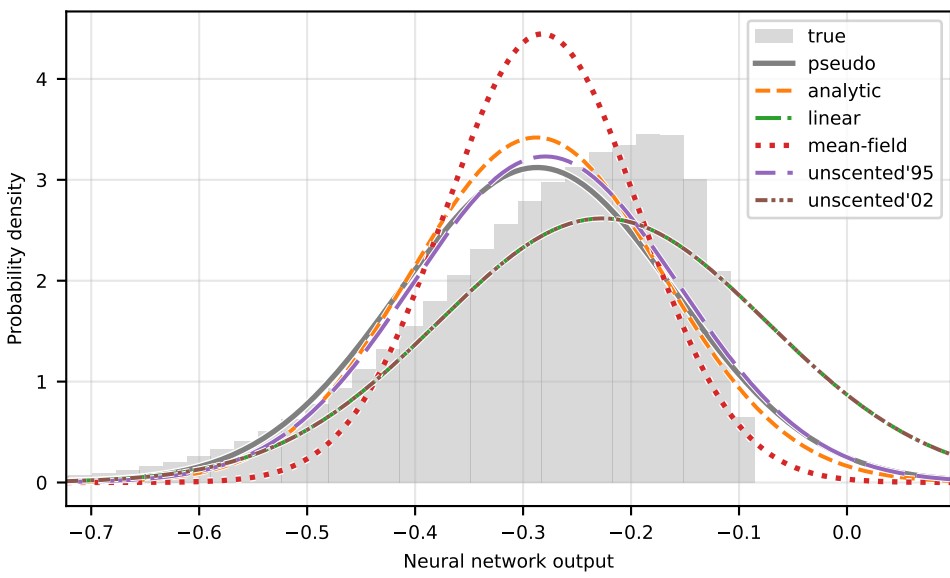

Figure 51: Probability distributions for Network(architecture=wide, weights=trained, activation=relu), variance=small

| distribution | $\mu$ | $\sigma^2$ |
|---|---|---|
| pseudo-true $(Y_1)$ | $-1.894 \times 10^{-1} \pm 8.2 \times 10^{-6}$ | $5.868 \times 10^{-1} \pm 2.1 \times 10^{-5}$ |
| analytic | $-1.771 \times 10^{-1}$ | $6.492 \times 10^{-1}$ |
| mean-field | $-3.079 \times 10^{-1}$ | $4.012 \times 10^{-1}$ |
| linear | $-2.263 \times 10^{-1}$ | $2.322 \times 10^{0}$ |
| unscented'95 | $-1.181 \times 10^{-1}$ | $4.681 \times 10^{-1}$ |
| unscented'02 | $-8.244 \times 10^{0}$ | $1.309 \times 10^{+2}$ |

Table 88: Comparison of moments for Network(architecture=wide, weights=trained, activation=relu), variance=medium

| distribution | $d_{\mathrm{W}}(\cdot, Y_0)$ | $D_{\mathrm{KL}}(Y_1 \parallel \cdot)$ |
|---|---|---|
| pseudo-true $(Y_1)$ | $1.485 \times 10^{-1} \pm 1.3 \times 10^{-5}$ | $0$ |
| analytic | $1.619 \times 10^{-1} \pm 1.5 \times 10^{-5}$ | $2.772 \times 10^{-3} \pm 2.0 \times 10^{-6}$ |
| mean-field | $1.360 \times 10^{-1} \pm 1.1 \times 10^{-5}$ | $4.392 \times 10^{-2} \pm 6.1 \times 10^{-6}$ |
| linear | $7.349 \times 10^{-1} \pm 1.4 \times 10^{-5}$ | $7.923 \times 10^{-1} \pm 5.2 \times 10^{-5}$ |
| unscented'95 | $1.910 \times 10^{-1} \pm 1.2 \times 10^{-5}$ | $1.618 \times 10^{-2} \pm 3.1 \times 10^{-6}$ |
| unscented'02 | $1.233 \times 10^{+1} \pm 1.2 \times 10^{-4}$ | $1.636 \times 10^{+2} \pm 5.8 \times 10^{-3}$ |

Table 89: Comparison of statistical distances for Network(architecture=wide, weights=trained, activation=relu), variance=medium

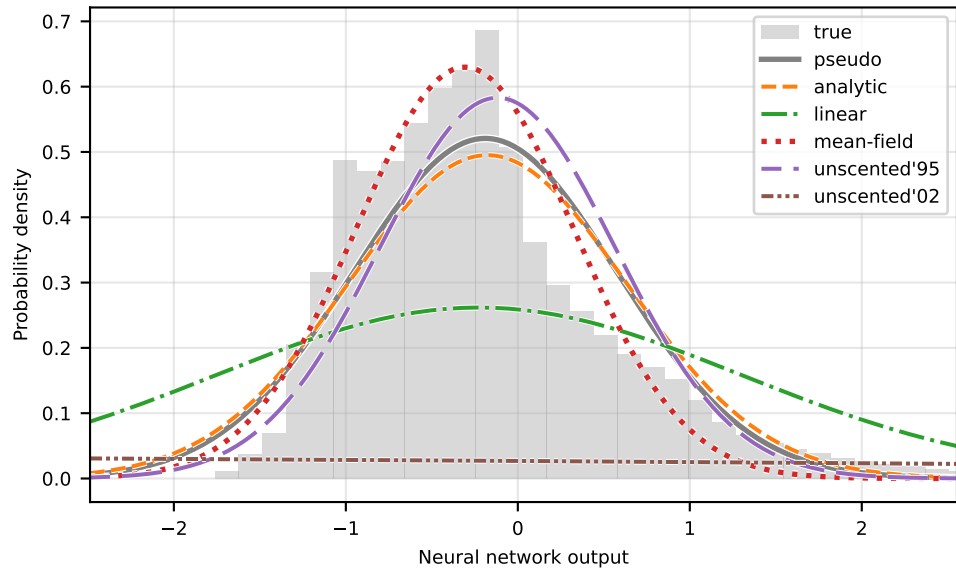

Figure 52: Probability distributions for Network(architecture=wide, weights=trained, activation=relu), variance=medium

| distribution | $\mu$ | $\sigma^2$ |
|---|---|---|
| pseudo-true ($Y_1$) | $+5.938 \times 10^0 \pm 6.5 \times 10^{-5}$ | $3.707 \times 10^{+1} \pm 1.3 \times 10^{-3}$ |
| analytic | $+5.670 \times 10^0$ | $2.890 \times 10^{+1}$ |
| mean-field | $+5.268 \times 10^0$ | $2.730 \times 10^{+1}$ |
| linear | $-2.263 \times 10^{-1}$ | $2.322 \times 10^{+2}$ |
| unscented'95 | $+8.636 \times 10^0$ | $3.085 \times 10^{+1}$ |
| unscented'02 | $-5.028 \times 10^{+2}$ | $5.054 \times 10^{+5}$ |

Table 90: Comparison of moments for Network(architecture=wide, weights=trained, activation=relu), variance=large

| distribution | $d_{\mathrm{W}}(\cdot, Y_0)$ | $D_{\mathrm{KL}}(Y_1 \parallel \cdot)$ |
|---|---|---|
| pseudo-true ($Y_1$) | $5.875 \times 10^{-1} \pm 4.0 \times 10^{-5}$ | $0$ |
| analytic | $5.658 \times 10^{-1} \pm 3.0 \times 10^{-5}$ | $1.523 \times 10^{-2} \pm 4.1 \times 10^{-6}$ |
| mean-field | $5.534 \times 10^{-1} \pm 2.9 \times 10^{-5}$ | $2.724 \times 10^{-2} \pm 5.0 \times 10^{-6}$ |
| linear | $3.429 \times 10^0 \pm 5.0 \times 10^{-5}$ | $2.228 \times 10^0 \pm 1.1 \times 10^{-4}$ |
| unscented'95 | $1.235 \times 10^0 \pm 3.1 \times 10^{-5}$ | $1.061 \times 10^{-1} \pm 4.9 \times 10^{-6}$ |
| unscented'02 | $2.846 \times 10^{+2} \pm 2.6 \times 10^{-3}$ | $1.030 \times 10^{+4} \pm 3.7 \times 10^{-1}$ |

Table 91: Comparison of statistical distances for Network(architecture=wide, weights=trained, activation=relu), variance=large

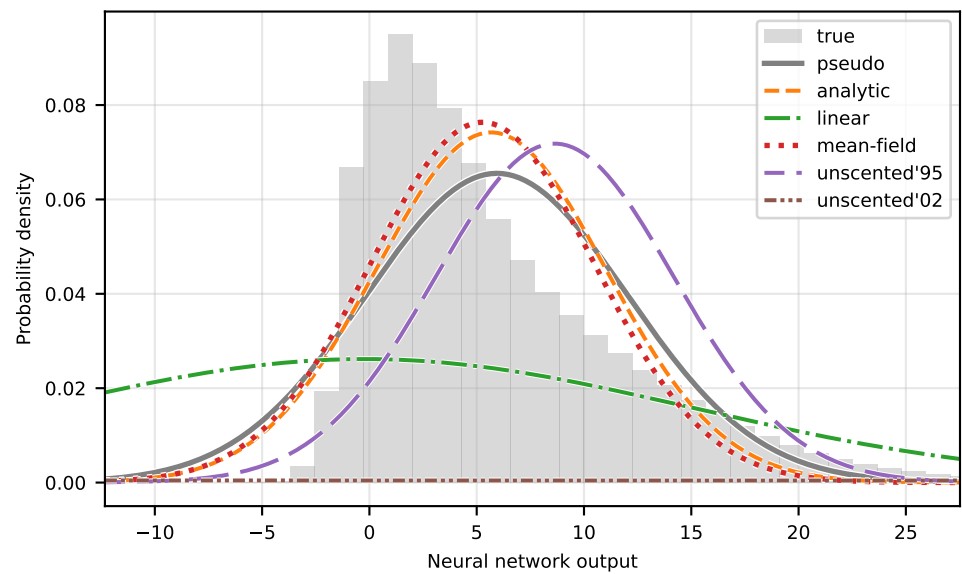

Figure 53: Probability distributions for Network(architecture=wide, weights=trained, activation=relu), variance=large

| distribution | $\mu$ | $\sigma^2$ |
|---|---|---|
| pseudo-true $(Y_1)$ | $-5.429 \times 10^0 \pm 8.7 \times 10^{-7}$ | $2.035 \times 10^{-1} \pm 1.8 \times 10^{-6}$ |
| analytic | $-5.427 \times 10^0$ | $1.967 \times 10^{-1}$ |
| mean-field | $-5.403 \times 10^0$ | $1.344 \times 10^{-1}$ |
| linear | $-5.335 \times 10^0$ | $2.681 \times 10^{-1}$ |
| unscented'95 | $-5.439 \times 10^0$ | $2.195 \times 10^{-1}$ |
| unscented'02 | $-5.335 \times 10^0$ | $2.681 \times 10^{-1}$ |

Table 92: Comparison of moments for Network(architecture=wide, weights=initialized, activation=relu residual), variance=small

| distribution | $d_{\mathrm{W}}(\cdot, Y_0)$ | $D_{\mathrm{KL}}(Y_1 \parallel \cdot)$ |
|---|---|---|
| pseudo-true $(Y_1)$ | $2.655 \times 10^{-2} \pm 3.6 \times 10^{-6}$ | $0$ |
| analytic | $2.999 \times 10^{-2} \pm 3.1 \times 10^{-6}$ | $2.979 \times 10^{-4} \pm 1.5 \times 10^{-7}$ |
| mean-field | $1.121 \times 10^{-1} \pm 2.0 \times 10^{-6}$ | $3.923 \times 10^{-2} \pm 1.5 \times 10^{-6}$ |
| linear | $1.608 \times 10^{-1} \pm 1.9 \times 10^{-6}$ | $4.236 \times 10^{-2} \pm 1.6 \times 10^{-6}$ |
| unscented'95 | $2.274 \times 10^{-2} \pm 3.1 \times 10^{-6}$ | $1.712 \times 10^{-3} \pm 3.6 \times 10^{-7}$ |
| unscented'02 | $1.608 \times 10^{-1} \pm 1.9 \times 10^{-6}$ | $4.236 \times 10^{-2} \pm 1.6 \times 10^{-6}$ |

Table 93: Comparison of statistical distances for Network(architecture=wide, weights=initialized, activation=relu residual), variance=small

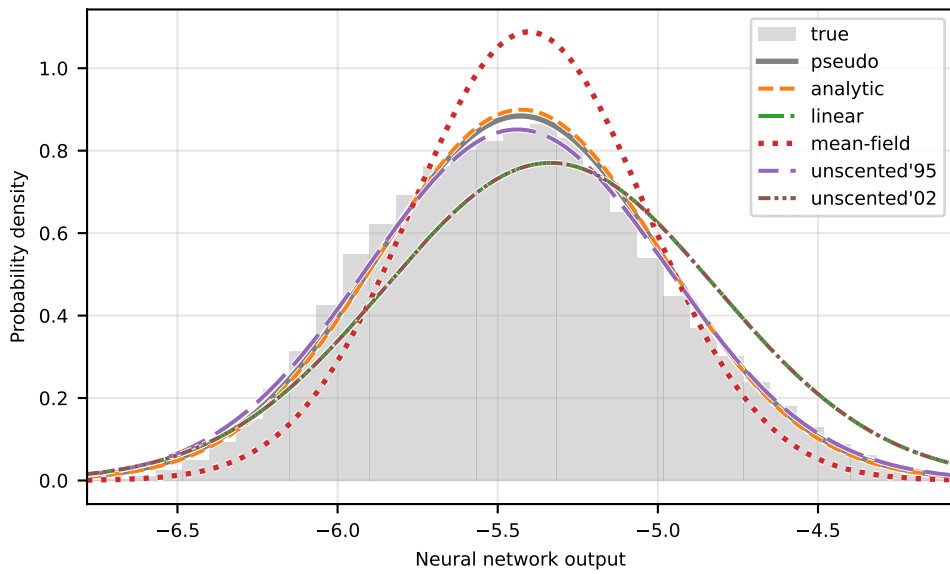

Figure 54: Probability distributions for Network(architecture=wide, weights=initialized, activation=relu residual), variance=small

| distribution | $\mu$ | $\sigma^2$ |
|---|---|---|
| pseudo-true ($Y_1$) | $-6.570 \times 10^0 \pm 2.1 \times 10^{-5}$ | $8.554 \times 10^0 \pm 1.4 \times 10^{-4}$ |
| analytic | $-6.581 \times 10^0$ | $6.679 \times 10^0$ |
| mean-field | $-6.661 \times 10^0$ | $8.103 \times 10^0$ |
| linear | $-5.335 \times 10^0$ | $2.681 \times 10^{+1}$ |
| unscented'95 | $-6.634 \times 10^0$ | $5.961 \times 10^0$ |
| unscented'02 | $-6.031 \times 10^{+1}$ | $6.072 \times 10^{+3}$ |

Table 94: Comparison of moments for Network(architecture=wide, weights=initialized, activation=relu residual), variance=medium

| distribution | $d_{\mathrm{W}}(\cdot, Y_0)$ | $D_{\mathrm{KL}}(Y_1 \parallel \cdot)$ |
|---|---|---|
| pseudo-true ($Y_1$) | $9.062 \times 10^{-2} \pm 2.6 \times 10^{-5}$ | $0$ |
| analytic | $2.083 \times 10^{-1} \pm 1.5 \times 10^{-5}$ | $1.413 \times 10^{-2} \pm 1.8 \times 10^{-6}$ |
| mean-field | $1.237 \times 10^{-1} \pm 1.5 \times 10^{-5}$ | $1.206 \times 10^{-3} \pm 3.6 \times 10^{-7}$ |
| linear | $1.183 \times 10^0 \pm 1.9 \times 10^{-5}$ | $5.852 \times 10^{-1} \pm 1.8 \times 10^{-5}$ |
| unscented'95 | $2.634 \times 10^{-1} \pm 1.4 \times 10^{-5}$ | $2.924 \times 10^{-2} \pm 2.4 \times 10^{-6}$ |
| unscented'02 | $4.361 \times 10^{+1} \pm 1.8 \times 10^{-4}$ | $5.200 \times 10^{+2} \pm 8.7 \times 10^{-3}$ |

Table 95: Comparison of statistical distances for Network(architecture=wide, weights=initialized, activation=relu residual), variance=medium

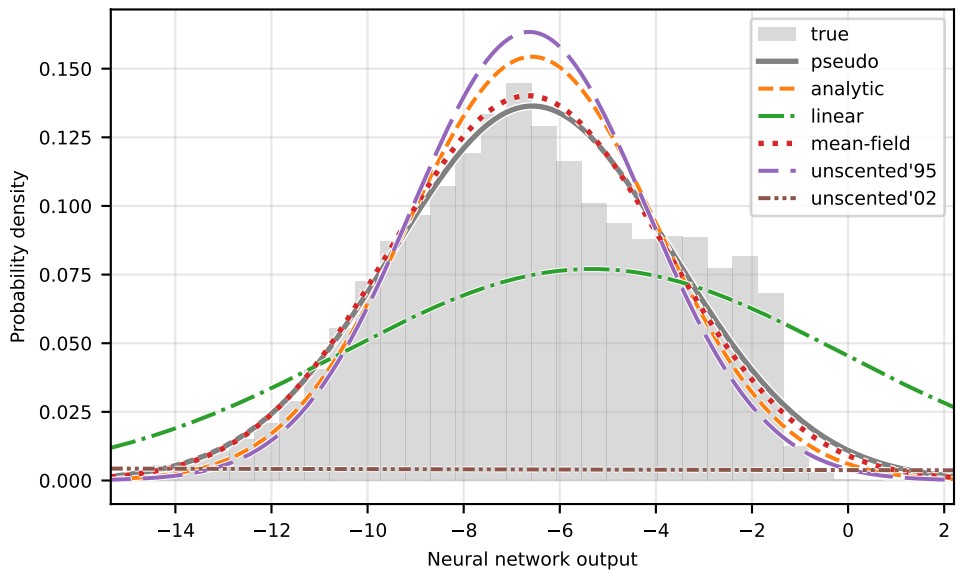

Figure 55: Probability distributions for Network(architecture=wide, weights=initialized, activation=relu residual), variance=medium

| distribution | $\mu$ | $\sigma^2$ |
|---|---|---|
| pseudo-true ($Y_1$) | $-3.755 \times 10^{+1} \pm 2.5 \times 10^{-4}$ | $7.896 \times 10^{+2} \pm 1.8 \times 10^{-2}$ |
| analytic | $-3.858 \times 10^{+1}$ | $5.527 \times 10^{+2}$ |
| mean-field | $-3.508 \times 10^{+1}$ | $7.094 \times 10^{+2}$ |
| linear | $-5.335 \times 10^{0}$ | $2.681 \times 10^{+3}$ |
| unscented'95 | $-3.855 \times 10^{+1}$ | $6.209 \times 10^{+2}$ |
| unscented'02 | $-6.382 \times 10^{+3}$ | $8.134 \times 10^{+7}$ |

Table 96: Comparison of moments for Network(architecture=wide, weights=initialized, activation=relu residual), variance=large

| distribution | $d_{\mathrm{W}}(\cdot, Y_0)$ | $D_{\mathrm{KL}}(Y_1 \parallel \cdot)$ |
|---|---|---|
| pseudo-true ($Y_1$) | $5.563 \times 10^{-1} \pm 6.6 \times 10^{-5}$ | $0$ |
| analytic | $9.409 \times 10^{-1} \pm 4.3 \times 10^{-5}$ | $2.904 \times 10^{-2} \pm 3.2 \times 10^{-6}$ |
| mean-field | $6.105 \times 10^{-1} \pm 5.9 \times 10^{-5}$ | $6.612 \times 10^{-3} \pm 1.6 \times 10^{-6}$ |
| linear | $6.261 \times 10^{0} \pm 5.5 \times 10^{-5}$ | $1.243 \times 10^{0} \pm 3.8 \times 10^{-5}$ |
| unscented'95 | $8.038 \times 10^{-1} \pm 3.5 \times 10^{-5}$ | $1.400 \times 10^{-2} \pm 2.3 \times 10^{-6}$ |
| unscented'02 | $1.677 \times 10^{+3} \pm 9.5 \times 10^{-3}$ | $7.699 \times 10^{+4} \pm 1.7 \times 10^{0}$ |

Table 97: Comparison of statistical distances for Network(architecture=wide, weights=initialized, activation=relu residual), variance=large

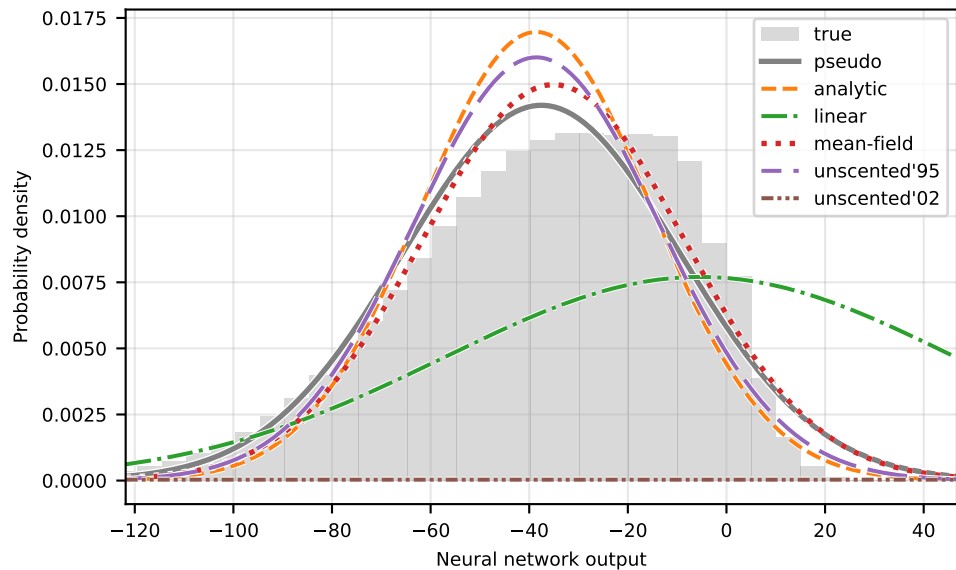

Figure 56: Probability distributions for Network(architecture=wide, weights=initialized, activation=relu residual), variance=large

| distribution | $\mu$ | $\sigma^2$ |
|---|---|---|
| pseudo-true ($Y_1$) | $+4.943 \times 10^{-2} \pm 8.5 \times 10^{-7}$ | $5.939 \times 10^{-2} \pm 1.2 \times 10^{-6}$ |
| analytic | $+5.056 \times 10^{-2}$ | $5.561 \times 10^{-2}$ |
| mean-field | $+6.542 \times 10^{-2}$ | $1.351 \times 10^{-1}$ |
| linear | $+1.866 \times 10^{-1}$ | $7.861 \times 10^{-2}$ |
| unscented'95 | $+3.719 \times 10^{-2}$ | $6.147 \times 10^{-2}$ |
| unscented'02 | $+1.866 \times 10^{-1}$ | $7.861 \times 10^{-2}$ |

Table 98: Comparison of moments for Network(architecture=wide, weights=trained, activation=relu residual), variance=small

| distribution | $d_{\mathrm{W}}(\cdot, Y_0)$ | $D_{\mathrm{KL}}(Y_1 \parallel \cdot)$ |
|---|---|---|
| pseudo-true ($Y_1$) | $1.563 \times 10^{-2} \pm 5.2 \times 10^{-6}$ | $0$ |
| analytic | $1.974 \times 10^{-2} \pm 3.4 \times 10^{-6}$ | $1.072 \times 10^{-3} \pm 6.6 \times 10^{-7}$ |
| mean-field | $1.966 \times 10^{-1} \pm 3.8 \times 10^{-6}$ | $2.287 \times 10^{-1} \pm 1.3 \times 10^{-5}$ |
| linear | $2.778 \times 10^{-1} \pm 2.2 \times 10^{-6}$ | $1.800 \times 10^{-1} \pm 6.8 \times 10^{-6}$ |
| unscented'95 | $2.904 \times 10^{-2} \pm 3.2 \times 10^{-6}$ | $1.560 \times 10^{-3} \pm 4.4 \times 10^{-7}$ |
| unscented'02 | $2.778 \times 10^{-1} \pm 2.2 \times 10^{-6}$ | $1.800 \times 10^{-1} \pm 6.8 \times 10^{-6}$ |

Table 99: Comparison of statistical distances for Network(architecture=wide, weights=trained, activation=relu residual), variance=small

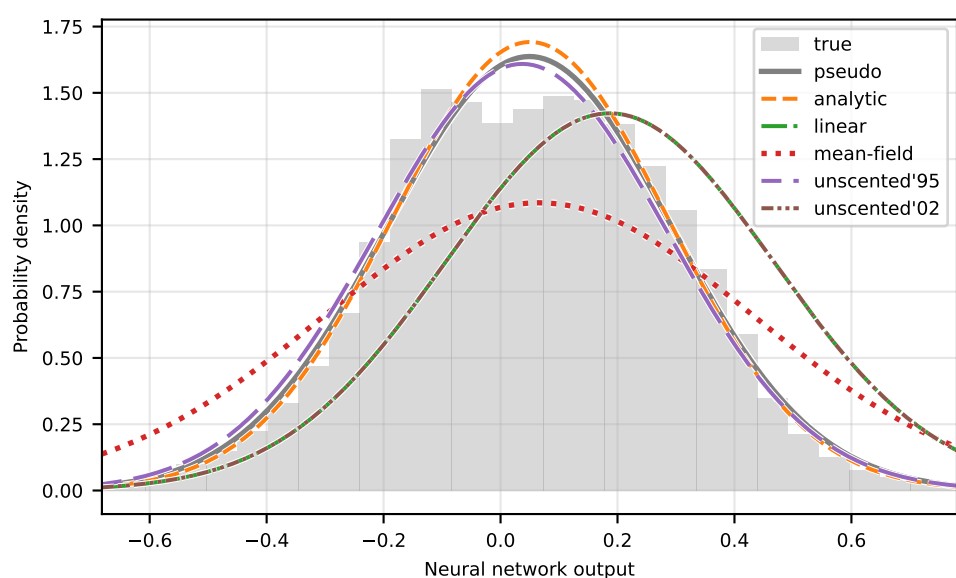

Figure 57: Probability distributions for Network(architecture=wide, weights=trained, activation=relu residual), variance=small

| distribution | $\mu$ | $\sigma^2$ |
|---|---|---|
| pseudo-true $(Y_1)$ | $-5.646 \times 10^{-1} \pm 2.4 \times 10^{-5}$ | $2.438 \times 10^0 \pm 1.0 \times 10^{-4}$ |
| analytic | $-6.082 \times 10^{-1}$ | $2.762 \times 10^0$ |
| mean-field | $-6.252 \times 10^{-1}$ | $8.171 \times 10^0$ |
| linear | $+1.866 \times 10^{-1}$ | $7.861 \times 10^0$ |
| unscented'95 | $-6.606 \times 10^{-1}$ | $1.274 \times 10^0$ |
| unscented'02 | $+3.824 \times 10^{+1}$ | $2.904 \times 10^{+3}$ |

Table 100: Comparison of moments for Network(architecture=wide, weights=trained, activation=relu residual), variance=medium

| distribution | $d_{\mathrm{W}}(\cdot, Y_0)$ | $D_{\mathrm{KL}}(Y_1 \parallel \cdot)$ |
|---|---|---|
| pseudo-true $(Y_1)$ | $5.621 \times 10^{-2} \pm 3.8 \times 10^{-5}$ | $0$ |
| analytic | $1.021 \times 10^{-1} \pm 3.2 \times 10^{-5}$ | $4.452 \times 10^{-3} \pm 2.8 \times 10^{-6}$ |
| mean-field | $8.479 \times 10^{-1} \pm 2.7 \times 10^{-5}$ | $5.720 \times 10^{-1} \pm 5.0 \times 10^{-5}$ |
| linear | $9.518 \times 10^{-1} \pm 2.9 \times 10^{-5}$ | $6.428 \times 10^{-1} \pm 5.5 \times 10^{-5}$ |
| unscented'95 | $2.614 \times 10^{-1} \pm 1.7 \times 10^{-5}$ | $8.772 \times 10^{-2} \pm 1.0 \times 10^{-5}$ |
| unscented'02 | $4.221 \times 10^{+1} \pm 4.7 \times 10^{-4}$ | $9.004 \times 10^{+2} \pm 3.9 \times 10^{-2}$ |

Table 101: Comparison of statistical distances for Network(architecture=wide, weights=trained, activation=relu residual), variance=medium

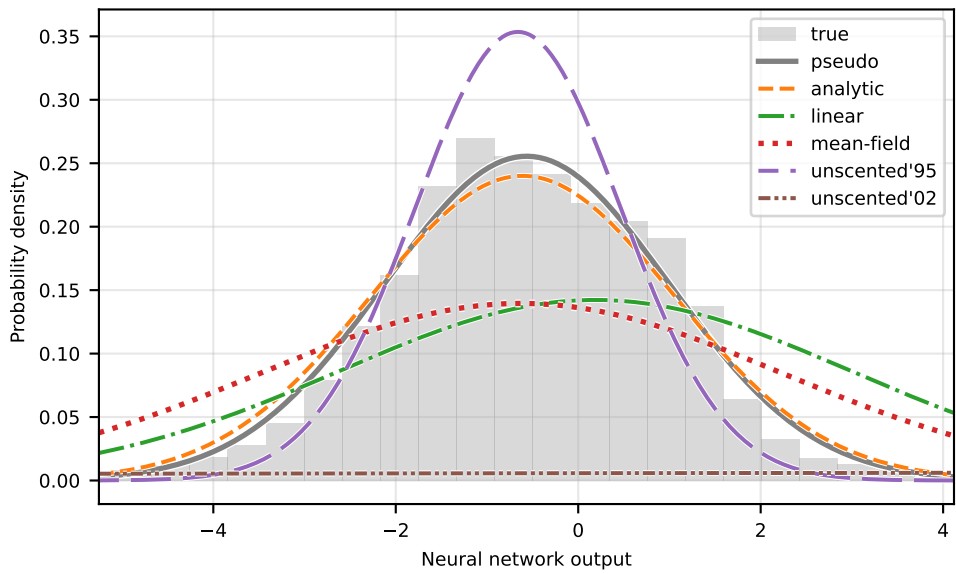

Figure 58: Probability distributions for Network(architecture=wide, weights=trained, activation=relu residual), variance=medium

| distribution | $\mu$ | $\sigma^2$ |
|---|---|---|
| pseudo-true ($Y_1$) | $-4.090 \times 10^0 \pm 2.5 \times 10^{-4}$ | $5.138 \times 10^{+2} \pm 1.2 \times 10^{-2}$ |
| analytic | $-4.302 \times 10^0$ | $4.533 \times 10^{+2}$ |
| mean-field | $-5.117 \times 10^{-1}$ | $7.093 \times 10^{+2}$ |
| linear | $+1.866 \times 10^{-1}$ | $7.861 \times 10^{+2}$ |
| unscented'95 | $-2.672 \times 10^0$ | $5.364 \times 10^{+2}$ |
| unscented'02 | $-3.175 \times 10^{+3}$ | $2.017 \times 10^{+7}$ |

Table 102: Comparison of moments for Network(architecture=wide, weights=trained, activation=relu residual), variance=large

| distribution | $d_{\mathrm{W}}(\cdot, Y_0)$ | $D_{\mathrm{KL}}(Y_1 \parallel \cdot)$ |
|---|---|---|
| pseudo-true ($Y_1$) | $5.767 \times 10^{-1} \pm 7.1 \times 10^{-5}$ | $0$ |
| analytic | $5.756 \times 10^{-1} \pm 6.8 \times 10^{-5}$ | $3.818 \times 10^{-3} \pm 1.4 \times 10^{-6}$ |
| mean-field | $1.220 \times 10^0 \pm 7.1 \times 10^{-5}$ | $4.147 \times 10^{-2} \pm 5.1 \times 10^{-6}$ |
| linear | $1.477 \times 10^0 \pm 6.7 \times 10^{-5}$ | $7.014 \times 10^{-2} \pm 7.0 \times 10^{-6}$ |
| unscented'95 | $6.572 \times 10^{-1} \pm 7.3 \times 10^{-5}$ | $2.426 \times 10^{-3} \pm 9.3 \times 10^{-7}$ |
| unscented'02 | $9.296 \times 10^{+2} \pm 5.3 \times 10^{-3}$ | $2.941 \times 10^{+4} \pm 6.7 \times 10^{-1}$ |

Table 103: Comparison of statistical distances for Network(architecture=wide, weights=trained, activation=relu residual), variance=large

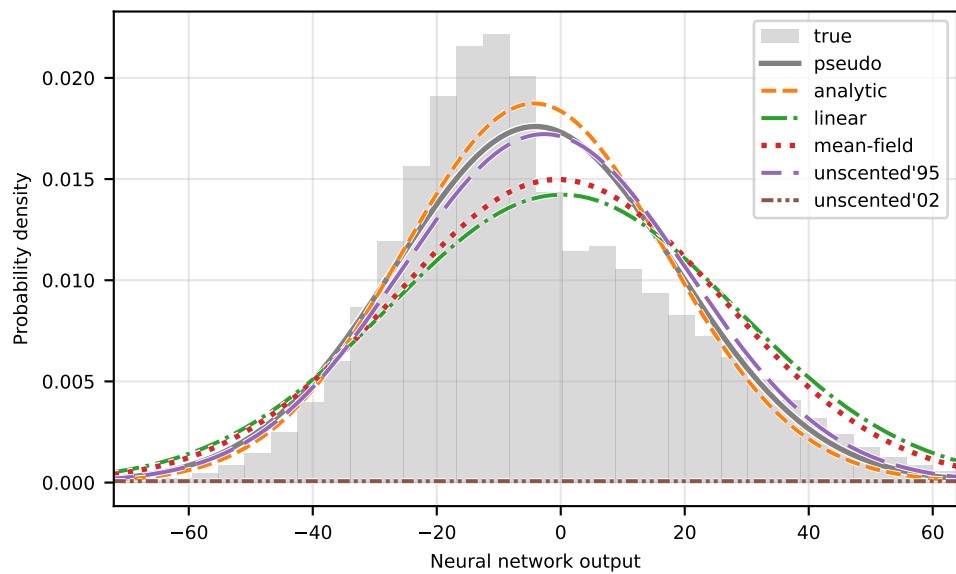

Figure 59: Probability distributions for Network(architecture=wide, weights=trained, activation=relu residual), variance=large

| distribution | $\mu$ | $\sigma^2$ |
|---|---|---|
| pseudo-true ($Y_1$) | $-6.998 \times 10^{-1} \pm 1.0 \times 10^{-4}$ | $4.743 \times 10^{-2} \pm 2.3 \times 10^{-5}$ |
| analytic | $-7.013 \times 10^{-1}$ | $4.170 \times 10^{-2}$ |
| mean-field | $-7.007 \times 10^{-1}$ | $3.827 \times 10^{-2}$ |
| linear | $-4.078 \times 10^{-1}$ | $0$ |
| unscented'95 | $-9.009 \times 10^{-1}$ | $8.230 \times 10^{-2}$ |
| unscented'02 | $-4.078 \times 10^{-1}$ | $4.322 \times 10^{-21}$ |

Table 104: Comparison of moments for Network(architecture=wide, weights=initialized, activation=heaviside), variance=small

| distribution | $d_{\mathrm{W}}(\cdot, Y_0)$ | $D_{\mathrm{KL}}(Y_1 \parallel \cdot)$ |
|---|---|---|
| pseudo-true ($Y_1$) | $8.746 \times 10^{-3} \pm 1.0 \times 10^{-4}$ | $0$ |
| analytic | $2.274 \times 10^{-2} \pm 1.1 \times 10^{-4}$ | $3.998 \times 10^{-3} \pm 2.9 \times 10^{-5}$ |
| mean-field | $3.737 \times 10^{-2} \pm 1.1 \times 10^{-4}$ | $1.074 \times 10^{-2} \pm 4.6 \times 10^{-5}$ |
| linear | $- \pm -$ | $\infty \pm -$ |
| unscented'95 | $4.309 \times 10^{-1} \pm 2.4 \times 10^{-4}$ | $5.181 \times 10^{-1} \pm 6.5 \times 10^{-4}$ |
| unscented'02 | $6.680 \times 10^{-1} \pm 1.8 \times 10^{-4}$ | $2.232 \times 10^{+1} \pm 6.2 \times 10^{-4}$ |

Table 105: Comparison of statistical distances for Network(architecture=wide, weights=initialized, activation=heaviside), variance=small

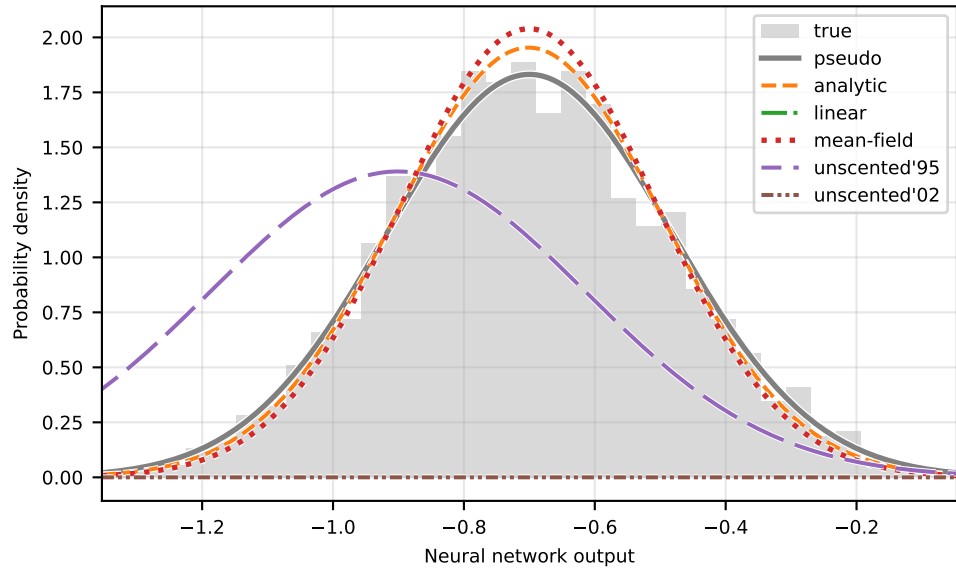

Figure 60: Probability distributions for Network(architecture=wide, weights=initialized, activation=heaviside), variance=small

| distribution | $\mu$ | $\sigma^2$ |
|---|---|---|
| pseudo-true ($Y_1$) | $-6.270 \times 10^{-1} \pm 1.9 \times 10^{-4}$ | $5.427 \times 10^{-2} \pm 6.3 \times 10^{-5}$ |
| analytic | $-6.293 \times 10^{-1}$ | $5.005 \times 10^{-2}$ |
| mean-field | $-6.308 \times 10^{-1}$ | $4.576 \times 10^{-2}$ |
| linear | $-4.078 \times 10^{-1}$ | $0$ |
| unscented'95 | $-6.559 \times 10^{-1}$ | $9.895 \times 10^{-2}$ |
| unscented'02 | $-1.626 \times 10^{+5}$ | $5.287 \times 10^{+10}$ |

Table 106: Comparison of moments for Network(architecture=wide, weights=initialized, activation=heaviside), variance=medium

| distribution | $d_{\mathrm{W}}(\cdot, Y_0)$ | $D_{\mathrm{KL}}(Y_1 \parallel \cdot)$ |
|---|---|---|
| pseudo-true ($Y_1$) | $1.975 \times 10^{-3} \pm 1.1 \times 10^{-4}$ | $0$ |
| analytic | $1.647 \times 10^{-2} \pm 2.3 \times 10^{-4}$ | $1.661 \times 10^{-3} \pm 4.5 \times 10^{-5}$ |
| mean-field | $3.274 \times 10^{-2} \pm 2.4 \times 10^{-4}$ | $7.035 \times 10^{-3} \pm 9.0 \times 10^{-5}$ |
| linear | $- \pm -$ | $\infty \pm -$ |
| unscented'95 | $1.435 \times 10^{-1} \pm 3.4 \times 10^{-4}$ | $1.190 \times 10^{-1} \pm 5.1 \times 10^{-4}$ |
| unscented'02 | $4.714 \times 10^{+5} \pm 1.4 \times 10^{+2}$ | $7.306 \times 10^{+11} \pm 8.5 \times 10^{+8}$ |

Table 107: Comparison of statistical distances for Network(architecture=wide, weights=initialized, activation=heaviside), variance=medium

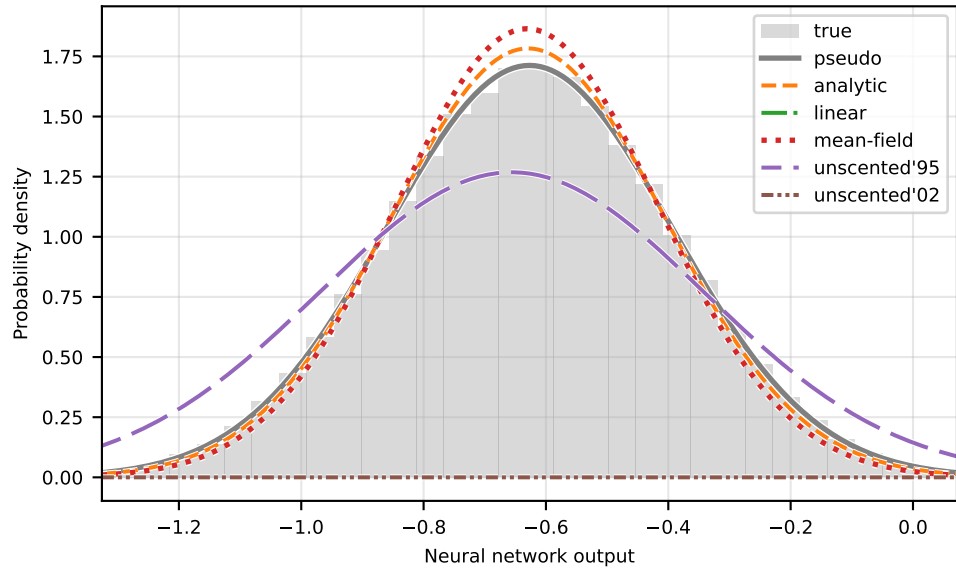

Figure 61: Probability distributions for Network(architecture=wide, weights=initialized, activation=heaviside), variance=medium

| distribution | $\mu$ | $\sigma^2$ |
|---|---|---|
| pseudo-true ($Y_1$) | $-6.306 \times 10^{-1} \pm 1.6 \times 10^{-4}$ | $5.625 \times 10^{-2} \pm 7.4 \times 10^{-5}$ |
| analytic | $-6.308 \times 10^{-1}$ | $5.223 \times 10^{-2}$ |
| mean-field | $-6.383 \times 10^{-1}$ | $4.932 \times 10^{-2}$ |
| linear | $-4.078 \times 10^{-1}$ | $0$ |
| unscented'95 | $-7.735 \times 10^{-1}$ | $5.789 \times 10^{-2}$ |
| unscented'02 | $-3.475 \times 10^{+5}$ | $2.416 \times 10^{+11}$ |

Table 108: Comparison of moments for Network(architecture=wide, weights=initialized, activation=heaviside), variance=large

| distribution | $d_{\mathrm{W}}(\cdot, Y_0)$ | $D_{\mathrm{KL}}(Y_1 \parallel \cdot)$ |
|---|---|---|
| pseudo-true ($Y_1$) | $2.546 \times 10^{-3} \pm 1.5 \times 10^{-4}$ | $0$ |
| analytic | $1.505 \times 10^{-2} \pm 2.3 \times 10^{-4}$ | $1.355 \times 10^{-3} \pm 4.7 \times 10^{-5}$ |
| mean-field | $2.879 \times 10^{-2} \pm 2.7 \times 10^{-4}$ | $4.681 \times 10^{-3} \pm 8.4 \times 10^{-5}$ |
| linear | $- \pm -$ | $\infty \pm -$ |
| unscented'95 | $2.935 \times 10^{-1} \pm 3.4 \times 10^{-4}$ | $1.818 \times 10^{-1} \pm 4.7 \times 10^{-4}$ |
| unscented'02 | $9.985 \times 10^{+5} \pm 3.3 \times 10^{+2}$ | $3.221 \times 10^{+12} \pm 4.2 \times 10^{+9}$ |

Table 109: Comparison of statistical distances for Network(architecture=wide, weights=initialized, activation=heaviside), variance=large

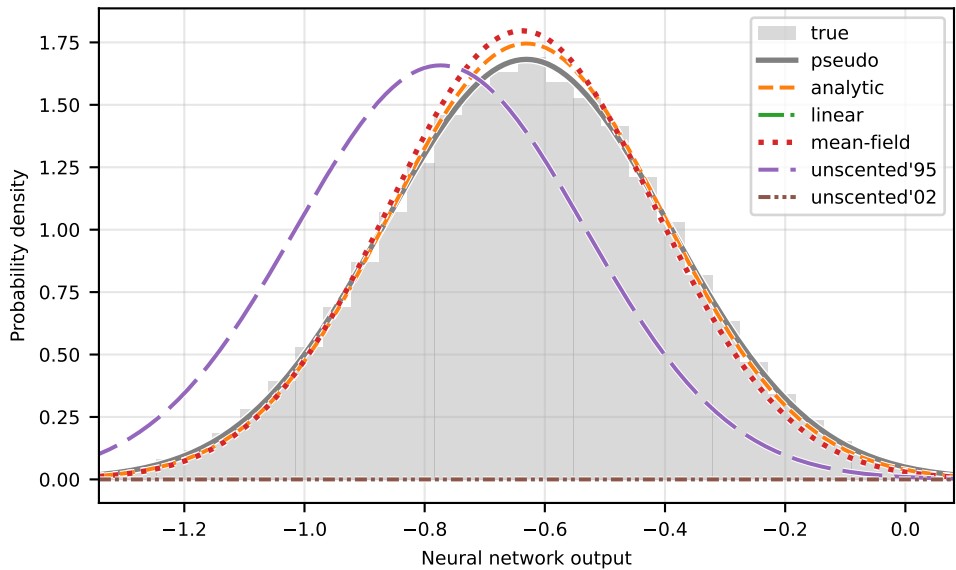

Figure 62: Probability distributions for Network(architecture=wide, weights=initialized, activation=heaviside), variance=large

| distribution | $\mu$ | $\sigma^2$ |
|---|---|---|
| pseudo-true $(Y_1)$ | $-2.766 \times 10^0 \pm 1.1 \times 10^{-4}$ | $2.197 \times 10^{-1} \pm 1.0 \times 10^{-4}$ |
| analytic | $-2.774 \times 10^0$ | $1.893 \times 10^{-1}$ |
| mean-field | $-2.734 \times 10^0$ | $1.686 \times 10^{-1}$ |
| linear | $-2.896 \times 10^0$ | $0$ |
| unscented'95 | $-2.997 \times 10^0$ | $1.551 \times 10^{-1}$ |
| unscented'02 | $-2.896 \times 10^0$ | $8.872 \times 10^{-20}$ |

Table 110: Comparison of moments for Network(architecture=wide, weights=initialized, activation=heaviside residual), variance=small

| distribution | $d_{\mathrm{W}}(\cdot, Y_0)$ | $D_{\mathrm{KL}}(Y_1 \parallel \cdot)$ |
|---|---|---|
| pseudo-true $(Y_1)$ | $3.325 \times 10^{-2} \pm 1.2 \times 10^{-4}$ | $0$ |
| analytic | $5.316 \times 10^{-2} \pm 1.6 \times 10^{-4}$ | $5.447 \times 10^{-3} \pm 3.3 \times 10^{-5}$ |
| mean-field | $6.836 \times 10^{-2} \pm 1.4 \times 10^{-4}$ | $1.844 \times 10^{-2} \pm 5.0 \times 10^{-5}$ |
| linear | $- \pm -$ | $\infty \pm -$ |
| unscented'95 | $3.367 \times 10^{-1} \pm 1.6 \times 10^{-4}$ | $1.481 \times 10^{-1} \pm 1.2 \times 10^{-4}$ |
| unscented'02 | $5.758 \times 10^{-1} \pm 9.8 \times 10^{-5}$ | $2.072 \times 10^{+1} \pm 2.4 \times 10^{-4}$ |

Table 111: Comparison of statistical distances for Network(architecture=wide, weights=initialized, activation=heaviside residual), variance=small

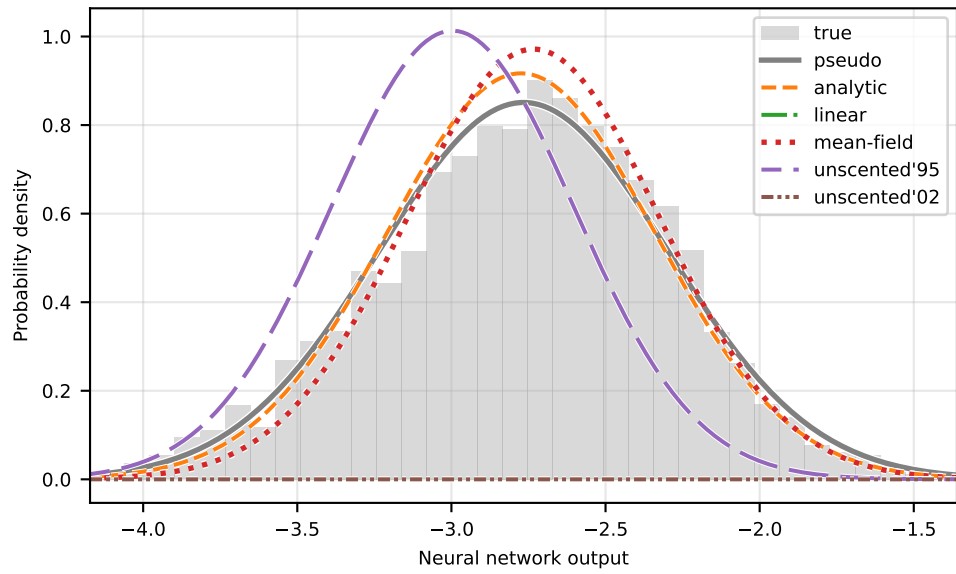

Figure 63: Probability distributions for Network(architecture=wide, weights=initialized, activation=heaviside residual), variance=small

| distribution | $\mu$ | $\sigma^2$ |
|---|---|---|
| pseudo-true $(Y_1)$ | $-2.330 \times 10^0 \pm 2.3 \times 10^{-4}$ | $3.146 \times 10^{-1} \pm 2.3 \times 10^{-4}$ |
| analytic | $-2.338 \times 10^0$ | $3.152 \times 10^{-1}$ |
| mean-field | $-2.383 \times 10^0$ | $4.215 \times 10^{-1}$ |
| linear | $-2.896 \times 10^0$ | $0$ |
| unscented'95 | $-2.077 \times 10^0$ | $1.681 \times 10^{-1}$ |
| unscented'02 | $+5.462 \times 10^{+4}$ | $5.967 \times 10^{+9}$ |

Table 112: Comparison of moments for Network(architecture=wide, weights=initialized, activation=heaviside residual), variance=medium

| distribution | $d_{\mathrm{W}}(\cdot, Y_0)$ | $D_{\mathrm{KL}}(Y_1 \parallel \cdot)$ |
|---|---|---|
| pseudo-true $(Y_1)$ | $4.107 \times 10^{-3} \pm 1.3 \times 10^{-4}$ | $0$ |
| analytic | $1.199 \times 10^{-2} \pm 3.0 \times 10^{-4}$ | $1.223 \times 10^{-4} \pm 6.6 \times 10^{-6}$ |
| mean-field | $1.095 \times 10^{-1} \pm 2.9 \times 10^{-4}$ | $2.808 \times 10^{-2} \pm 1.4 \times 10^{-4}$ |
| linear | $- \pm -$ | $\infty \pm -$ |
| unscented'95 | $3.453 \times 10^{-1} \pm 3.1 \times 10^{-4}$ | $1.823 \times 10^{-1} \pm 2.1 \times 10^{-4}$ |
| unscented'02 | $1.020 \times 10^{+5} \pm 1.9 \times 10^{+1}$ | $1.422 \times 10^{+10} \pm 1.0 \times 10^{+7}$ |

Table 113: Comparison of statistical distances for Network(architecture=wide, weights=initialized, activation=heaviside residual), variance=medium

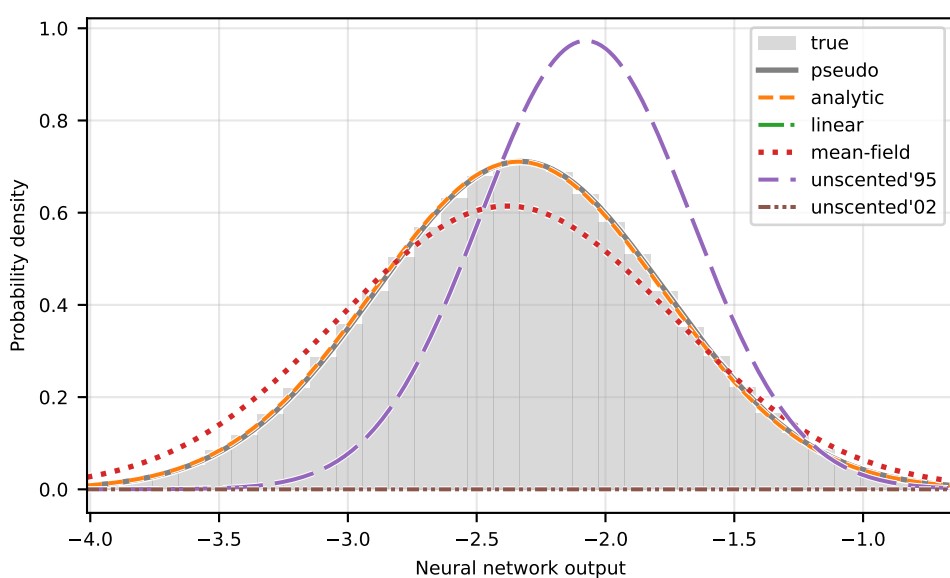

Figure 64: Probability distributions for Network(architecture=wide, weights=initialized, activation=heaviside residual), variance=medium

| distribution | $\mu$ | $\sigma^2$ |
|---|---|---|
| pseudo-true $(Y_1)$ | $-2.018 \times 10^0 \pm 3.0 \times 10^{-4}$ | $3.376 \times 10^{-1} \pm 1.9 \times 10^{-4}$ |
| analytic | $-2.035 \times 10^0$ | $3.613 \times 10^{-1}$ |
| mean-field | $-2.143 \times 10^0$ | $5.100 \times 10^{-1}$ |
| linear | $-2.896 \times 10^0$ | $0$ |
| unscented'95 | $-2.078 \times 10^0$ | $4.143 \times 10^{-1}$ |
| unscented'02 | $-4.102 \times 10^{+4}$ | $3.364 \times 10^{+9}$ |

Table 114: Comparison of moments for Network(architecture=wide, weights=initialized, activation=heaviside residual), variance=large

| distribution | $d_{\mathrm{W}}(\cdot, Y_0)$ | $D_{\mathrm{KL}}(Y_1 \parallel \cdot)$ |
|---|---|---|
| pseudo-true $(Y_1)$ | $1.042 \times 10^{-2} \pm 1.8 \times 10^{-4}$ | $0$ |
| analytic | $3.051 \times 10^{-2} \pm 3.2 \times 10^{-4}$ | $1.643 \times 10^{-3} \pm 2.9 \times 10^{-5}$ |
| mean-field | $1.986 \times 10^{-1} \pm 3.1 \times 10^{-4}$ | $7.239 \times 10^{-2} \pm 2.2 \times 10^{-4}$ |
| linear | $- \pm -$ | $\infty \pm -$ |
| unscented'95 | $9.586 \times 10^{-2} \pm 3.1 \times 10^{-4}$ | $1.669 \times 10^{-2} \pm 9.7 \times 10^{-5}$ |
| unscented'02 | $7.529 \times 10^{+4} \pm 1.1 \times 10^{+1}$ | $7.473 \times 10^{+9} \pm 4.3 \times 10^{+6}$ |

Table 115: Comparison of statistical distances for Network(architecture=wide, weights=initialized, activation=heaviside residual), variance=large

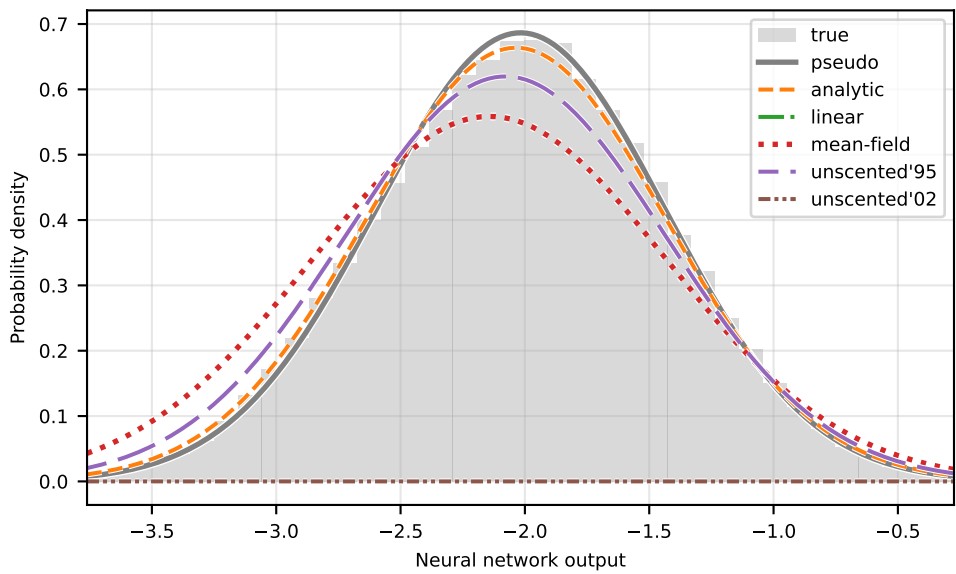

Figure 65: Probability distributions for Network(architecture=wide, weights=initialized, activation=heaviside residual), variance=large

| distribution | $\mu$ | $\sigma^2$ |
|---|---|---|
| pseudo-true ($Y_1$) | $+9.823 \times 10^{-1} \pm 2.3 \times 10^{-10}$ | $1.938 \times 10^{-7} \pm 9.1 \times 10^{-13}$ |
| analytic | $+9.823 \times 10^{-1}$ | $1.944 \times 10^{-7}$ |
| mean-field | $+9.823 \times 10^{-1}$ | $3.562 \times 10^{-7}$ |
| linear | $+9.823 \times 10^{-1}$ | $2.069 \times 10^{-7}$ |
| unscented'95 | $+9.823 \times 10^{-1}$ | $1.956 \times 10^{-7}$ |
| unscented'02 | $+9.823 \times 10^{-1}$ | $2.069 \times 10^{-7}$ |

Table 116: Comparison of moments for Network(architecture=deep, weights=initialized, activation=probit), variance=small

| distribution | $d_{\mathrm{W}}(\cdot, Y_0)$ | $D_{\mathrm{KL}}(Y_1 \parallel \cdot)$ |
|---|---|---|
| pseudo-true ($Y_1$) | $3.046 \times 10^{-4} \pm 1.2 \times 10^{-7}$ | $0$ |
| analytic | $3.010 \times 10^{-4} \pm 1.2 \times 10^{-7}$ | $2.877 \times 10^{-6} \pm 6.9 \times 10^{-9}$ |
| mean-field | $5.874 \times 10^{-3} \pm 8.7 \times 10^{-8}$ | $1.155 \times 10^{-1} \pm 2.0 \times 10^{-6}$ |
| linear | $5.152 \times 10^{-4} \pm 7.2 \times 10^{-8}$ | $1.088 \times 10^{-3} \pm 1.6 \times 10^{-7}$ |
| unscented'95 | $2.856 \times 10^{-4} \pm 1.2 \times 10^{-7}$ | $2.159 \times 10^{-5} \pm 2.2 \times 10^{-8}$ |
| unscented'02 | $5.579 \times 10^{-4} \pm 7.3 \times 10^{-8}$ | $1.099 \times 10^{-3} \pm 1.6 \times 10^{-7}$ |

Table 117: Comparison of statistical distances for Network(architecture=deep, weights=initialized, activation=probit), variance=small

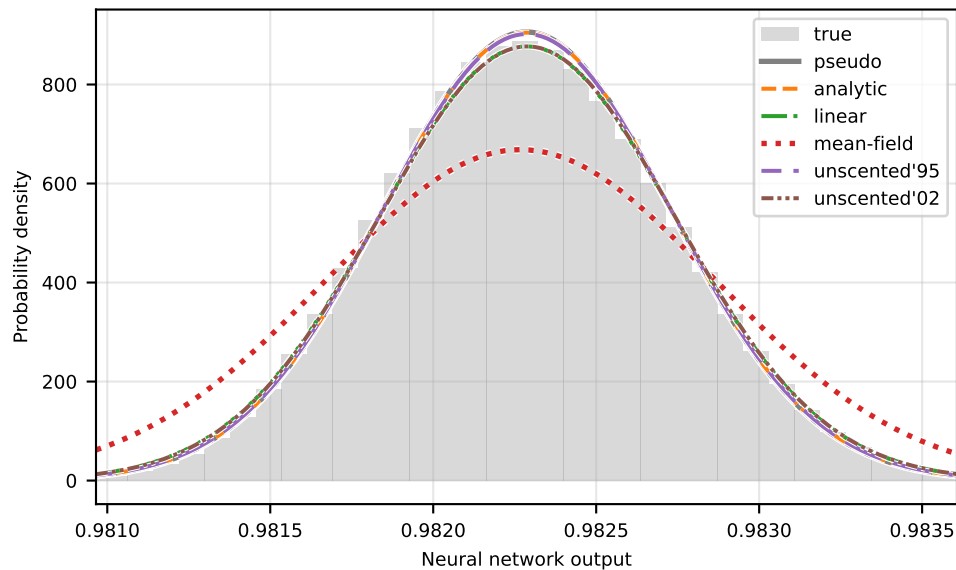

Figure 66: Probability distributions for Network(architecture=deep, weights=initialized, activation=probit), variance=small

| distribution | $\mu$ | $\sigma^2$ |
|---|---|---|
| pseudo-true ($Y_1$) | $+9.817 \times 10^{-1} \pm 8.1 \times 10^{-9}$ | $4.914 \times 10^{-6} \pm 5.4 \times 10^{-11}$ |
| analytic | $+9.817 \times 10^{-1}$ | $4.609 \times 10^{-6}$ |
| mean-field | $+9.814 \times 10^{-1}$ | $1.063 \times 10^{-5}$ |
| linear | $+9.823 \times 10^{-1}$ | $2.069 \times 10^{-5}$ |
| unscented'95 | $+9.816 \times 10^{-1}$ | $4.722 \times 10^{-6}$ |
| unscented'02 | $+9.825 \times 10^{-1}$ | $2.077 \times 10^{-5}$ |

Table 118: Comparison of moments for Network(architecture=deep, weights=initialized, activation=probit), variance=medium

| distribution | $d_{\mathrm{W}}(\cdot, Y_0)$ | $D_{\mathrm{KL}}(Y_1 \parallel \cdot)$ |
|---|---|---|
| pseudo-true ($Y_1$) | $2.313 \times 10^{-3} \pm 3.1 \times 10^{-7}$ | 0 |
| analytic | $3.086 \times 10^{-3} \pm 2.6 \times 10^{-7}$ | $1.014 \times 10^{-3} \pm 3.3 \times 10^{-7}$ |
| mean-field | $1.787 \times 10^{-2} \pm 3.2 \times 10^{-7}$ | $2.081 \times 10^{-1} \pm 6.9 \times 10^{-6}$ |
| linear | $3.995 \times 10^{-2} \pm 5.1 \times 10^{-7}$ | $9.210 \times 10^{-1} \pm 1.8 \times 10^{-5}$ |
| unscented'95 | $3.355 \times 10^{-3} \pm 3.5 \times 10^{-7}$ | $1.433 \times 10^{-3} \pm 1.7 \times 10^{-7}$ |
| unscented'02 | $4.112 \times 10^{-2} \pm 5.9 \times 10^{-7}$ | $9.546 \times 10^{-1} \pm 1.8 \times 10^{-5}$ |

Table 119: Comparison of statistical distances for Network(architecture=deep, weights=initialized, activation=probit), variance=medium

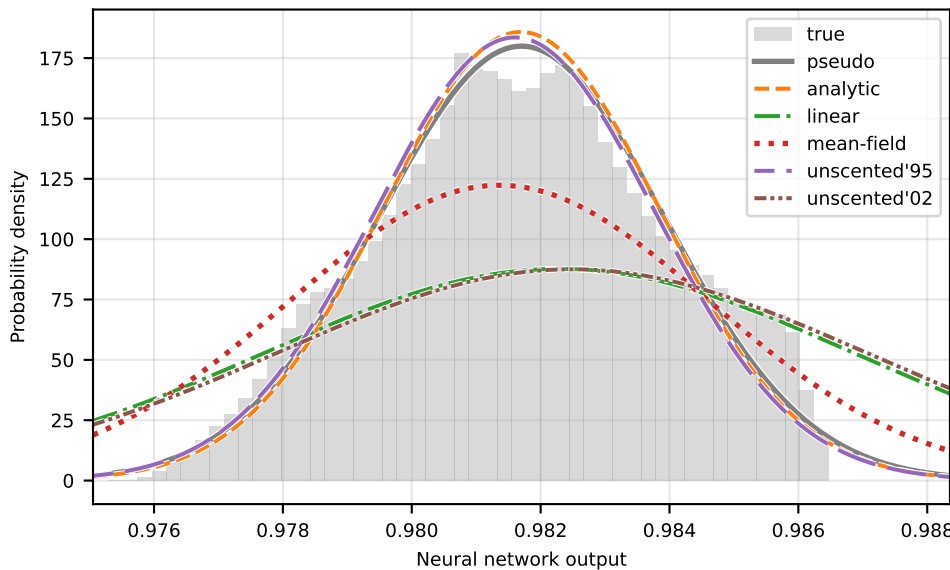

Figure 67: Probability distributions for Network(architecture=deep, weights=initialized, activation=probit), variance=medium

| distribution | $\mu$ | $\sigma^2$ |
|---|---|---|
| pseudo-true ($Y_1$) | $+9.802 \times 10^{-1} \pm 1.1 \times 10^{-7}$ | $1.055 \times 10^{-5} \pm 6.2 \times 10^{-10}$ |
| analytic | $+9.802 \times 10^{-1}$ | $9.472 \times 10^{-6}$ |
| mean-field | $+9.794 \times 10^{-1}$ | $1.949 \times 10^{-5}$ |
| linear | $+9.823 \times 10^{-1}$ | $2.069 \times 10^{-3}$ |
| unscented'95 | $+9.801 \times 10^{-1}$ | $1.202 \times 10^{-5}$ |
| unscented'02 | $+1.002 \times 10^{0}$ | $2.841 \times 10^{-3}$ |

Table 120: Comparison of moments for Network(architecture=deep, weights=initialized, activation=probit), variance=large

| distribution | $d_{\mathrm{W}}(\cdot, Y_0)$ | $D_{\mathrm{KL}}(Y_1 \parallel \cdot)$ |
|---|---|---|
| pseudo-true ($Y_1$) | $6.341 \times 10^{-3} \pm 1.2 \times 10^{-6}$ | $0$ |
| analytic | $7.496 \times 10^{-3} \pm 1.6 \times 10^{-6}$ | $2.916 \times 10^{-3} \pm 3.1 \times 10^{-6}$ |
| mean-field | $2.005 \times 10^{-2} \pm 2.4 \times 10^{-6}$ | $1.412 \times 10^{-1} \pm 2.8 \times 10^{-5}$ |
| linear | $5.893 \times 10^{-1} \pm 1.1 \times 10^{-5}$ | $9.509 \times 10^{+1} \pm 5.7 \times 10^{-3}$ |
| unscented'95 | $5.693 \times 10^{-3} \pm 1.9 \times 10^{-6}$ | $4.550 \times 10^{-3} \pm 4.2 \times 10^{-6}$ |
| unscented'02 | $7.639 \times 10^{-1} \pm 1.2 \times 10^{-5}$ | $1.538 \times 10^{+2} \pm 9.2 \times 10^{-3}$ |

Table 121: Comparison of statistical distances for Network(architecture=deep, weights=initialized, activation=probit), variance=large

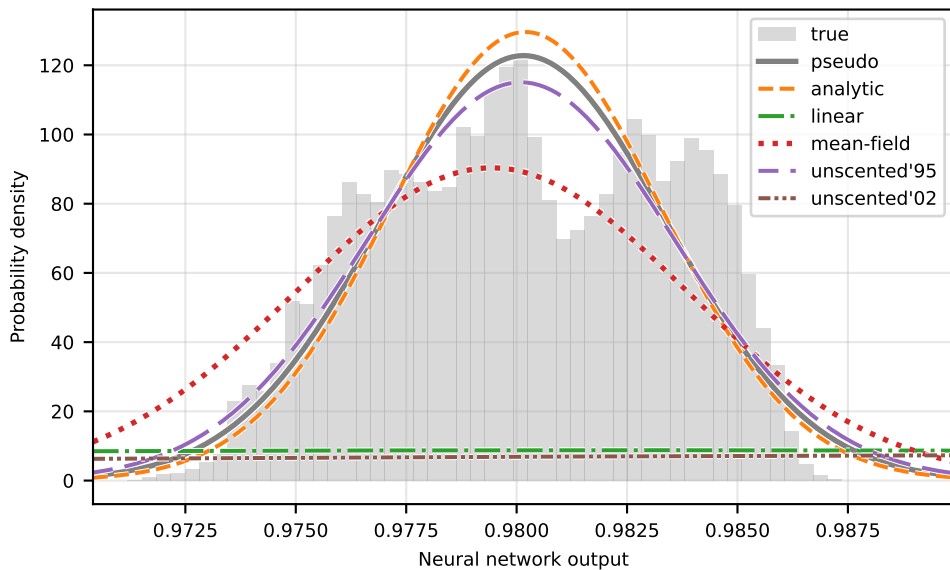

Figure 68: Probability distributions for Network(architecture=deep, weights=initialized, activation=probit), variance=large

| distribution | $\mu$ | $\sigma^2$ |
|---|---|---|
| pseudo-true $(Y_1)$ | $-2.932 \times 10^{-3} \pm 2.9 \times 10^{-7}$ | $5.431 \times 10^{-2} \pm 5.2 \times 10^{-7}$ |
| analytic | $-4.371 \times 10^{-3}$ | $4.186 \times 10^{-2}$ |
| mean-field | $+3.967 \times 10^{-2}$ | $1.949 \times 10^{-6}$ |
| linear | $+8.458 \times 10^{-2}$ | $3.872 \times 10^{-2}$ |
| unscented'95 | $-5.370 \times 10^{-3}$ | $4.734 \times 10^{-2}$ |
| unscented'02 | $-7.562 \times 10^{-3}$ | $5.570 \times 10^{-2}$ |

Table 122: Comparison of moments for Network(architecture=deep, weights=trained, activation=probit), variance=small

| distribution | $d_{\mathrm{W}}(\cdot, Y_0)$ | $D_{\mathrm{KL}}(Y_1 \parallel \cdot)$ |
|---|---|---|
| pseudo-true $(Y_1)$ | $1.147 \times 10^{-1} \pm 2.3 \times 10^{-6}$ | $0$ |
| analytic | $1.129 \times 10^{-1} \pm 1.9 \times 10^{-6}$ | $1.559 \times 10^{-2} \pm 1.1 \times 10^{-6}$ |
| mean-field | $3.569 \times 10^{-1} \pm 1.2 \times 10^{-6}$ | $4.634 \times 10^{0} \pm 4.8 \times 10^{-6}$ |
| linear | $1.813 \times 10^{-1} \pm 4.8 \times 10^{-7}$ | $9.615 \times 10^{-2} \pm 1.1 \times 10^{-6}$ |
| unscented'95 | $1.132 \times 10^{-1} \pm 2.2 \times 10^{-6}$ | $4.562 \times 10^{-3} \pm 6.1 \times 10^{-7}$ |
| unscented'02 | $1.191 \times 10^{-1} \pm 2.6 \times 10^{-6}$ | $3.580 \times 10^{-4} \pm 1.4 \times 10^{-7}$ |

Table 123: Comparison of statistical distances for Network(architecture=deep, weights=trained, activation=probit), variance=small

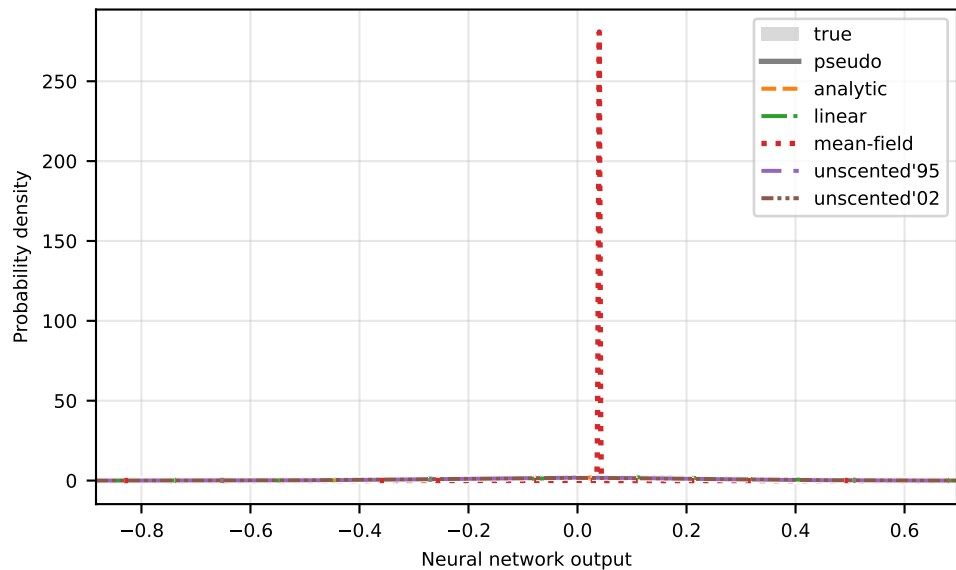

Figure 69: Probability distributions for Network(architecture=deep, weights=trained, activation=probit), variance=small

| distribution | $\mu$ | $\sigma^2$ |
|---|---|---|
| pseudo-true ($Y_1$) | $-5.472 \times 10^{-1} \pm 1.3 \times 10^{-5}$ | $8.328 \times 10^{-1} \pm 2.3 \times 10^{-5}$ |
| analytic | $-8.363 \times 10^{-1}$ | $3.840 \times 10^{-1}$ |
| mean-field | $-1.277 \times 10^{0}$ | $5.188 \times 10^{-5}$ |
| linear | $+8.458 \times 10^{-2}$ | $3.872 \times 10^{0}$ |
| unscented'95 | $-5.163 \times 10^{-1}$ | $1.106 \times 10^{0}$ |
| unscented'02 | $-9.129 \times 10^{0}$ | $1.737 \times 10^{+2}$ |

Table 124: Comparison of moments for Network(architecture=deep, weights=trained, activation=probit), variance=medium

| distribution | $d_{\mathrm{W}}(\cdot, Y_0)$ | $D_{\mathrm{KL}}(Y_1 \parallel \cdot)$ |
|---|---|---|
| pseudo-true ($Y_1$) | $2.094 \times 10^{-1} \pm 1.2 \times 10^{-5}$ | $0$ |
| analytic | $4.192 \times 10^{-1} \pm 2.0 \times 10^{-5}$ | $1.678 \times 10^{-1} \pm 9.0 \times 10^{-6}$ |
| mean-field | $9.681 \times 10^{-1} \pm 1.5 \times 10^{-5}$ | $4.661 \times 10^{0} \pm 1.4 \times 10^{-5}$ |
| linear | $1.094 \times 10^{0} \pm 1.8 \times 10^{-5}$ | $1.296 \times 10^{0} \pm 6.1 \times 10^{-5}$ |
| unscented'95 | $2.019 \times 10^{-1} \pm 2.0 \times 10^{-5}$ | $2.269 \times 10^{-2} \pm 4.7 \times 10^{-6}$ |
| unscented'02 | $1.262 \times 10^{+1} \pm 9.6 \times 10^{-5}$ | $1.453 \times 10^{+2} \pm 4.0 \times 10^{-3}$ |

Table 125: Comparison of statistical distances for Network(architecture=deep, weights=trained, activation=probit), variance=medium

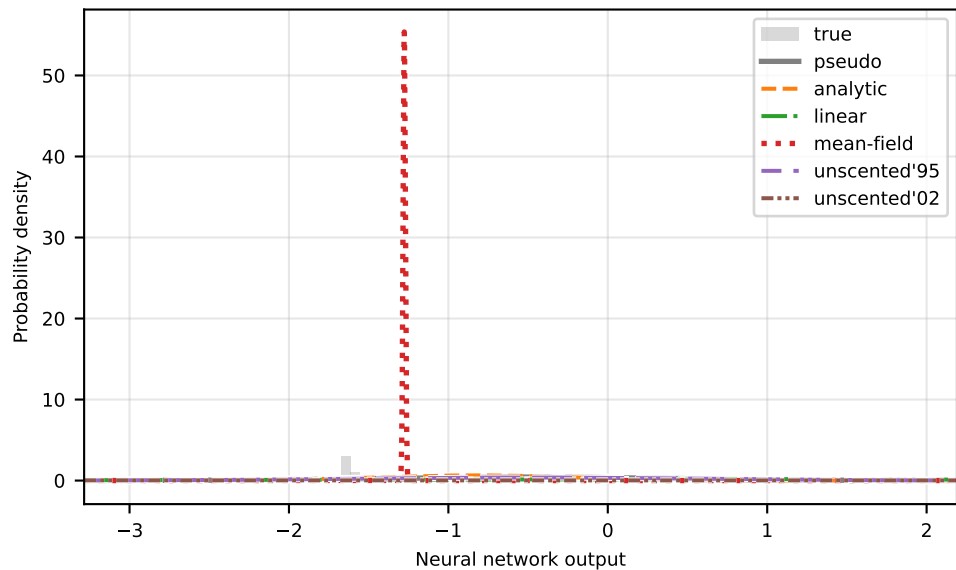

Figure 70: Probability distributions for Network(architecture=deep, weights=trained, activation=probit), variance=medium

| distribution | $\mu$ | $\sigma^2$ |
|---|---|---|
| pseudo-true ($Y_1$) | $-8.683 \times 10^{-1} \pm 3.2 \times 10^{-5}$ | $8.802 \times 10^{-1} \pm 5.5 \times 10^{-5}$ |
| analytic | $-9.489 \times 10^{-1}$ | $3.595 \times 10^{-1}$ |
| mean-field | $-1.559 \times 10^{0}$ | $5.722 \times 10^{-5}$ |
| linear | $+8.458 \times 10^{-2}$ | $3.872 \times 10^{+2}$ |
| unscented'95 | $-8.587 \times 10^{-1}$ | $9.796 \times 10^{-1}$ |
| unscented'02 | $-9.212 \times 10^{+2}$ | $1.698 \times 10^{+6}$ |

Table 126: Comparison of moments for Network(architecture=deep, weights=trained, activation=probit), variance=large

| distribution | $d_{\mathrm{W}}(\cdot, Y_0)$ | $D_{\mathrm{KL}}(Y_1 \parallel \cdot)$ |
|---|---|---|
| pseudo-true ($Y_1$) | $3.908 \times 10^{-1} \pm 3.4 \times 10^{-5}$ | $0$ |
| analytic | $4.425 \times 10^{-1} \pm 3.4 \times 10^{-5}$ | $1.557 \times 10^{-1} \pm 2.0 \times 10^{-5}$ |
| mean-field | $7.749 \times 10^{-1} \pm 2.8 \times 10^{-5}$ | $4.591 \times 10^{0} \pm 3.5 \times 10^{-5}$ |
| linear | $1.549 \times 10^{+1} \pm 2.6 \times 10^{-4}$ | $2.169 \times 10^{+2} \pm 1.4 \times 10^{-2}$ |
| unscented'95 | $3.925 \times 10^{-1} \pm 3.4 \times 10^{-5}$ | $3.018 \times 10^{-3} \pm 3.7 \times 10^{-6}$ |
| unscented'02 | $1.330 \times 10^{+3} \pm 2.1 \times 10^{-2}$ | $1.446 \times 10^{+6} \pm 9.0 \times 10^{+1}$ |

Table 127: Comparison of statistical distances for Network(architecture=deep, weights=trained, activation=probit), variance=large

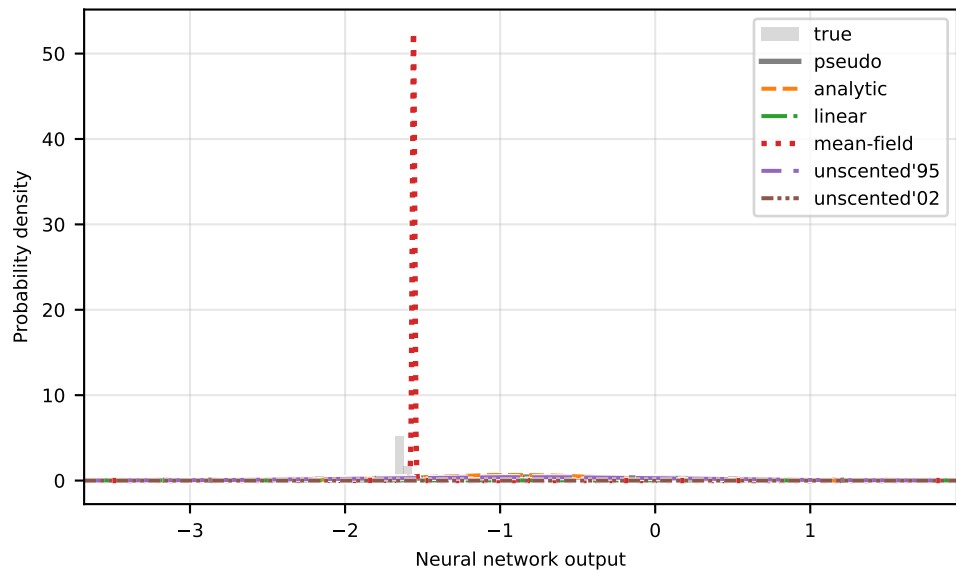

Figure 71: Probability distributions for Network(architecture=deep, weights=trained, activation=probit), variance=large

| distribution | $\mu$ | $\sigma^2$ |
|---|---|---|
| pseudo-true ($Y_1$) | $+7.399 \times 10^0 \pm 3.2 \times 10^{-6}$ | $4.314 \times 10^{-1} \pm 9.8 \times 10^{-6}$ |
| analytic | $+7.409 \times 10^0$ | $3.469 \times 10^{-1}$ |
| mean-field | $+7.582 \times 10^0$ | $7.771 \times 10^{-1}$ |
| linear | $+7.650 \times 10^0$ | $7.550 \times 10^{-1}$ |
| unscented'95 | $+7.308 \times 10^0$ | $1.613 \times 10^{-1}$ |
| unscented'02 | $+6.993 \times 10^0$ | $1.616 \times 10^0$ |

Table 128: Comparison of moments for Network(architecture=deep, weights=initialized, activation=probit residual), variance=small

| distribution | $d_{\mathrm{W}}(\cdot, Y_0)$ | $D_{\mathrm{KL}}(Y_1 \parallel \cdot)$ |
|---|---|---|
| pseudo-true ($Y_1$) | $6.736 \times 10^{-2} \pm 1.3 \times 10^{-5}$ | $0$ |
| analytic | $7.097 \times 10^{-2} \pm 1.1 \times 10^{-5}$ | $1.115 \times 10^{-2} \pm 2.2 \times 10^{-6}$ |
| mean-field | $3.262 \times 10^{-1} \pm 9.6 \times 10^{-6}$ | $1.452 \times 10^{-1} \pm 1.0 \times 10^{-5}$ |
| linear | $3.753 \times 10^{-1} \pm 1.1 \times 10^{-5}$ | $1.678 \times 10^{-1} \pm 1.0 \times 10^{-5}$ |
| unscented'95 | $2.501 \times 10^{-1} \pm 4.8 \times 10^{-6}$ | $1.885 \times 10^{-1} \pm 6.9 \times 10^{-6}$ |
| unscented'02 | $7.446 \times 10^{-1} \pm 1.0 \times 10^{-5}$ | $9.041 \times 10^{-1} \pm 3.6 \times 10^{-5}$ |

Table 129: Comparison of statistical distances for Network(architecture=deep, weights=initialized, activation=probit residual), variance=small

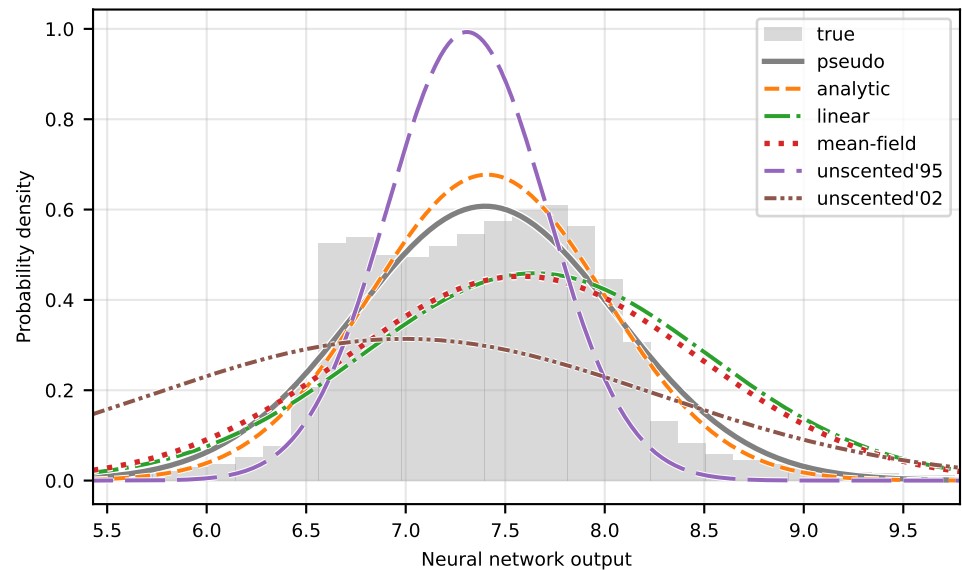

Figure 72: Probability distributions for Network(architecture=deep, weights=initialized, activation=probit residual), variance=small

| distribution | $\mu$ | $\sigma^2$ |
|---|---|---|
| pseudo-true ($Y_1$) | $+6.488 \times 10^0 \pm 6.5 \times 10^{-5}$ | $8.614 \times 10^0 \pm 4.0 \times 10^{-4}$ |
| analytic | $+6.419 \times 10^0$ | $7.410 \times 10^0$ |
| mean-field | $+6.860 \times 10^0$ | $8.171 \times 10^0$ |
| linear | $+7.650 \times 10^0$ | $7.550 \times 10^{+1}$ |
| unscented'95 | $+6.199 \times 10^0$ | $4.770 \times 10^0$ |
| unscented'02 | $-5.797 \times 10^{+1}$ | $8.688 \times 10^{+3}$ |

Table 130: Comparison of moments for Network(architecture=deep, weights=initialized, activation=probit residual), variance=medium

| distribution | $d_{\mathrm{W}}(\cdot, Y_0)$ | $D_{\mathrm{KL}}(Y_1 \parallel \cdot)$ |
|---|---|---|
| pseudo-true ($Y_1$) | $2.098 \times 10^{-1} \pm 4.8 \times 10^{-5}$ | $0$ |
| analytic | $2.348 \times 10^{-1} \pm 4.2 \times 10^{-5}$ | $5.680 \times 10^{-3} \pm 3.0 \times 10^{-6}$ |
| mean-field | $2.172 \times 10^{-1} \pm 3.8 \times 10^{-5}$ | $8.719 \times 10^{-3} \pm 3.3 \times 10^{-6}$ |
| linear | $2.719 \times 10^0 \pm 7.2 \times 10^{-5}$ | $2.875 \times 10^0 \pm 1.8 \times 10^{-4}$ |
| unscented'95 | $4.377 \times 10^{-1} \pm 3.4 \times 10^{-5}$ | $7.725 \times 10^{-2} \pm 9.3 \times 10^{-6}$ |
| unscented'02 | $5.243 \times 10^{+1} \pm 6.3 \times 10^{-4}$ | $7.415 \times 10^{+2} \pm 3.5 \times 10^{-2}$ |

Table 131: Comparison of statistical distances for Network(architecture=deep, weights=initialized, activation=probit residual), variance=medium

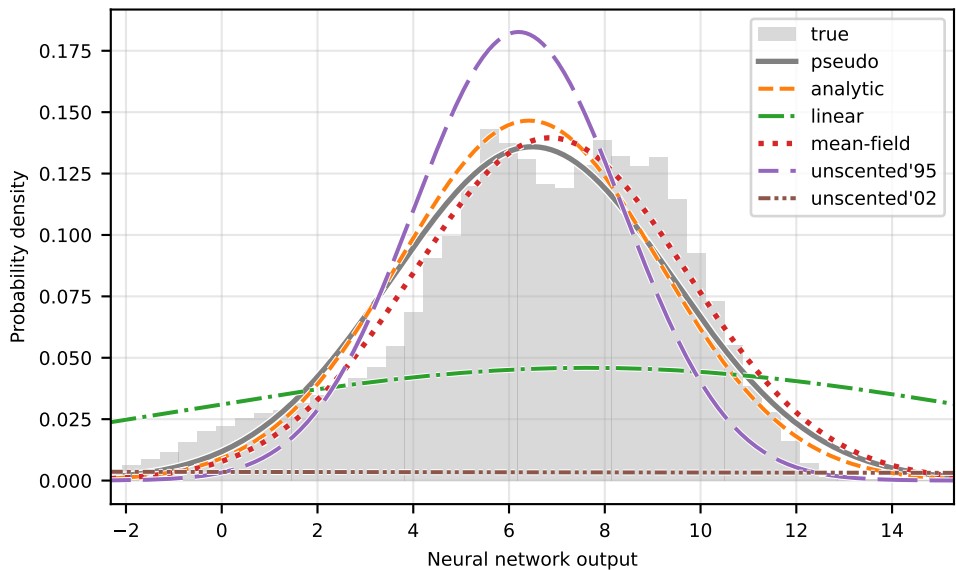

Figure 73: Probability distributions for Network(architecture=deep, weights=initialized, activation=probit residual), variance=medium

| distribution | $\mu$ | $\sigma^2$ |
|---|---|---|
| pseudo-true $(Y_1)$ | $+3.429 \times 10^0 \pm 3.0 \times 10^{-4}$ | $1.417 \times 10^{+1} \pm 1.8 \times 10^{-3}$ |
| analytic | $+3.589 \times 10^0$ | $1.220 \times 10^{+1}$ |
| mean-field | $+4.069 \times 10^0$ | $1.242 \times 10^{+1}$ |
| linear | $+7.650 \times 10^0$ | $7.550 \times 10^{+3}$ |
| unscented'95 | $+3.591 \times 10^0$ | $6.302 \times 10^0$ |
| unscented'02 | $-6.533 \times 10^{+3}$ | $8.557 \times 10^{+7}$ |

Table 132: Comparison of moments for Network(architecture=deep, weights=initialized, activation=probit residual), variance=large

| distribution | $d_{\mathrm{W}}(\cdot, Y_0)$ | $D_{\mathrm{KL}}(Y_1 \parallel \cdot)$ |
|---|---|---|
| pseudo-true $(Y_1)$ | $6.358 \times 10^{-2} \pm 1.4 \times 10^{-4}$ | $0$ |
| analytic | $1.549 \times 10^{-1} \pm 1.2 \times 10^{-4}$ | $6.207 \times 10^{-3} \pm 8.1 \times 10^{-6}$ |
| mean-field | $3.305 \times 10^{-1} \pm 1.6 \times 10^{-4}$ | $1.857 \times 10^{-2} \pm 1.3 \times 10^{-5}$ |
| linear | $3.423 \times 10^{+1} \pm 1.2 \times 10^{-3}$ | $2.635 \times 10^{+2} \pm 3.4 \times 10^{-2}$ |
| unscented'95 | $5.266 \times 10^{-1} \pm 1.2 \times 10^{-4}$ | $1.284 \times 10^{-1} \pm 3.4 \times 10^{-5}$ |
| unscented'02 | $4.715 \times 10^{+3} \pm 1.5 \times 10^{-1}$ | $4.528 \times 10^{+6} \pm 5.7 \times 10^{+2}$ |

Table 133: Comparison of statistical distances for Network(architecture=deep, weights=initialized, activation=probit residual), variance=large

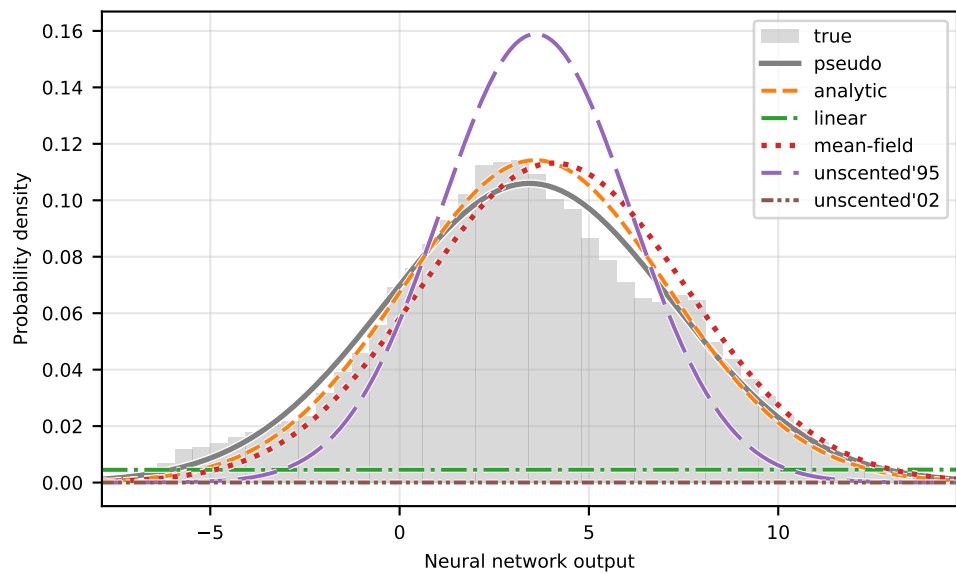

Figure 74: Probability distributions for Network(architecture=deep, weights=initialized, activation=probit residual), variance=large

| distribution | $\mu$ | $\sigma^2$ |
|---|---|---|
| pseudo-true $(Y_1)$ | $+4.098 \times 10^{-1} \pm 3.1 \times 10^{-6}$ | $3.397 \times 10^{-1} \pm 1.1 \times 10^{-5}$ |
| analytic | $+3.878 \times 10^{-1}$ | $3.891 \times 10^{-1}$ |
| mean-field | $+5.497 \times 10^{-1}$ | $8.286 \times 10^{-1}$ |
| linear | $+6.993 \times 10^{-1}$ | $4.761 \times 10^{-1}$ |
| unscented'95 | $+2.598 \times 10^{-1}$ | $3.086 \times 10^{-1}$ |
| unscented'02 | $-2.036 \times 10^{-1}$ | $2.107 \times 10^{0}$ |

Table 134: Comparison of moments for Network(architecture=deep, weights=trained, activation=probit residual), variance=small

| distribution | $d_{\mathrm{W}}(\cdot, Y_0)$ | $D_{\mathrm{KL}}(Y_1 \parallel \cdot)$ |
|---|---|---|
| pseudo-true $(Y_1)$ | $4.032 \times 10^{-2} \pm 1.2 \times 10^{-5}$ | $0$ |
| analytic | $7.162 \times 10^{-2} \pm 9.2 \times 10^{-6}$ | $5.537 \times 10^{-3} \pm 2.3 \times 10^{-6}$ |
| mean-field | $3.717 \times 10^{-1} \pm 8.8 \times 10^{-6}$ | $3.026 \times 10^{-1} \pm 2.5 \times 10^{-5}$ |
| linear | $3.796 \times 10^{-1} \pm 5.8 \times 10^{-6}$ | $1.554 \times 10^{-1} \pm 1.2 \times 10^{-5}$ |
| unscented'95 | $2.047 \times 10^{-1} \pm 9.2 \times 10^{-6}$ | $3.535 \times 10^{-2} \pm 1.6 \times 10^{-6}$ |
| unscented'02 | $1.141 \times 10^{0} \pm 1.6 \times 10^{-5}$ | $2.242 \times 10^{0} \pm 1.0 \times 10^{-4}$ |

Table 135: Comparison of statistical distances for Network(architecture=deep, weights=trained, activation=probit residual), variance=small

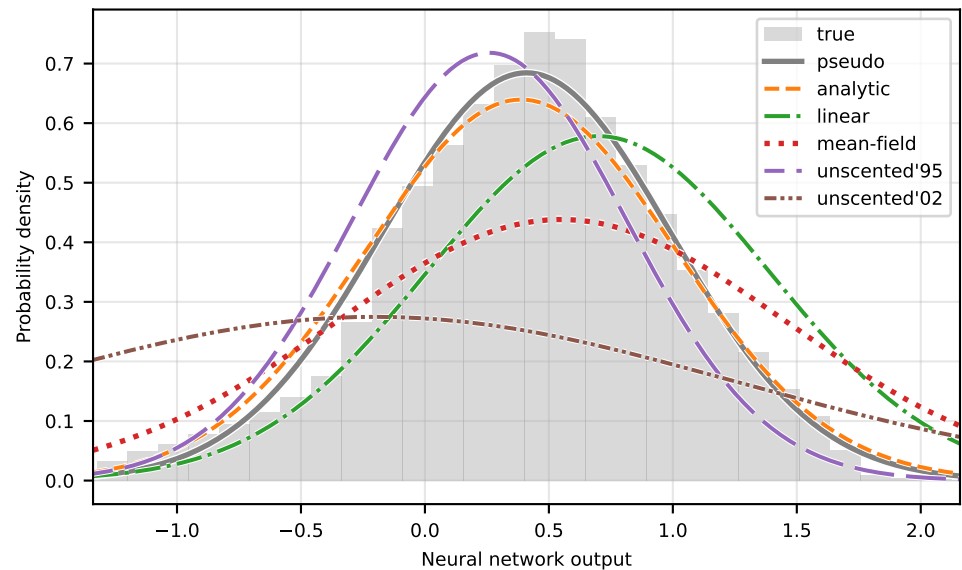

Figure 75: Probability distributions for Network(architecture=deep, weights=trained, activation=probit residual), variance=small

| distribution | $\mu$ | $\sigma^2$ |
|---|---|---|
| pseudo-true $(Y_1)$ | $+2.459 \times 10^0 \pm 5.9 \times 10^{-5}$ | $5.535 \times 10^0 \pm 2.8 \times 10^{-4}$ |
| analytic | $+2.739 \times 10^0$ | $6.167 \times 10^0$ |
| mean-field | $+3.668 \times 10^0$ | $8.286 \times 10^0$ |
| linear | $+6.993 \times 10^{-1}$ | $4.761 \times 10^{+1}$ |
| unscented'95 | $+2.535 \times 10^0$ | $4.672 \times 10^0$ |
| unscented'02 | $-8.959 \times 10^{+1}$ | $1.635 \times 10^{+4}$ |

Table 136: Comparison of moments for Network(architecture=deep, weights=trained, activation=probit residual), variance=medium

| distribution | $d_{\mathrm{W}}(\cdot, Y_0)$ | $D_{\mathrm{KL}}(Y_1 \parallel \cdot)$ |
|---|---|---|
| pseudo-true $(Y_1)$ | $1.405 \times 10^{-1} \pm 5.3 \times 10^{-5}$ | $0$ |
| analytic | $2.169 \times 10^{-1} \pm 3.4 \times 10^{-5}$ | $1.010 \times 10^{-2} \pm 4.0 \times 10^{-6}$ |
| mean-field | $8.063 \times 10^{-1} \pm 3.5 \times 10^{-5}$ | $1.788 \times 10^{-1} \pm 2.1 \times 10^{-5}$ |
| linear | $2.477 \times 10^0 \pm 7.8 \times 10^{-5}$ | $3.005 \times 10^0 \pm 2.1 \times 10^{-4}$ |
| unscented'95 | $1.861 \times 10^{-1} \pm 4.6 \times 10^{-5}$ | $7.311 \times 10^{-3} \pm 4.1 \times 10^{-6}$ |
| unscented'02 | $8.207 \times 10^{+1} \pm 1.1 \times 10^{-3}$ | $2.238 \times 10^{+3} \pm 1.1 \times 10^{-1}$ |

Table 137: Comparison of statistical distances for Network(architecture=deep, weights=trained, activation=probit residual), variance=medium

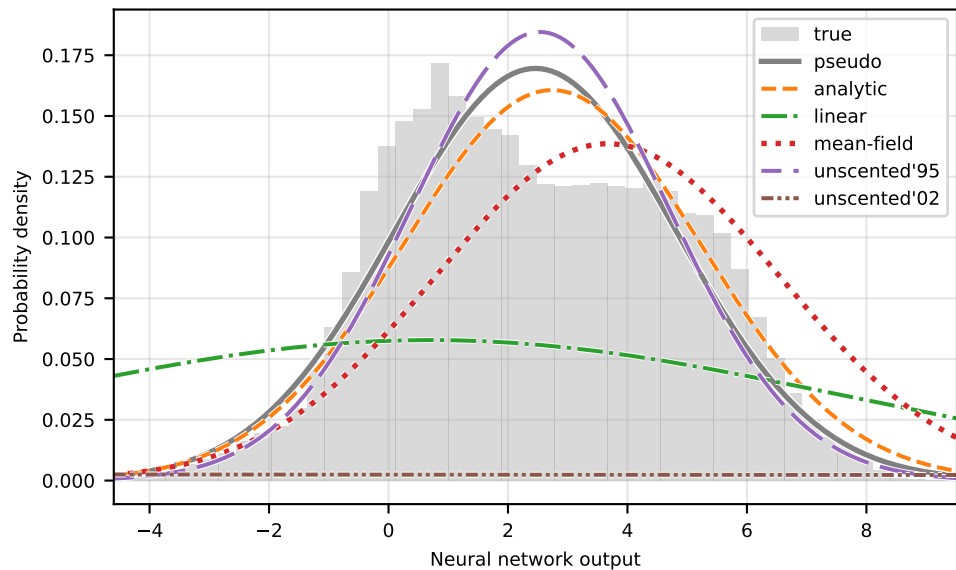

Figure 76: Probability distributions for Network(architecture=deep, weights=trained, activation=probit residual), variance=medium

| distribution | $\mu$ | $\sigma^2$ |
|---|---|---|
| pseudo-true ($Y_1$) | $+1.793 \times 10^0 \pm 2.6 \times 10^{-4}$ | $9.967 \times 10^0 \pm 1.7 \times 10^{-3}$ |
| analytic | $+1.831 \times 10^0$ | $1.005 \times 10^{+1}$ |
| mean-field | $+2.630 \times 10^0$ | $1.248 \times 10^{+1}$ |
| linear | $+6.993 \times 10^{-1}$ | $4.761 \times 10^{+3}$ |
| unscented'95 | $+2.269 \times 10^0$ | $2.237 \times 10^0$ |
| unscented'02 | $-8.971 \times 10^{+3}$ | $1.610 \times 10^{+8}$ |

Table 138: Comparison of moments for Network(architecture=deep, weights=trained, activation=probit residual), variance=large

| distribution | $d_{\mathrm{W}}(\cdot, Y_0)$ | $D_{\mathrm{KL}}(Y_1 \parallel \cdot)$ |
|---|---|---|
| pseudo-true ($Y_1$) | $4.560 \times 10^{-2} \pm 1.5 \times 10^{-4}$ | $0$ |
| analytic | $4.399 \times 10^{-2} \pm 1.5 \times 10^{-4}$ | $9.368 \times 10^{-5} \pm 1.4 \times 10^{-6}$ |
| mean-field | $4.740 \times 10^{-1} \pm 1.5 \times 10^{-4}$ | $4.881 \times 10^{-2} \pm 3.9 \times 10^{-5}$ |
| linear | $2.957 \times 10^{+1} \pm 1.3 \times 10^{-3}$ | $2.353 \times 10^{+2} \pm 4.1 \times 10^{-2}$ |
| unscented'95 | $7.773 \times 10^{-1} \pm 1.2 \times 10^{-4}$ | $3.707 \times 10^{-1} \pm 6.2 \times 10^{-5}$ |
| unscented'02 | $7.065 \times 10^{+3} \pm 3.0 \times 10^{-1}$ | $1.212 \times 10^{+7} \pm 2.1 \times 10^{+3}$ |

Table 139: Comparison of statistical distances for Network(architecture=deep, weights=trained, activation=probit residual), variance=large

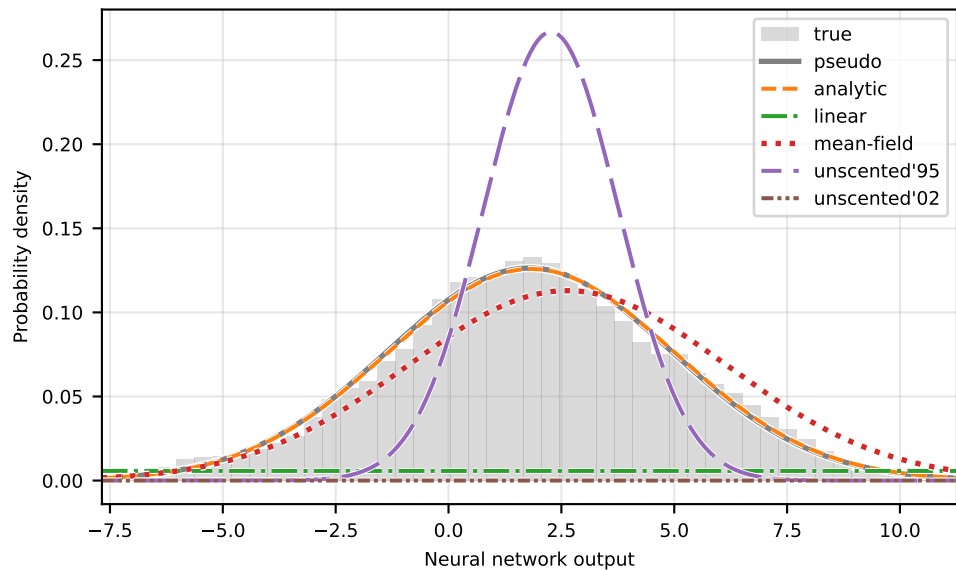

Figure 77: Probability distributions for Network(architecture=deep, weights=trained, activation=probit residual), variance=large

| distribution | $\mu$ | $\sigma^2$ |
|---|---|---|
| pseudo-true ($Y_1$) | $+8.681 \times 10^{-2} \pm 1.7 \times 10^{-7}$ | $1.227 \times 10^{-2} \pm 9.2 \times 10^{-8}$ |
| analytic | $+8.729 \times 10^{-2}$ | $1.224 \times 10^{-2}$ |
| mean-field | $+7.827 \times 10^{-2}$ | $7.985 \times 10^{-3}$ |
| linear | $+1.006 \times 10^{-1}$ | $1.994 \times 10^{-2}$ |
| unscented'95 | $+8.267 \times 10^{-2}$ | $1.311 \times 10^{-2}$ |
| unscented'02 | $+8.112 \times 10^{-2}$ | $2.070 \times 10^{-2}$ |

Table 140: Comparison of moments for Network(architecture=deep, weights=initialized, activation=sine), variance=small

| distribution | $d_{\mathrm{W}}(\cdot, Y_0)$ | $D_{\mathrm{KL}}(Y_1 \parallel \cdot)$ |
|---|---|---|
| pseudo-true ($Y_1$) | $7.001 \times 10^{-2} \pm 3.1 \times 10^{-6}$ | $0$ |
| analytic | $6.975 \times 10^{-2} \pm 3.3 \times 10^{-6}$ | $1.113 \times 10^{-5} \pm 1.1 \times 10^{-8}$ |
| mean-field | $9.025 \times 10^{-2} \pm 3.3 \times 10^{-6}$ | $4.323 \times 10^{-2} \pm 1.3 \times 10^{-6}$ |
| linear | $8.098 \times 10^{-2} \pm 3.8 \times 10^{-6}$ | $7.746 \times 10^{-2} \pm 2.4 \times 10^{-6}$ |
| unscented'95 | $7.278 \times 10^{-2} \pm 3.1 \times 10^{-6}$ | $1.795 \times 10^{-3} \pm 2.5 \times 10^{-7}$ |
| unscented'02 | $1.092 \times 10^{-1} \pm 1.6 \times 10^{-6}$ | $8.327 \times 10^{-2} \pm 2.6 \times 10^{-6}$ |

Table 141: Comparison of statistical distances for Network(architecture=deep, weights=initialized, activation=sine), variance=small

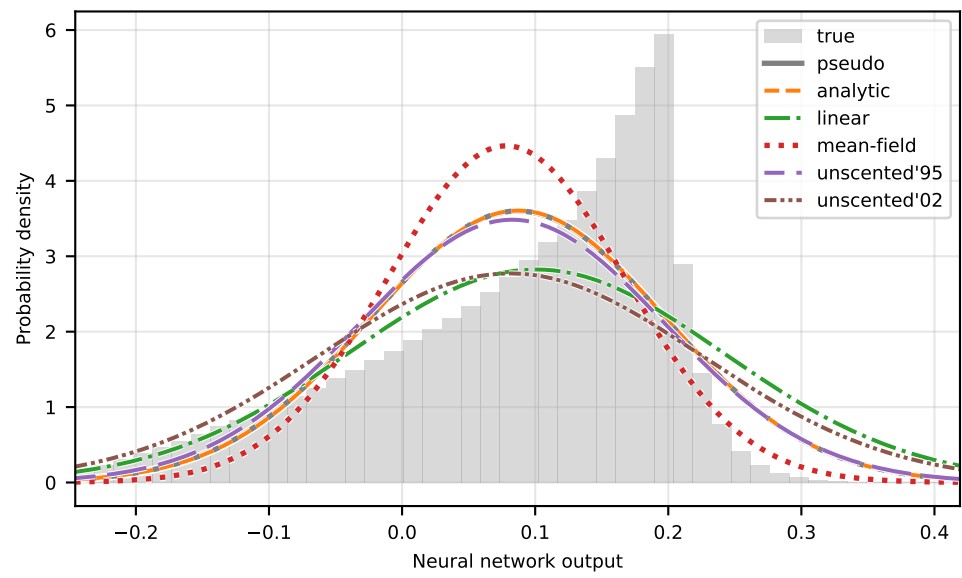

Figure 78: Probability distributions for Network(architecture=deep, weights=initialized, activation=sine), variance=small

| distribution | $\mu$ | $\sigma^2$ |
|---|---|---|
| pseudo-true $(Y_1)$ | $-7.468 \times 10^{-2} \pm 1.8 \times 10^{-5}$ | $3.835 \times 10^{-2} \pm 7.9 \times 10^{-6}$ |
| analytic | $-8.706 \times 10^{-2}$ | $3.731 \times 10^{-2}$ |
| mean-field | $-1.600 \times 10^{-1}$ | $3.821 \times 10^{-2}$ |
| linear | $+1.006 \times 10^{-1}$ | $1.994 \times 10^{0}$ |
| unscented'95 | $-1.997 \times 10^{-1}$ | $4.768 \times 10^{-2}$ |
| unscented'02 | $-1.848 \times 10^{0}$ | $9.588 \times 10^{0}$ |

Table 142: Comparison of moments for Network(architecture=deep, weights=initialized, activation=sine), variance=medium

| distribution | $d_{\mathrm{W}}(\cdot, Y_0)$ | $D_{\mathrm{KL}}(Y_1 \parallel \cdot)$ |
|---|---|---|
| pseudo-true $(Y_1)$ | $3.136 \times 10^{-2} \pm 4.1 \times 10^{-5}$ | $0$ |
| analytic | $4.568 \times 10^{-2} \pm 3.7 \times 10^{-5}$ | $2.187 \times 10^{-3} \pm 5.0 \times 10^{-6}$ |
| mean-field | $1.934 \times 10^{-1} \pm 4.6 \times 10^{-5}$ | $9.502 \times 10^{-2} \pm 5.3 \times 10^{-5}$ |
| linear | $2.210 \times 10^{0} \pm 1.5 \times 10^{-4}$ | $2.393 \times 10^{+1} \pm 5.3 \times 10^{-3}$ |
| unscented'95 | $2.834 \times 10^{-1} \pm 4.9 \times 10^{-5}$ | $2.164 \times 10^{-1} \pm 1.1 \times 10^{-4}$ |
| unscented'02 | $6.191 \times 10^{0} \pm 3.4 \times 10^{-4}$ | $1.627 \times 10^{+2} \pm 3.4 \times 10^{-2}$ |

Table 143: Comparison of statistical distances for Network(architecture=deep, weights=initialized, activation=sine), variance=medium

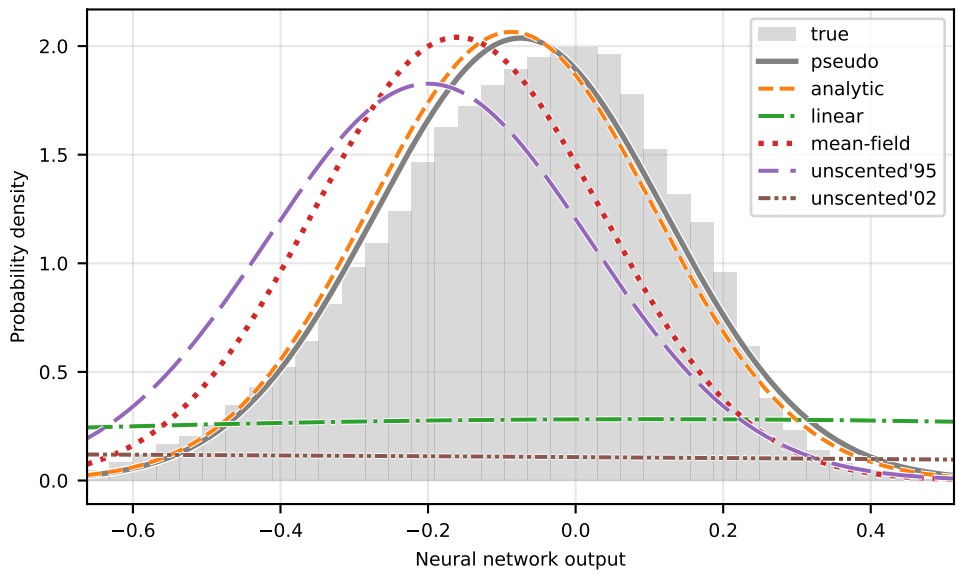

Figure 79: Probability distributions for Network(architecture=deep, weights=initialized, activation=sine), variance=medium

| distribution | $\mu$ | $\sigma^2$ |
|---|---|---|
| pseudo-true ($Y_1$) | $-1.076 \times 10^{-1} \pm 1.4 \times 10^{-4}$ | $3.749 \times 10^{-2} \pm 5.1 \times 10^{-5}$ |
| analytic | $-1.069 \times 10^{-1}$ | $3.680 \times 10^{-2}$ |
| mean-field | $-1.584 \times 10^{-1}$ | $3.866 \times 10^{-2}$ |
| linear | $+1.006 \times 10^{-1}$ | $1.994 \times 10^{+2}$ |
| unscented'95 | $-7.389 \times 10^{-2}$ | $3.900 \times 10^{-2}$ |
| unscented'02 | $-1.946 \times 10^{+2}$ | $7.604 \times 10^{+4}$ |

Table 144: Comparison of moments for Network(architecture=deep, weights=initialized, activation=sine), variance=large

| distribution | $d_{\mathrm{W}}(\cdot, Y_0)$ | $D_{\mathrm{KL}}(Y_1 \parallel \cdot)$ |
|---|---|---|
| pseudo-true ($Y_1$) | $2.691 \times 10^{-2} \pm 2.1 \times 10^{-4}$ | $0$ |
| analytic | $2.623 \times 10^{-2} \pm 2.3 \times 10^{-4}$ | $1.076 \times 10^{-4} \pm 1.3 \times 10^{-5}$ |
| mean-field | $1.175 \times 10^{-1} \pm 2.9 \times 10^{-4}$ | $3.469 \times 10^{-2} \pm 2.2 \times 10^{-4}$ |
| linear | $2.526 \times 10^{+1} \pm 8.9 \times 10^{-3}$ | $2.656 \times 10^{+3} \pm 3.7 \times 10^{0}$ |
| unscented'95 | $7.668 \times 10^{-2} \pm 3.0 \times 10^{-4}$ | $1.559 \times 10^{-2} \pm 1.2 \times 10^{-4}$ |
| unscented'02 | $6.193 \times 10^{+2} \pm 2.1 \times 10^{-1}$ | $1.519 \times 10^{+6} \pm 2.1 \times 10^{+3}$ |

Table 145: Comparison of statistical distances for Network(architecture=deep, weights=initialized, activation=sine), variance=large

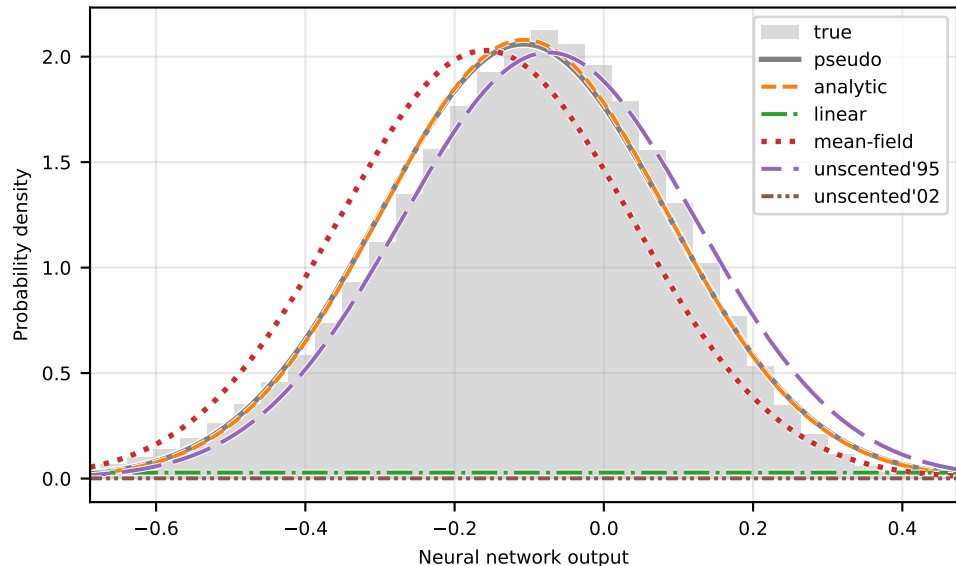

Figure 80: Probability distributions for Network(architecture=deep, weights=initialized, activation=sine), variance=large

| distribution | $\mu$ | $\sigma^2$ |
|---|---|---|
| pseudo-true ($Y_1$) | $-3.309 \times 10^{-1} \pm 3.4 \times 10^{-7}$ | $2.760 \times 10^{-2} \pm 4.2 \times 10^{-7}$ |
| analytic | $-3.310 \times 10^{-1}$ | $2.474 \times 10^{-2}$ |
| mean-field | $-2.899 \times 10^{-1}$ | $9.013 \times 10^{-3}$ |
| linear | $-2.892 \times 10^{-1}$ | $2.971 \times 10^{-2}$ |
| unscented'95 | $-3.366 \times 10^{-1}$ | $2.708 \times 10^{-2}$ |
| unscented'02 | $-3.311 \times 10^{-1}$ | $3.322 \times 10^{-2}$ |

Table 146: Comparison of moments for Network(architecture=deep, weights=trained, activation=sine), variance=small

| distribution | $d_{\mathrm{W}}(\cdot, Y_0)$ | $D_{\mathrm{KL}}(Y_1 \parallel \cdot)$ |
|---|---|---|
| pseudo-true ($Y_1$) | $4.504 \times 10^{-2} \pm 3.2 \times 10^{-6}$ | $0$ |
| analytic | $4.779 \times 10^{-2} \pm 3.8 \times 10^{-6}$ | $2.875 \times 10^{-3} \pm 7.9 \times 10^{-7}$ |
| mean-field | $1.453 \times 10^{-1} \pm 2.4 \times 10^{-6}$ | $2.534 \times 10^{-1} \pm 4.7 \times 10^{-6}$ |
| linear | $1.023 \times 10^{-1} \pm 9.3 \times 10^{-7}$ | $3.288 \times 10^{-2} \pm 1.2 \times 10^{-6}$ |
| unscented'95 | $5.108 \times 10^{-2} \pm 4.4 \times 10^{-6}$ | $6.692 \times 10^{-4} \pm 1.5 \times 10^{-7}$ |
| unscented'02 | $5.361 \times 10^{-2} \pm 4.3 \times 10^{-6}$ | $9.147 \times 10^{-3} \pm 1.6 \times 10^{-6}$ |

Table 147: Comparison of statistical distances for Network(architecture=deep, weights=trained, activation=sine), variance=small

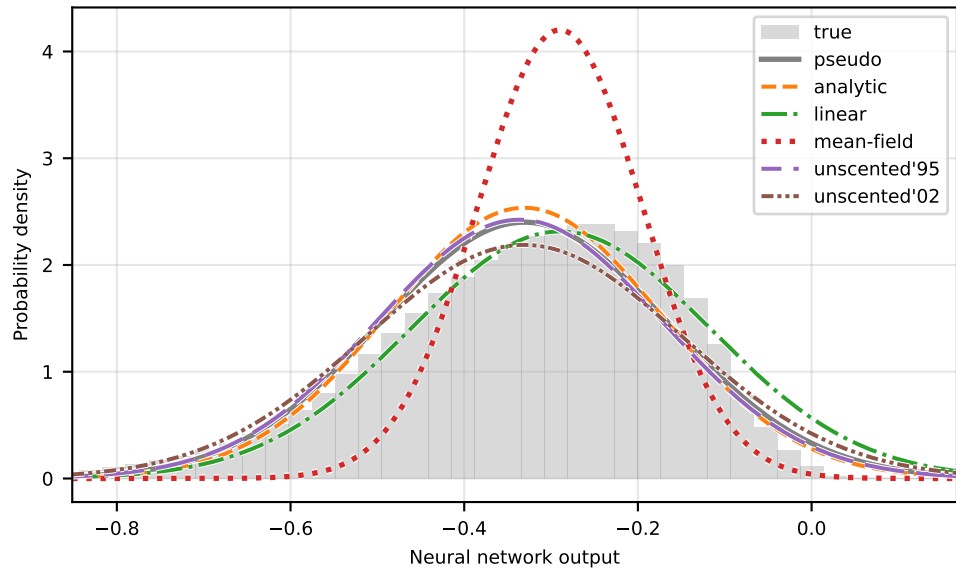

Figure 81: Probability distributions for Network(architecture=deep, weights=trained, activation=sine), variance=small

| distribution | $\mu$ | $\sigma^2$ |
|---|---|---|
| pseudo-true ($Y_1$) | $-2.705 \times 10^{-2} \pm 2.4 \times 10^{-5}$ | $1.056 \times 10^{-1} \pm 1.7 \times 10^{-5}$ |
| analytic | $+3.115 \times 10^{-2}$ | $5.628 \times 10^{-2}$ |
| mean-field | $-2.939 \times 10^{-2}$ | $4.324 \times 10^{-2}$ |
| linear | $-2.892 \times 10^{-1}$ | $2.971 \times 10^{0}$ |
| unscented'95 | $-1.594 \times 10^{-1}$ | $5.871 \times 10^{-2}$ |
| unscented'02 | $-4.479 \times 10^{0}$ | $3.808 \times 10^{+1}$ |

Table 148: Comparison of moments for Network(architecture=deep, weights=trained, activation=sine), variance=medium

| distribution | $d_{\mathrm{W}}(\cdot, Y_0)$ | $D_{\mathrm{KL}}(Y_1 \parallel \cdot)$ |
|---|---|---|
| pseudo-true ($Y_1$) | $2.870 \times 10^{-2} \pm 3.2 \times 10^{-5}$ | $0$ |
| analytic | $1.373 \times 10^{-1} \pm 4.2 \times 10^{-5}$ | $9.720 \times 10^{-2} \pm 3.5 \times 10^{-5}$ |
| mean-field | $1.568 \times 10^{-1} \pm 3.6 \times 10^{-5}$ | $1.513 \times 10^{-1} \pm 4.9 \times 10^{-5}$ |
| linear | $2.004 \times 10^{0} \pm 1.2 \times 10^{-4}$ | $1.222 \times 10^{+1} \pm 2.3 \times 10^{-3}$ |
| unscented'95 | $2.508 \times 10^{-1} \pm 3.8 \times 10^{-5}$ | $1.545 \times 10^{-1} \pm 4.1 \times 10^{-5}$ |
| unscented'02 | $1.046 \times 10^{+1} \pm 4.7 \times 10^{-4}$ | $2.706 \times 10^{+2} \pm 4.5 \times 10^{-2}$ |

Table 149: Comparison of statistical distances for Network(architecture=deep, weights=trained, activation=sine), variance=medium

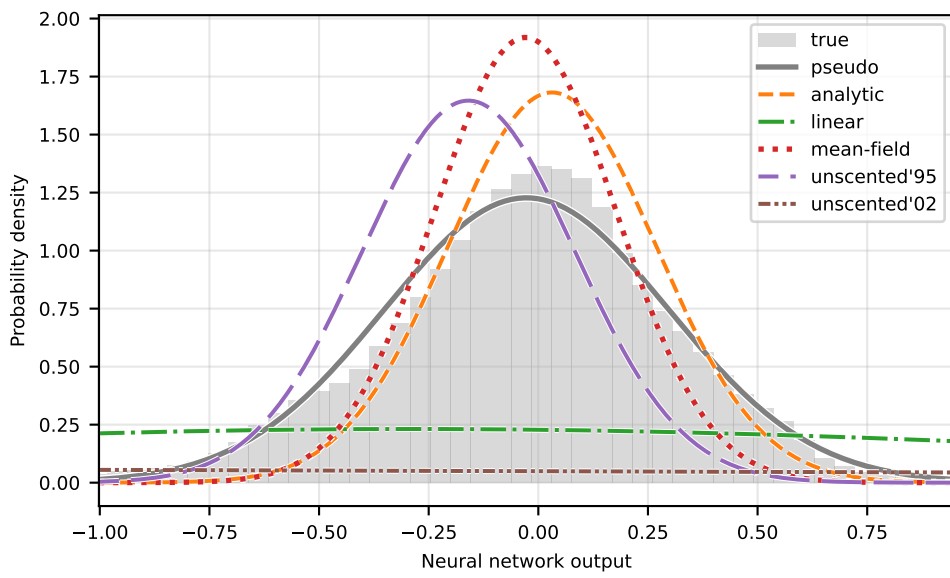

Figure 82: Probability distributions for Network(architecture=deep, weights=trained, activation=sine), variance=medium

| distribution | $\mu$ | $\sigma^2$ |
|---|---|---|
| pseudo-true ($Y_1$) | $+2.810 \times 10^{-2} \pm 1.6 \times 10^{-4}$ | $7.180 \times 10^{-2} \pm 5.9 \times 10^{-5}$ |
| analytic | $+3.082 \times 10^{-2}$ | $5.672 \times 10^{-2}$ |
| mean-field | $-4.625 \times 10^{-3}$ | $4.329 \times 10^{-2}$ |
| linear | $-2.892 \times 10^{-1}$ | $2.971 \times 10^{+2}$ |
| unscented'95 | $+5.393 \times 10^{-2}$ | $2.043 \times 10^{-2}$ |
| unscented'02 | $-4.205 \times 10^{+2}$ | $3.534 \times 10^{+5}$ |

Table 150: Comparison of moments for Network(architecture=deep, weights=trained, activation=sine), variance=large

| distribution | $d_{\mathrm{W}}(\cdot, Y_0)$ | $D_{\mathrm{KL}}(Y_1 \parallel \cdot)$ |
|---|---|---|
| pseudo-true ($Y_1$) | $4.247 \times 10^{-3} \pm 1.3 \times 10^{-4}$ | $0$ |
| analytic | $4.589 \times 10^{-2} \pm 1.8 \times 10^{-4}$ | $1.292 \times 10^{-2} \pm 8.6 \times 10^{-5}$ |
| mean-field | $1.065 \times 10^{-1} \pm 2.0 \times 10^{-4}$ | $6.192 \times 10^{-2} \pm 1.8 \times 10^{-4}$ |
| linear | $2.616 \times 10^{+1} \pm 5.5 \times 10^{-3}$ | $2.065 \times 10^{+3} \pm 1.7 \times 10^{0}$ |
| unscented'95 | $1.966 \times 10^{-1} \pm 1.5 \times 10^{-4}$ | $2.754 \times 10^{-1} \pm 2.9 \times 10^{-4}$ |
| unscented'02 | $1.136 \times 10^{+3} \pm 2.3 \times 10^{-1}$ | $3.692 \times 10^{+6} \pm 3.0 \times 10^{+3}$ |

Table 151: Comparison of statistical distances for Network(architecture=deep, weights=trained, activation=sine), variance=large

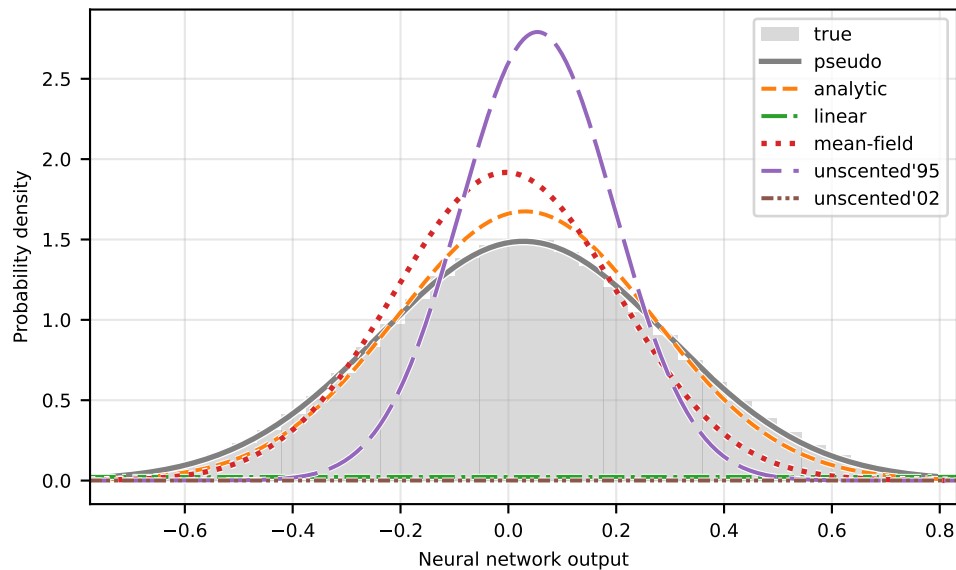

Figure 83: Probability distributions for Network(architecture=deep, weights=trained, activation=sine), variance=large

| distribution | $\mu$ | $\sigma^2$ |
|---|---|---|
| pseudo-true ($Y_1$) | $+1.833 \times 10^0 \pm 9.3 \times 10^{-4}$ | $5.519 \times 10^0 \pm 3.5 \times 10^{-3}$ |
| analytic | $+1.634 \times 10^0$ | $6.420 \times 10^0$ |
| mean-field | $+1.506 \times 10^0$ | $7.240 \times 10^0$ |
| linear | $+5.635 \times 10^0$ | $4.854 \times 10^{+3}$ |
| unscented'95 | $+1.465 \times 10^0$ | $3.321 \times 10^0$ |
| unscented'02 | $+2.030 \times 10^{+2}$ | $8.267 \times 10^{+4}$ |

Table 152: Comparison of moments for Network(architecture=deep, weights=initialized, activation=sine residual), variance=small

| distribution | $d_{\mathrm{W}}(\cdot, Y_0)$ | $D_{\mathrm{KL}}(Y_1 \parallel \cdot)$ |
|---|---|---|
| pseudo-true ($Y_1$) | $1.564 \times 10^{-2} \pm 4.7 \times 10^{-4}$ | $0$ |
| analytic | $1.521 \times 10^{-1} \pm 4.8 \times 10^{-4}$ | $9.624 \times 10^{-3} \pm 6.7 \times 10^{-5}$ |
| mean-field | $2.590 \times 10^{-1} \pm 4.8 \times 10^{-4}$ | $2.989 \times 10^{-2} \pm 1.3 \times 10^{-4}$ |
| linear | $3.510 \times 10^{+1} \pm 6.0 \times 10^{-3}$ | $4.371 \times 10^{+2} \pm 2.8 \times 10^{-1}$ |
| unscented'95 | $3.414 \times 10^{-1} \pm 3.5 \times 10^{-4}$ | $6.712 \times 10^{-2} \pm 1.3 \times 10^{-4}$ |
| unscented'02 | $1.839 \times 10^{+2} \pm 3.0 \times 10^{-2}$ | $1.115 \times 10^{+4} \pm 7.1 \times 10^0$ |

Table 153: Comparison of statistical distances for Network(architecture=deep, weights=initialized, activation=sine residual), variance=small

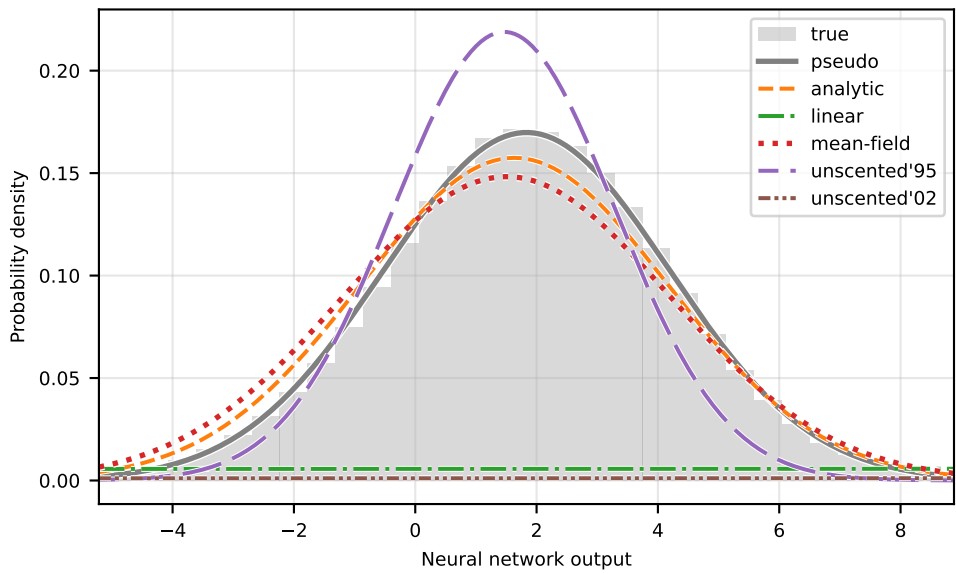

Figure 84: Probability distributions for Network(architecture=deep, weights=initialized, activation=sine residual), variance=small

| distribution | $\mu$ | $\sigma^2$ |
|---|---|---|
| pseudo-true $(Y_1)$ | $+5.948 \times 10^{-1} \pm 1.9 \times 10^{-3}$ | $1.022 \times 10^{+1} \pm 1.0 \times 10^{-2}$ |
| analytic | $+6.953 \times 10^{-1}$ | $1.046 \times 10^{+1}$ |
| mean-field | $+8.278 \times 10^{-1}$ | $1.010 \times 10^{+1}$ |
| linear | $+5.635 \times 10^{0}$ | $4.854 \times 10^{+5}$ |
| unscented'95 | $-2.237 \times 10^{-2}$ | $8.705 \times 10^{0}$ |
| unscented'02 | $+1.935 \times 10^{+4}$ | $7.487 \times 10^{+8}$ |

Table 154: Comparison of moments for Network(architecture=deep, weights=initialized, activation=sine residual), variance=medium

| distribution | $d_{\mathrm{W}}(\cdot, Y_0)$ | $D_{\mathrm{KL}}(Y_1 \parallel \cdot)$ |
|---|---|---|
| pseudo-true $(Y_1)$ | $1.137 \times 10^{-2} \pm 5.1 \times 10^{-4}$ | $0$ |
| analytic | $5.630 \times 10^{-2} \pm 1.0 \times 10^{-3}$ | $6.376 \times 10^{-4} \pm 2.5 \times 10^{-5}$ |
| mean-field | $1.303 \times 10^{-1} \pm 1.0 \times 10^{-3}$ | $2.700 \times 10^{-3} \pm 4.1 \times 10^{-5}$ |
| linear | $3.095 \times 10^{+2} \pm 8.0 \times 10^{-2}$ | $2.374 \times 10^{+4} \pm 2.4 \times 10^{+1}$ |
| unscented'95 | $3.467 \times 10^{-1} \pm 1.0 \times 10^{-3}$ | $2.476 \times 10^{-2} \pm 1.4 \times 10^{-4}$ |
| unscented'02 | $1.514 \times 10^{+4} \pm 3.9 \times 10^{0}$ | $5.494 \times 10^{+7} \pm 5.6 \times 10^{+4}$ |

Table 155: Comparison of statistical distances for Network(architecture=deep, weights=initialized, activation=sine residual), variance=medium

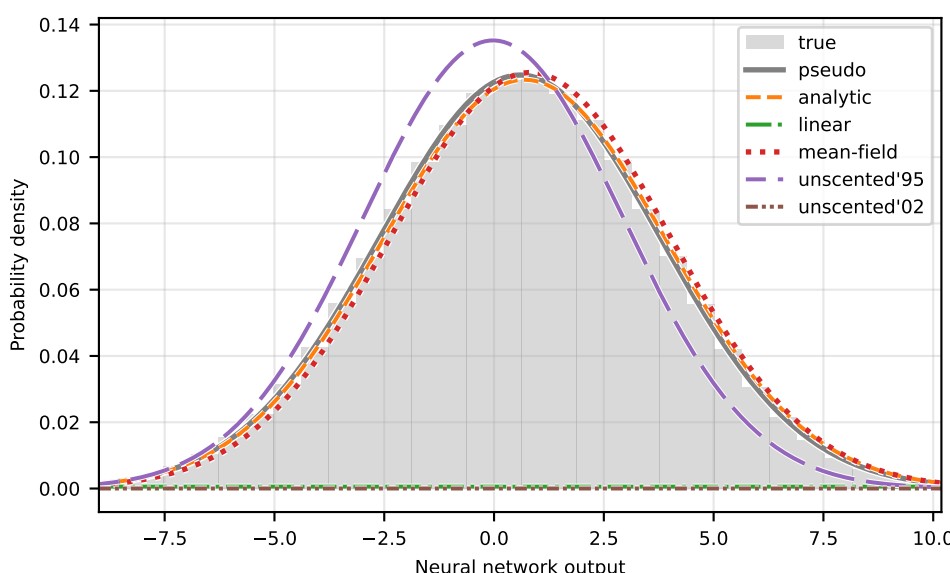

Figure 85: Probability distributions for Network(architecture=deep, weights=initialized, activation=sine residual), variance=medium

| distribution | $\mu$ | $\sigma^2$ |
|---|---|---|
| pseudo-true ($Y_1$) | $+7.725 \times 10^{-1} \pm 2.2 \times 10^{-3}$ | $1.062 \times 10^{+1} \pm 1.2 \times 10^{-2}$ |
| analytic | $+7.669 \times 10^{-1}$ | $1.060 \times 10^{+1}$ |
| mean-field | $+6.359 \times 10^{-1}$ | $1.038 \times 10^{+1}$ |
| linear | $+5.635 \times 10^{0}$ | $4.854 \times 10^{+7}$ |
| unscented'95 | $-2.682 \times 10^{0}$ | $9.806 \times 10^{0}$ |
| unscented'02 | $-1.840 \times 10^{+6}$ | $6.771 \times 10^{+12}$ |

Table 156: Comparison of moments for Network(architecture=deep, weights=initialized, activation=sine residual), variance=large

| distribution | $d_{\mathrm{W}}(\cdot, Y_0)$ | $D_{\mathrm{KL}}(Y_1 \parallel \cdot)$ |
|---|---|---|
| pseudo-true ($Y_1$) | $5.743 \times 10^{-3} \pm 2.9 \times 10^{-4}$ | $0$ |
| analytic | $8.862 \times 10^{-3} \pm 6.3 \times 10^{-4}$ | $1.424 \times 10^{-5} \pm 3.0 \times 10^{-6}$ |
| mean-field | $7.593 \times 10^{-2} \pm 1.2 \times 10^{-3}$ | $1.020 \times 10^{-3} \pm 2.7 \times 10^{-5}$ |
| linear | $3.077 \times 10^{+3} \pm 8.8 \times 10^{-1}$ | $2.284 \times 10^{+6} \pm 2.6 \times 10^{+3}$ |
| unscented'95 | $1.913 \times 10^{0} \pm 1.5 \times 10^{-3}$ | $5.632 \times 10^{-1} \pm 1.1 \times 10^{-3}$ |
| unscented'02 | $1.426 \times 10^{+6} \pm 4.1 \times 10^{+2}$ | $4.780 \times 10^{+11} \pm 5.4 \times 10^{+8}$ |

Table 157: Comparison of statistical distances for Network(architecture=deep, weights=initialized, activation=sine residual), variance=large

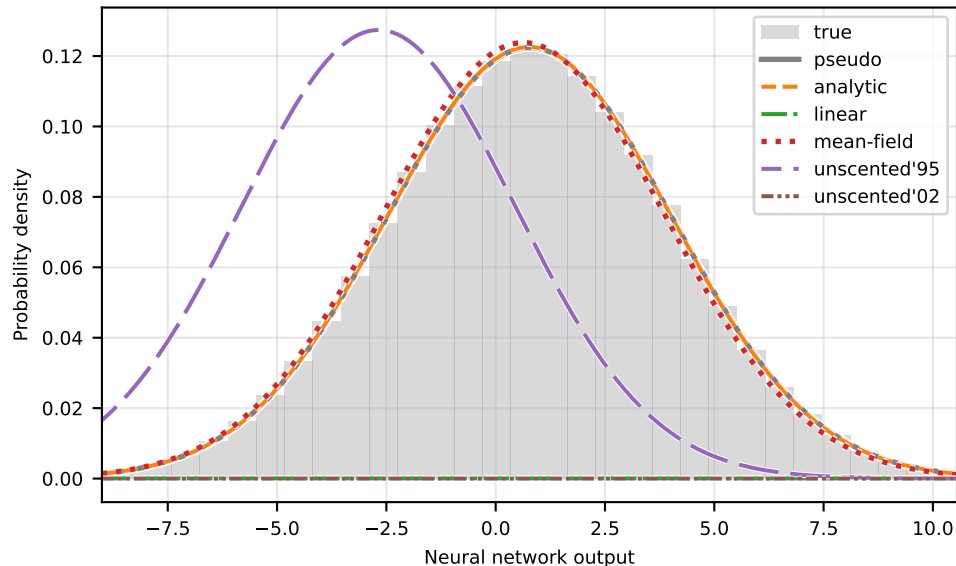

Figure 86: Probability distributions for Network(architecture=deep, weights=initialized, activation=sine residual), variance=large

| distribution | $\mu$ | $\sigma^2$ |
|---|---|---|
| pseudo-true $(Y_1)$ | $+1.816 \times 10^0 \pm 1.1 \times 10^{-3}$ | $5.545 \times 10^0 \pm 4.2 \times 10^{-3}$ |
| analytic | $+1.657 \times 10^0$ | $6.417 \times 10^0$ |
| mean-field | $+1.526 \times 10^0$ | $7.238 \times 10^0$ |
| linear | $+2.756 \times 10^0$ | $2.222 \times 10^{+3}$ |
| unscented'95 | $+1.666 \times 10^0$ | $3.020 \times 10^0$ |
| unscented'02 | $-1.425 \times 10^{+3}$ | $4.076 \times 10^{+6}$ |

Table 158: Comparison of moments for Network(architecture=deep, weights=trained, activation=sine residual), variance=small

| distribution | $d_{\mathrm{W}}(\cdot, Y_0)$ | $D_{\mathrm{KL}}(Y_1 \parallel \cdot)$ |
|---|---|---|
| pseudo-true $(Y_1)$ | $1.305 \times 10^{-2} \pm 3.9 \times 10^{-4}$ | $0$ |
| analytic | $1.324 \times 10^{-1} \pm 4.7 \times 10^{-4}$ | $7.895 \times 10^{-3} \pm 6.0 \times 10^{-5}$ |
| mean-field | $2.408 \times 10^{-1} \pm 4.8 \times 10^{-4}$ | $2.701 \times 10^{-2} \pm 1.2 \times 10^{-4}$ |
| linear | $2.330 \times 10^{+1} \pm 4.9 \times 10^{-3}$ | $1.970 \times 10^{+2} \pm 1.5 \times 10^{-1}$ |
| unscented'95 | $3.278 \times 10^{-1} \pm 5.5 \times 10^{-4}$ | $7.811 \times 10^{-2} \pm 1.8 \times 10^{-4}$ |
| unscented'02 | $1.300 \times 10^{+3} \pm 2.5 \times 10^{-1}$ | $5.511 \times 10^{+5} \pm 4.2 \times 10^{+2}$ |

Table 159: Comparison of statistical distances for Network(architecture=deep, weights=trained, activation=sine residual), variance=small

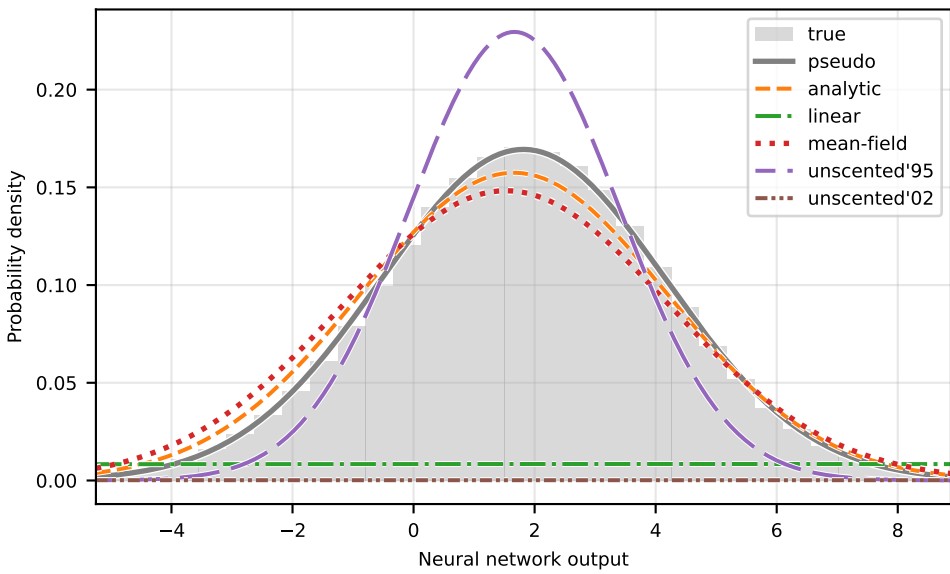

Figure 87: Probability distributions for Network(architecture=deep, weights=trained, activation=sine residual), variance=small

| distribution | $\mu$ | $\sigma^2$ |
|---|---|---|
| pseudo-true ($Y_1$) | $+5.940 \times 10^{-1} \pm 2.0 \times 10^{-3}$ | $1.022 \times 10^{+1} \pm 1.1 \times 10^{-2}$ |
| analytic | $+6.930 \times 10^{-1}$ | $1.046 \times 10^{+1}$ |
| mean-field | $+8.258 \times 10^{-1}$ | $1.010 \times 10^{+1}$ |
| linear | $+2.756 \times 10^{0}$ | $2.222 \times 10^{+5}$ |
| unscented'95 | $-1.853 \times 10^{-1}$ | $9.735 \times 10^{0}$ |
| unscented'02 | $+4.562 \times 10^{+3}$ | $4.183 \times 10^{+7}$ |

Table 160: Comparison of moments for Network(architecture=deep, weights=trained, activation=sine residual), variance=medium

| distribution | $d_{\mathrm{W}}(\cdot, Y_0)$ | $D_{\mathrm{KL}}(Y_1 \parallel \cdot)$ |
|---|---|---|
| pseudo-true ($Y_1$) | $1.207 \times 10^{-2} \pm 5.1 \times 10^{-4}$ | $0$ |
| analytic | $5.551 \times 10^{-2} \pm 1.1 \times 10^{-3}$ | $6.270 \times 10^{-4} \pm 2.7 \times 10^{-5}$ |
| mean-field | $1.297 \times 10^{-1} \pm 1.1 \times 10^{-3}$ | $2.673 \times 10^{-3} \pm 4.4 \times 10^{-5}$ |
| linear | $2.090 \times 10^{+2} \pm 5.8 \times 10^{-2}$ | $1.087 \times 10^{+4} \pm 1.2 \times 10^{+1}$ |
| unscented'95 | $4.359 \times 10^{-1} \pm 1.1 \times 10^{-3}$ | $3.031 \times 10^{-2} \pm 1.5 \times 10^{-4}$ |
| unscented'02 | $3.575 \times 10^{+3} \pm 9.8 \times 10^{-1}$ | $3.065 \times 10^{+6} \pm 3.4 \times 10^{+3}$ |

Table 161: Comparison of statistical distances for Network(architecture=deep, weights=trained, activation=sine residual), variance=medium

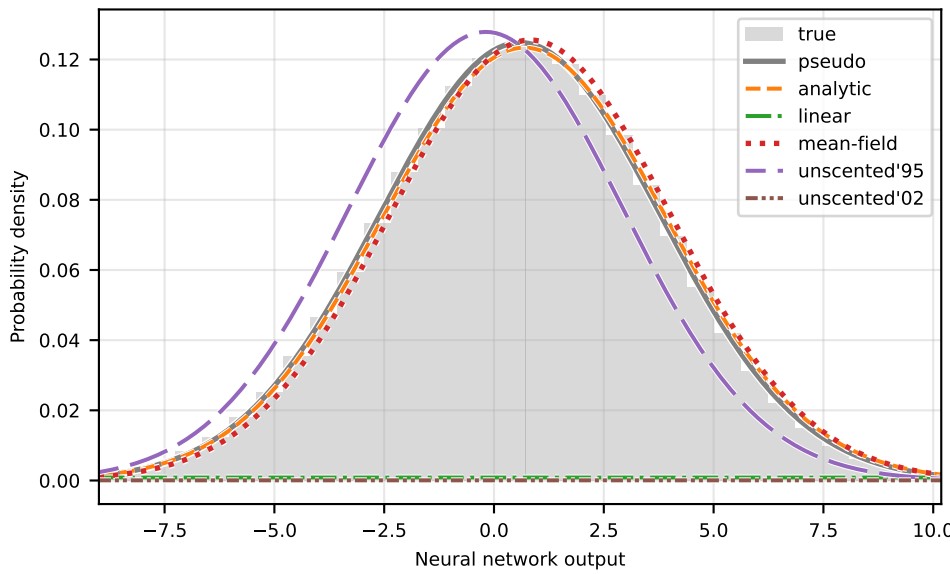

Figure 88: Probability distributions for Network(architecture=deep, weights=trained, activation=sine residual), variance=medium

| distribution | $\mu$ | $\sigma^2$ |
|---|---|---|
| pseudo-true ($Y_1$) | $+7.718 \times 10^{-1} \pm 2.1 \times 10^{-3}$ | $1.063 \times 10^{+1} \pm 1.2 \times 10^{-2}$ |
| analytic | $+7.654 \times 10^{-1}$ | $1.059 \times 10^{+1}$ |
| mean-field | $+6.345 \times 10^{-1}$ | $1.038 \times 10^{+1}$ |
| linear | $+2.756 \times 10^{0}$ | $2.222 \times 10^{+7}$ |
| unscented'95 | $-2.734 \times 10^{0}$ | $9.988 \times 10^{0}$ |
| unscented'02 | $-2.299 \times 10^{+4}$ | $1.059 \times 10^{+9}$ |

Table 162: Comparison of moments for Network(architecture=deep, weights=trained, activation=sine residual), variance=large

| distribution | $d_{\mathrm{W}}(\cdot, Y_0)$ | $D_{\mathrm{KL}}(Y_1 \parallel \cdot)$ |
|---|---|---|
| pseudo-true ($Y_1$) | $5.821 \times 10^{-3} \pm 3.1 \times 10^{-4}$ | $0$ |
| analytic | $8.940 \times 10^{-3} \pm 6.7 \times 10^{-4}$ | $1.504 \times 10^{-5} \pm 3.2 \times 10^{-6}$ |
| mean-field | $7.640 \times 10^{-2} \pm 1.2 \times 10^{-3}$ | $1.033 \times 10^{-3} \pm 2.5 \times 10^{-5}$ |
| linear | $2.082 \times 10^{+3} \pm 5.9 \times 10^{-1}$ | $1.046 \times 10^{+6} \pm 1.2 \times 10^{+3}$ |
| unscented'95 | $1.942 \times 10^{0} \pm 1.5 \times 10^{-3}$ | $5.794 \times 10^{-1} \pm 1.1 \times 10^{-3}$ |
| unscented'02 | $1.782 \times 10^{+4} \pm 5.1 \times 10^{0}$ | $7.468 \times 10^{+7} \pm 8.5 \times 10^{+4}$ |

Table 163: Comparison of statistical distances for Network(architecture=deep, weights=trained, activation=sine residual), variance=large

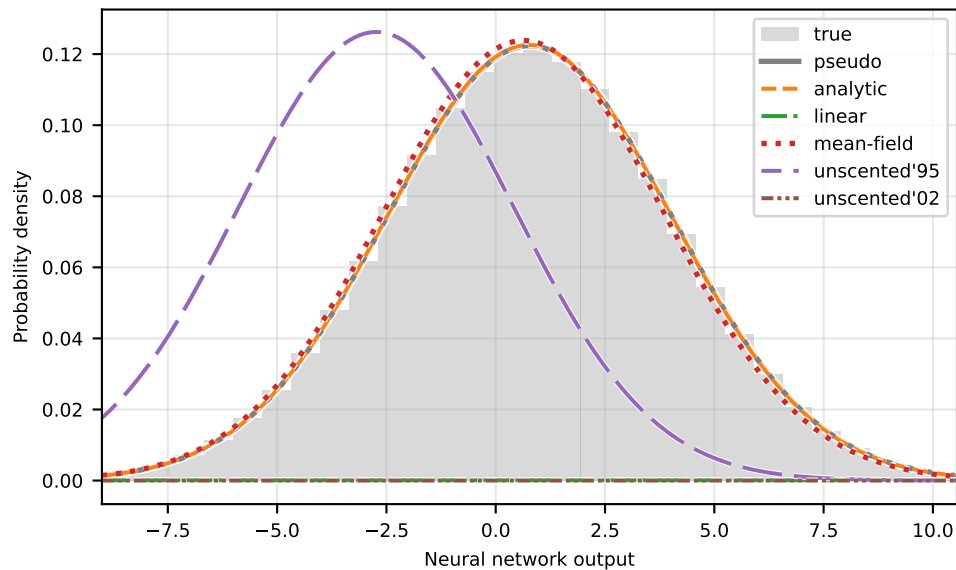

Figure 89: Probability distributions for Network(architecture=deep, weights=trained, activation=sine residual), variance=large

| distribution | $\mu$ | $\sigma^2$ |
|---|---|---|
| pseudo-true ($Y_1$) | $+2.886 \times 10^0 \pm 1.8 \times 10^{-7}$ | $1.239 \times 10^{-2} \pm 1.5 \times 10^{-7}$ |
| analytic | $+2.886 \times 10^0$ | $1.254 \times 10^{-2}$ |
| mean-field | $+2.894 \times 10^0$ | $1.642 \times 10^{-2}$ |
| linear | $+2.870 \times 10^0$ | $1.262 \times 10^{-2}$ |
| unscented'95 | $+2.886 \times 10^0$ | $1.281 \times 10^{-2}$ |
| unscented'02 | $+2.886 \times 10^0$ | $1.316 \times 10^{-2}$ |

Table 164: Comparison of moments for Network(architecture=deep, weights=initialized, activation=gelu), variance=small

| distribution | $d_{\mathrm{W}}(\cdot, Y_0)$ | $D_{\mathrm{KL}}(Y_1 \parallel \cdot)$ |
|---|---|---|
| pseudo-true ($Y_1$) | $1.205 \times 10^{-2} \pm 2.2 \times 10^{-6}$ | $0$ |
| analytic | $1.165 \times 10^{-2} \pm 2.8 \times 10^{-6}$ | $4.131 \times 10^{-5} \pm 7.0 \times 10^{-8}$ |
| mean-field | $4.870 \times 10^{-2} \pm 2.7 \times 10^{-6}$ | $2.410 \times 10^{-2} \pm 2.1 \times 10^{-6}$ |
| linear | $5.025 \times 10^{-2} \pm 4.9 \times 10^{-7}$ | $1.143 \times 10^{-2} \pm 2.5 \times 10^{-7}$ |
| unscented'95 | $1.236 \times 10^{-2} \pm 3.5 \times 10^{-6}$ | $2.818 \times 10^{-4} \pm 2.1 \times 10^{-7}$ |
| unscented'02 | $1.358 \times 10^{-2} \pm 4.0 \times 10^{-6}$ | $9.330 \times 10^{-4} \pm 3.8 \times 10^{-7}$ |

Table 165: Comparison of statistical distances for Network(architecture=deep, weights=initialized, activation=gelu), variance=small

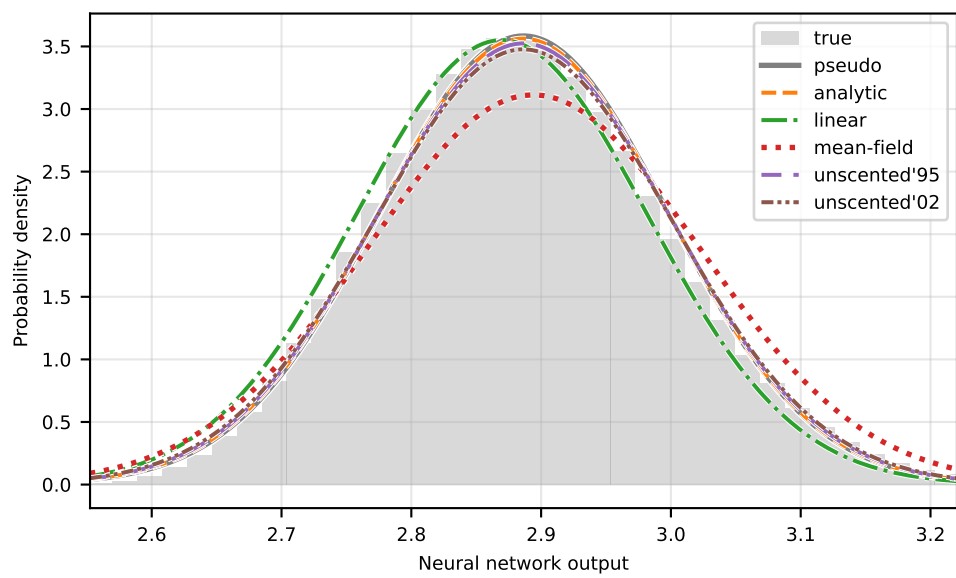

Figure 90: Probability distributions for Network(architecture=deep, weights=initialized, activation=gelu), variance=small

| distribution | $\mu$ | $\sigma^2$ |
|---|---|---|
| pseudo-true ($Y_1$) | $+2.815 \times 10^0 \pm 7.4 \times 10^{-6}$ | $2.040 \times 10^{-1} \pm 1.0 \times 10^{-5}$ |
| analytic | $+2.834 \times 10^0$ | $2.371 \times 10^{-1}$ |
| mean-field | $+2.635 \times 10^0$ | $5.798 \times 10^{-1}$ |
| linear | $+2.870 \times 10^0$ | $1.262 \times 10^0$ |
| unscented'95 | $+2.732 \times 10^0$ | $1.270 \times 10^{-1}$ |
| unscented'02 | $+4.517 \times 10^0$ | $6.690 \times 10^0$ |

Table 166: Comparison of moments for Network(architecture=deep, weights=initialized, activation=gelu), variance=medium

| distribution | $d_{\mathrm{W}}(\cdot, Y_0)$ | $D_{\mathrm{KL}}(Y_1 \parallel \cdot)$ |
|---|---|---|
| pseudo-true ($Y_1$) | $6.632 \times 10^{-2} \pm 1.8 \times 10^{-5}$ | $0$ |
| analytic | $7.115 \times 10^{-2} \pm 2.2 \times 10^{-5}$ | $6.835 \times 10^{-3} \pm 3.8 \times 10^{-6}$ |
| mean-field | $4.028 \times 10^{-1} \pm 1.4 \times 10^{-5}$ | $4.783 \times 10^{-1} \pm 5.3 \times 10^{-5}$ |
| linear | $7.860 \times 10^{-1} \pm 2.6 \times 10^{-5}$ | $1.689 \times 10^0 \pm 1.3 \times 10^{-4}$ |
| unscented'95 | $1.395 \times 10^{-1} \pm 1.8 \times 10^{-5}$ | $6.516 \times 10^{-2} \pm 7.7 \times 10^{-6}$ |
| unscented'02 | $3.322 \times 10^0 \pm 5.4 \times 10^{-5}$ | $2.125 \times 10^{+1} \pm 1.1 \times 10^{-3}$ |

Table 167: Comparison of statistical distances for Network(architecture=deep, weights=initialized, activation=gelu), variance=medium

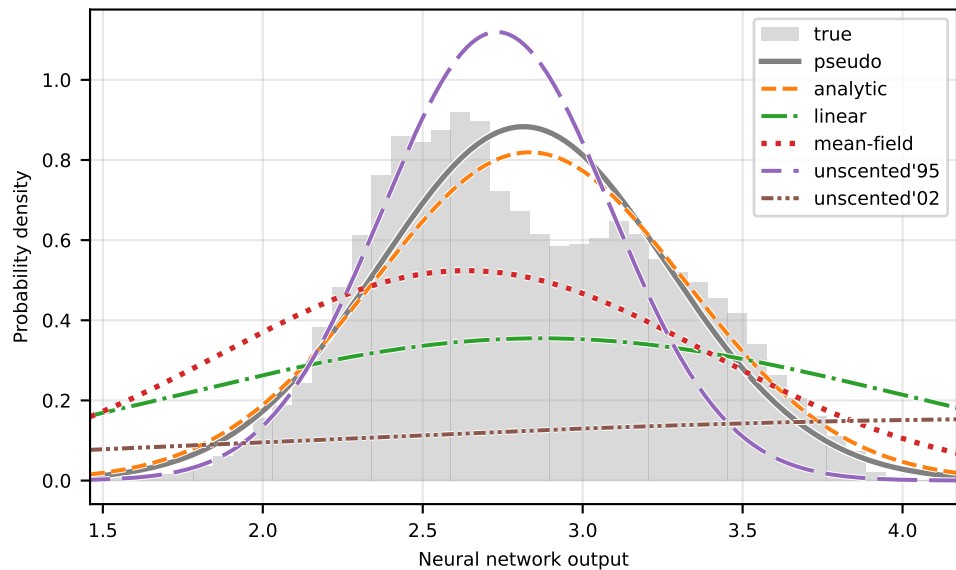

Figure 91: Probability distributions for Network(architecture=deep, weights=initialized, activation=gelu), variance=medium

| distribution | $\mu$ | $\sigma^2$ |
|---|---|---|
| pseudo-true $(Y_1)$ | $+7.526 \times 10^0 \pm 1.1 \times 10^{-4}$ | $1.930 \times 10^{+1} \pm 2.9 \times 10^{-3}$ |
| analytic | $+8.032 \times 10^0$ | $1.048 \times 10^{+1}$ |
| mean-field | $+7.986 \times 10^0$ | $1.344 \times 10^{+1}$ |
| linear | $+2.870 \times 10^0$ | $1.262 \times 10^{+2}$ |
| unscented'95 | $+8.265 \times 10^0$ | $1.355 \times 10^{+1}$ |
| unscented'02 | $+1.676 \times 10^{+2}$ | $5.441 \times 10^{+4}$ |

Table 168: Comparison of moments for Network(architecture=deep, weights=initialized, activation=gelu), variance=large

| distribution | $d_{\mathrm{W}}(\cdot, Y_0)$ | $D_{\mathrm{KL}}(Y_1 \parallel \cdot)$ |
|---|---|---|
| pseudo-true $(Y_1)$ | $4.622 \times 10^{-1} \pm 8.9 \times 10^{-5}$ | $0$ |
| analytic | $6.447 \times 10^{-1} \pm 7.4 \times 10^{-5}$ | $8.356 \times 10^{-2} \pm 3.2 \times 10^{-5}$ |
| mean-field | $5.737 \times 10^{-1} \pm 7.7 \times 10^{-5}$ | $3.468 \times 10^{-2} \pm 2.1 \times 10^{-5}$ |
| linear | $3.070 \times 10^0 \pm 1.7 \times 10^{-4}$ | $2.392 \times 10^0 \pm 4.9 \times 10^{-4}$ |
| unscented'95 | $6.487 \times 10^{-1} \pm 6.9 \times 10^{-5}$ | $4.204 \times 10^{-2} \pm 1.9 \times 10^{-5}$ |
| unscented'02 | $1.079 \times 10^{+2} \pm 4.1 \times 10^{-3}$ | $2.069 \times 10^{+3} \pm 3.1 \times 10^{-1}$ |

Table 169: Comparison of statistical distances for Network(architecture=deep, weights=initialized, activation=gelu), variance=large

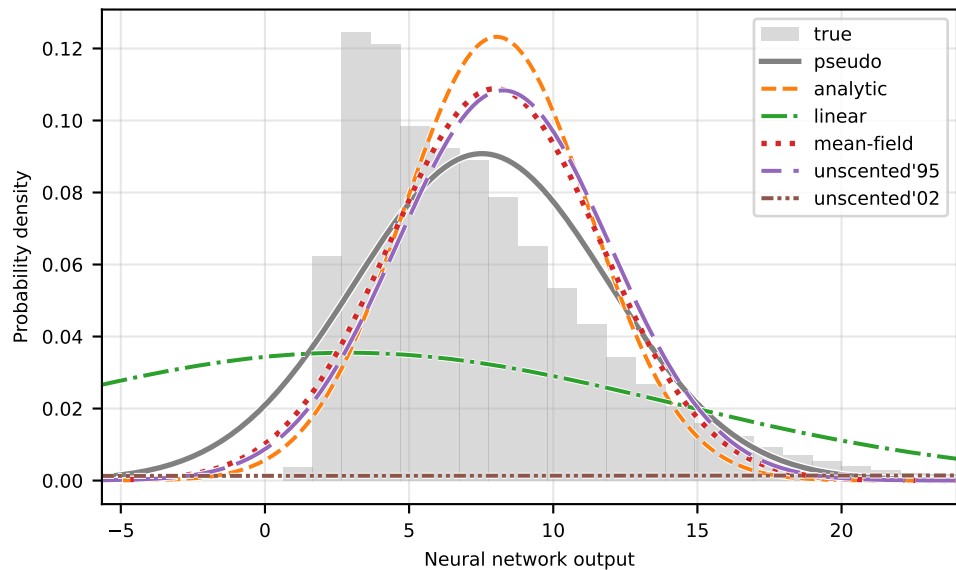

Figure 92: Probability distributions for Network(architecture=deep, weights=initialized, activation=gelu), variance=large

| distribution | $\mu$ | $\sigma^2$ |
|---|---|---|
| pseudo-true ($Y_1$) | $-5.675 \times 10^{-1} \pm 3.1 \times 10^{-7}$ | $6.305 \times 10^{-2} \pm 5.0 \times 10^{-7}$ |
| analytic | $-5.702 \times 10^{-1}$ | $6.299 \times 10^{-2}$ |
| mean-field | $-5.180 \times 10^{-1}$ | $2.304 \times 10^{-2}$ |
| linear | $-5.470 \times 10^{-1}$ | $8.199 \times 10^{-2}$ |
| unscented'95 | $-5.727 \times 10^{-1}$ | $6.783 \times 10^{-2}$ |
| unscented'02 | $-5.755 \times 10^{-1}$ | $8.361 \times 10^{-2}$ |

Table 170: Comparison of moments for Network(architecture=deep, weights=trained, activation=gelu), variance=small

| distribution | $d_{\mathrm{W}}(\cdot, Y_0)$ | $D_{\mathrm{KL}}(Y_1 \parallel \cdot)$ |
|---|---|---|
| pseudo-true ($Y_1$) | $4.070 \times 10^{-2} \pm 4.1 \times 10^{-6}$ | $0$ |
| analytic | $4.193 \times 10^{-2} \pm 4.1 \times 10^{-6}$ | $5.721 \times 10^{-5} \pm 1.4 \times 10^{-8}$ |
| mean-field | $1.656 \times 10^{-1} \pm 2.1 \times 10^{-6}$ | $2.056 \times 10^{-1} \pm 2.3 \times 10^{-6}$ |
| linear | $5.535 \times 10^{-2} \pm 7.2 \times 10^{-6}$ | $2.221 \times 10^{-2} \pm 1.2 \times 10^{-6}$ |
| unscented'95 | $4.372 \times 10^{-2} \pm 5.3 \times 10^{-6}$ | $1.577 \times 10^{-3} \pm 3.0 \times 10^{-7}$ |
| unscented'02 | $7.483 \times 10^{-2} \pm 2.8 \times 10^{-6}$ | $2.243 \times 10^{-2} \pm 1.3 \times 10^{-6}$ |

Table 171: Comparison of statistical distances for Network(architecture=deep, weights=trained, activation=gelu), variance=small

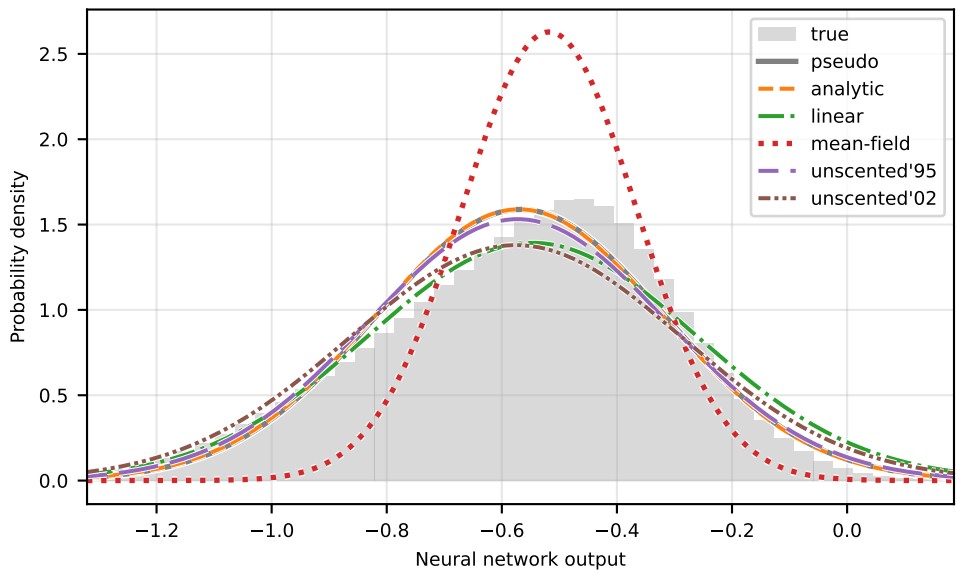

Figure 93: Probability distributions for Network(architecture=deep, weights=trained, activation=gelu), variance=small

| distribution | $\mu$ | $\sigma^2$ |
|---|---|---|
| pseudo-true ($Y_1$) | $-2.262 \times 10^{-1} \pm 1.0 \times 10^{-5}$ | $3.663 \times 10^{-1} \pm 1.7 \times 10^{-5}$ |
| analytic | $-7.138 \times 10^{-2}$ | $3.740 \times 10^{-1}$ |
| mean-field | $+7.814 \times 10^{-3}$ | $7.079 \times 10^{-1}$ |
| linear | $-5.470 \times 10^{-1}$ | $8.199 \times 10^{0}$ |
| unscented'95 | $-3.967 \times 10^{-1}$ | $5.968 \times 10^{-1}$ |
| unscented'02 | $-3.399 \times 10^{0}$ | $2.446 \times 10^{+1}$ |

Table 172: Comparison of moments for Network(architecture=deep, weights=trained, activation=gelu), variance=medium

| distribution | $d_{\mathrm{W}}(\cdot, Y_0)$ | $D_{\mathrm{KL}}(Y_1 \parallel \cdot)$ |
|---|---|---|
| pseudo-true ($Y_1$) | $6.726 \times 10^{-2} \pm 2.0 \times 10^{-5}$ | $0$ |
| analytic | $2.096 \times 10^{-1} \pm 1.5 \times 10^{-5}$ | $3.284 \times 10^{-2} \pm 6.0 \times 10^{-6}$ |
| mean-field | $3.801 \times 10^{-1} \pm 2.1 \times 10^{-5}$ | $2.117 \times 10^{-1} \pm 3.1 \times 10^{-5}$ |
| linear | $2.320 \times 10^{0} \pm 4.2 \times 10^{-5}$ | $9.278 \times 10^{0} \pm 5.0 \times 10^{-4}$ |
| unscented'95 | $2.379 \times 10^{-1} \pm 1.5 \times 10^{-5}$ | $1.103 \times 10^{-1} \pm 1.4 \times 10^{-5}$ |
| unscented'02 | $5.556 \times 10^{0} \pm 7.7 \times 10^{-5}$ | $4.454 \times 10^{+1} \pm 2.1 \times 10^{-3}$ |

Table 173: Comparison of statistical distances for Network(architecture=deep, weights=trained, activation=gelu), variance=medium

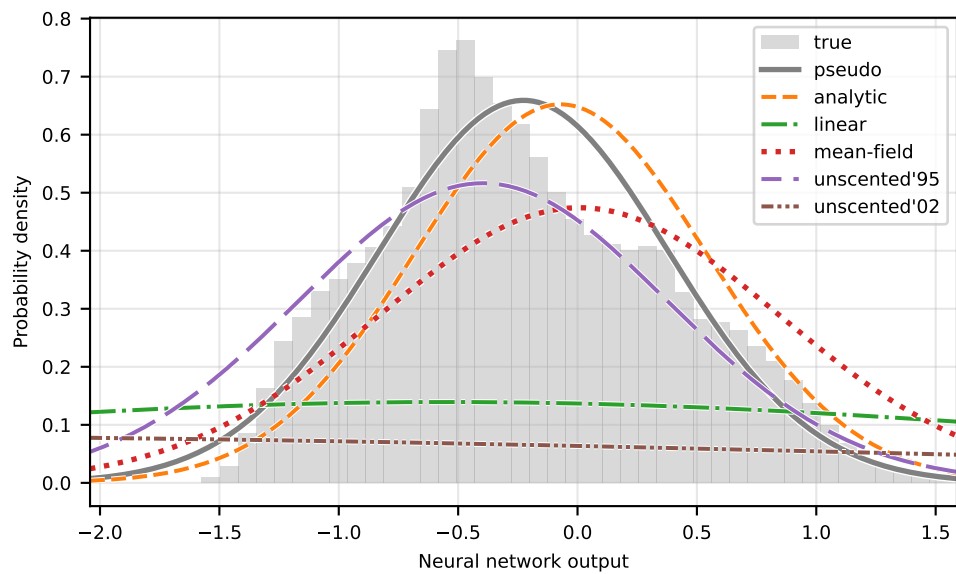

Figure 94: Probability distributions for Network(architecture=deep, weights=trained, activation=gelu), variance=medium

| distribution | $\mu$ | $\sigma^2$ |
|---|---|---|
| pseudo-true $(Y_1)$ | $+3.776 \times 10^0 \pm 1.7 \times 10^{-4}$ | $1.469 \times 10^{+1} \pm 2.2 \times 10^{-3}$ |
| analytic | $+4.338 \times 10^0$ | $8.886 \times 10^0$ |
| mean-field | $+4.591 \times 10^0$ | $1.377 \times 10^{+1}$ |
| linear | $-5.470 \times 10^{-1}$ | $8.199 \times 10^{+2}$ |
| unscented'95 | $+3.805 \times 10^0$ | $1.557 \times 10^{+1}$ |
| unscented'02 | $-2.854 \times 10^{+2}$ | $1.632 \times 10^{+5}$ |

Table 174: Comparison of moments for Network(architecture=deep, weights=trained, activation=gelu), variance=large

| distribution | $d_{\mathrm{W}}(\cdot, Y_0)$ | $D_{\mathrm{KL}}(Y_1 \parallel \cdot)$ |
|---|---|---|
| pseudo-true $(Y_1)$ | $3.440 \times 10^{-1} \pm 9.7 \times 10^{-5}$ | $0$ |
| analytic | $5.409 \times 10^{-1} \pm 8.8 \times 10^{-5}$ | $6.445 \times 10^{-2} \pm 2.5 \times 10^{-5}$ |
| mean-field | $5.635 \times 10^{-1} \pm 4.6 \times 10^{-5}$ | $2.363 \times 10^{-2} \pm 8.6 \times 10^{-6}$ |
| linear | $1.029 \times 10^{+1} \pm 4.6 \times 10^{-4}$ | $2.604 \times 10^{+1} \pm 4.2 \times 10^{-3}$ |
| unscented'95 | $3.634 \times 10^{-1} \pm 7.7 \times 10^{-5}$ | $9.051 \times 10^{-4} \pm 4.8 \times 10^{-6}$ |
| unscented'02 | $2.038 \times 10^{+2} \pm 7.8 \times 10^{-3}$ | $8.398 \times 10^{+3} \pm 1.3 \times 10^0$ |

Table 175: Comparison of statistical distances for Network(architecture=deep, weights=trained, activation=gelu), variance=large

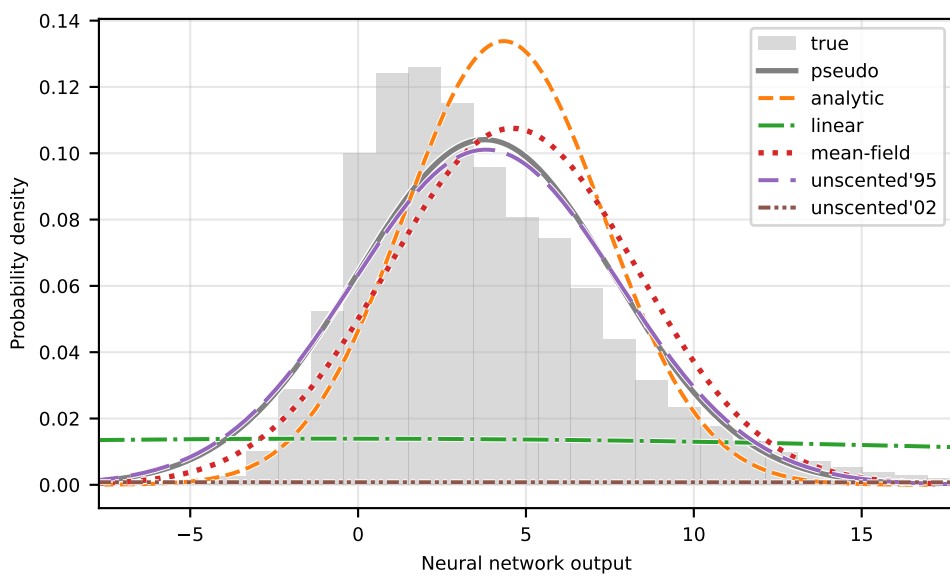

Figure 95: Probability distributions for Network(architecture=deep, weights=trained, activation=gelu), variance=large

| distribution | $\mu$ | $\sigma^2$ |
|---|---|---|
| pseudo-true ($Y_1$) | $-7.048 \times 10^{+1} \pm 2.5 \times 10^{-4}$ | $7.272 \times 10^{+3} \pm 9.5 \times 10^{-2}$ |
| analytic | $-7.235 \times 10^{+1}$ | $7.475 \times 10^{+3}$ |
| mean-field | $-6.834 \times 10^{+1}$ | $9.767 \times 10^{+3}$ |
| linear | $-9.331 \times 10^{+1}$ | $1.011 \times 10^{+4}$ |
| unscented'95 | $-6.667 \times 10^{+1}$ | $6.940 \times 10^{+3}$ |
| unscented'02 | $-4.291 \times 10^{+1}$ | $1.518 \times 10^{+4}$ |

Table 176: Comparison of moments for Network(architecture=deep, weights=initialized, activation=gelu residual), variance=small

| distribution | $d_{\mathrm{W}}(\cdot, Y_0)$ | $D_{\mathrm{KL}}(Y_1 \parallel \cdot)$ |
|---|---|---|
| pseudo-true ($Y_1$) | $6.311 \times 10^{-1} \pm 7.9 \times 10^{-5}$ | $0$ |
| analytic | $5.682 \times 10^{-1} \pm 5.8 \times 10^{-5}$ | $4.297 \times 10^{-4} \pm 1.9 \times 10^{-7}$ |
| mean-field | $1.403 \times 10^{0} \pm 6.2 \times 10^{-5}$ | $2.437 \times 10^{-2} \pm 2.3 \times 10^{-6}$ |
| linear | $2.471 \times 10^{0} \pm 2.7 \times 10^{-5}$ | $6.610 \times 10^{-2} \pm 3.0 \times 10^{-6}$ |
| unscented'95 | $8.604 \times 10^{-1} \pm 7.9 \times 10^{-5}$ | $1.541 \times 10^{-3} \pm 3.0 \times 10^{-7}$ |
| unscented'02 | $4.445 \times 10^{0} \pm 5.8 \times 10^{-5}$ | $2.282 \times 10^{-1} \pm 8.0 \times 10^{-6}$ |

Table 177: Comparison of statistical distances for Network(architecture=deep, weights=initialized, activation=gelu residual), variance=small

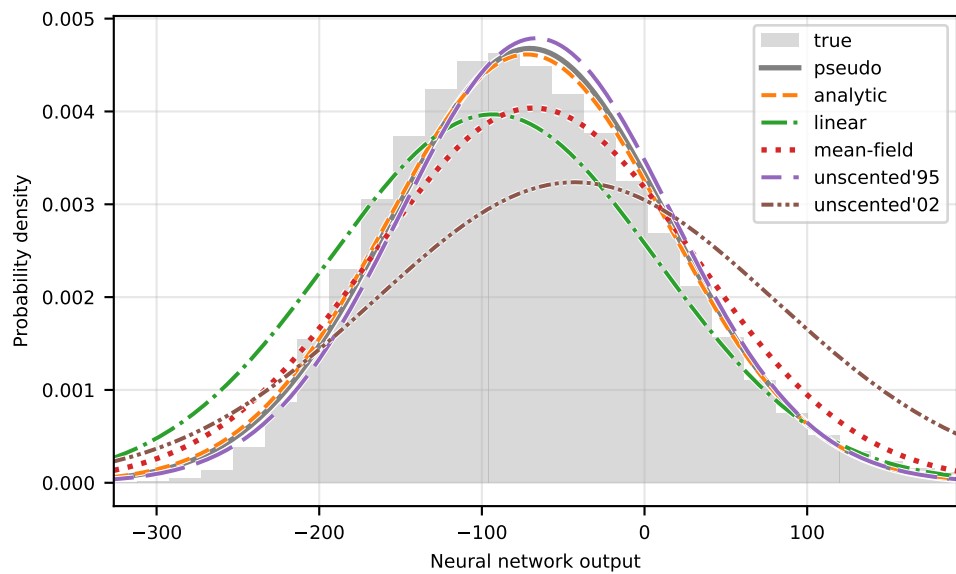

Figure 96: Probability distributions for Network(architecture=deep, weights=initialized, activation=gelu residual), variance=small

| distribution | $\mu$ | $\sigma^2$ |
|---|---|---|
| pseudo-true ($Y_1$) | $+3.447 \times 10^{+1} \pm 8.2 \times 10^{-3}$ | $3.583 \times 10^{+5} \pm 1.4 \times 10^{+1}$ |
| analytic | $+8.599 \times 10^{+1}$ | $4.080 \times 10^{+5}$ |
| mean-field | $+1.463 \times 10^{+2}$ | $5.186 \times 10^{+5}$ |
| linear | $-9.331 \times 10^{+1}$ | $1.011 \times 10^{+6}$ |
| unscented'95 | $+1.674 \times 10^{+2}$ | $3.425 \times 10^{+5}$ |
| unscented'02 | $+4.944 \times 10^{+3}$ | $5.175 \times 10^{+7}$ |

Table 178: Comparison of moments for Network(architecture=deep, weights=initialized, activation=gelu residual), variance=medium

| distribution | $d_{\mathrm{W}}(\cdot, Y_0)$ | $D_{\mathrm{KL}}(Y_1 \parallel \cdot)$ |
|---|---|---|
| pseudo-true ($Y_1$) | $1.827 \times 10^0 \pm 3.2 \times 10^{-4}$ | $0$ |
| analytic | $3.593 \times 10^0 \pm 3.9 \times 10^{-4}$ | $8.122 \times 10^{-3} \pm 3.0 \times 10^{-6}$ |
| mean-field | $6.621 \times 10^0 \pm 4.0 \times 10^{-4}$ | $5.629 \times 10^{-2} \pm 9.6 \times 10^{-6}$ |
| linear | $1.377 \times 10^{+1} \pm 5.0 \times 10^{-4}$ | $4.146 \times 10^{-1} \pm 3.6 \times 10^{-5}$ |
| unscented'95 | $5.953 \times 10^0 \pm 4.4 \times 10^{-4}$ | $2.517 \times 10^{-2} \pm 3.1 \times 10^{-6}$ |
| unscented'02 | $2.730 \times 10^{+2} \pm 2.7 \times 10^{-3}$ | $1.029 \times 10^{+2} \pm 4.1 \times 10^{-3}$ |

Table 179: Comparison of statistical distances for Network(architecture=deep, weights=initialized, activation=gelu residual), variance=medium

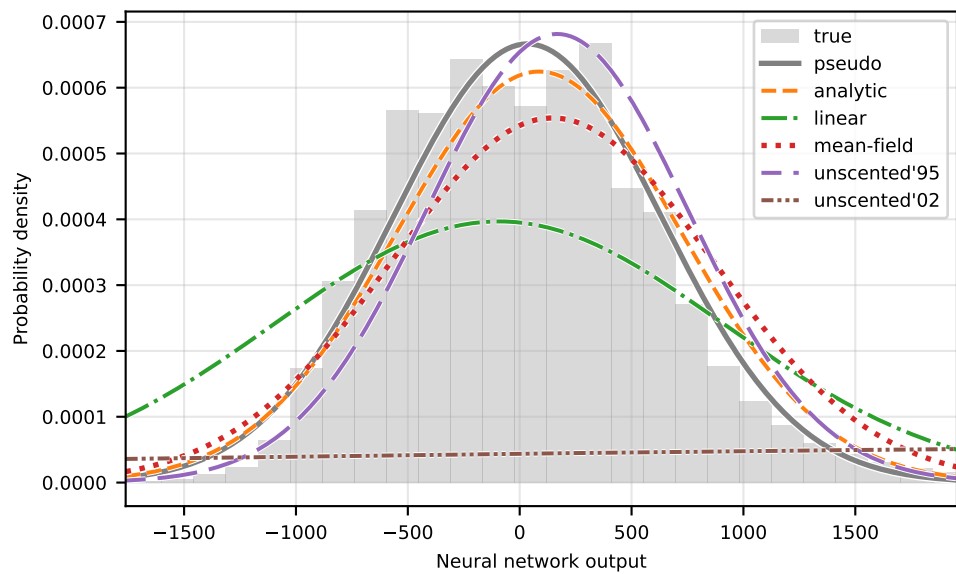

Figure 97: Probability distributions for Network(architecture=deep, weights=initialized, activation=gelu residual), variance=medium

| distribution | $\mu$ | $\sigma^2$ |
|---|---|---|
| pseudo-true ($Y_1$) | $+5.290 \times 10^{+3} \pm 8.3 \times 10^{-2}$ | $4.372 \times 10^{+7} \pm 2.1 \times 10^{+3}$ |
| analytic | $+5.838 \times 10^{+3}$ | $3.727 \times 10^{+7}$ |
| mean-field | $+4.691 \times 10^{+3}$ | $3.766 \times 10^{+7}$ |
| linear | $-9.331 \times 10^{+1}$ | $1.011 \times 10^{+8}$ |
| unscented'95 | $+7.423 \times 10^{+3}$ | $1.861 \times 10^{+7}$ |
| unscented'02 | $+4.778 \times 10^{+5}$ | $4.569 \times 10^{+11}$ |

Table 180: Comparison of moments for Network(architecture=deep, weights=initialized, activation=gelu residual), variance=large

| distribution | $d_{\mathrm{W}}(\cdot, Y_0)$ | $D_{\mathrm{KL}}(Y_1 \parallel \cdot)$ |
|---|---|---|
| pseudo-true ($Y_1$) | $1.207 \times 10^{+1} \pm 1.7 \times 10^{-3}$ | $0$ |
| analytic | $1.529 \times 10^{+1} \pm 1.5 \times 10^{-3}$ | $9.480 \times 10^{-3} \pm 3.3 \times 10^{-6}$ |
| mean-field | $1.129 \times 10^{+1} \pm 1.6 \times 10^{-3}$ | $9.404 \times 10^{-3} \pm 3.5 \times 10^{-6}$ |
| linear | $6.621 \times 10^{+1} \pm 1.2 \times 10^{-3}$ | $5.681 \times 10^{-1} \pm 4.6 \times 10^{-5}$ |
| unscented'95 | $3.666 \times 10^{+1} \pm 1.1 \times 10^{-3}$ | $1.919 \times 10^{-1} \pm 1.1 \times 10^{-5}$ |
| unscented'02 | $8.146 \times 10^{+3} \pm 9.8 \times 10^{-2}$ | $7.773 \times 10^{+3} \pm 3.7 \times 10^{-1}$ |

Table 181: Comparison of statistical distances for Network(architecture=deep, weights=initialized, activation=gelu residual), variance=large

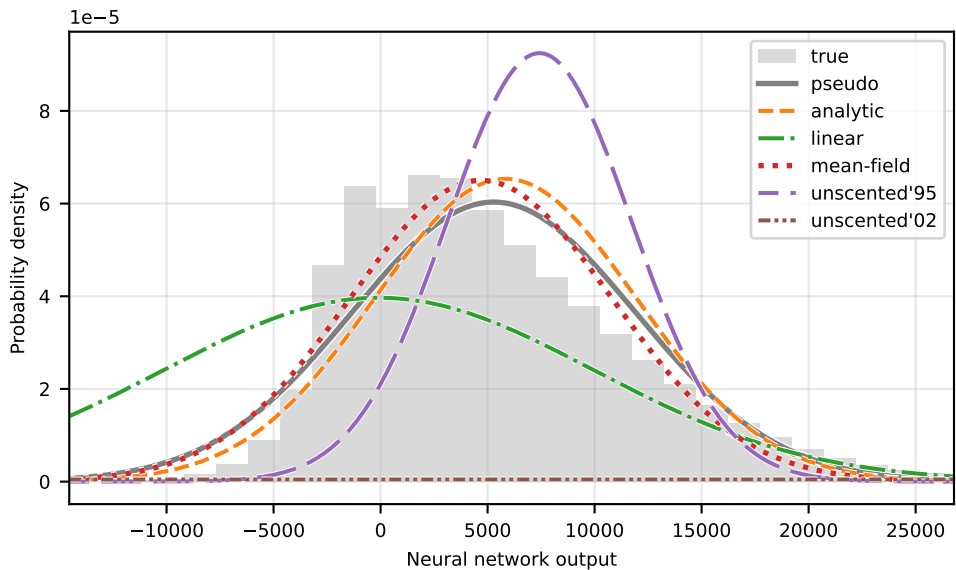

Figure 98: Probability distributions for Network(architecture=deep, weights=initialized, activation=gelu residual), variance=large

| distribution | $\mu$ | $\sigma^2$ |
|---|---|---|
| pseudo-true ($Y_1$) | $+5.117 \times 10^{+1} \pm 1.9 \times 10^{-4}$ | $1.663 \times 10^{+3} \pm 6.1 \times 10^{-2}$ |
| analytic | $+4.852 \times 10^{+1}$ | $1.777 \times 10^{+3}$ |
| mean-field | $+3.612 \times 10^{+1}$ | $9.507 \times 10^{+3}$ |
| linear | $+4.609 \times 10^{+1}$ | $6.393 \times 10^{+2}$ |
| unscented'95 | $+5.265 \times 10^{+1}$ | $1.452 \times 10^{+3}$ |
| unscented'02 | $+7.208 \times 10^{+1}$ | $1.990 \times 10^{+3}$ |

Table 182: Comparison of moments for Network(architecture=deep, weights=trained, activation=gelu residual), variance=small

| distribution | $d_{\mathrm{W}}(\cdot, Y_0)$ | $D_{\mathrm{KL}}(Y_1 \parallel \cdot)$ |
|---|---|---|
| pseudo-true ($Y_1$) | $7.140 \times 10^{-1} \pm 1.0 \times 10^{-4}$ | $0$ |
| analytic | $8.277 \times 10^{-1} \pm 1.2 \times 10^{-4}$ | $3.241 \times 10^{-3} \pm 1.3 \times 10^{-6}$ |
| mean-field | $7.635 \times 10^{0} \pm 1.3 \times 10^{-4}$ | $1.554 \times 10^{0} \pm 8.9 \times 10^{-5}$ |
| linear | $1.621 \times 10^{0} \pm 6.2 \times 10^{-5}$ | $1.780 \times 10^{-1} \pm 1.1 \times 10^{-5}$ |
| unscented'95 | $6.240 \times 10^{-1} \pm 8.5 \times 10^{-5}$ | $5.081 \times 10^{-3} \pm 2.3 \times 10^{-6}$ |
| unscented'02 | $3.337 \times 10^{0} \pm 7.9 \times 10^{-5}$ | $1.400 \times 10^{-1} \pm 9.3 \times 10^{-6}$ |

Table 183: Comparison of statistical distances for Network(architecture=deep, weights=trained, activation=gelu residual), variance=small

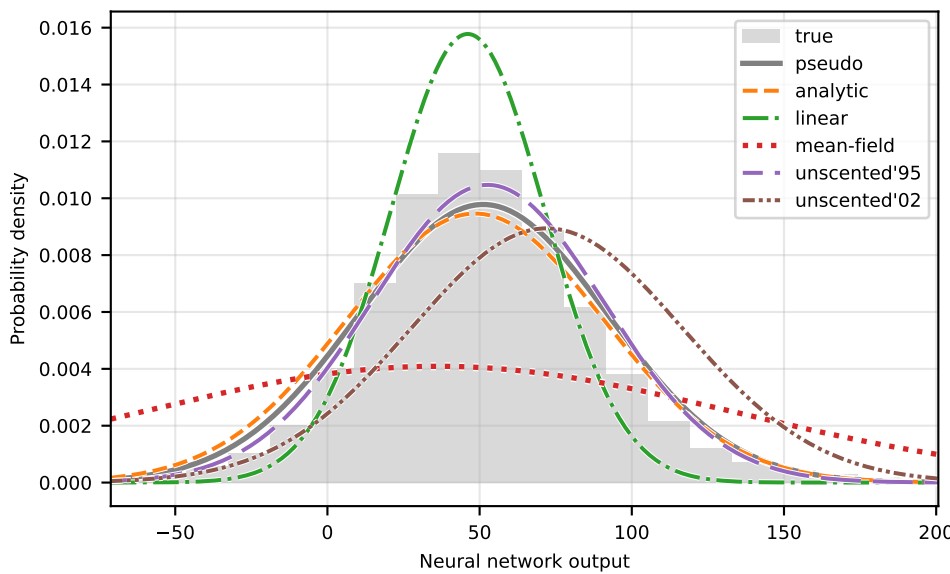

Figure 99: Probability distributions for Network(architecture=deep, weights=trained, activation=gelu residual), variance=small

| distribution | $\mu$ | $\sigma^2$ |
|---|---|---|
| pseudo-true $(Y_1)$ | $-1.195 \times 10^{+2} \pm 5.4 \times 10^{-3}$ | $1.574 \times 10^{+5} \pm 1.4 \times 10^{+1}$ |
| analytic | $-1.198 \times 10^{+2}$ | $2.787 \times 10^{+5}$ |
| mean-field | $-7.058 \times 10^{+1}$ | $5.193 \times 10^{+5}$ |
| linear | $+4.609 \times 10^{+1}$ | $6.393 \times 10^{+4}$ |
| unscented'95 | $-3.022 \times 10^{+1}$ | $1.027 \times 10^{+5}$ |
| unscented'02 | $+2.644 \times 10^{+3}$ | $1.357 \times 10^{+7}$ |

Table 184: Comparison of moments for Network(architecture=deep, weights=trained, activation=gelu residual), variance=medium

| distribution | $d_{\mathrm{W}}(\cdot, Y_0)$ | $D_{\mathrm{KL}}(Y_1 \parallel \cdot)$ |
|---|---|---|
| pseudo-true $(Y_1)$ | $2.730 \times 10^0 \pm 5.6 \times 10^{-4}$ | $0$ |
| analytic | $6.343 \times 10^0 \pm 4.1 \times 10^{-4}$ | $9.965 \times 10^{-2} \pm 3.3 \times 10^{-5}$ |
| mean-field | $1.362 \times 10^{+1} \pm 6.8 \times 10^{-4}$ | $5.604 \times 10^{-1} \pm 9.9 \times 10^{-5}$ |
| linear | $8.801 \times 10^0 \pm 4.3 \times 10^{-4}$ | $2.407 \times 10^{-1} \pm 2.0 \times 10^{-5}$ |
| unscented'95 | $4.771 \times 10^0 \pm 3.7 \times 10^{-4}$ | $6.496 \times 10^{-2} \pm 1.4 \times 10^{-5}$ |
| unscented'02 | $1.746 \times 10^{+2} \pm 3.9 \times 10^{-3}$ | $6.463 \times 10^{+1} \pm 5.7 \times 10^{-3}$ |

Table 185: Comparison of statistical distances for Network(architecture=deep, weights=trained, activation=gelu residual), variance=medium

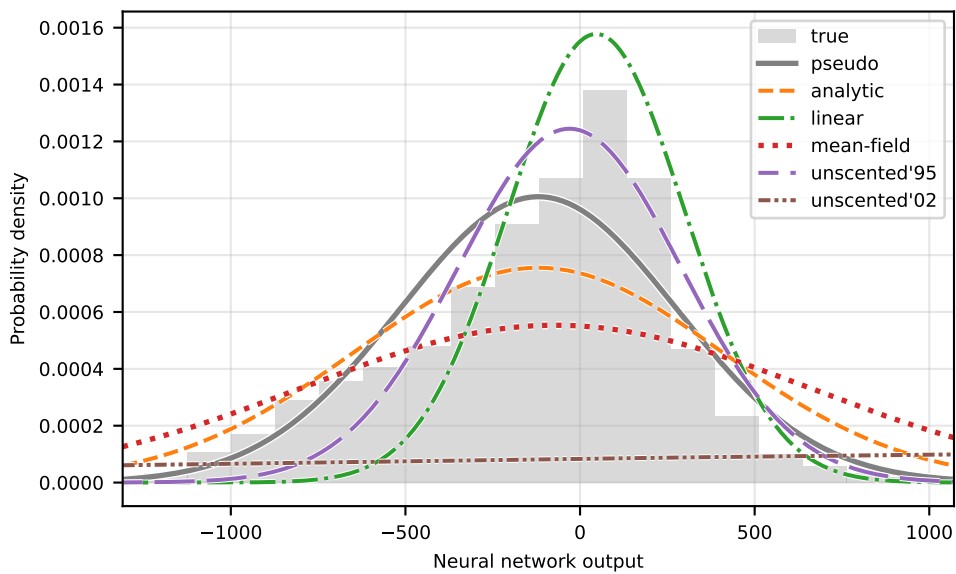

Figure 100: Probability distributions for Network(architecture=deep, weights=trained, activation=gelu residual), variance=medium

| distribution | $\mu$ | $\sigma^2$ |
|---|---|---|
| pseudo-true ($Y_1$) | $+2.667 \times 10^{+3} \pm 7.4 \times 10^{-2}$ | $3.189 \times 10^{+7} \pm 1.7 \times 10^{+3}$ |
| analytic | $+2.912 \times 10^{+3}$ | $3.073 \times 10^{+7}$ |
| mean-field | $+1.762 \times 10^{+3}$ | $3.778 \times 10^{+7}$ |
| linear | $+4.609 \times 10^{+1}$ | $6.393 \times 10^{+6}$ |
| unscented'95 | $+4.419 \times 10^{+3}$ | $1.409 \times 10^{+7}$ |
| unscented'02 | $+2.537 \times 10^{+5}$ | $1.287 \times 10^{+11}$ |

Table 186: Comparison of moments for Network(architecture=deep, weights=trained, activation=gelu residual), variance=large

| distribution | $d_{\mathrm{W}}(\cdot, Y_0)$ | $D_{\mathrm{KL}}(Y_1 \parallel \cdot)$ |
|---|---|---|
| pseudo-true ($Y_1$) | $6.428 \times 10^0 \pm 1.3 \times 10^{-3}$ | $0$ |
| analytic | $7.260 \times 10^0 \pm 1.4 \times 10^{-3}$ | $1.279 \times 10^{-3} \pm 1.0 \times 10^{-6}$ |
| mean-field | $1.205 \times 10^{+1} \pm 9.7 \times 10^{-4}$ | $2.044 \times 10^{-2} \pm 5.8 \times 10^{-6}$ |
| linear | $4.083 \times 10^{+1} \pm 8.4 \times 10^{-4}$ | $5.115 \times 10^{-1} \pm 1.7 \times 10^{-5}$ |
| unscented'95 | $3.094 \times 10^{+1} \pm 8.9 \times 10^{-4}$ | $1.774 \times 10^{-1} \pm 1.3 \times 10^{-5}$ |
| unscented'02 | $4.662 \times 10^{+3} \pm 6.4 \times 10^{-2}$ | $3.001 \times 10^{+3} \pm 1.6 \times 10^{-1}$ |

Table 187: Comparison of statistical distances for Network(architecture=deep, weights=trained, activation=gelu residual), variance=large

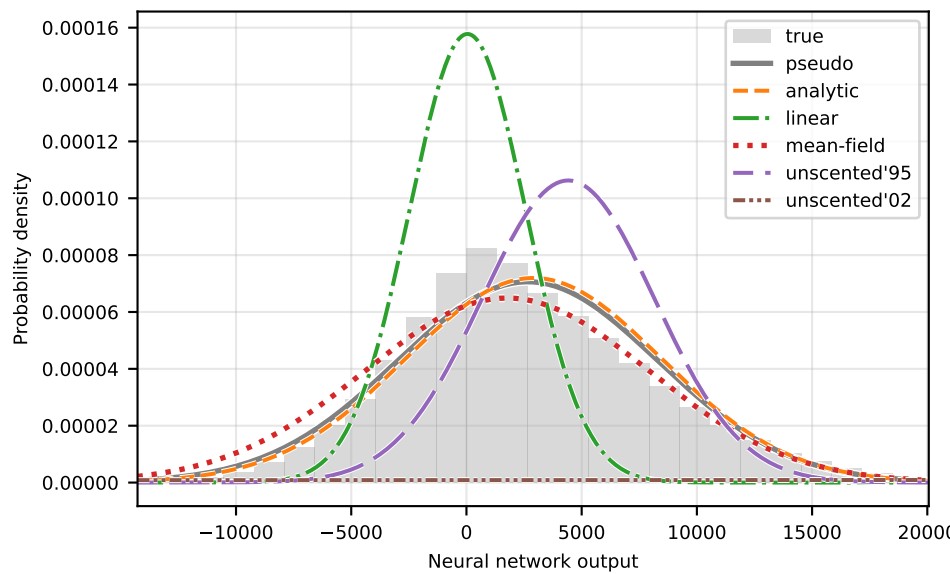

Figure 101: Probability distributions for Network(architecture=deep, weights=trained, activation=gelu residual), variance=large

| distribution | $\mu$ | $\sigma^2$ |
|---|---|---|
| pseudo-true $(Y_1)$ | $+2.616 \times 10^0 \pm 5.0 \times 10^{-7}$ | $5.245 \times 10^{-3} \pm 1.4 \times 10^{-7}$ |
| analytic | $+2.615 \times 10^0$ | $5.611 \times 10^{-3}$ |
| mean-field | $+2.634 \times 10^0$ | $1.284 \times 10^{-2}$ |
| linear | $+2.611 \times 10^0$ | $1.690 \times 10^{-2}$ |
| unscented'95 | $+2.603 \times 10^0$ | $5.793 \times 10^{-3}$ |
| unscented'02 | $+2.611 \times 10^0$ | $1.690 \times 10^{-2}$ |

Table 188: Comparison of moments for Network(architecture=deep, weights=initialized, activation=relu), variance=small

| distribution | $d_{\mathrm{W}}(\cdot, Y_0)$ | $D_{\mathrm{KL}}(Y_1 \parallel \cdot)$ |
|---|---|---|
| pseudo-true $(Y_1)$ | $2.790 \times 10^{-2} \pm 3.4 \times 10^{-6}$ | $0$ |
| analytic | $2.852 \times 10^{-2} \pm 4.4 \times 10^{-6}$ | $1.186 \times 10^{-3} \pm 8.9 \times 10^{-7}$ |
| mean-field | $1.245 \times 10^{-1} \pm 2.6 \times 10^{-6}$ | $3.103 \times 10^{-1} \pm 2.0 \times 10^{-5}$ |
| linear | $1.724 \times 10^{-1} \pm 3.3 \times 10^{-6}$ | $5.280 \times 10^{-1} \pm 2.9 \times 10^{-5}$ |
| unscented'95 | $4.575 \times 10^{-2} \pm 3.0 \times 10^{-6}$ | $1.660 \times 10^{-2} \pm 1.8 \times 10^{-6}$ |
| unscented'02 | $1.724 \times 10^{-1} \pm 3.3 \times 10^{-6}$ | $5.280 \times 10^{-1} \pm 2.9 \times 10^{-5}$ |

Table 189: Comparison of statistical distances for Network(architecture=deep, weights=initialized, activation=relu), variance=small

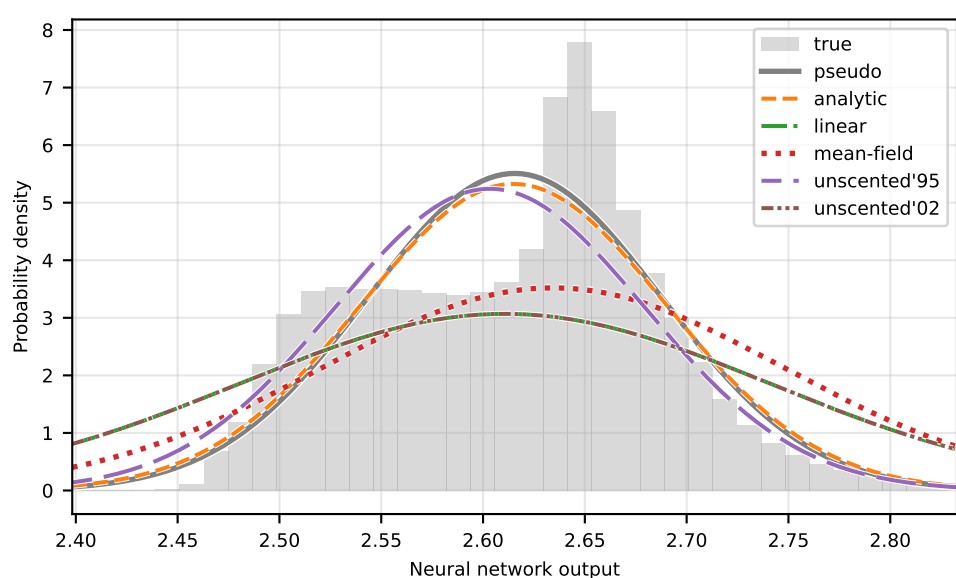

Figure 102: Probability distributions for Network(architecture=deep, weights=initialized, activation=relu), variance=small

| distribution | $\mu$ | $\sigma^2$ |
|---|---|---|
| pseudo-true ($Y_1$) | $+2.429 \times 10^0 \pm 6.6 \times 10^{-6}$ | $7.734 \times 10^{-2} \pm 6.4 \times 10^{-6}$ |
| analytic | $+2.411 \times 10^0$ | $7.888 \times 10^{-2}$ |
| mean-field | $+2.392 \times 10^0$ | $3.227 \times 10^{-1}$ |
| linear | $+2.611 \times 10^0$ | $1.690 \times 10^0$ |
| unscented'95 | $+2.355 \times 10^0$ | $3.979 \times 10^{-2}$ |
| unscented'02 | $+2.611 \times 10^0$ | $1.690 \times 10^0$ |

Table 190: Comparison of moments for Network(architecture=deep, weights=initialized, activation=relu), variance=medium

| distribution | $d_{\mathrm{W}}(\cdot, Y_0)$ | $D_{\mathrm{KL}}(Y_1 \parallel \cdot)$ |
|---|---|---|
| pseudo-true ($Y_1$) | $6.887 \times 10^{-2} \pm 1.6 \times 10^{-5}$ | $0$ |
| analytic | $6.803 \times 10^{-2} \pm 1.4 \times 10^{-5}$ | $2.270 \times 10^{-3} \pm 1.9 \times 10^{-6}$ |
| mean-field | $4.132 \times 10^{-1} \pm 2.6 \times 10^{-5}$ | $8.809 \times 10^{-1} \pm 1.3 \times 10^{-4}$ |
| linear | $1.555 \times 10^0 \pm 4.6 \times 10^{-5}$ | $9.094 \times 10^0 \pm 8.8 \times 10^{-4}$ |
| unscented'95 | $1.607 \times 10^{-1} \pm 1.9 \times 10^{-5}$ | $1.250 \times 10^{-1} \pm 1.8 \times 10^{-5}$ |
| unscented'02 | $1.555 \times 10^0 \pm 4.6 \times 10^{-5}$ | $9.094 \times 10^0 \pm 8.8 \times 10^{-4}$ |

Table 191: Comparison of statistical distances for Network(architecture=deep, weights=initialized, activation=relu), variance=medium

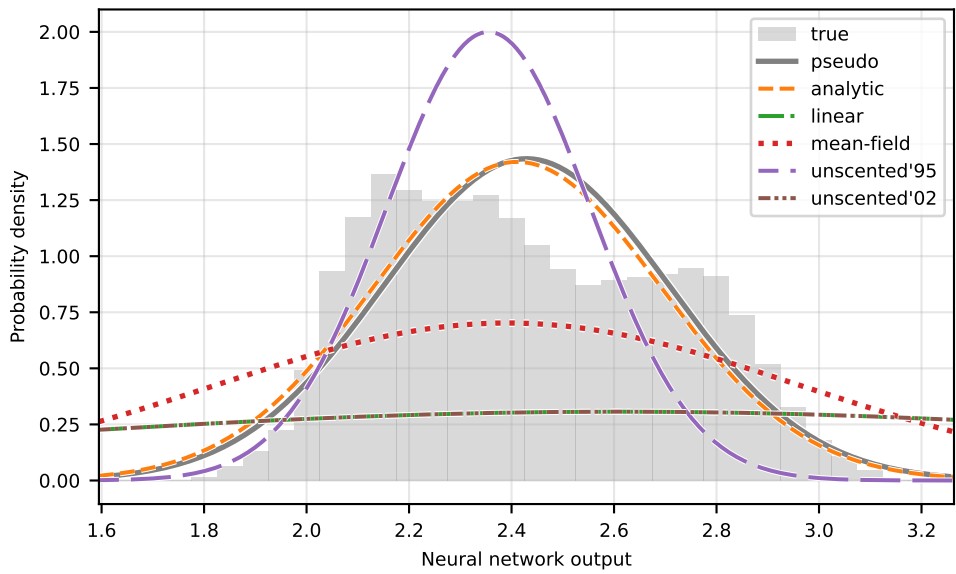

Figure 103: Probability distributions for Network(architecture=deep, weights=initialized, activation=relu), variance=medium

| distribution | $\mu$ | $\sigma^2$ |
|---|---|---|
| pseudo-true $(Y_1)$ | $+7.572 \times 10^0 \pm 1.0 \times 10^{-4}$ | $1.890 \times 10^{+1} \pm 2.5 \times 10^{-3}$ |
| analytic | $+8.042 \times 10^0$ | $9.281 \times 10^0$ |
| mean-field | $+7.961 \times 10^0$ | $1.191 \times 10^{+1}$ |
| linear | $+2.611 \times 10^0$ | $1.690 \times 10^{+2}$ |
| unscented'95 | $+8.271 \times 10^0$ | $1.248 \times 10^{+1}$ |
| unscented'02 | $-3.548 \times 10^{+1}$ | $3.074 \times 10^{+3}$ |

Table 192: Comparison of moments for Network(architecture=deep, weights=initialized, activation=relu), variance=large

| distribution | $d_{\mathrm{W}}(\cdot, Y_0)$ | $D_{\mathrm{KL}}(Y_1 \parallel \cdot)$ |
|---|---|---|
| pseudo-true $(Y_1)$ | $4.696 \times 10^{-1} \pm 8.3 \times 10^{-5}$ | $0$ |
| analytic | $6.699 \times 10^{-1} \pm 7.2 \times 10^{-5}$ | $1.070 \times 10^{-1} \pm 3.2 \times 10^{-5}$ |
| mean-field | $5.803 \times 10^{-1} \pm 6.7 \times 10^{-5}$ | $4.992 \times 10^{-2} \pm 2.3 \times 10^{-5}$ |
| linear | $3.750 \times 10^0 \pm 1.6 \times 10^{-4}$ | $3.525 \times 10^0 \pm 6.0 \times 10^{-4}$ |
| unscented'95 | $6.525 \times 10^{-1} \pm 6.9 \times 10^{-5}$ | $5.053 \times 10^{-2} \pm 1.9 \times 10^{-5}$ |
| unscented'02 | $2.595 \times 10^{+1} \pm 9.0 \times 10^{-4}$ | $1.273 \times 10^{+2} \pm 1.7 \times 10^{-2}$ |

Table 193: Comparison of statistical distances for Network(architecture=deep, weights=initialized, activation=relu), variance=large

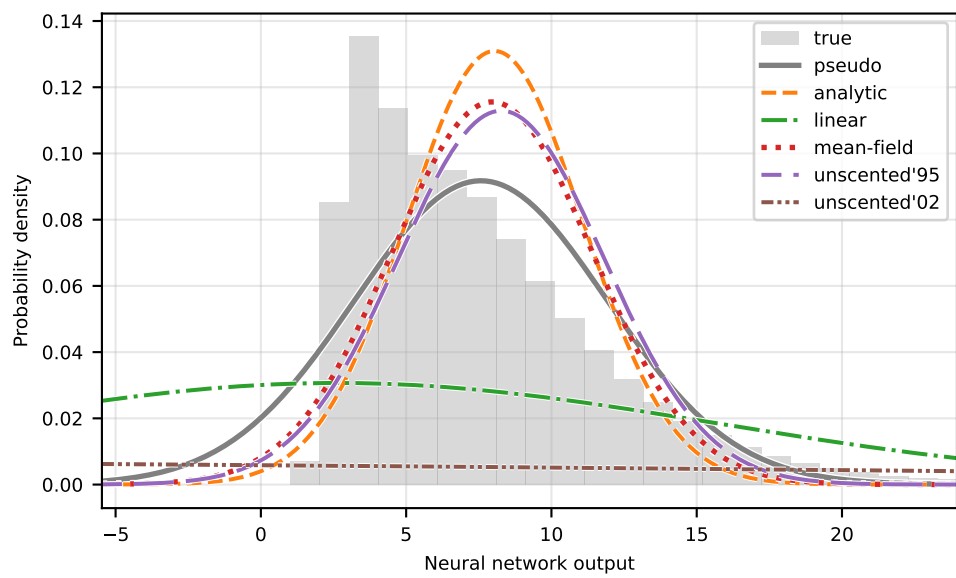

Figure 104: Probability distributions for Network(architecture=deep, weights=initialized, activation=relu), variance=large

| distribution | $\mu$ | $\sigma^2$ |
|---|---|---|
| pseudo-true ($Y_1$) | $-6.083 \times 10^{-1} \pm 8.5 \times 10^{-7}$ | $4.457 \times 10^{-2} \pm 6.2 \times 10^{-7}$ |
| analytic | $-6.074 \times 10^{-1}$ | $4.083 \times 10^{-2}$ |
| mean-field | $-6.079 \times 10^{-1}$ | $1.516 \times 10^{-2}$ |
| linear | $-6.030 \times 10^{-1}$ | $5.371 \times 10^{-2}$ |
| unscented'95 | $-5.946 \times 10^{-1}$ | $6.185 \times 10^{-2}$ |
| unscented'02 | $-6.030 \times 10^{-1}$ | $5.371 \times 10^{-2}$ |

Table 194: Comparison of moments for Network(architecture=deep, weights=trained, activation=relu), variance=small

| distribution | $d_{\mathrm{W}}(\cdot, Y_0)$ | $D_{\mathrm{KL}}(Y_1 \parallel \cdot)$ |
|---|---|---|
| pseudo-true ($Y_1$) | $2.680 \times 10^{-2} \pm 3.6 \times 10^{-6}$ | $0$ |
| analytic | $3.606 \times 10^{-2} \pm 4.0 \times 10^{-6}$ | $1.875 \times 10^{-3} \pm 5.8 \times 10^{-7}$ |
| mean-field | $1.644 \times 10^{-1} \pm 3.0 \times 10^{-6}$ | $2.092 \times 10^{-1} \pm 4.6 \times 10^{-6}$ |
| linear | $2.810 \times 10^{-2} \pm 2.6 \times 10^{-6}$ | $9.589 \times 10^{-3} \pm 1.5 \times 10^{-6}$ |
| unscented'95 | $5.845 \times 10^{-2} \pm 2.8 \times 10^{-6}$ | $3.215 \times 10^{-2} \pm 2.8 \times 10^{-6}$ |
| unscented'02 | $2.810 \times 10^{-2} \pm 2.6 \times 10^{-6}$ | $9.589 \times 10^{-3} \pm 1.5 \times 10^{-6}$ |

Table 195: Comparison of statistical distances for Network(architecture=deep, weights=trained, activation=relu), variance=small

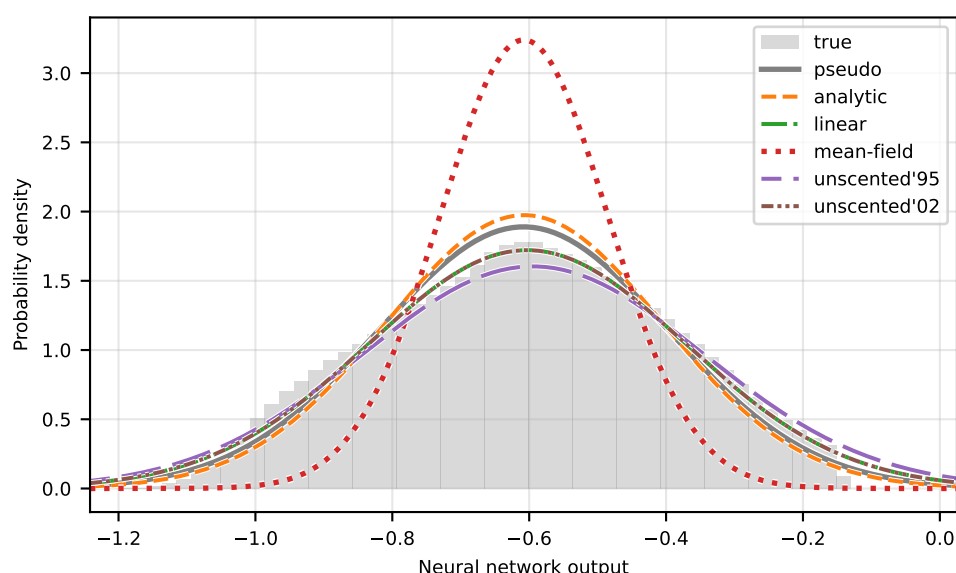

Figure 105: Probability distributions for Network(architecture=deep, weights=trained, activation=relu), variance=small

| distribution | $\mu$ | $\sigma^2$ |
|---|---|---|
| pseudo-true ($Y_1$) | $-2.734 \times 10^{-1} \pm 1.3 \times 10^{-5}$ | $2.430 \times 10^{-1} \pm 1.9 \times 10^{-5}$ |
| analytic | $-1.693 \times 10^{-1}$ | $1.785 \times 10^{-1}$ |
| mean-field | $-1.942 \times 10^{-1}$ | $3.867 \times 10^{-1}$ |
| linear | $-6.030 \times 10^{-1}$ | $5.371 \times 10^{0}$ |
| unscented'95 | $-2.971 \times 10^{-1}$ | $3.028 \times 10^{-1}$ |
| unscented'02 | $-6.030 \times 10^{-1}$ | $5.371 \times 10^{0}$ |

Table 196: Comparison of moments for Network(architecture=deep, weights=trained, activation=relu), variance=medium

| distribution | $d_{\mathrm{W}}(\cdot, Y_0)$ | $D_{\mathrm{KL}}(Y_1 \parallel \cdot)$ |
|---|---|---|
| pseudo-true ($Y_1$) | $4.927 \times 10^{-2} \pm 2.3 \times 10^{-5}$ | $0$ |
| analytic | $1.696 \times 10^{-1} \pm 1.8 \times 10^{-5}$ | $4.375 \times 10^{-2} \pm 8.1 \times 10^{-6}$ |
| mean-field | $1.932 \times 10^{-1} \pm 2.4 \times 10^{-5}$ | $7.632 \times 10^{-2} \pm 2.6 \times 10^{-5}$ |
| linear | $2.091 \times 10^{0} \pm 6.3 \times 10^{-5}$ | $9.229 \times 10^{0} \pm 8.3 \times 10^{-4}$ |
| unscented'95 | $6.066 \times 10^{-2} \pm 2.1 \times 10^{-5}$ | $1.422 \times 10^{-2} \pm 9.2 \times 10^{-6}$ |
| unscented'02 | $2.091 \times 10^{0} \pm 6.3 \times 10^{-5}$ | $9.229 \times 10^{0} \pm 8.3 \times 10^{-4}$ |

Table 197: Comparison of statistical distances for Network(architecture=deep, weights=trained, activation=relu), variance=medium

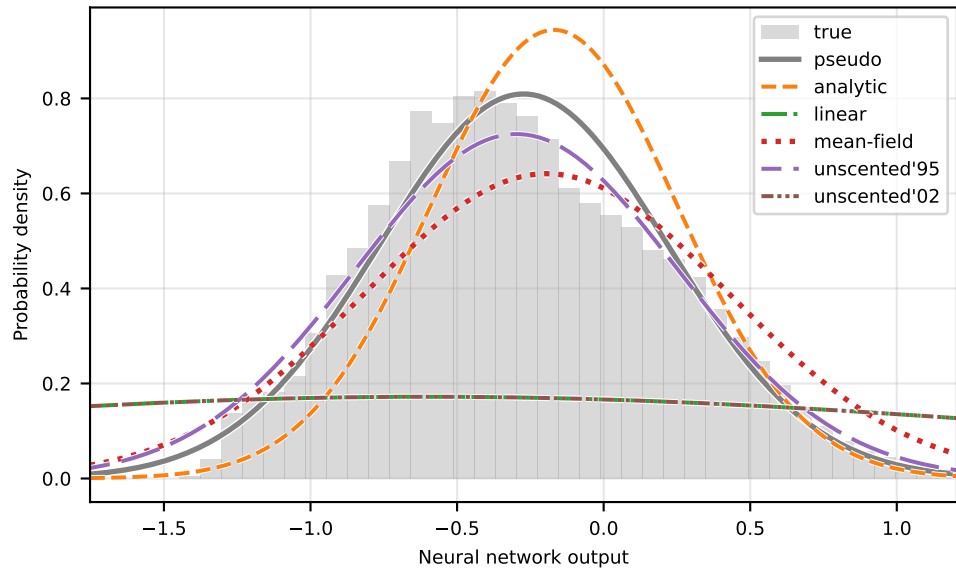

Figure 106: Probability distributions for Network(architecture=deep, weights=trained, activation=relu), variance=medium

| distribution | $\mu$ | $\sigma^2$ |
|---|---|---|
| pseudo-true $(Y_1)$ | $+3.361 \times 10^0 \pm 1.5 \times 10^{-4}$ | $1.637 \times 10^{+1} \pm 2.3 \times 10^{-3}$ |
| analytic | $+3.930 \times 10^0$ | $8.832 \times 10^0$ |
| mean-field | $+4.131 \times 10^0$ | $1.254 \times 10^{+1}$ |
| linear | $-6.030 \times 10^{-1}$ | $5.371 \times 10^{+2}$ |
| unscented'95 | $+3.242 \times 10^0$ | $1.518 \times 10^{+1}$ |
| unscented'02 | $+1.249 \times 10^{+3}$ | $3.124 \times 10^{+6}$ |

Table 198: Comparison of moments for Network(architecture=deep, weights=trained, activation=relu), variance=large

| distribution | $d_{\mathrm{W}}(\cdot, Y_0)$ | $D_{\mathrm{KL}}(Y_1 \parallel \cdot)$ |
|---|---|---|
| pseudo-true $(Y_1)$ | $3.964 \times 10^{-1} \pm 1.0 \times 10^{-4}$ | $0$ |
| analytic | $6.038 \times 10^{-1} \pm 1.1 \times 10^{-4}$ | $8.819 \times 10^{-2} \pm 3.3 \times 10^{-5}$ |
| mean-field | $5.667 \times 10^{-1} \pm 8.2 \times 10^{-5}$ | $3.443 \times 10^{-2} \pm 1.7 \times 10^{-5}$ |
| linear | $7.814 \times 10^0 \pm 3.5 \times 10^{-4}$ | $1.464 \times 10^{+1} \pm 2.3 \times 10^{-3}$ |
| unscented'95 | $3.663 \times 10^{-1} \pm 9.9 \times 10^{-5}$ | $1.817 \times 10^{-3} \pm 5.0 \times 10^{-6}$ |
| unscented'02 | $8.674 \times 10^{+2} \pm 3.1 \times 10^{-2}$ | $1.428 \times 10^{+5} \pm 2.0 \times 10^{+1}$ |

Table 199: Comparison of statistical distances for Network(architecture=deep, weights=trained, activation=relu), variance=large

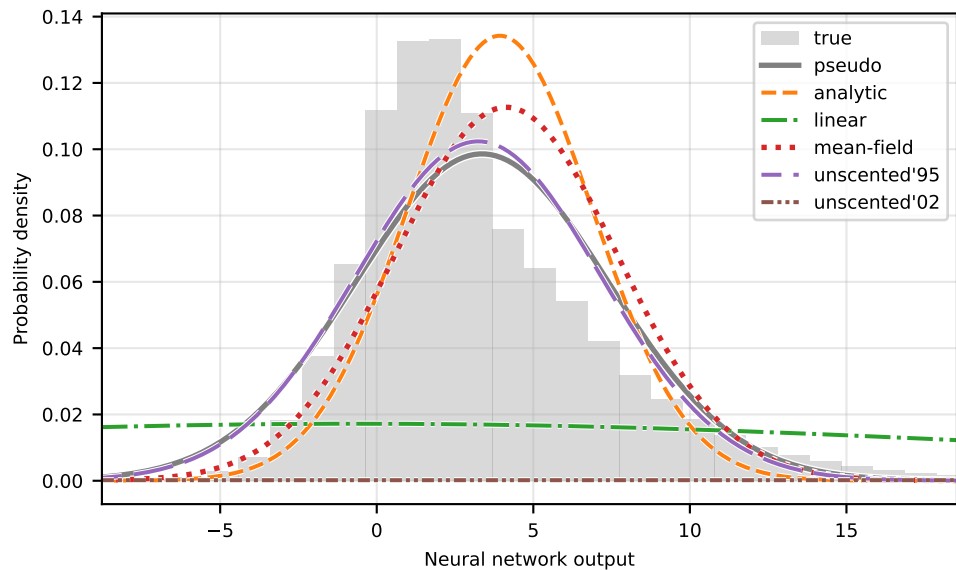

Figure 107: Probability distributions for Network(architecture=deep, weights=trained, activation=relu), variance=large

| distribution | $\mu$ | $\sigma^2$ |
| --- | --- | --- |
| pseudo-true $(Y_1)$ | $+6.455 \times 10^{+1} \pm 5.0 \times 10^{-4}$ | $6.699 \times 10^{+3} \pm 1.8 \times 10^{-1}$ |
| analytic | $+6.526 \times 10^{+1}$ | $6.392 \times 10^{+3}$ |
| mean-field | $+6.421 \times 10^{+1}$ | $8.737 \times 10^{+3}$ |
| linear | $+4.588 \times 10^{+1}$ | $3.313 \times 10^{+3}$ |
| unscented'95 | $+6.665 \times 10^{+1}$ | $5.455 \times 10^{+3}$ |
| unscented'02 | $+4.588 \times 10^{+1}$ | $3.313 \times 10^{+3}$ |

Table 200: Comparison of moments for Network(architecture=deep, weights=initialized, activation=relu residual), variance=small

| distribution | $d_{\mathrm{W}}(\cdot, Y_0)$ | $D_{\mathrm{KL}}(Y_1 \parallel \cdot)$ |
| --- | --- | --- |
| pseudo-true $(Y_1)$ | $1.234 \times 10^0 \pm 8.6 \times 10^{-5}$ | $0$ |
| analytic | $1.204 \times 10^0 \pm 6.8 \times 10^{-5}$ | $5.794 \times 10^{-4} \pm 5.9 \times 10^{-7}$ |
| mean-field | $1.820 \times 10^0 \pm 7.3 \times 10^{-5}$ | $1.932 \times 10^{-2} \pm 4.1 \times 10^{-6}$ |
| linear | $2.105 \times 10^0 \pm 5.6 \times 10^{-5}$ | $1.254 \times 10^{-1} \pm 7.0 \times 10^{-6}$ |
| unscented'95 | $1.224 \times 10^0 \pm 8.9 \times 10^{-5}$ | $1.020 \times 10^{-2} \pm 2.4 \times 10^{-6}$ |
| unscented'02 | $2.105 \times 10^0 \pm 5.6 \times 10^{-5}$ | $1.254 \times 10^{-1} \pm 7.0 \times 10^{-6}$ |

Table 201: Comparison of statistical distances for Network(architecture=deep, weights=initialized, activation=relu residual), variance=small

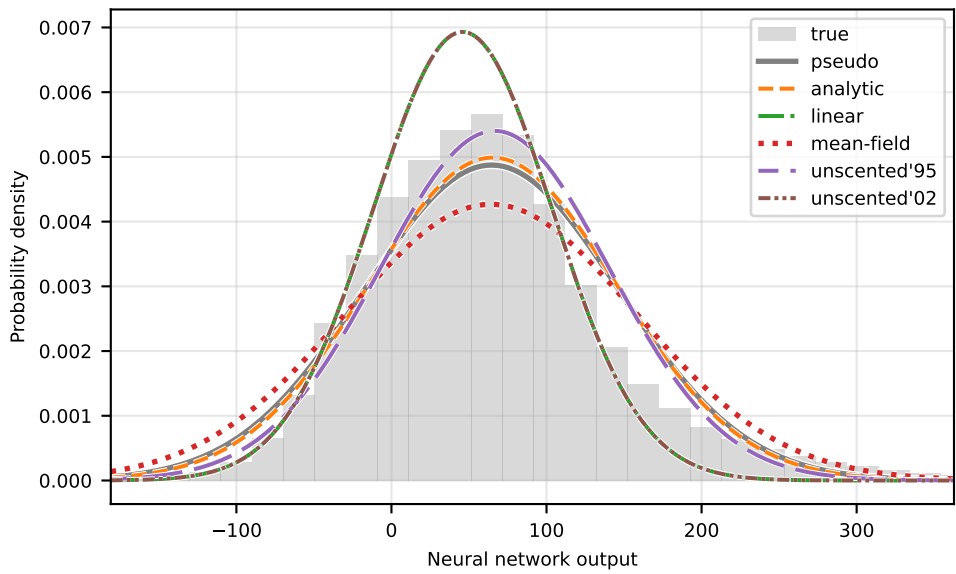

Figure 108: Probability distributions for Network(architecture=deep, weights=initialized, activation=relu residual), variance=small

| distribution | $\mu$ | $\sigma^2$ |
|---|---|---|
| pseudo-true ($Y_1$) | $+1.366 \times 10^{+2} \pm 7.3 \times 10^{-3}$ | $3.198 \times 10^{+5} \pm 1.8 \times 10^{+1}$ |
| analytic | $+1.641 \times 10^{+2}$ | $3.460 \times 10^{+5}$ |
| mean-field | $+2.213 \times 10^{+2}$ | $4.806 \times 10^{+5}$ |
| linear | $+4.588 \times 10^{+1}$ | $3.313 \times 10^{+5}$ |
| unscented'95 | $+2.183 \times 10^{+2}$ | $3.264 \times 10^{+5}$ |
| unscented'02 | $+8.620 \times 10^{+4}$ | $1.484 \times 10^{+10}$ |

Table 202: Comparison of moments for Network(architecture=deep, weights=initialized, activation=relu residual), variance=medium

| distribution | $d_{\mathrm{W}}(\cdot, Y_0)$ | $D_{\mathrm{KL}}(Y_1 \parallel \cdot)$ |
|---|---|---|
| pseudo-true ($Y_1$) | $2.050 \times 10^0 \pm 4.1 \times 10^{-4}$ | $0$ |
| analytic | $2.867 \times 10^0 \pm 4.1 \times 10^{-4}$ | $2.761 \times 10^{-3} \pm 2.3 \times 10^{-6}$ |
| mean-field | $6.383 \times 10^0 \pm 3.3 \times 10^{-4}$ | $5.890 \times 10^{-2} \pm 1.5 \times 10^{-5}$ |
| linear | $3.815 \times 10^0 \pm 3.2 \times 10^{-4}$ | $1.318 \times 10^{-2} \pm 2.9 \times 10^{-6}$ |
| unscented'95 | $4.264 \times 10^0 \pm 4.0 \times 10^{-4}$ | $1.054 \times 10^{-2} \pm 2.0 \times 10^{-6}$ |
| unscented'02 | $5.053 \times 10^{+3} \pm 7.2 \times 10^{-2}$ | $3.478 \times 10^{+4} \pm 2.0 \times 10^0$ |

Table 203: Comparison of statistical distances for Network(architecture=deep, weights=initialized, activation=relu residual), variance=medium

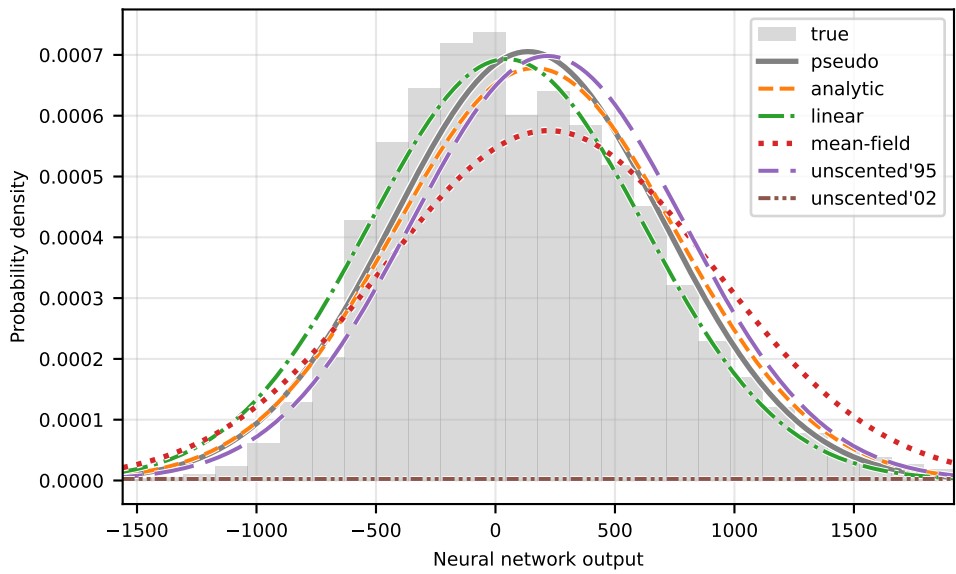

Figure 109: Probability distributions for Network(architecture=deep, weights=initialized, activation=relu residual), variance=medium

| distribution | $\mu$ | $\sigma^2$ |
|---|---|---|
| pseudo-true ($Y_1$) | $+5.307 \times 10^{+3} \pm 7.5 \times 10^{-2}$ | $4.356 \times 10^{+7} \pm 2.1 \times 10^{+3}$ |
| analytic | $+5.810 \times 10^{+3}$ | $3.855 \times 10^{+7}$ |
| mean-field | $+4.672 \times 10^{+3}$ | $3.783 \times 10^{+7}$ |
| linear | $+4.588 \times 10^{+1}$ | $3.313 \times 10^{+7}$ |
| unscented'95 | $+7.418 \times 10^{+3}$ | $1.873 \times 10^{+7}$ |
| unscented'02 | $+1.113 \times 10^{+6}$ | $2.477 \times 10^{+12}$ |

Table 204: Comparison of moments for Network(architecture=deep, weights=initialized, activation=relu residual), variance=large

| distribution | $d_{\mathrm{W}}(\cdot, Y_0)$ | $D_{\mathrm{KL}}(Y_1 \parallel \cdot)$ |
|---|---|---|
| pseudo-true ($Y_1$) | $1.207 \times 10^{+1} \pm 1.6 \times 10^{-3}$ | $0$ |
| analytic | $1.484 \times 10^{+1} \pm 1.4 \times 10^{-3}$ | $6.496 \times 10^{-3} \pm 2.6 \times 10^{-6}$ |
| mean-field | $1.125 \times 10^{+1} \pm 1.6 \times 10^{-3}$ | $9.378 \times 10^{-3} \pm 3.2 \times 10^{-6}$ |
| linear | $6.476 \times 10^{+1} \pm 1.1 \times 10^{-3}$ | $3.349 \times 10^{-1} \pm 1.2 \times 10^{-5}$ |
| unscented'95 | $3.635 \times 10^{+1} \pm 9.3 \times 10^{-4}$ | $1.881 \times 10^{-1} \pm 1.1 \times 10^{-5}$ |
| unscented'02 | $1.909 \times 10^{+4} \pm 2.3 \times 10^{-1}$ | $4.251 \times 10^{+4} \pm 2.0 \times 10^{0}$ |

Table 205: Comparison of statistical distances for Network(architecture=deep, weights=initialized, activation=relu residual), variance=large

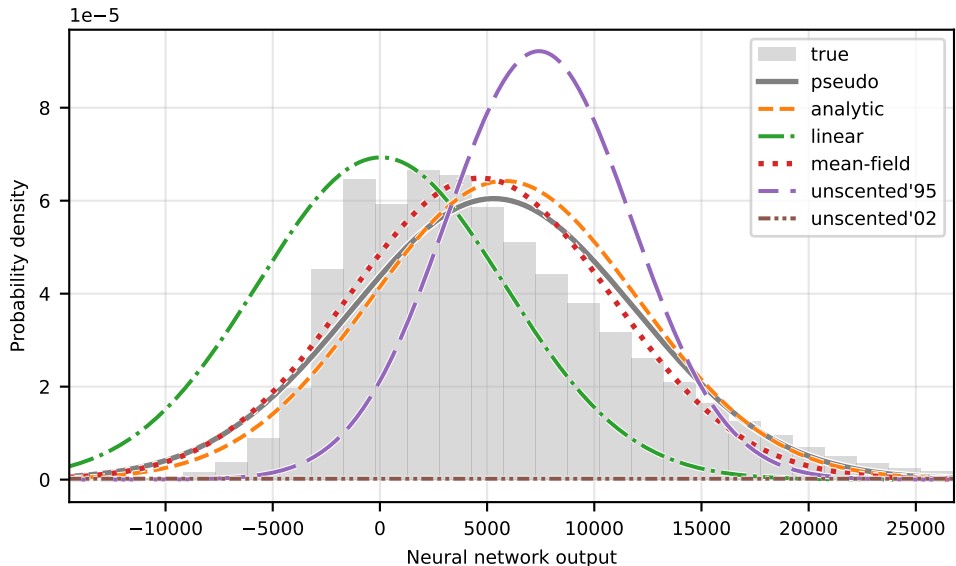

Figure 110: Probability distributions for Network(architecture=deep, weights=initialized, activation=relu residual), variance=large

| distribution | $\mu$ | $\sigma^2$ |
|---|---|---|
| pseudo-true ($Y_1$) | $-2.520 \times 10^{+1} \pm 4.7 \times 10^{-4}$ | $1.799 \times 10^{+3} \pm 1.2 \times 10^{-1}$ |
| analytic | $-2.716 \times 10^{+1}$ | $1.354 \times 10^{+3}$ |
| mean-field | $-3.235 \times 10^{+1}$ | $9.135 \times 10^{+3}$ |
| linear | $-4.096 \times 10^{+1}$ | $6.231 \times 10^{+3}$ |
| unscented'95 | $-1.945 \times 10^{+1}$ | $1.267 \times 10^{+3}$ |
| unscented'02 | $-4.096 \times 10^{+1}$ | $6.231 \times 10^{+3}$ |

Table 206: Comparison of moments for Network(architecture=deep, weights=trained, activation=relu residual), variance=small

| distribution | $d_{\mathrm{W}}(\cdot, Y_0)$ | $D_{\mathrm{KL}}(Y_1 \parallel \cdot)$ |
|---|---|---|
| pseudo-true ($Y_1$) | $7.204 \times 10^{-1} \pm 1.7 \times 10^{-4}$ | $0$ |
| analytic | $7.507 \times 10^{-1} \pm 1.2 \times 10^{-4}$ | $1.946 \times 10^{-2} \pm 8.3 \times 10^{-6}$ |
| mean-field | $6.635 \times 10^{0} \pm 2.0 \times 10^{-4}$ | $1.241 \times 10^{0} \pm 1.3 \times 10^{-4}$ |
| linear | $4.870 \times 10^{0} \pm 1.6 \times 10^{-4}$ | $6.797 \times 10^{-1} \pm 8.4 \times 10^{-5}$ |
| unscented'95 | $1.278 \times 10^{0} \pm 6.3 \times 10^{-5}$ | $3.666 \times 10^{-2} \pm 8.5 \times 10^{-6}$ |
| unscented'02 | $4.870 \times 10^{0} \pm 1.6 \times 10^{-4}$ | $6.797 \times 10^{-1} \pm 8.4 \times 10^{-5}$ |

Table 207: Comparison of statistical distances for Network(architecture=deep, weights=trained, activation=relu residual), variance=small

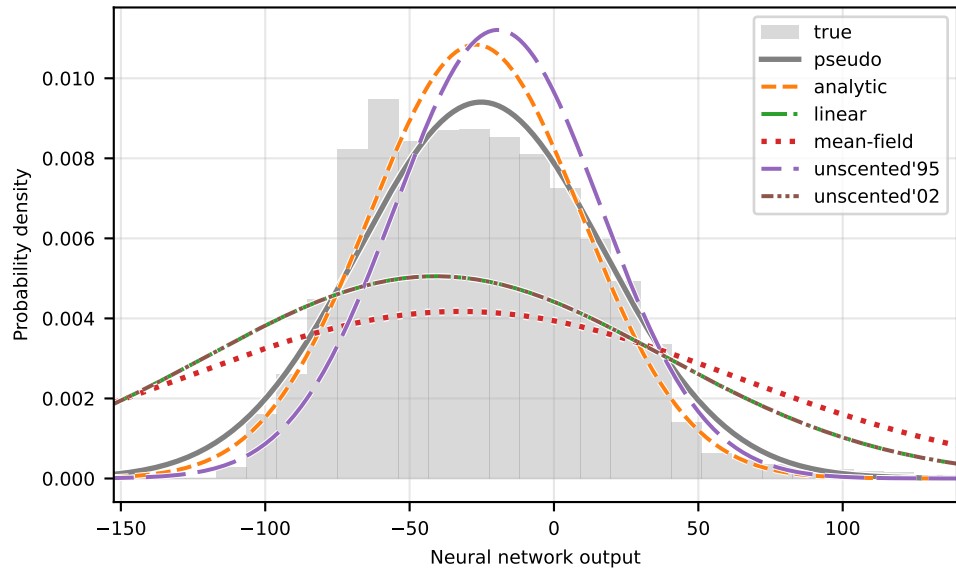

Figure 111: Probability distributions for Network(architecture=deep, weights=trained, activation=relu residual), variance=small

| distribution | $\mu$ | $\sigma^2$ |
|---|---|---|
| pseudo-true ($Y_1$) | $-1.372 \times 10^{+1} \pm 7.8 \times 10^{-3}$ | $1.394 \times 10^{+5} \pm 1.6 \times 10^{+1}$ |
| analytic | $-2.415 \times 10^{+1}$ | $2.175 \times 10^{+5}$ |
| mean-field | $+2.939 \times 10^{+1}$ | $4.827 \times 10^{+5}$ |
| linear | $-4.096 \times 10^{+1}$ | $6.231 \times 10^{+5}$ |
| unscented'95 | $+6.028 \times 10^{+1}$ | $9.951 \times 10^{+4}$ |
| unscented'02 | $-4.096 \times 10^{+1}$ | $6.231 \times 10^{+5}$ |

Table 208: Comparison of moments for Network(architecture=deep, weights=trained, activation=relu residual), variance=medium

| distribution | $d_{\mathrm{W}}(\cdot, Y_0)$ | $D_{\mathrm{KL}}(Y_1 \parallel \cdot)$ |
|---|---|---|
| pseudo-true ($Y_1$) | $2.432 \times 10^0 \pm 8.2 \times 10^{-4}$ | $0$ |
| analytic | $5.344 \times 10^0 \pm 7.5 \times 10^{-4}$ | $5.820 \times 10^{-2} \pm 3.1 \times 10^{-5}$ |
| mean-field | $1.467 \times 10^{+1} \pm 8.9 \times 10^{-4}$ | $6.173 \times 10^{-1} \pm 1.4 \times 10^{-4}$ |
| linear | $1.861 \times 10^{+1} \pm 1.0 \times 10^{-3}$ | $9.894 \times 10^{-1} \pm 1.9 \times 10^{-4}$ |
| unscented'95 | $4.410 \times 10^0 \pm 3.7 \times 10^{-4}$ | $4.508 \times 10^{-2} \pm 1.5 \times 10^{-5}$ |
| unscented'02 | $1.861 \times 10^{+1} \pm 1.0 \times 10^{-3}$ | $9.894 \times 10^{-1} \pm 1.9 \times 10^{-4}$ |

Table 209: Comparison of statistical distances for Network(architecture=deep, weights=trained, activation=relu residual), variance=medium

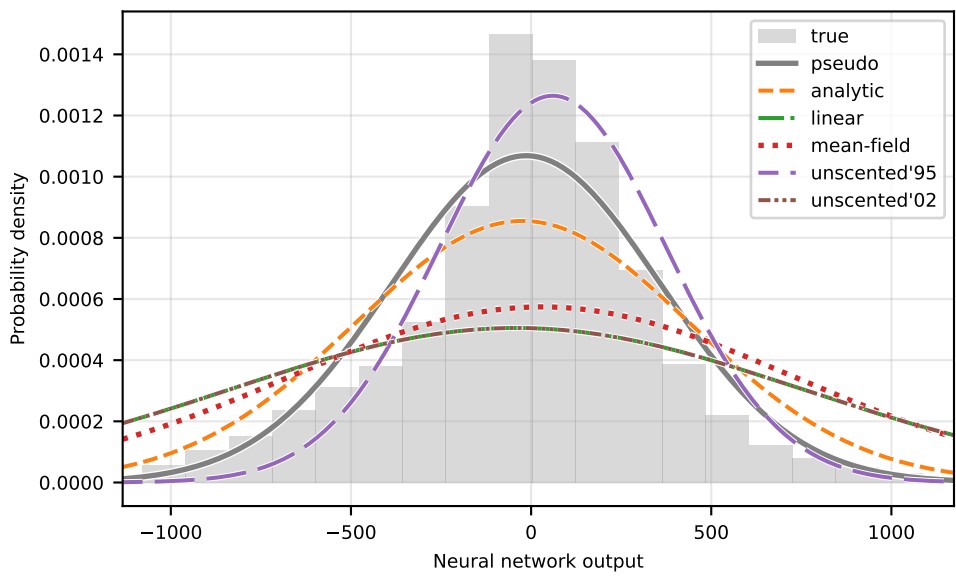

Figure 112: Probability distributions for Network(architecture=deep, weights=trained, activation=relu residual), variance=medium

| distribution | $\mu$ | $\sigma^2$ |
|---|---|---|
| pseudo-true ($Y_1$) | $+3.974 \times 10^{+3} \pm 6.7 \times 10^{-2}$ | $3.397 \times 10^{+7} \pm 1.7 \times 10^{+3}$ |
| analytic | $+4.166 \times 10^{+3}$ | $3.255 \times 10^{+7}$ |
| mean-field | $+2.983 \times 10^{+3}$ | $3.785 \times 10^{+7}$ |
| linear | $-4.096 \times 10^{+1}$ | $6.231 \times 10^{+7}$ |
| unscented'95 | $+5.757 \times 10^{+3}$ | $1.384 \times 10^{+7}$ |
| unscented'02 | $-3.249 \times 10^{+4}$ | $2.168 \times 10^{+9}$ |

Table 210: Comparison of moments for Network(architecture=deep, weights=trained, activation=relu residual), variance=large

| distribution | $d_{\mathrm{W}}(\cdot, Y_0)$ | $D_{\mathrm{KL}}(Y_1 \parallel \cdot)$ |
|---|---|---|
| pseudo-true ($Y_1$) | $9.128 \times 10^0 \pm 1.1 \times 10^{-3}$ | $0$ |
| analytic | $9.622 \times 10^0 \pm 1.2 \times 10^{-3}$ | $9.921 \times 10^{-4} \pm 1.0 \times 10^{-6}$ |
| mean-field | $1.297 \times 10^{+1} \pm 8.6 \times 10^{-4}$ | $1.747 \times 10^{-2} \pm 3.6 \times 10^{-6}$ |
| linear | $5.259 \times 10^{+1} \pm 9.7 \times 10^{-4}$ | $3.510 \times 10^{-1} \pm 3.2 \times 10^{-5}$ |
| unscented'95 | $3.335 \times 10^{+1} \pm 9.4 \times 10^{-4}$ | $1.995 \times 10^{-1} \pm 1.2 \times 10^{-5}$ |
| unscented'02 | $5.818 \times 10^{+2} \pm 7.9 \times 10^{-3}$ | $4.890 \times 10^{+1} \pm 2.5 \times 10^{-3}$ |

Table 211: Comparison of statistical distances for Network(architecture=deep, weights=trained, activation=relu residual), variance=large

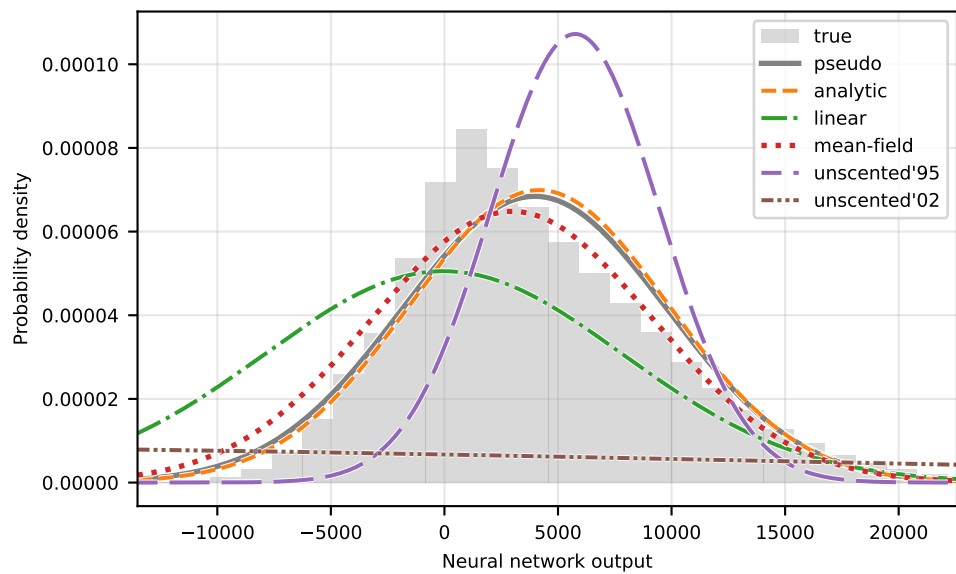

Figure 113: Probability distributions for Network(architecture=deep, weights=trained, activation=relu residual), variance=large

| distribution | $\mu$ | $\sigma^2$ |
|---|---|---|
| pseudo-true ($Y_1$) | $+5.368 \times 10^{-1} \pm 1.1 \times 10^{-4}$ | $9.674 \times 10^{-2} \pm 3.7 \times 10^{-5}$ |
| analytic | $+5.948 \times 10^{-1}$ | $9.177 \times 10^{-2}$ |
| mean-field | $+5.946 \times 10^{-1}$ | $8.172 \times 10^{-2}$ |
| linear | $+8.223 \times 10^{-2}$ | $0$ |
| unscented'95 | $+5.531 \times 10^{-1}$ | $9.697 \times 10^{-2}$ |
| unscented'02 | $+8.223 \times 10^{-2}$ | $1.303 \times 10^{-22}$ |

Table 212: Comparison of moments for Network(architecture=deep, weights=initialized, activation=heaviside), variance=small

| distribution | $d_{\mathrm{W}}(\cdot, Y_0)$ | $D_{\mathrm{KL}}(Y_1 \parallel \cdot)$ |
|---|---|---|
| pseudo-true ($Y_1$) | $5.623 \times 10^{-2} \pm 4.9 \times 10^{-5}$ | $0$ |
| analytic | $1.172 \times 10^{-1} \pm 1.6 \times 10^{-4}$ | $1.811 \times 10^{-2} \pm 6.4 \times 10^{-5}$ |
| mean-field | $1.170 \times 10^{-1} \pm 1.5 \times 10^{-4}$ | $2.400 \times 10^{-2} \pm 6.4 \times 10^{-5}$ |
| linear | $- \pm -$ | $\infty \pm -$ |
| unscented'95 | $6.479 \times 10^{-2} \pm 1.1 \times 10^{-4}$ | $1.379 \times 10^{-3} \pm 1.8 \times 10^{-5}$ |
| unscented'02 | $8.272 \times 10^{-1} \pm 1.8 \times 10^{-4}$ | $2.460 \times 10^{+1} \pm 5.1 \times 10^{-4}$ |

Table 213: Comparison of statistical distances for Network(architecture=deep, weights=initialized, activation=heaviside), variance=small

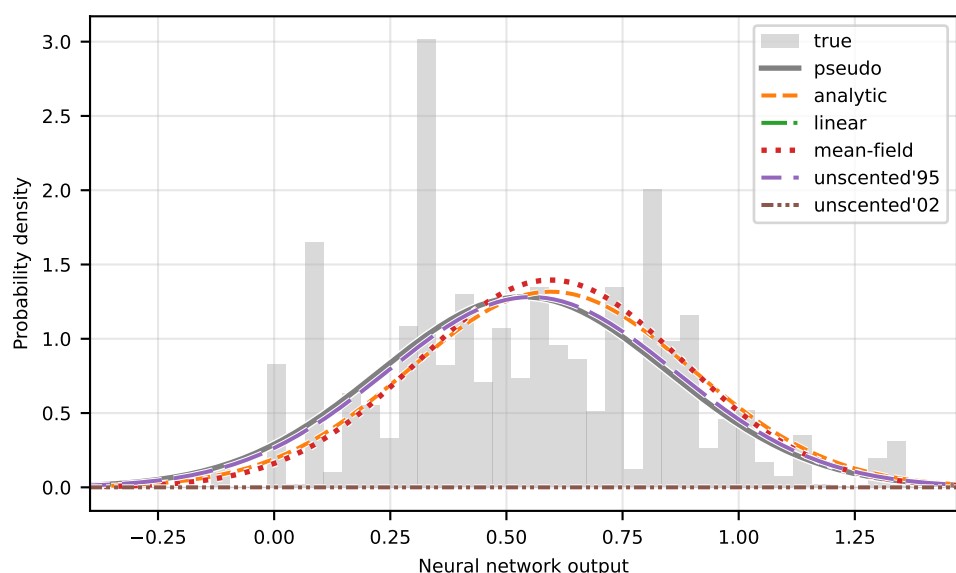

Figure 114: Probability distributions for Network(architecture=deep, weights=initialized, activation=heaviside), variance=small

| distribution | $\mu$ | $\sigma^2$ |
|---|---|---|
| pseudo-true ($Y_1$) | $+5.383 \times 10^{-1} \pm 1.2 \times 10^{-4}$ | $9.035 \times 10^{-2} \pm 7.9 \times 10^{-5}$ |
| analytic | $+5.949 \times 10^{-1}$ | $9.201 \times 10^{-2}$ |
| mean-field | $+5.957 \times 10^{-1}$ | $8.197 \times 10^{-2}$ |
| linear | $+8.223 \times 10^{-2}$ | $0$ |
| unscented'95 | $+4.318 \times 10^{-1}$ | $6.143 \times 10^{-2}$ |
| unscented'02 | $+8.223 \times 10^{-2}$ | $1.303 \times 10^{-22}$ |

Table 214: Comparison of moments for Network(architecture=deep, weights=initialized, activation=heaviside), variance=medium

| distribution | $d_{\mathrm{W}}(\cdot, Y_0)$ | $D_{\mathrm{KL}}(Y_1 \parallel \cdot)$ |
|---|---|---|
| pseudo-true ($Y_1$) | $4.662 \times 10^{-2} \pm 1.3 \times 10^{-4}$ | $0$ |
| analytic | $1.058 \times 10^{-1} \pm 2.1 \times 10^{-4}$ | $1.777 \times 10^{-2} \pm 7.7 \times 10^{-5}$ |
| mean-field | $1.085 \times 10^{-1} \pm 2.0 \times 10^{-4}$ | $2.049 \times 10^{-2} \pm 8.4 \times 10^{-5}$ |
| linear | $- \pm -$ | $\infty \pm -$ |
| unscented'95 | $1.951 \times 10^{-1} \pm 2.2 \times 10^{-4}$ | $9.570 \times 10^{-2} \pm 1.6 \times 10^{-4}$ |
| unscented'02 | $8.522 \times 10^{-1} \pm 2.5 \times 10^{-4}$ | $2.465 \times 10^{+1} \pm 8.9 \times 10^{-4}$ |

Table 215: Comparison of statistical distances for Network(architecture=deep, weights=initialized, activation=heaviside), variance=medium

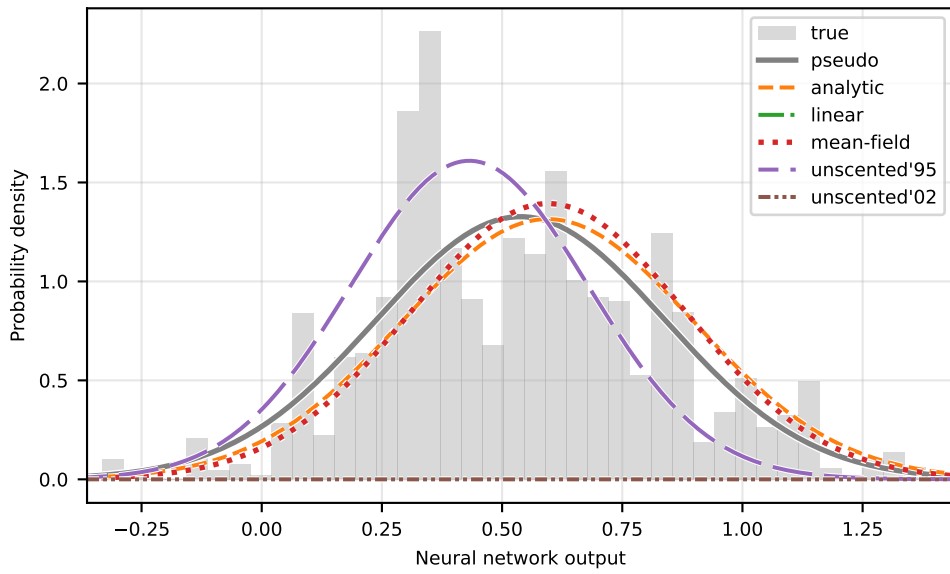

Figure 115: Probability distributions for Network(architecture=deep, weights=initialized, activation=heaviside), variance=medium

| distribution | $\mu$ | $\sigma^2$ |
|---|---|---|
| pseudo-true $(Y_1)$ | $+5.414 \times 10^{-1} \pm 2.1 \times 10^{-4}$ | $8.995 \times 10^{-2} \pm 8.5 \times 10^{-5}$ |
| analytic | $+5.949 \times 10^{-1}$ | $9.189 \times 10^{-2}$ |
| mean-field | $+5.952 \times 10^{-1}$ | $8.185 \times 10^{-2}$ |
| linear | $+8.223 \times 10^{-2}$ | $0$ |
| unscented'95 | $+6.727 \times 10^{-1}$ | $1.985 \times 10^{-1}$ |
| unscented'02 | $+8.223 \times 10^{-2}$ | $1.303 \times 10^{-22}$ |

Table 216: Comparison of moments for Network(architecture=deep, weights=initialized, activation=heaviside), variance=large

| distribution | $d_{\mathrm{W}}(\cdot, Y_0)$ | $D_{\mathrm{KL}}(Y_1 \parallel \cdot)$ |
|---|---|---|
| pseudo-true $(Y_1)$ | $5.311 \times 10^{-2} \pm 1.5 \times 10^{-4}$ | $0$ |
| analytic | $1.025 \times 10^{-1} \pm 3.5 \times 10^{-4}$ | $1.604 \times 10^{-2} \pm 1.3 \times 10^{-4}$ |
| mean-field | $1.057 \times 10^{-1} \pm 3.4 \times 10^{-4}$ | $1.827 \times 10^{-2} \pm 1.3 \times 10^{-4}$ |
| linear | $— \pm —$ | $\infty \pm —$ |
| unscented'95 | $3.236 \times 10^{-1} \pm 3.6 \times 10^{-4}$ | $3.036 \times 10^{-1} \pm 7.7 \times 10^{-4}$ |
| unscented'02 | $8.558 \times 10^{-1} \pm 3.6 \times 10^{-4}$ | $2.466 \times 10^{+1} \pm 1.2 \times 10^{-3}$ |

Table 217: Comparison of statistical distances for Network(architecture=deep, weights=initialized, activation=heaviside), variance=large

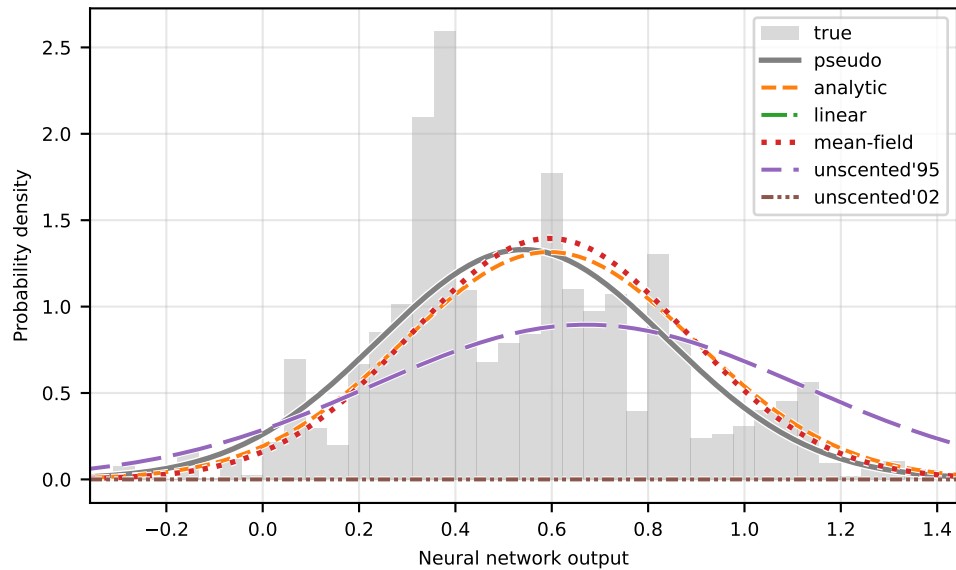

Figure 116: Probability distributions for Network(architecture=deep, weights=initialized, activation=heaviside), variance=large

| distribution | $\mu$ | $\sigma^2$ |
|---|---|---|
| pseudo-true ($Y_1$) | $+8.067 \times 10^{-1} \pm 1.6 \times 10^{-4}$ | $4.988 \times 10^{-1} \pm 2.0 \times 10^{-4}$ |
| analytic | $+7.753 \times 10^{-1}$ | $4.018 \times 10^{-1}$ |
| mean-field | $+8.268 \times 10^{-1}$ | $5.413 \times 10^{-1}$ |
| linear | $+7.963 \times 10^{-1}$ | $0$ |
| unscented'95 | $+5.844 \times 10^{-1}$ | $5.985 \times 10^{-1}$ |
| unscented'02 | $+7.963 \times 10^{-1}$ | $6.996 \times 10^{-21}$ |

Table 218: Comparison of moments for Network(architecture=deep, weights=initialized, activation=heaviside residual), variance=small

| distribution | $d_{\mathrm{W}}(\cdot, Y_0)$ | $D_{\mathrm{KL}}(Y_1 \parallel \cdot)$ |
|---|---|---|
| pseudo-true ($Y_1$) | $4.940 \times 10^{-2} \pm 5.1 \times 10^{-5}$ | $0$ |
| analytic | $8.399 \times 10^{-2} \pm 1.0 \times 10^{-4}$ | $1.190 \times 10^{-2} \pm 3.3 \times 10^{-5}$ |
| mean-field | $5.953 \times 10^{-2} \pm 6.8 \times 10^{-5}$ | $2.123 \times 10^{-3} \pm 1.4 \times 10^{-5}$ |
| linear | $- \pm -$ | $\infty \pm -$ |
| unscented'95 | $2.677 \times 10^{-1} \pm 2.1 \times 10^{-4}$ | $5.836 \times 10^{-2} \pm 1.2 \times 10^{-4}$ |
| unscented'02 | $6.676 \times 10^{-1} \pm 9.1 \times 10^{-5}$ | $2.236 \times 10^{+1} \pm 2.0 \times 10^{-4}$ |

Table 219: Comparison of statistical distances for Network(architecture=deep, weights=initialized, activation=heaviside residual), variance=small

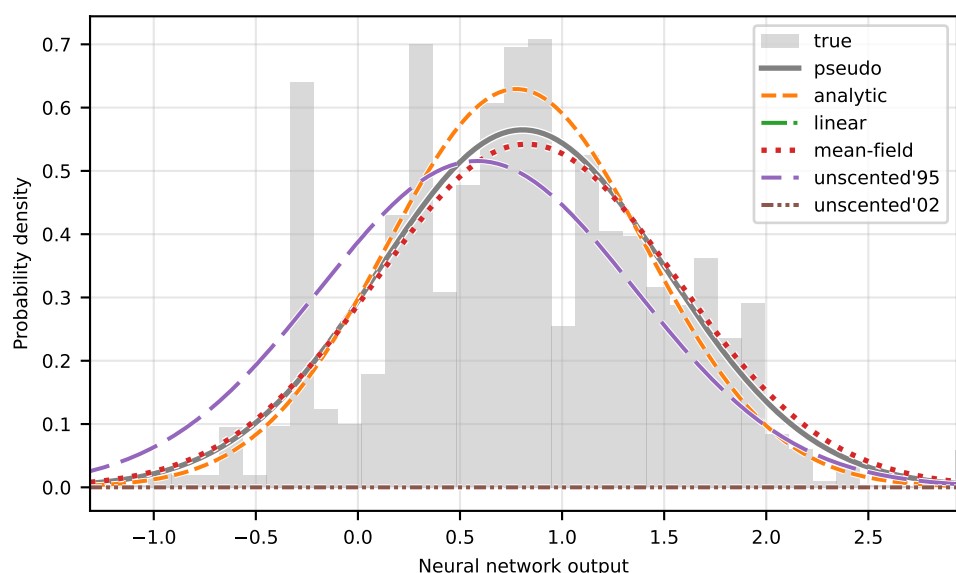

Figure 117: Probability distributions for Network(architecture=deep, weights=initialized, activation=heaviside residual), variance=small

| distribution | $\mu$ | $\sigma^2$ |
|---|---|---|
| pseudo-true $(Y_1)$ | $+7.707 \times 10^{-1} \pm 3.1 \times 10^{-4}$ | $1.283 \times 10^0 \pm 5.4 \times 10^{-4}$ |
| analytic | $+6.602 \times 10^{-1}$ | $1.241 \times 10^0$ |
| mean-field | $+5.408 \times 10^{-1}$ | $1.006 \times 10^0$ |
| linear | $+7.963 \times 10^{-1}$ | $0$ |
| unscented'95 | $+1.122 \times 10^0$ | $8.345 \times 10^{-1}$ |
| unscented'02 | $+7.963 \times 10^{-1}$ | $6.996 \times 10^{-21}$ |

Table 220: Comparison of moments for Network(architecture=deep, weights=initialized, activation=heaviside residual), variance=medium

| distribution | $d_{\mathrm{W}}(\cdot, Y_0)$ | $D_{\mathrm{KL}}(Y_1 \parallel \cdot)$ |
|---|---|---|
| pseudo-true $(Y_1)$ | $4.528 \times 10^{-2} \pm 2.2 \times 10^{-4}$ | $0$ |
| analytic | $1.078 \times 10^{-1} \pm 2.9 \times 10^{-4}$ | $5.025 \times 10^{-3} \pm 2.7 \times 10^{-5}$ |
| mean-field | $2.201 \times 10^{-1} \pm 2.3 \times 10^{-4}$ | $3.429 \times 10^{-2} \pm 6.8 \times 10^{-5}$ |
| linear | $- \pm -$ | $\infty \pm -$ |
| unscented'95 | $3.297 \times 10^{-1} \pm 3.0 \times 10^{-4}$ | $8.825 \times 10^{-2} \pm 9.9 \times 10^{-5}$ |
| unscented'02 | $8.681 \times 10^{-1} \pm 1.3 \times 10^{-4}$ | $2.283 \times 10^{+1} \pm 2.1 \times 10^{-4}$ |

Table 221: Comparison of statistical distances for Network(architecture=deep, weights=initialized, activation=heaviside residual), variance=medium

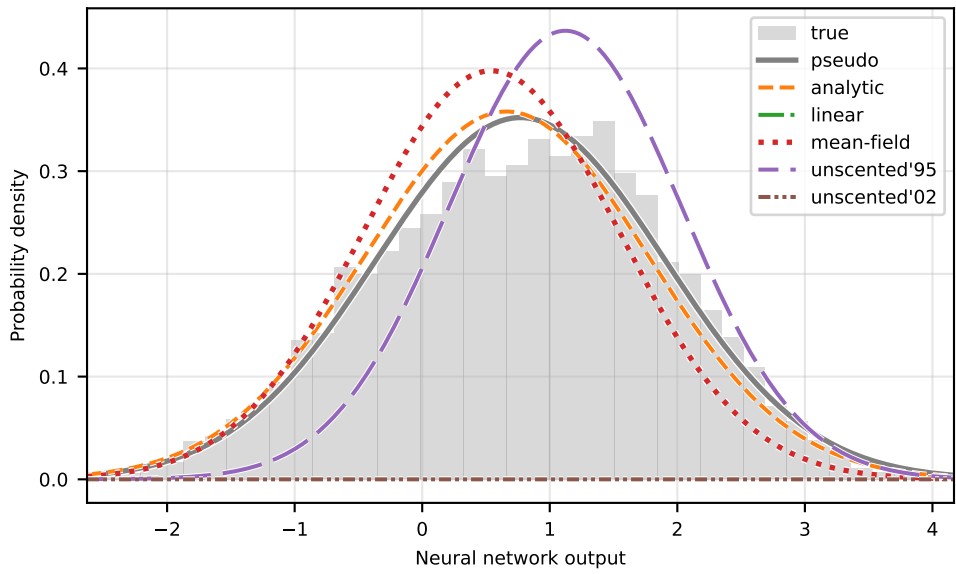

Figure 118: Probability distributions for Network(architecture=deep, weights=initialized, activation=heaviside residual), variance=medium

| distribution | $\mu$ | $\sigma^2$ |
|---|---|---|
| pseudo-true $(Y_1)$ | $+5.804 \times 10^{-1} \pm 4.0 \times 10^{-4}$ | $1.098 \times 10^0 \pm 7.2 \times 10^{-4}$ |
| analytic | $+5.070 \times 10^{-1}$ | $1.071 \times 10^0$ |
| mean-field | $+4.866 \times 10^{-1}$ | $1.190 \times 10^0$ |
| linear | $+7.963 \times 10^{-1}$ | $0$ |
| unscented'95 | $+7.638 \times 10^{-1}$ | $1.589 \times 10^0$ |
| unscented'02 | $+7.963 \times 10^{-1}$ | $6.996 \times 10^{-21}$ |

Table 222: Comparison of moments for Network(architecture=deep, weights=initialized, activation=heaviside residual), variance=large

| distribution | $d_{\mathrm{W}}(\cdot, Y_0)$ | $D_{\mathrm{KL}}(Y_1 \parallel \cdot)$ |
|---|---|---|
| pseudo-true $(Y_1)$ | $3.778 \times 10^{-2} \pm 2.7 \times 10^{-4}$ | $0$ |
| analytic | $7.241 \times 10^{-2} \pm 3.8 \times 10^{-4}$ | $2.611 \times 10^{-3} \pm 2.8 \times 10^{-5}$ |
| mean-field | $9.352 \times 10^{-2} \pm 3.7 \times 10^{-4}$ | $5.700 \times 10^{-3} \pm 4.3 \times 10^{-5}$ |
| linear | $— \pm —$ | $\infty \pm —$ |
| unscented'95 | $2.401 \times 10^{-1} \pm 3.8 \times 10^{-4}$ | $5.412 \times 10^{-2} \pm 1.8 \times 10^{-4}$ |
| unscented'02 | $8.522 \times 10^{-1} \pm 2.3 \times 10^{-4}$ | $2.277 \times 10^{+1} \pm 3.1 \times 10^{-4}$ |

Table 223: Comparison of statistical distances for Network(architecture=deep, weights=initialized, activation=heaviside residual), variance=large

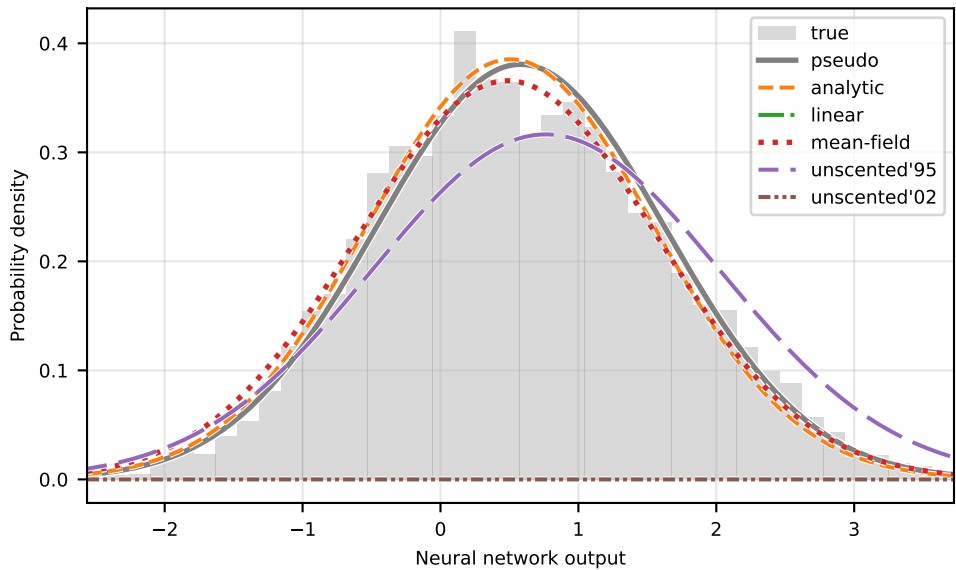

Figure 119: Probability distributions for Network(architecture=deep, weights=initialized, activation=heaviside residual), variance=large

## O  COMMENTS ON COMPUTATIONAL COMPLEXITY

The cost of moment matching, like that of linearized covariance propagation, is cubic in the number of hidden neurons due to the need to evaluate "sandwich" covariances such as $A\Sigma A^\intercal$. [6] By contrast, the computational complexity of evaluating a neural network is quadratic in the number of hidden neurons. Thus, theoretically, moment matching is one polynomial order of magnitude slower than point prediction, on the same order as linearization. For neural networks with high input dimension, moment matching is much faster than Monte Carlo, which scales exponentially with dimension.

We performed a simple experiment using Jupyter Notebook's `%%timeit` to compare the computational cost of moment matching and Monte Carlo. We compare our method with the ground truth method used in §1, which runs quasi-Monte Carlo (QMC) on $2^{16}$ samples. We use GeLU because it has the most complicated covariance expression (see Appendix E).

On "wide residual" networks (defined in Appendix N):

- Our method takes 88.4 ms ± 5.96 ms.
- QMC takes 1.79 s ± 83 ms.

On "deep residual" networks (defined in Appendix N):

- Our method takes 23.7 ms ± 957 $\mu$s.
- QMC takes 878 ms ± 61.8 ms.

---

[6]Contrary to occasional claims that the cost of propagating a covariance matrix scales *quadratically* with the number of hidden neurons (Akgül et al., 2025, §5).

## P LLM USAGE STATEMENT

We consider that LLMs were not involved "at the level of a contributing author." We nevertheless include a usage statement for full transparency. LLM usage consisted of early ideation involved in checking whether $\Phi$ had exact integrals, as well as autocomplete (executable code and LaTeX) and limited code generation (executable code). We take full responsibility for the content of this manuscript.

