# OpenReview forum: "Closed-form uncertainty quantification of deep residual neural networks"
_ICLR.cc/2026/Conference — Submitted to ICLR 2026_

### Official Review · Reviewer_L8SC · 2025-10-31

**Soundness:** 3
**Presentation:** 2
**Contribution:** 2
**Rating:** 4
**Confidence:** 4

**Summary:**

The authors aim to address the problem of propagating the mean and covariance of a general multivariate Gaussian distribution through a deep (residual) neural network using layer-by-layer moment matching. Technically, they derive analytical representations for the first two moments when using sine and probit activations in the neural network. Theoretical guanrantees and adversarial examples are provided for further analysis. Experimentally, they find competitive results on statistical calibration for inference under noisy or missing features on real data.

**Strengths:**

- The problem to be addressed is relevant for uncertainty estimation based on the first and second moment propagation.

- The idea is novel in the sense of deriving analytical solution for neural networks with sine or prohit activation functions.

- Theorectical guarantees have been dervied and analyzed for attribution of sources of error in this approach.

- The analysis of adversarial examples can shed light in future investigation in this direction.

- The carefully desgined experiments are comprehensive in terms of the tasks and statistical significance.

**Weaknesses:**

- The major concern regarding this idea is the scalibility and generalizability of the specific activation functions on which this work is built. As sin and prohit activation functions are less widely used than others such as relu, etc., the analytical repsentations derived in this work seems limited in this sense.

- The presentation flow is somehow hard to follow, especailly the section of method. I would suggest the authors to shorten the method part and merge the problem formulation, theorectical guarantees into this part to improve the coherence.

- Though it is insightful for the readers to provide theorectical guarantees and adversarial examples. The actual implication of them for the main claim in the paper is not very clear. Therefore these two parts could be reduced to avoid distraction to the readers.

- There are two suggestions for the experimental part:
 1. The result presentation layout can be improved, for example, Fig1 and 2 can be merged to avoid occupaying the whole page.
 2. The proposed solution would be more convincing to be evaluated on more challenging tasks with larger network architectures and datasets, such as image classification.

**Questions:**

See above

---

> ### Author Response · Authors · 2025-11-21
>
> Thank you for your useful feedback.
> We address general areas of concern in depth in the [top-level comment](https://openreview.net/forum?id=CHWjetQzYS&noteId=L1pHRQSKEy), but provide an itemized response in this thread.
>
> > The major concern regarding this idea is the scalibility and generalizability of the specific activation functions on which this work is built.
> > As sin and [probit] activation functions are less widely used than others such as relu, etc., the analytical [representations] derived in this work seems limited in this sense.
>
> We have added methodologically innovative derivations for ReLU and GeLU.
> As the latest revision stands, the analytical representations are not limited by choice of activation function.

---

> ### Author Response · Authors · 2025-11-21
>
> > The presentation flow is somehow hard to follow, [especially] the section of method. I would suggest the authors to shorten the method part and merge the problem formulation, [theoretical] guarantees into this part to improve the coherence.
>
> Thank you for this suggestion. We have implemented it as part of reorganizing the paper into a more linear narrative.
> See also the top-level comment, Weakness 4.

---

> ### Author Response · Authors · 2025-11-21
>
> > Though it is insightful for the readers to provide [theoretical] guarantees and adversarial examples. The actual implication of them for the main claim in the paper is not very clear. Therefore these two parts could be reduced to avoid distraction to the readers.
>
> The contributions of the theoretical guarantees are:
> 1. They provide an _a priori_ bound on the distributional approximation error of every layer in the network. The most similar result of this kind in the literature is a bound on total variation distance for a single ReLU neuron with Gaussian input, [Petersen et al. (2024), "Uncertainty Quantification via Stable Distribution Propagation"](https://openreview.net/forum?id=cZttUMTiPL).
> 2. The looseness of this bound reflects a hard limit of the problem itself: P =? NP is an obstruction to computing exact moments for deep networks with many inputs; see also the top-level comment, Weakness 3.

---

> ### Author Response · Authors · 2025-11-21
>
> > The result presentation layout can be improved, for example, Fig1 and 2 can be merged to avoid [occupying] the whole page.
>
> Thank you for this suggestion.
> We have adjusted the LaTeX float settings to place figures more efficiently.

---

> ### Author Response · Authors · 2025-11-21
>
> > The proposed solution would be more convincing to be evaluated on more challenging tasks with larger network architectures and datasets, such as image classification.
>
> Our method has no loss of accuracy with wider networks; we choose 400 neurons as an arbitrary cutoff due to computational constraints in our numerical experiments.
> But as we discuss in the top-level comment, Weakness 3, if [P != #P](https://complexityzoo.net/Complexity_Zoo:Symbols#sharpp), then no efficient method (ours included) can compute exact moments for deep networks with many inputs.

---

### Official Review · Reviewer_usVg · 2025-10-31

**Soundness:** 2
**Presentation:** 2
**Contribution:** 2
**Rating:** 4
**Confidence:** 3

**Summary:**

This paper proposes an analytic method (“Yana”) for propagating Gaussian input uncertainty through deep residual neural networks. By deriving closed-form expressions for the mean and covariance of sine and probit activations under Gaussian input, the method allows exact moment matching at each layer. Experiments on synthetic random networks and two small real datasets (California Housing regression, Taiwanese Bankruptcy classification) show that the proposed method can achieve orders-of-magnitude improvements in approximation accuracy over linearization, mean-field, and unscented transform baselines. The paper also includes theoretical error bounds and provides code for reproducibility.

**Strengths:**

- **Novelty**: Closed-form propagation formulas for probit and sine activations are mathematically elegant and, to my knowledge, new.
- **Theoretical contribution**: Includes detailed derivations, moment identities, and a theoretical error bound.
- **Synthetic evaluation**: Stress tests across 72 random networks, showing clear gains over other analytic approximations.
- **Reproducibility**: Code and detailed appendices are provided.

**Weaknesses:**

1. **Ambiguity of contribution**
   The paper presents itself as proposing a *new method for uncertainty quantification*. If that is the claim, the evaluation seems inadequate, since there are no comparisons to standard UQ baselines such as heteroscedastic neural networks [5], ensembles [6], dropout-based Bayesian approximations [2], or fully Bayesian neural networks [1].
   If instead the true contribution is a *new analytic Gaussian propagation approximation*, then the chosen baselines (linearization, mean-field, unscented transform) are more reasonable, but the framing should be explicit about this narrower scope.
   As written, the paper makes claims about introducing a novel UQ method, while in practice it only proposes a new way of propagating Gaussian input distributions through a network. Moreover, all reported baselines are themselves approximation schemes of this propagation task, rather than established UQ methods, which makes the positioning and evaluation misaligned.


2. **Missing ground-truth baseline** A clear benchmark should be included, a simple Monte Carlo sampling: draw many samples from the Gaussian input distribution, run them through the network, and empirically compute the output mean and covariance. This baseline is straightforward, works with any activation, and—given enough samples—provides a gold standard approximation. While more costly due to multiple forward passes, it would contextualize the claimed accuracy improvements of the proposed analytic method. Its absence makes it difficult to judge whether the gains are practically meaningful compared to this simple and applicable approach.

3. **Computational cost considerations**
   The method scales as \(O(n^3)\) in layer width (due to covariance propagation), as noted by the authors, similar to linearization methods but higher than a single forward pass (\(O(n^2)\)). While cheaper than Monte Carlo with many samples (\(O(Mn^2)\)), the trade-off is unclear in practice. Runtime comparisons against Monte Carlo sampling vs the proposed method would clarify the actual efficiency benefits.

4. **Limited scope of experiments**
   - Real datasets are small (≤20k samples, ≤95 features).
   - Networks are shallow (≤3 layers, width ≤200).
   - Evaluations are therefore small for modern NNs, not representative of modern deep learning architectures (e.g., ResNets, Transformers).

5. **Narrow focus on input Gaussian uncertainty**
The method appears to model only *aleatoric* uncertainty, while ignoring *epistemic* uncertainty. It is unclear why a practitioner would choose this approach over standard aleatoric uncertainty-aware models—such as heteroscedastic neural networks [5]—where the network learns to predict both the mean and the variance (as opposed to only the mean in standard regression tasks).



6. **Restrictive activations**
   Exact formulas are derived only for sine and probit activations, which are rarely used in modern practice compared to ReLU or GELU. This strongly limits applicability.


---

### **References**
[1] Blundell, C., Cornebise, J., Kavukcuoglu, K., & Wierstra, D. (2015). *Weight Uncertainty in Neural Networks.* ICML.
[2] Gal, Y., & Ghahramani, Z. (2016). *Dropout as a Bayesian Approximation: Representing Model Uncertainty in Deep Learning.* ICML.
[5] Nix, D. A., & Weigend, A. S. (1994). *Estimating the Mean and Variance of the Target Probability Distribution.* IEEE International Conference on Neural Networks (ICNN).
[6] Lakshminarayanan, B., Pritzel, A., & Blundell, C. (2017). *Simple and Scalable Predictive Uncertainty Estimation using Deep Ensembles.* NeurIPS.
[7] Wright, O., Nakahira, Y., & Moura, J. M. F. (2024). *An Analytic Solution to Covariance Propagation in Neural Networks.* AISTATS.

**Questions:**

1. Can the method be generalized or approximated for ReLU/GELU activations?
2. How does the method compare in runtime and accuracy to Monte Carlo sampling?
3. Is the contribution intended as a general UQ method or a propagation approximation?
4. How would the method scale to modern architectures with hundreds of layers?

---

> ### Author Response · Authors · 2025-11-21
>
> Thank you for your useful feedback.
> We address general areas of concern in depth in the [top-level comment](https://openreview.net/forum?id=CHWjetQzYS&noteId=L1pHRQSKEy), but provide an itemized response in this thread.
>
> > The paper presents itself as proposing a new method for uncertainty quantification.
> >
> > ...
> >
> > If instead the true contribution is a new analytic Gaussian propagation approximation, then the chosen baselines (linearization, mean-field, unscented transform) are more reasonable, but the framing should be explicit about this narrower scope.
> > As written, the paper makes claims about introducing a novel UQ method, while in practice it only proposes a new way of propagating Gaussian input distributions through a network. Moreover, all reported baselines are themselves approximation schemes of this propagation task, rather than established UQ methods, which makes the positioning and evaluation misaligned.
>
> The top-level comment clarifies that our contribution is to distribution propagation, which is a subproblem within uncertainty quantification.
> While the idea of layer-wise moment matching is not new _per se_, our contribution lies in providing exact covariances for activations in settings where many works have only derived approximations.
> We clarify this positioning and comparison in the additions to Section 2.1 (Methodology).

---

> ### Author Response · Authors · 2025-11-21
>
> > If that is the claim, the evaluation seems inadequate, since there are no comparisons to standard UQ baselines such as heteroscedastic neural networks [5], ensembles [6], dropout-based Bayesian approximations [2], or fully Bayesian neural networks [1].
>
> In the latest version, we implement variational Bayesian neural networks in Section 5.4 as a testbed for approximate Gaussian moment matching.
> We share an emphasis on properly aligned evaluation; accordingly, Footnote 3 discusses the importance of benchmarking against Monte Carlo variational inference.

---

> ### Author Response · Authors · 2025-11-21
>
> > A clear benchmark should be included, a simple Monte Carlo sampling: draw many samples from the Gaussian input distribution, run them through the network, and empirically compute the output mean and covariance. This baseline is straightforward, works with any activation, and—given enough samples—provides a gold standard approximation. While more costly due to multiple forward passes, it would contextualize the claimed accuracy improvements of the proposed analytic method. Its absence makes it difficult to judge whether the gains are practically meaningful compared to this simple and applicable approach.
> >
> > ...
> >
> > How does the method compare in ... accuracy to Monte Carlo sampling?
>
> **In the random neural networks suite (Section 5.1)**, the latest version retains the Monte Carlo baselines $Y_0$ and $Y_1$. Paraphrasing Section 2.2,
> the true distribution is $Y_0 = f(X)$, and the pseudo-true distribution is $Y_1 = \mathcal N(\mathbb E Y_0, \mathrm{Cov} Y_0)$.
> Whereas $Y_0$ is the ground-truth (not necessarily Normal) distribution, $Y_1$ is the closest Gaussian approximation (by KL divergence) to $Y_0$.
> We obtain $Y_0$ and $Y_1$ in the baselines by quasi-Monte Carlo (QMC) simulation.
>
> The evaluation metrics are:
> - KL divergence between the true distribution $Y_0$ and the Normal approximation $Y_1$.
>     - On single layers, our method would, by construction, achieve 0 in this metric, whereas, generically, no other reviewed or tested method would.
>     - For this reason, all of our random neural networks are designed to be challenging by virtue of their depth (multiple layers) and large weight initializations.
> - Wasserstein distance between the true distribution $Y_0$ and the pseudo-true distribution $Y_1$.
>     - Our method is not exact in this metric (even for a single layer), as a nonlinearly transformed Gaussian is generally not Gaussian.
>
> Across diverse settings, our method provides a better approximation than the baselines.
> Appendix N.1 groups the KL divergence results by variance level in boxplots and provides tables of Wasserstein distance and KL divergence evaluations for all combinations of networks, methods, and variances.
>
> **In the new Section 5.4 on variational Bayesian inference**, we compare our method and a leading deterministic approximation to Monte Carlo sampling from the predictive distribution, which we treat as ground truth.
> We demonstrate that our method achieves substantially more accurate moments (Figure 3 and Appendix L.1, Table 6, Figures 6–8).

---

> ### Author Response · Authors · 2025-11-21
>
> > The method scales as (O(n^3)) in layer width (due to covariance propagation), as noted by the authors, similar to linearization methods but higher than a single forward pass ((O(n^2))). While cheaper than Monte Carlo with many samples ((O(Mn^2))), the trade-off is unclear in practice. Runtime comparisons against Monte Carlo sampling vs the proposed method would clarify the actual efficiency benefits.
> >
> > ...
> >
> > How does the method compare in runtime ... to Monte Carlo sampling?
>
> Compared to ground-truth-quality quasi-Monte Carlo (QMC), our method is much faster.
> We benchmark the most complex analytical case (GeLU) against QMC in Appendix O.
> On "wide residual" networks, our method takes 88.4 ms ± 5.96 ms, whereas QMC takes 1.79 s ± 83 ms.
> On "deep residual" networks, our method takes 23.7 ms ± 957 μs, whereas QMC takes 878 ms ± 61.8 ms.

---

> ### Author Response · Authors · 2025-11-21
>
> > Real datasets are small (≤20k samples, ≤95 features).
>
> In our most realistic example, we use a topology similar to that in the UCI dataset experiments of Wright et al. (2024), "An Analytic Solution to Covariance Propagation in Neural Networks," Section 4.3.
> We believe that these datasets are rich enough to demonstrate the capabilities of approximate Gaussian moment matching.

---

> ### Author Response · Authors · 2025-11-21
>
> > Networks are shallow (≤3 layers, width ≤200).
>
> In the latest version, we increase the depth of `deep` random networks (Section 5.1) from eight layers to 20 layers.
> - For an example network (ReLU), see p. 148, Figure 105.
>
> We preserve the width of the `wide` networks at 400 neurons.
> - For an example network (GeLU with skip connections), see p. 88, Figure 45.

---

> ### Author Response · Authors · 2025-11-21
>
> > Evaluations are therefore small for modern NNs, not representative of modern deep learning architectures (e.g., ResNets, Transformers).
> >
> > ...
> >
> > How would the method scale to modern architectures with hundreds of layers?
>
> In short, layer-wise moment matching is a poor fit for very deep networks and architectures such as attention.
> It is likely that in the case of strong nonlinearity (large variances and/or weights), our method does not scale to modern architectures with hundreds of layers, due to fundamental limitations discussed in our top-level comment, Weakness 3.

---

> ### Author Response · Authors · 2025-11-21
>
> > The method appears to model only aleatoric uncertainty, while ignoring epistemic uncertainty. It is unclear why a practitioner would choose this approach over standard aleatoric uncertainty-aware models—such as heteroscedastic neural networks [5]—where the network learns to predict both the mean and the variance (as opposed to only the mean in standard regression tasks).
> >
> > ...
> >
> > Is the contribution intended as a general UQ method or a propagation approximation?
>
> The main innovation of our paper is that, for single layers, our propagation is not an approximation; it is exact.
> Our top-level comment, Weakness 1, clarifies the kinds of uncertainty that are explored in our examples.
> In short, moment matching can be applied to problems of uncertainty both in the input $x$ and in the function $f$ in $f(x)$.

---

> ### Author Response · Authors · 2025-11-21
>
> > Exact formulas are derived only for sine and probit activations, which are rarely used in modern practice compared to ReLU or GELU. This strongly limits applicability.
> >
> > ...
> >
> > Can the method be generalized or approximated for ReLU/GELU activations?
>
> Yes, we have extended the method to ReLU and GeLU in the latest revision.
> For a multivariate Gaussian input, our calculations are not merely approximate.
> They are exact.
>
> We hope that the comparisons to prior work in Section 2.1 (Methodology) make clear both the significance of our contribution (exact vs. approximate) and the novel methodology involved in its derivation.
>
> In the case of GeLU, our variational Bayes example (Section 5.4) shows that our method, even when combined with approximate multiplication of Gaussians, outperforms a leading GeLU moment matching method.

---

### Official Review · Reviewer_EoSR · 2025-11-02

**Soundness:** 2
**Presentation:** 1
**Contribution:** 1
**Rating:** 2
**Confidence:** 3

**Summary:**

This paper proposes a deterministic method for propagating input uncertainty through deep neural networks by computing closed-form expressions for mean and covariance transformations under specific activations (probit and sine) that admit analytic bivariate Gaussian moments. The method enables full-covariance propagation through deep and residual layers without resorting to Monte Carlo or linearization.

**Strengths:**

- The core mathematical derivations seem to be correct and technically consistent.
- Exact propagation of mean and covariance for specific activations is neat and removes the need for local linearization.

**Weaknesses:**

- In my opinion, the paper lacks a coherent narrative, with abrupt paragraph breaks, redundant notation, and sections that feel disconnected. Key ideas are scattered rather than developed progressively, making the reading experience not smooth.
- Current experiments are limited, with no demonstration on modern architectures, applications (e.g., uncertainty quantification), or interesting benchmarks. As of now, these experiments are not up to the standard of a conference like ICLR.
- The paper proposes a method that only works in sin and probit activations. These are not used in practice, so people will end up reverting to methods that already exist in the literature.
- The bound in section 4 is underwhelming.
- The authors do not discuss two recent works that seem to be relevant and actually discuss useful applications of why someone would want to propagate uncertainty across networks (including residual networks). Authors should cite these works [1, 2] and compare them in more serious uncertainty quantification tasks.

[1] Post-Hoc Uncertainty Quantification in Pre-Trained Neural Networks via Activation-Level Gaussian Processes. Bergna et al., 2025.
[2] Streamlining Prediction in Bayesian Deep Learning. Li et al., 2025.

**Questions:**

- Why would people use this method (Y_ana) in practice if they use common activation functions? Won't people revert to approximations than seem to work well in practice and exist already? [1, 2]

---

> ### Author Response · Authors · 2025-11-21
>
> Thank you for your useful feedback.
> We address general areas of concern in depth in the [top-level comment](https://openreview.net/forum?id=CHWjetQzYS&noteId=L1pHRQSKEy), but provide an itemized response in this thread.
>
> > In my opinion, the paper lacks a coherent narrative,
>
> We have added material to the Introduction and Methodology to reinforce the narrative that our paper emphasizes exactness of moment propagation.
> We support this narrative by discussing in Section 5.4 the contrast between our approach to benchmarking variational Bayes and that pursued in prior work.
>
> > with abrupt paragraph breaks, redundant notation, and sections that feel disconnected.
>
> We have added transitions that ease abrupt paragraph breaks and connect previously disconnected sections.
> We have eliminated the $\mathrm N^*$ notation, which was rarely used, and are open to further input on notation.
>
> > Key ideas are scattered rather than developed progressively, making the reading experience not smooth.
>
> The latest version of this paper consolidates our technical innovation in the Methodology section.
> We believe that, in addition to other changes, transferring the exposition of moment propagation baselines to the appendix makes the paper smoother.
> We are open to further suggestions on areas of improvement to the presentation.

---

> ### Author Response · Authors · 2025-11-21
>
> > Current experiments are limited, with no demonstration on modern architectures,
>
> The top-level comment qualifies our contribution to feedforward and residual neural networks.
> We are aware that these are simpler than architectures used in applied ML today.
> However, as we emphasize in the revised Introduction,
> "even in the simplest case of a single hidden layer with two neurons, no existing method can compute the mean and covariance exactly, in closed form."

---

> ### Author Response · Authors · 2025-11-21
>
> > applications (e.g., uncertainty quantification),
>
> The original manuscript presented the problem as propagating the uncertainty of $x$ in the neural network inference $f(x)$.
> We have reworded the manuscript to reserve the term "uncertainty quantification" for uncertainty in $f$.
> Furthermore, we have also added a new example on variational Bayes inference that deals precisely with uncertainty in $f$, and demonstrates the superior accuracy of our moment matching against a state-of-the-art analytical approach.

---

> ### Author Response · Authors · 2025-11-21
>
> > or interesting benchmarks. As of now, these experiments are not up to the standard of a conference like ICLR.
>
> We believe that the benchmarks presented in Figure 3 and Appendix L are narratively interesting, as they decisively juxtapose our method with the successive approximations of [Wright et al. (2024), "An Analytic Solution to Covariance Propagation in Neural Networks"](https://proceedings.mlr.press/v238/wright24a.html).

---

> ### Author Response · Authors · 2025-11-21
>
> > The paper proposes a method that only works in sin and probit activations.
> >
> > ...
> >
> > Why would people use this method (Y_ana) in practice if they use common activation functions?
>
> Our method now applies to GeLU, ReLU, and Heaviside activations.
>
> > These are not used in practice, so people will end up reverting to methods that already exist in the literature.
>
> In addition to a new suite of experimentation on random neural networks,
> in Section 5.4, we directly compare our method to Wright et al. (2024), which to our knowledge is the only method of its kind that already exists in the literature.

---

> ### Author Response · Authors · 2025-11-21
>
> > The bound in section 4 is underwhelming.
>
> If "underwhelming" refers to the fact that this bound is exponentially loose in deep networks, then we agree; however, the top-level comment explains why we expect looseness from _any_ polynomial-time (in the number of neurons) method for moment propagation through neural networks, which includes moment matching.
> If anything, the experiments on random networks of up to 20 layers suggest that hidden representations are more Gaussian in practice than in theory, which can be a fruitful question for further investigation.
>
> In our opinion, the strength of the bound is that it bounds the statistical distance between the true multivariate distribution of a hidden layer and its Gaussian approximation, at every layer.
> By comparison, the theoretical result of [Petersen et al. (2024), "Uncertainty Quantification via Stable Distribution Propagation"](https://openreview.net/forum?id=cZttUMTiPL) is applicable to a single neuron with Gaussian input.
>
>
> Our Wasserstein recursion is one of relatively few applications of [Nourdin et al. (2009), "Second order Poincaré inequalities and CLTs on Wiener space"](https://www.sciencedirect.com/science/article/pii/S0022123608005387) to realistic inference problems. (See [Ivan Nourdin's website](https://sites.google.com/site/malliavinstein/home) for a list of applications.)

---

> ### Author Response · Authors · 2025-11-21
>
> > The authors do not discuss two recent works that seem to be relevant and actually discuss useful applications of why someone would want to propagate uncertainty across networks (including residual networks). Authors should cite these works [1, 2]
>
> Thank you for bringing these related works to our attention.
> In the latest version, we cite [Li et al. (2025)](https://openreview.net/forum?id=pW387D5OUN) in the Introduction as well as in Section 5.4.
> We cite both of these works in the context of linearized uncertainty propagation in Related work, as well as in Appendix A, Table 4.
> As the authors acknowledge, Jacobian linearization is inaccurate in the large-variance regime:
> - We demonstrate this by an explicit calculation in Section 4, Example 1.
> - We show experimentally that even in the small-variance regime which favors linearization, our moment matching has superior accuracy (Appendix N.1, Figure 9).

---

> ### Author Response · Authors · 2025-11-21
>
> > and compare them in more serious uncertainty quantification tasks.
> >
> > ...
> >
> > Won't people revert to approximations than seem to work well in practice and exist already? [1, 2]
>
> While we find the Gaussian process methodology of [Bergna et al. (2025)](https://openreview.net/forum?id=y78LfYhlNy) interesting, it arises in the context of Laplace approximation, which is outside the current scope of our paper.
> We commend Li et al. (2025) for tackling modern architectures such as attention, and for the practical effectiveness of their approximations.
> They are outside the current scope of our paper, which emphasizes exact moment matching.
>
> On an area of common ground, Li et al. (2025) makes a mean-field assumption on pre-activations: "ignore correlation between $h$ for computational reason" (Appendix A.6).
> Likewise, [a very recent work](https://openreview.net/forum?id=utVraPnnZw) that builds on Bergna et al. (2025) also uses a mean-field assumption on neurons.
> The mean-field assumption can lead to inaccurate moment propagation.
> - We demonstrate this by an explicit calculation in Section 4, Example 3.
> - We show this experimentally in Appendix N.1, Figures 9–11.
> - Even though the mean-field assumption is not accurate for moment propagation, it can still be used for valid inference if the same assumption is made in training. We discuss this phenomenon in Section 5.4, footnote 3.
> - For this reason, our emphasis on distributional correctness leads us to benchmark against Monte Carlo as ground truth (Section 5.4).

---

> ### Author Response · Authors · 2025-11-21
>
> In fact, our work addresses all of the distribution propagation limitations surfaced in Li et al. (2025, p. 10).
>
> > The local linearisation of activation functions induces an error that depends on both the activation function and the location and scale of the distribution over the input to the activation function.
>
> Our experiments (Sections 5.1–3.5) show the superior accuracy of moment matching to linearization.
>
> > Moreover, we assume independence between the activations and model parameters for the local Gaussian approximation in linear layers and residual connections, which may incur a loss of information in the propagation. Especially, the independence assumption in the residual block is potentially harmful, and relaxing it would be a valuable future direction.
>
> We directly calculate the covariance between the nonlinearity and the residual using Stein's Lemma:
>  - probit (Appendix D, Lemma 9)
>  - GeLU (Appendix E, Lemma 12)
>  - ReLU (Appendix F, Lemma 15)
>  - Heaviside (Appendix G, Lemma 16)
>  - sine (Appendix H, Lemma 19)
>
>
>
> > Further, it would be interesting to estimate the induced approximation error to identify potential failure modes
>
> We identify failure modes of linearization and independence assumptions in our Section 4.

---

### Author Response · Authors · 2025-11-21
**Overview of major revision**

We are grateful for the attention of the reviewers and the concrete suggestions.
The latest revision in OpenReview highlights substantive additions to the main text in blue.

In this top-level comment, we enumerate major weaknesses raised in the first round of reviews, demonstrate that they are mitigated in the latest OpenReview revision, and reiterate what we consider to be the strengths of this work.
We will use separate threaded responses to address itemized reviewer feedback.

## References
1. [A. Wu, S. Nowozin, E. Meeds, R. E. Turner, J. M. Hernández-Lobato, and A. L. Gaunt, “Deterministic Variational Inference for Robust Bayesian Neural Networks,” Mar. 07, 2019.](https://openreview.net/forum?id=B1l08oAct7)

2. [O. Wright, Y. Nakahira, and J. M. F. Moura, “An Analytic Solution to Covariance Propagation in Neural Networks,” in Proceedings of The 27th International Conference on Artificial Intelligence and Statistics, PMLR, Apr. 2024, pp. 4087–4095.](https://proceedings.mlr.press/v238/wright24a.html)

3. [F. Petersen, A. Mishra, H. Kuehne, C. Borgelt, O. Deussen, and M. Yurochkin, “Uncertainty Quantification via Stable Distribution Propagation,” Feb. 13, 2024.](https://openreview.net/forum?id=cZttUMTiPL)

4. [B. J. Frey and G. E. Hinton, “Variational Learning in Nonlinear Gaussian Belief Networks,” Neural Computation, vol. 11, no. 1, pp. 193–213, Jan. 1999.](https://www.mitpressjournals.org/doi/abs/10.1162/089976699300016872)

5. [Michael Feischl and Fabian Zehetgruber. Computational Math with Neural Networks is Hard, May 2025. arXiv:2505.17751](https://arxiv.org/abs/2505.17751).


#### LLM usage statement

We used an LLM to suggest grammatical copyedits to our official comments.
Otherwise the text was drafted by hand, and we take full responsibility for its correctness.

---

> ### Author Response · Authors · 2025-11-21
>
> ## Weakness 1: no uncertainty _quantification_
> The original manuscript claims to address uncertainty quantification, but does not engage with some state-of-the-art methods (such as activation Gaussian processes) and common baselines (such as dropout) for representing the epistemic uncertainty of a trained neural network, i.e. of the $f$ of $f(x)$.
>
> #### Response
> 1. We acknowledge that our contribution is primarily located in uncertainty _propagation_, and that it is semantically loose to call this task UQ.
> We have removed references to UQ when we only mean the propagation of epistemic uncertainty in the $x$ of $f(x)$.
>
> 1. We have added examples (Section 5.4) that apply our method of distribution propagation to variational Bayesian neural networks.
> This is the application that connects the basic methodology of deterministic distribution propagation to epistemic uncertainty quantification, expressed as the predictive distribution of a neural network with weights drawn from a posterior distribution.
> References [1–2] derive moment-matching approximations to the evidence lower bound, of which [2] is more recent and accurate.
>     - We argue in Section 5.4 that the correct evaluation for distribution propagation is to measure discrepancy from Monte Carlo predictive distributions, which are treated as ground truth.
>     - In Figure 3, we compare our method to the covariance approximation of [2].
>     We observe that as more terms in the GeLU covariance series expansion [2] are added, the approximation [2] becomes more accurate.
>     Yet our approximation is two orders of magnitude more accurate than the longest partial sum provided in the work [2].
>
> 1. We consider other methods of epistemic uncertainty quantification to be outside the scope of this work. Comparisons of variational Bayes to other UQ methods can be found in [1, Appendix D].
>
> 1. We retain two examples (California Housing and Taiwan Bankruptcy) that use input noise, which is a form of epistemic uncertainty in the input $x$ of $f(x)$.

---

> > ### Author Response · Authors · 2025-11-21
> >
> > ## Weakness 2: not applicable to common activation functions
> > The original manuscript derives moment matching for two activation functions, probit and sine, and argues that they should be selected as an architecture choice if distribution propagation is desired.
> > But probit and sine are not used in large-scale machine learning today, so the scope of the original manuscript is limited to new networks that are trained with the express objective of input distribution propagation.
> >
> > #### Response
> > 1. We have derived moment matching for three additional activation functions: ReLU, GeLU, and Heaviside.
> >     - GeLU is used in frontier models such as DeepSeek-VL.
> >     - Heaviside is discontinuous and therefore not trainable in deterministic networks, but has nicely-behaved moments, and has historically been used for stochastic networks [4].
> >     - These three activation functions have heretofore resisted attempts at exact moment matching. We have added new material in the Introduction on historical attempts to derive moment matching for these activation functions.
> >     Moreover, in Section 2, Methodology, we accompany each activation function with the main idea of its moment derivation, as well as citations to papers that have approximated moment matching for these activation functions but stop short of deriving exact full covariance matrices.
> >     - The probabilistic derivations of ReLU (Appendix F) and Heaviside (Appendix G) moments use novel methods based on dominated limits of GeLU and probit, respectively.
> >
> > 1. Our paper also derives exact moment matching for layers with an affine superposition, such as residual layers. Whereas [1, Appendix C.1] also allows for skip connections, the covariance for Heaviside is mistaken (compare to the expression for $L$ in our Appendix G, Lemma 16). While the ReLU covariance is exact, the activation-function covariance term with which it is superimposed is not.
> >
> >     - Our strategy for computing skip covariances with Stein's Lemma (Appendix C, Lemma) is based on probabilistic ideas, which can be more elegant than improper Riemann integrals.

---

> ### Author Response · Authors · 2025-11-21
>
> ## Weakness 3: not applicable to state-of-the-art architectures
> The original manuscript tests analytic approximation on neural networks with at most eight layers and hundreds of neurons.
> These networks are not representative of state-of-the-art architectures such as attention, or of deep networks such as the state-of-the-art vision ResNets.
>
> #### Response
> 1. We have increased the depth of our "deep" random network ensemble from 8 to 20 layers (Appendix N).
>
> 2. Our methodology has not extended to attention, because we do not know of exact moment matching for the softmax or its derivatives. (cf. [a very recent work](https://arxiv.org/abs/2509.10695) that uses a local approximation to attention)
>
> 3. Depth is a weakness of layer-by-layer moment matching, as the distributional approximation becomes more wrong with each layer.
> This fact emerges from our deep random network examples (pp. 109–162).
> Our method's relative advantage over other distribution propagation methods becomes smaller (less than 10x KL divergence), and in some cases vanishes (e.g. on p. 138).
> Other works that apply moment matching to variational Bayes neural networks use one layer [1, Section 6] [2, Section 4.3].
>
>     - Our theoretical analysis in Appendix I supports this observation.
>     Thm. 1 degrades exponentially in the number of layers.
>     - Our adversarial example Section 4, Example 4, is an illustration of what happens when the distributional approximation becomes more wrong with each layer.
>     - While we are forthcoming about this fundamental limitation with layer-by-layer moment matching, this limitation should be understood in light of a deeper obstruction to matching moments across multiple layers:
>     exact, efficient moment matching for deep networks with many inputs would _a fortiori_ resolve P = NP in the affirmative [5].
>     - Despite theoretical challenges, moment matching can be practically accurate in networks of even 20 layers, e.g. for the GeLU activation function on p. 137.
>     This is a more demanding test than 5 layers [1, Appendix C].

---

> ### Author Response · Authors · 2025-11-21
>
> ## Weakness 4: disconnected organization/presentation flow
> The organization of the original manuscript is disconnected, and the presentation flow is not clear.
> In particular, there are abrupt paragraph and section breaks and unintroduced equations and theorem environments.
> The Adversarial Examples section is disconnected from the rest of the manuscript.
>
> #### Response
> 1. We have added transitions to introduce mathematical entities such as theorems and equations.
>
> 2. We have streamlined the theoretical sections into a new Methodology section (Section 2).
> In particular,
>     - We have transferred the baseline methods into Appendix B in order to ensure that our method is presented in a continuous narrative from exposition to theory to examples.
>     - We have clarified the ground truth for distribution propagation in Section 2.2.
>
> 3. We have added transition material to Section 4, Adversarial Examples, that relates the examples to claims in the Introduction.

---

> ### Author Response · Authors · 2025-11-21
>
> ## Strength: neat, exact derivations
> We thank the reviewers for their appreciation of the mathematical elegance of the derivations of $M$, $K$, and $L$ in the appendices of the original manuscript.
> We owe much of the inspiration for these techniques to the works cited in the introduction.
>
> We believe that the new derivations that support GeLU moments in the current version are even more interesting.
> For example, the differentiated Gaussian integrals in Appendix C, Lemmas 5–6 involve the trick of manipulating an unknown integral into a linear recursion, similar to the trigonometric integrals one encounters when first learning to integrate by parts.
>
> Creative ideas such as this one and the GeLU-to-ReLU limit were required to derive exact moment matching that is lacking in works cited.
>
> ## Strength: comprehensive experiments
> We thank the reviewers for their appreciation of the exhaustive and decisive test suite of random neural networks.
> Given that our method is exact on single hidden layers, we felt obliged to stress-test it back-to-back with other approximations on diverse multilayer networks in which the distributional approximation degrades.
>
> In the new example on variational Bayes networks, we again obtain decisive improvements in the easy-to-interpret KL divergence metric that compares approximate moments to Monte Carlo ground truth.

---

### Meta-Review · Area_Chair_NvXc · 2026-01-03

**Summary:**

The paper derives exact formulas for moment matching when passing a Normal distribution through one layer of a neural network (possibly residual, with several different activation functions allowed). These formulas can be used to estimate the distribution of the output of a deep neural network with noise applied to either the inputs or the weights, or both. The authors consider several applications of this methodology, such as in regression with input uncertainty and Bayesian neural nets.

The reviewers raised several core concerns.

1. The original manuscript only included a derviation for Probit and Sine activation functions.
2. The paper did not include experiments actually related to uncertainty quantification.
3. The experiments in the paper are limited to relatively toy and small networks.
4. The presentation of the paper was hard to follow, with disconnected organization.

Overall, I believe this paper addresses a well-defined question, which is relevant to several areas of deep learning (e.g. it can be useful in uncertainty quantification). However, the original manuscript submitted by the authors had significant issues and limitations, and was updated in major ways during the rebuttal. While I believe this paper has potential, in my opinion it needs to go through a proper review process again, given the major updates to the presentation, theoretical results (completely new results on new activation types) and experiments.

**Reviewer Concerns:**

During the rebuttal period, the authors significantly updated the manuscript, potentially addressing many of the concerns of the reviewers. Unfortunately, the reviewers did not have a chance to respond to the rebuttal.

In particular, the authors addressed the concern 1, and derived the moment matching formulas for ReLU and GeLU, as well as some other common activation functions. This was probably the main concern of the reviewers, and it is likely that it would be addressed by the new derivations, although I think the paper needs to go through another round of review to make sure that the new derivations are correct, and that the presentation is appropriate.

For the concern 2, the authors clarified the scope of the paper as "uncertainty propagation" instead of uncertainty quantification. They also added an experiment on Bayesian neural networks, which connects the paper more closely to UQ.

For concern 3, the contributions of the authors are necessarily related to the size of the models used: they derive exact moment-matching formulas for propagating Gaussian distributions through a neural network. The authors admit that their results do not apply to some architectures, such as attention. They extended the experiments to include deeper models. The concern may be partially addressed by the authors.

For the presentation, the authors performed a significant revision. It may partially address concerns with presentation.

Generally, I do believe that the authors may have addressed the core concern of the reviewers, which was the restriction on the activation functions. However, the paper underwent a major revision, and many aspects of the paper changed significantly. Given this, I believe it would be appropriate for the paper to go through another round of reviews.

**Reviewer Scores:**

Reviewer EoSR: 2 -> 4

Reviewer usVg: 4 -> 4/6

Reviewer L8SC: 4 -> 4/6

---

### Decision · Program_Chairs · 2026-01-26

Reject